# Beyond Weisfeiler-Lehman: A Quantitative Framework for GNN Expressiveness

**Bohang Zhang**[1][*][†] **Jingchu Gai**[2][*] **Yiheng Du**[3] **Qiwei Ye**[4] **Di He**[1][✉] **Liwei Wang**[1,5][✉]

[1]National Key Laboratory of General Artificial Intelligence, SIST, Peking University
[2]School of Mathematical Science, Peking University [3]Yuanpei College, Peking University
[4]Beijing Academy of Artificial Intelligence [5]Center for Machine Learning Research, Peking University
zhangbohang@pku.edu.cn, {gaijingchu,duyiheng}@stu.pku.edu.cn
qwye@baai.ac.cn, {dihe,wanglw}@pku.edu.cn

## Abstract

Designing expressive Graph Neural Networks (GNNs) is a fundamental topic in the graph learning community. So far, GNN expressiveness has been primarily assessed via the Weisfeiler-Lehman (WL) hierarchy. However, such an expressivity measure has notable limitations: it is inherently *coarse*, *qualitative*, and may not well reflect practical requirements (e.g., the ability to encode substructures). In this paper, we introduce a novel framework for *quantitatively* studying the expressiveness of GNN architectures, addressing all the above limitations. Specifically, we identify a fundamental expressivity measure termed *homomorphism expressivity*, which quantifies the ability of GNN models to count graphs under homomorphism. Homomorphism expressivity offers a complete and practical assessment tool: the completeness enables *direct* expressivity comparisons between GNN models, while the practicality allows for understanding concrete GNN abilities such as subgraph counting. By examining four classes of prominent GNNs as case studies, we derive simple, unified, and elegant descriptions of their homomorphism expressivity for both invariant and equivariant settings. Our results provide novel insights into a series of previous work, unify the landscape of different subareas in the community, and settle several open questions. Empirically, extensive experiments on both synthetic and real-world tasks verify our theory, showing that the practical performance of GNN models aligns well with the proposed metric.

## 1 Introduction

Owing to the ubiquity of graph-structured data in numerous applications, Graph Neural Networks (GNNs) have achieved enormous success in the field of machine learning over the past few years. However, one of the most prominent drawbacks of popular GNNs lies in the limited expressive power. In particular, Morris et al. (2019); Xu et al. (2019) showed that Message Passing GNNs (MPNNs) are intrinsically bounded by the 1-dimensional Weisfeiler-Lehman test (1-WL) in distinguishing non-isomorphic graphs (Weisfeiler & Lehman, 1968). Since then, the Weisfeiler-Lehman hierarchy has become a yardstick to measure the expressiveness and guide designing more powerful GNN architectures (see Appendix A.1 for an overview of representative approaches in this area).

However, as more and more architectures have been proposed, the limitations of the WL hierarchy are becoming increasingly evident. First, the WL hierarchy is arguably too coarse to evaluate the expressive power of *practical* GNN models (Morris et al., 2022; Puny et al., 2023). On one hand, architectures inspired by higher-order WL tests (Maron et al., 2019b;a; Morris et al., 2019) often suffer from substantial computation/memory costs. On the other hand, most practical and efficient GNNs are only proved to be strictly more expressive than 1-WL by leveraging toy example graphs (e.g., Zhang & Li, 2021; Bevilacqua et al., 2022; Wijesinghe & Wang, 2022a). Such a *qualitative* characterization may provide little insight into the models' true expressiveness. Besides, the expressive power brought from the WL hierarchy often does not align well with the one required in practice (Veličković, 2022). Hence, how to study the expressiveness of GNN models in a *quantitative*, *systematic*, and *practical* way remains a central research direction for the GNN community.

---

[*]Equal technical contributions.
[†]Project lead.

To address the above limitations, this paper takes a different approach by studying GNN expressivity from the following practical angle: *What structural information can a GNN model encode*? Since the ability to detect/count graph substructures is crucial in various real-world applications (Chen et al., 2020; Huang et al., 2023; Tahmasebi et al., 2023), many expressive GNNs have been proposed based on preprocessing substructure information (Bouritsas et al., 2022; Barceló et al., 2021; Bodnar et al., 2021b;a). However, instead of augmenting GNNs by manually preprocessed (task-specific) substructures, it is nowadays more desirable to design generic, domain-agnostic GNNs that can end-to-end *learn* different structural information suitable for diverse applications. This naturally gives rise to the fundamental question of characterizing the complete set of substructures prevalent GNN models can encode. Unfortunately, this problem is widely recognized as challenging even when examining simple structures like cycles (Fürer, 2017; Arvind et al., 2020; Huang et al., 2023).

**Our contributions**. Motivated by GNNs' ability to encode substructures, this paper presents a novel framework for quantitatively analyzing the expressive power of GNN models. Our approach is rooted in a critical discovery: given a GNN model $M$, the model's output representation for any graph $G$ can be fully determined by the structural information of $G$ over some pattern family $\mathcal{F}^M$, where $\mathcal{F}^M$ corresponds to precisely all (and only) those substructures that can be "encoded" by model $M$. In this way, the set $\mathcal{F}^M$ can be naturally viewed as an expressivity description of $M$: after identifying $\mathcal{F}^M$ for each model $M$, the expressivity of different models can then be qualitatively/quantitatively compared by simply looking at their *set inclusion relation* and *set difference*.

The crux here is to define an appropriate notion of "encodability" so that $\mathcal{F}^M$ can admit a simple description. We identify that a good candidate is the *homomorphism expressivity*: i.e., $\mathcal{F}^M$ consists of all substructures that can be counted by model $M$ under homomorphism (see Section 2 for a formal definition). Homomorphism is a foundational concept in graph theory (Lovász, 2012) and is linked to many important topics such as graph coloring, graph matching, and *subgraph counting*. With this concept, we are able to give complete, unified, and surprisingly elegant descriptions of the pattern family $\mathcal{F}^M$ for a wide range of mainstream GNN architectures listed below:

- **MPNN** (e.g., Gilmer et al., 2017; Hamilton et al., 2017; Kipf & Welling, 2017; Xu et al., 2019);
- **Subgraph GNN** (You et al., 2021; Zhang & Li, 2021; Bevilacqua et al., 2022; Qian et al., 2022);
- **Local GNN** (Morris et al., 2020; 2022; Zhang et al., 2023a; Frasca et al., 2022);
- **Folklore-type GNN** (Maron et al., 2019a; Zhang et al., 2023a; Feng et al., 2023).

Technically, the descriptions are based on a novel application and extension of the concept of *nested ear decomposition* (NED) in graph theory (Eppstein, 1992). We prove that: (i) (*necessity*) each model $M$ above can count (under homomorphism) a specific family of patterns $\mathcal{F}^M$, characterized by a specific type of NED; (ii) (*sufficiency*) any pattern $F \notin \mathcal{F}^M$ cannot be counted under homomorphism by model $M$; (iii) (*completeness*) for any graph, information collected from the homomorphism count in pattern family $\mathcal{F}^M$ determines its representation computed by model $M$. Therefore, *homomorphism expressivity is well-defined and is a complete expressivity measure for GNN models*.

Our theory can be generalized in various aspects. One significant extension is the node-level and edge-level expressivity for equivariant GNNs (Azizian & Lelarge, 2021; Geerts & Reutter, 2022), which can be naturally tackled by a fine-grained analysis of NED. As another non-trivial generalization, we study higher-order GNN variants for several of the above architectures and derive results by defining higher-order NED. Both aspects demonstrate the flexibility of our proposed framework, suggesting it as a general recipe for analyzing future architectures.

**Implications**. Homomorphism expressivity serves as a powerful toolbox for bridging different sub-areas in the GNN community, providing fresh understandings of a series of known results that were previously proved in complex ways, and answering a set of unresolved open problems. **First**, our results can readily establish a complete expressiveness *hierarchy* among all the aforementioned architectures and their higher-order extensions. This recovers and extends a number of results in Morris et al. (2020); Qian et al. (2022); Zhang et al. (2023a); Frasca et al. (2022) and answers their open problems (Section 4.1). In fact, our results go far beyond revealing the expressivity gap between models: we essentially answer *how large* the gap is and establish a systematic approach to constructing counterexample graphs. **Second**, based on the relation between homomorphism and subgraph count, we are able to characterize the subgraph counting power of GNN models for *all* patterns at graph, node, and edge levels, significantly advancing an open direction initiated in Fürer (2017); Arvind et al. (2020) (Section 4.2). As a special case, our results extend recent findings in

Huang et al. (2023) about the cycle counting power of GNN models, highlighting that Local 2-GNN can already subgraph-count all cycles/paths within 7 nodes (even at edge-level). **Third**, our results provide a new toolbox for studying the polynomial expressivity proposed recently in Puny et al. (2023), extending it to various practical architectures and answering an open question (Section 4.3). Empirically, an extensive set of experiments verifies our theory, showing that the homomorphism expressivity of different models matches well with their practical performance in diverse tasks.

## 2 PRELIMINARY

**Notations**. We use { } and {{ }} to denote sets and multisets, respectively. Given a (multi)set $S$, its cardinality is denoted as $|S|$. In this paper, we consider finite, undirected, vertex-labeled graphs with no self-loops or repeated edges. Let $G = (V_G, E_G, \ell_G)$ be a graph with vertex set $V_G$, edge set $E_G$, and label function $\ell_G$, where each edge in $E_G$ is a set $\{u, v\} \subset V_G$ of cardinality two, and $\ell_G(u)$ is the label of vertex $u$. The *rooted graph* $G^u$ is a graph obtained from $G$ by marking the special vertex $u \in V_G$; we can similarly consider marking two special vertices $u, v \in V_G$ (denote by $G^{uv}$). The *neighbors* of vertex $u$ is denoted as $N_G(u) := \{v \in V_G : \{u, v\} \in E_G\}$. A graph $F = (V_F, E_F, \ell_F)$ is a *subgraph* of $G$ if $V_F \subset V_G$, $E_F \subset E_G$, and $\ell_F(u) = \ell_G(u)$ for all $u \in V_F$. A *simple path* $P$ in $G$ is an edge set of the form $\{\{w_0, w_1\}, \cdots, \{w_{k-1}, w_k\}\} \subset E_G$ where $w_i \neq w_j$ for all $i \neq j$. Here, $w_0$ and $w_k$ are called endpoints of $P$ and other vertices are called internal points.

**Homomorphism, isomorphism, and subgraph count**. Given two graphs $F$ and $G$, a homomorphism from $F$ to $G$ is a mapping $f : V_F \to V_G$ that preserves edges and labels, i.e., $\ell_F(u) = \ell_G(f(u))$ for all $u \in V_F$, and $\{f(u), f(v)\} \in E_G$ for all $\{u, v\} \in E_F$. When the mapping $f$ exists, we say $F$ is homomorphic to $G$. We denote by $\mathsf{Hom}(F, G)$ the set of all homomorphisms from $F$ to $G$ and define $\mathsf{hom}(F, G) = |\mathsf{Hom}(F, G)|$, which counts the number of homomorphisms for pattern $F$ in graph $G$. If $f$ is further surjective on both vertices and edges, we call $G$ a *homomorphic image* of $F$. Denote by $\mathsf{Spasm}(F)$ the set of all homomorphic images of $F$, called the spasm of $F$. For rooted graphs, homomorphism should additionally preserve vertex marking: i.e., if $f$ is a homomorphism from $F^{uv}$ to $G^{xy}$, then $f(u) = x$ and $f(v) = y$.

A mapping $f : V_F \to V_G$ is called an isomorphism if $f$ is a bijection and both $f$ and its inverse $f^{-1}$ are homomorphisms. We denote by $\mathsf{Sub}(F, G)$ the set of all subgraphs of $G$ isomorphic to $F$ and define $\mathsf{sub}(F, G) = |\mathsf{Sub}(F, G)|$, which counts the number of patterns $F$ occurred in graph $G$ as a subgraph. We note that a similar definition holds for rooted graphs (e.g., $\mathsf{sub}(F^{uv}, G^{xy})$).

**Graph neural networks**. GNNs can be generally described as graph functions that are invariant under isomorphism. To achieve such invariance, most popular GNN models follow a *color refinement* (CR) paradigm: they maintain a feature representation (color) for each vertex or vertex tuples and iteratively refine these features through equivariant aggregation layers. Finally, there is a global pooling layer to merge all features and obtain the graph representation. Below, we separately define the corresponding CR algorithms for four mainstream classes of GNNs studied in this paper.

- **MPNN**. Given a graph $G$, MPNN maintains a color $\chi_G^{\mathsf{MP}}(u)$ for each vertex $u \in V_G$. Initially, the color only depends on the vertex label, i.e., $\chi_G^{\mathsf{MP},(0)}(u) = \ell_G(u)$. Then, in each iteration, the color is refined by the following update formula (where hash is a perfect hash function):

$$\chi_G^{\mathsf{MP},(t+1)}(u) = \mathsf{hash}\left(\chi_G^{\mathsf{MP},(t)}(u), \{\!\{\chi_G^{\mathsf{MP},(t)}(v) : v \in N_G(u)\}\!\}\right). \qquad (1)$$

After a sufficient number of iterations, the colors become stable. We denote by $\chi_G^{\mathsf{MP}}(u)$ the stable color of $u$, which is also the node feature of $u$ computed by the MPNN. The graph representation is defined as the multiset of node colors, i.e., $\chi_G^{\mathsf{MP}}(G) = \{\!\{\chi_G^{\mathsf{MP}}(u) : u \in V_G\}\!\}$.

- **Subgraph GNN**. It treats a graph $G$ as a set of subgraphs $\{\!\{G^u : u \in V_G\}\!\}$, each obtained from $G$ by marking a special vertex $u \in V_G$. Subgraph GNN maintains a color $\chi_G^{\mathsf{Sub}}(u, v)$ for each vertex $v$ in graph $G^u$. Initially, $\chi_G^{\mathsf{Sub},(0)}(u, v) = (\ell_G(v), \mathbb{I}[u = v])$, where the latter term distinguishes the special mark. It then runs MPNNs independently on each graph $G^u$:

$$\chi_G^{\mathsf{Sub},(t+1)}(u, v) = \mathsf{hash}\left(\chi_G^{\mathsf{Sub},(t)}(u, v), \{\!\{\chi_G^{\mathsf{Sub},(t)}(u, w) : w \in N_G(v)\}\!\}\right). \qquad (2)$$

Denote the stable color of $(u, v)$ as $\chi_G^{\mathsf{Sub}}(u, v)$. The node feature of $u$ computed by Subgraph GNN is defined by merging all colors in $G^u$, i.e., $\chi_G^{\mathsf{Sub}}(u) := \mathsf{hash}\left(\{\!\{\chi_G^{\mathsf{Sub}}(u, v) : v \in V_G\}\!\}\right)$. Finally, the graph representation is defined as $\chi_G^{\mathsf{Sub}}(G) = \{\!\{\chi_G^{\mathsf{Sub}}(u) : u \in V_G\}\!\}$.

- **Local GNN**. Inspired by the $k$-WL test (Grohe, 2017), Local $k$-GNN is defined by replacing all global aggregations in $k$-WL by *sparse* ones that only aggregate local neighbors, yielding a much more efficient CR algorithm. As an example, the iteration of Local 2-GNN has the following form and enjoys the same computational complexity as a Subgraph GNN.

$$\chi_G^{\mathsf{L},(t+1)}(u,v) = \mathsf{hash}\left(\chi_G^{\mathsf{L},(t)}(u,v), \{\!\!\{\chi_G^{\mathsf{L},(t)}(w,v) : w \in N_G(u)\}\!\!\}, \{\!\!\{\chi_G^{\mathsf{L},(t)}(u,w) : w \in N_G(v)\}\!\!\}\right). \tag{3}$$

Initially, $\chi_G^{\mathsf{L},(0)}(u,v) = (\ell_G(u), \ell_G(v), \mathbb{I}[u=v], \mathbb{I}[\{u,v\} \in E_G])$, which is called the *isomorphism type* of vertex pair $(u,v)$). We similarly denote the stable color as $\chi_G^{\mathsf{L}}(u,v)$ and define the node feature $\chi_G^{\mathsf{L}}(u)$ and graph representation $\chi_G^{\mathsf{L}}(G)$ as in the Subgraph GNN.

- **Folklore-type GNN**. The Folklore GNN (FGNN) is inspired by the standard $k$-FWL test (Cai et al., 1992). As an example, the iteration formula of 2-FGNN is written as follows:

$$\chi_G^{\mathsf{F},(t+1)}(u,v) = \mathsf{hash}\left(\chi_G^{\mathsf{F},(t)}(u,v), \{\!\!\{(\chi_G^{\mathsf{F},(t)}(w,v), \chi_G^{\mathsf{F},(t)}(u,w)) : w \in V_G\}\!\!\}\right). \tag{4}$$

One can similarly consider the more efficient Local 2-FGNN by only aggregating local neighbors, which has the same computational complexity as Local 2-GNN and Subgraph GNN:

$$\chi_G^{\mathsf{LF},(t+1)}(u,v) = \mathsf{hash}\left(\chi_G^{\mathsf{LF},(t)}(u,v), \{\!\!\{(\chi_G^{\mathsf{LF},(t)}(w,v), \chi_G^{\mathsf{LF},(t)}(u,w)) : w \in N_G(u) \cup N_G(v)\}\!\!\}\right). \tag{5}$$

The stable color, node feature, and graph representation can be similarly defined.

Finally, we note that the latter three types of GNNs can be naturally generalized into higher-order variants. We give a general definition of all these architectures in Appendix E.1. For the base case of $k = 1$, Subgraph $(k-1)$-GNN, Local $k$-GNN, and Local $k$-FGNN all reduce to the MPNN.

## 3 HOMOMORPHISM EXPRESSIVITY OF GRAPH NEURAL NETWORKS

### 3.1 HOMOMORPHISM EXPRESSIVITY

Given a GNN model $M$ and a substructure $F$, we say $M$ can count graph $F$ under homomorphism if, for any graph $G$, the graph representation $\chi_G^M(G)$ *determines* the homomorphism count $\mathsf{hom}(F, G)$. In other words, $\chi_G^M(G) = \chi_H^M(H)$ implies $\mathsf{hom}(F, G) = \mathsf{hom}(F, H)$ for any graphs $G, H$. The central question studied in this paper is, *what substructures $F$ can a GNN model $M$ count under homomorphism*? This gives rise to the notion of homomorphism expressivity defined below:

**Definition 3.1.** The homomorphism expressivity of a GNN model $M$, denoted by $\mathcal{F}^M$, is a family of (labeled) graphs satisfying the following conditions[1]:

a) For any two graphs $G, H$, $\chi_G^M(G) = \chi_H^M(H)$ *iff* $\mathsf{hom}(F, G) = \mathsf{hom}(F, H)$ for all $F \in \mathcal{F}^M$;

b) $\mathcal{F}^M$ is maximal, i.e., for any graph $F \notin \mathcal{F}^M$, there exists a pair of graphs $G, H$ such that $\chi_G^M(G) = \chi_H^M(H)$ and $\mathsf{hom}(F, G) \neq \mathsf{hom}(F, H)$.

**Example 3.2.** As a simple example, consider a maximally expressive GNN $M$ that can solve the graph isomorphism problem, i.e., it computes the same representation for two graphs *iff* they are isomorphic. Then, $\mathcal{F}^M$ contains all graphs. This is a classic result proved in Lovász (1967).

The significance of homomorphism expressivity can be justified in the following aspects. First, it is a *complete* expressivity measure. Based on item (a), the homomorphism count within $\mathcal{F}^M$ essentially captures *all* information embedded in the graph representation computed by model $M$. This contrasts with previously studied metrics such as the ability to compute biconnectivity properties (Zhang et al., 2023b) or count cycles (Huang et al., 2023), which only reflects restricted aspects of expressivity. Second, homomorphism expressivity is a *quantitative* measure and is much finer than qualitative expressivity results obtained from the graph isomorphism test. Specifically, by item (a), a GNN model $M_1$ is more expressive than another model $M_2$ in distinguishing non-isomorphic graphs *iff* $\mathcal{F}^{M_2} \subset \mathcal{F}^{M_1}$. Furthermore, by item (b), $M_1$ is strictly more expressive than $M_2$ *iff* $\mathcal{F}^{M_2} \subsetneq \mathcal{F}^{M_1}$, and the expressivity gap can be quantitatively understood via the set difference $\mathcal{F}^{M_1} \backslash \mathcal{F}^{M_2}$.

Consequently, by deriving which graphs are encompassed in the graph family $\mathcal{F}^M$, homomorphism expressivity provides a novel way to analyze and compare the expressivity of GNN models. In the next subsection, we will give exact characterizations of $\mathcal{F}^M$ for all models $M$ defined in Section 2.

---

[1]While homomorphism expressivity exists for all common GNNs such as the ones in Section 2, we note that it may not be well-defined for certain pathological GNNs. See Appendix F.1 for a deep discussion on it.

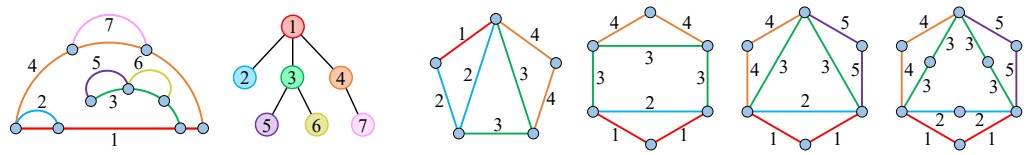

(a) Illustration of NED    (b) Examples of endpoint-shared/strong/almost-strong/general NED

Figure 1: Illustration of NED and its variants. The number $j$ next to each edge indicates that the edge belongs to the ear $P_j$. Different colors represent different ears. See Figure 9 for more examples.

## 3.2 MAIN RESULTS

To derive our main results, we leverage a concept in graph theory known as *nested ear decomposition* (NED), which is originally introduced in Eppstein (1992). Here, we adapt the definition as follows:

**Definition 3.3.** Given a graph $G$, a NED $\mathcal{P}$ is a partition of the edge set $E_G$ into a *sequence* of simple paths $P_1, \cdots, P_m$ (called ears), which satisfies the following conditions:

- Any two ears $P_i$ and $P_j$ with indices $1 \leq i < j \leq c$ do not intersect, where $c$ is the number of connected components of $G$.
- For each ear $P_j$ with index $j > c$, there is an ear $P_i$ with index $1 \leq i < j$ such that one or two endpoints of $P_j$ lie in ear $P_i$ (we say $P_j$ is *nested* on $P_i$). Moreover, except for the endpoints lying in ear $P_i$, no other vertices in $P_j$ are in any previous ear $P_k$ for $1 \leq k < j$. If both endpoints of $P_j$ lie in $P_i$, the subpath in $P_i$ that shares the endpoints of $P_j$ is called the *nested interval* of $P_j$ in $P_i$, denoted as $I(P_j) \subset P_i$. If only one endpoint lies in $P_i$, define $I(P_j) = \emptyset$.
- For all ears $P_j$, $P_k$ with $c < j < k \leq m$, either $I(P_j) \cap I(P_k) = \emptyset$ or $I(P_j) \subset I(P_k)$.

Intuitively, Definition 3.3 states that the relation between different ears forms a *forest*, in that each ear is nested on its parent. Moreover, the nested intervals either do not intersect or have inclusion relations for different children of the same parent ear. We give illustrations of NED in Figure 1.

In this paper, we considerably extend the concept of NED to several variants defined below:

- **Endpoint-shared NED**: a NED is called endpoint-shared if all ears with non-empty nested intervals share a common endpoint (see Figure 1(b,1)).
- **Strong NED**: a NED is called strong if for any two children $P_j$, $P_k$ ($j < k$) nested on the same parent ear, we have $I(P_j) \subset I(P_k)$ (see Figure 1(b,2)).
- **Almost-strong NED**: a NED is called almost-strong if for any children $P_j$, $P_k$ ($j < k$) nested on the same parent ear and $|I(P_j)| > 1$, we have $I(P_j) \subset I(P_k)$ (see Figure 1(b,3)).

We are now ready to present our main results:

**Theorem 3.4.** *For all GNN models $M$ defined in Section 2, the graph family $\mathcal{F}^M$ satisfying Definition 3.1 exists (and is unique). Moreover, each $\mathcal{F}^M$ can be separately described below:*

- *MPNN: $\mathcal{F}^{\mathsf{MP}} = \{F : F \text{ is a forest}\}$;*
- *Subgraph GNN: $\mathcal{F}^{\mathsf{Sub}} = \{F : F \text{ has an endpoint-shared NED}\}$;*
- *Local 2-GNN: $\mathcal{F}^{\mathsf{L}} = \{F : F \text{ has a strong NED}\}$;*
- *Local 2-FGNN: $\mathcal{F}^{\mathsf{LF}} = \{F : F \text{ has an almost-strong NED}\}$;*
- *2-FGNN: $\mathcal{F}^{\mathsf{F}} = \{F : F \text{ has a NED}\}$.*

Theorem 3.4 gives a unified description of the homomorphism expressivity for all popular GNN models defined in Section 2. Despite the elegant conclusion, the proof process is actually involved and represents a major technical contribution, so we present a proof sketch below. Our proof is divided into three parts, presented in Appendices C.2 to C.4. First, we show the existence of $\mathcal{F}^M$ for each model $M$ based on a beautiful theory developed in Dell et al. (2018). Using the technique of unfolding tree, we prove that $\mathcal{F}^M$ *at least* contains all graphs $F$ that allow a specific type of tree decomposition (Diestel, 2017), and the homomorphism information of these graphs determines the representation of any graph $G$ computed by $M$ (i.e., Definition 3.1(a) holds). However, characterizing $\mathcal{F}^M$ in terms of tree decomposition is sophisticated and not intuitive for most models $M$. In the next step, we give an *equivalent* description of $\mathcal{F}^M$ based on novel extensions of NED proposed in Definition 3.3, which is simpler and more elegant. In the last step, we prove that $\mathcal{F}^M$ does not contain other graphs. This is achieved by building non-trivial relations between three distinct theoretical tools: tree decomposition, pebble game (Cai et al., 1992), and Fürer graph (Fürer, 2001). Through a fine-grained analysis of the Fürer graphs expanded by $F \notin \mathcal{F}^M$ (see Theorems C.47 and C.53), we show they are precisely a pair of graphs satisfying Definition 3.1(b), thus concluding the proof.

**Discussions with Dvořák (2010); Dell et al. (2018).** Our work significantly extends a beautiful theory developed in Dvořák (2010); Dell et al. (2018), which showed that a pair of graphs $G$, $H$ are indistinguishable by 1-WL *iff* $\mathrm{hom}(F, G) = \mathrm{hom}(F, H)$ for all trees $F$, and more generally, they are indistinguishable by $k$-FWL *iff* $\mathrm{hom}(F, G) = \mathrm{hom}(F, H)$ for all graphs $F$ of bounded treewidth $k$. In this paper, we successfully generalize these results to a broad range of practical GNN models. Moreover, two distinct contributions are worth discussing. First, we highlight a key insight that homomorphism can serve as a fundamental *expressivity measure*, which has far-reaching consequences as will be elaborated in Section 4. To show that $\mathcal{F}^M$ is a valid expressivity measure, we prove an extra and non-trivial result that $\mathcal{F}^M$ is *maximal* (Definition 3.1(b)). Without this crucial property, $\mathcal{F}^{M_1} \supsetneq \mathcal{F}^{M_2}$ will not necessarily mean that model $M_1$ is strictly more expressive than $M_2$, thus preventing any quantitative comparison between models. Second, Dell et al. (2018) leveraged treewidth to describe results, which, unfortunately, cannot be applied to most GNN models studied here. Instead, we resort to the novel concept of NED, by which we successfully derive unified and elegant descriptions for all models. Moreover, as will be shown later, NED is quite flexible and can be naturally generalized to node/edge-level expressivity, which is not studied in prior work.

Finally, we remark that one can derive an equivalent (perhaps simpler) description of $\mathcal{F}^{\mathsf{Sub}}$, based on the fact that a graph $F$ has an endpoint-shared NED *iff* $F$ becomes a forest when deleting the shared endpoint. Formally, denoting by $F\backslash\{u\}$ the induced subgraph of $F$ over $V_F\backslash\{u\}$, we have

**Corollary 3.5.** $\mathcal{F}^{\mathsf{Sub}} = \{F : \exists u \in V_F \text{ s.t. } F\backslash\{u\} \text{ is a forest}\}$.

### 3.3 EXTENDING TO NODE/EDGE-LEVEL EXPRESSIVITY

So far, this paper mainly focuses on the graph-level expressivity, i.e., what information is encoded in the *graph representation*. In this subsection, we extend all results in Theorem 3.4 to the more fine-grained node/edge-level expressivity by answering what information is encoded in the node/edge features of a GNN (i.e., $\chi_G^M(u)$ or $\chi_G^M(u, v)$ in Section 2). This yields the following definition:

**Definition 3.6.** The node-level homomorphism expressivity of a GNN model $M$, denoted by $\mathcal{F}_\mathsf{n}^M$, is a family of connected *rooted* graphs satisfying the following conditions:

   a) For any connected graphs $G, H$ and vertices $u \in V_G$, $v \in V_H$, $\chi_G^M(u) = \chi_H^M(v)$ *iff* $\mathrm{hom}(F^w, G^u) = \mathrm{hom}(F^w, H^v)$ for all $F^w \in \mathcal{F}_\mathsf{n}^M$;

   b) For any connected rooted graph $F^w \notin \mathcal{F}_\mathsf{n}^M$, there exists a pair of connected graphs $G, H$ and vertices $u \in V_G$, $v \in V_H$ such that $\chi_G^M(u) = \chi_H^M(v)$ and $\mathrm{hom}(F^w, G^u) \neq \mathrm{hom}(F^w, H^v)$.

One can similarly define the edge-level homomorphism expressivity $\mathcal{F}_\mathsf{e}^M$ to be a family of connected rooted graphs, each marking two special vertices (we omit the definition for clarity). The following result exactly characterizes $\mathcal{F}_\mathsf{n}^M$ and $\mathcal{F}_\mathsf{e}^M$ for all models $M$ considered in this paper:

**Theorem 3.7.** *For all model $M$ defined in Section 2, $\mathcal{F}_\mathsf{n}^M$ and $\mathcal{F}_\mathsf{e}^M$ (except MPNN) exist. Moreover,*

   - *MPNN: $\mathcal{F}_\mathsf{n}^{\mathsf{MP}} = \{F^w : F \text{ is a tree}\}$;*

   - *Subgraph GNN:*
     $\mathcal{F}_\mathsf{n}^{\mathsf{Sub}} = \{F^w : F \text{ has a NED with shared endpoint } w\} = \{F^w : F\backslash\{w\} \text{ is a forest}\}$,
     $\mathcal{F}_\mathsf{e}^{\mathsf{Sub}} = \{F^{wx} : F \text{ has a NED with shared endpoint } w\} = \{F^{wx} : F\backslash\{w\} \text{ is a forest}\}$;

   - *2-FGNN: $\mathcal{F}_\mathsf{n}^{\mathsf{F}} = \{F^w : F \text{ has a NED where } w \text{ is an endpoint of the first ear}\}$,*
     $\mathcal{F}_\mathsf{e}^{\mathsf{F}} = \{F^{wx} : F \text{ has a NED where } w \text{ and } x \text{ are endpoints of the first ear}\}$.

*The cases of Local 2-GNN and Local 2-FGNN are similar to 2-FGNN by replacing "NED" with "strong NED" and "almost-strong NED", respectively.*

In summary, the node/edge-level homomorphism expressivity can be naturally described using NED by further specifying the endpoints of the first ear.

### 3.4 EXTENDING TO HIGHER-ORDER GNNS

Finally, we discuss how our results can be naturally extended to higher-order GNNs, thus providing a complete picture of the homomorphism expressivity hierarchy for infinitely many architectures. We focus on three representative examples: Subgraph $k$-GNN (Qian et al., 2022), Local $k$-GNN (Morris et al., 2020), and $k$-FGNN (Azizian & Lelarge, 2021). Subgraph $k$-GNN extracts a graph $G^{\boldsymbol{u}}$ for each vertex $k$-tuple $\boldsymbol{u} \in V_G^k$ and runs MPNNs independently, which recovers Subgraph GNN when $k = 1$. As the reader may have guessed, the following result exactly parallels Corollary 3.5:

**Theorem 3.8.** *The homomorphism expressivity of Subgraph $k$-GNN exists and can be described as $\mathcal{F}^{\mathsf{Sub}(k)} = \{F : \exists U \subset V_F \text{ s.t. } |U| \leq k \text{ and } F \backslash U \text{ is a forest}\}$.*

We next turn to Local $k$-GNN. To describe the result, we introduce a novel extension of Definition 3.3, called the $k$-order ear. Intuitively, it is formed by a graph of no more than $k$ vertices, plus $k$ paths each linking to a vertex in the graph (see Figure 2(a) for an illustration). Note that a 2-order ear is exactly a simple path. Then, we can naturally define the nested "interval" (see the solid orange lines in Figure 2(b) for an illustration) and thus define the concept of *$k$-order strong NED*. Due to space limit, a formal definition is deferred to Definition E.3. We have the following main result:

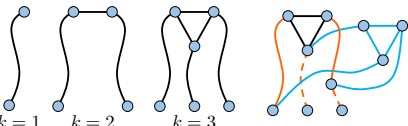

(a) $k$-order ears    (b) nested "interval"

Figure 2: Illustration of higher-order ears. Each curve indicates a path (with possibly zero length) and each straight segment indicates an edge.

**Theorem 3.9.** *The homomorphism expressivity of Local $k$-GNN exists and can be described as $\mathcal{F}^{\mathsf{L}(k)} = \{F : F \text{ has a } k\text{-order strong NED}\}$.*

Finally, let us consider the standard $k$-FGNN (or equivalently, the $k$-FWL). Unfortunately, we cannot find a description of its homomorphism expressivity based on some form of higher-order NED; nevertheless, it is easy to describe the results using the notion of treewidth (see Definition C.2). Specifically, denoting $\mathsf{tw}(F)$ to be the treewidth of graph $F$, we have the following result:

**Theorem 3.10.** *The homomorphism expressivity of $k$-FGNN exists and can be described as $\mathcal{F}^{\mathsf{F}(k)} = \{F : \mathsf{tw}(F) \leq k\}$.*

Interestingly, one can see that $\mathcal{F}^{\mathsf{Sub}(0)}$, $\mathcal{F}^{\mathsf{L}(1)}$, and $\mathcal{F}^{\mathsf{F}(1)}$ all degenerate to the family of forests, which coincides with the fact that all these higher-order GNNs reduces to MPNN for the base case.

## 4 IMPLICATIONS

The previous section has provided a complete description of the homomorphism expressivity for a variety of GNN models. In this section, we highlight the significance of these results through three different contexts. We will show how homomorphism expressivity can be used to link different GNN subareas, provide new insights into various known results, and answer a number of open problems.

### 4.1 QUALITATIVE EXPRESSIVITY COMPARISON

One direct corollary of Theorem 3.4 is that it readily enables expressivity comparison among all models in Section 2. This can be summarized below:

**Corollary 4.1.** *Under the notation of Theorem 3.4, $\mathcal{F}^{\mathsf{MP}} \subsetneq \mathcal{F}^{\mathsf{Sub}} \subsetneq \mathcal{F}^{\mathsf{L}} \subsetneq \mathcal{F}^{\mathsf{LF}} \subsetneq \mathcal{F}^{\mathsf{F}}$. Thus, the expressive power of the following GNN models strictly increases in order (in terms of distinguishing non-isomorphic graphs): MPNN, Subgraph GNN, Local 2-GNN, Local 2-FGNN, and 2-FGNN.*

*Proof.* $\mathcal{F}^{\mathsf{MP}} \subset \mathcal{F}^{\mathsf{Sub}}$ follows from Corollary 3.5 and the fact that deleting any vertex of a forest yields a forest. $\mathcal{F}^{\mathsf{Sub}} \subset \mathcal{F}^{\mathsf{L}}$ follows by the fact that any endpoint-shared NED is a strong NED. $\mathcal{F}^{\mathsf{L}} \subset \mathcal{F}^{\mathsf{LF}} \subset \mathcal{F}^{\mathsf{F}}$ follows similarly since any strong NED is an almost-strong NED and any almost-strong NED is a NED. To prove strict separation results, one can check that the four graphs in Figure 1(b) precisely reveal the gap between each pair of graph families, thus concluding the proof. $\square$

Corollary 4.1 recovers a series of results recently proved in Zhang et al. (2023a); Frasca et al. (2022). Compared to their results, our approach draws a much clearer picture of the expressivity gap between different architectures and essentially answers *how large* the gaps are. Moreover, we provide systematic guidance for finding counterexample graphs that unveil the expressivity gap: as shown in Corollary C.54, any graph $F \in \mathcal{F}^{M_2} \backslash \mathcal{F}^{M_1}$ immediately gives a pair of non-isomorphic graphs that reveals the gap between models $M_1$ and $M_2$. We note that this readily recovers the counterexamples constructed in Zhang et al. (2023a) and greatly simplifies their sophisticated case-by-case analysis.

We next turn to three types of higher-order GNNs studied in Section 3.4, for which we can establish a complete expressiveness hierarchy, as presented in Corollary 4.2. A graphical illustration of these results is given in Figure 3.

Figure 3: Expressiveness hierarchy of MPNN, Subgraph GNN, Local GNN, and FGNN.

**Corollary 4.2.** *Under the notations in Section 3.4, for any $k > 0$, the following hold:*

a) *$\mathcal{F}^{\mathsf{Sub}(k-1)} \subsetneq \mathcal{F}^{\mathsf{Sub}(k)}$. I.e., the expressive power of Subgraph $k$-GNN strictly increases with $k$;*

b) *$\mathcal{F}^{\mathsf{L}(k)} \subsetneq \mathcal{F}^{\mathsf{L}(k+1)}$. I.e., the expressive power of Local $k$-GNN strictly increases with $k$;*

c) *$\mathcal{F}^{\mathsf{Sub}(k)} \subsetneq \mathcal{F}^{\mathsf{L}(k+1)}$. I.e., Local $(k+1)$-GNN is strictly more expressive than Subgraph $k$-GNN;*

d) *$\mathcal{F}^{\mathsf{F}(k)} \subsetneq \mathcal{F}^{\mathsf{L}(k+1)} \subsetneq \mathcal{F}^{\mathsf{F}(k+1)}$. I.e., the expressive power of Local $(k+1)$-GNN lies strictly between $k$-FWL and $(k+1)$-FWL;*

e) *$\mathcal{F}^{\mathsf{Sub}(k)} \subsetneq \mathcal{F}^{\mathsf{F}(k+1)}$, and for all $k > 1$, $\mathcal{F}^{\mathsf{Sub}(k)} \backslash \mathcal{F}^{\mathsf{F}(k+1)} \neq \emptyset$ and $\mathcal{F}^{\mathsf{F}(k+1)} \backslash \mathcal{F}^{\mathsf{Sub}(k)} \neq \emptyset$. In other words, the expressive power of Subgraph $k$-GNN lies strictly within $(k+1)$-FWL, but it is incomparable to $k$-FWL when $k > 1$.*

Corollary 4.2 recovers results in Morris et al. (2020); Qian et al. (2022) and further answers two open problems. First, Corollary 4.2(c) is a new result that bridges Morris et al. (2020) with Qian et al. (2022) and partially answers an open question in Zhang et al. (2023a, Appendix C). Another new result is Corollary 4.2(d), which essentially answers a fundamental open problem raised in Frasca et al. (2022, Appendix E), showing that their proposed RelGN($k$) model is bounded by $k$-FWL with an inherent expressivity gap (see Appendix E.4 for a detailed discussion). To sum up, all these challenging open problems become straightforward through the lens of homomorphism expressivity.

## 4.2 SUBGRAPH COUNTING POWER

The significance of homomorphism expressivity can go much beyond qualitative comparisons between models. As another implication, it provides a systematic way to study GNNs' ability to encode structural information such as *subgraph count*, which has been found crucial in numerous practical applications. Specifically, a well-known result in graph theory states that, for any graphs $F, G$, the subgraph count $\mathsf{sub}(F, G)$ can be determined by $\mathsf{hom}(\tilde{F}, G)$ where $\tilde{F}$ ranges over all homomorphic images of $F$ (i.e., $\mathsf{Spasm}(F)$, see Section 2) (Lovász, 2012; Curticapean et al., 2017).

Mathematically, given any graph $F$, let $\mathsf{Spasm}^{\not\simeq}(F)$ be any maximal set of pairwise non-isomorphic graphs chosen from $\mathsf{Spasm}(F)$ (see Figure 4(a) for an illustration). Then, we have the following linear relation for all graph $G$:

$$\mathsf{sub}(F, G) = \sum_{\tilde{F} \in \mathsf{Spasm}^{\not\simeq}(F)} \alpha(F, \tilde{F}) \cdot \mathsf{hom}(\tilde{F}, G), \tag{6}$$

where $\alpha(F, \tilde{F}) \neq 0$ is a constant scalar coefficient independent of $G$. Based on this formula, we can easily study the subgraph counting power of GNN models as shown in Proposition 4.4.

**Definition 4.3.** Given a GNN model $M$, we say $M$ can subgraph-count graph $F$ at graph-level if $\chi_G^M(G) = \chi_H^M(H)$ implies $\mathsf{sub}(F, G) = \mathsf{sub}(F, H)$ for any graphs $G, H$. We say $M$ can subgraph-count rooted graph $F^w$ at node-level if $\chi_G^M(u) = \chi_H^M(v)$ implies $\mathsf{sub}(F^w, G^u) = \mathsf{sub}(F^w, H^v)$ for any graphs $G, H$ and vertices $u \in V_G, v \in V_H$. We can similarly define the edge-level subgraph counting ability for rooted graphs marking two special vertices.

**Proposition 4.4.** *For any GNN model $M$ defined in Section 2, it can subgraph-count graph $F$ (at graph-level) if $\tilde{F} \in \mathcal{F}^M$ for all $\tilde{F} \in \mathsf{Spasm}(F)$. It can subgraph-count $F^w$ (at node-level) if $\tilde{F}^w \in \mathcal{F}_{\mathsf{n}}^M$ for all $\tilde{F}^w \in \mathsf{Spasm}(F^w)$. A similar result holds for edge-level subgraph counting.*

The above proposition offers a simple way to affirm the ability of a GNN model $M$ to subgraph-count any pattern at graph/node/edge-level. On the other hand, one may wonder whether the converse direction also holds, i.e., $M$ cannot subgraph-count $F$ if there exists a homomorphic image $\tilde{F} \in \mathsf{Spasm}(F)$ such that $\tilde{F} \notin \mathcal{F}^M$. We find that it is indeed the case. Specifically, if the set $\mathsf{Spasm}(F) \backslash \mathcal{F}^M$ is not empty, then one can always find a pair of counterexample graphs $G, H$ such that $\chi_G^M(G) = \chi_H^M(H)$ but $\mathsf{sub}(F, G) \neq \mathsf{sub}(F, H)$. We eventually arrive at the following main theorem (see Appendix G.1 for a proof):

**Theorem 4.5.** *For any GNN model $M$ such that their homomorphism expressivity $\mathcal{F}^M$ exists, $M$ can subgraph-count $F$ iff $\mathsf{Spasm}(F) \subset \mathcal{F}^M$. Similar results hold for rooted graphs $F^u/F^{uv}$ by replacing $\mathcal{F}^M$ with node/edge-level homomorphism expressivity $\mathcal{F}_n^M/\mathcal{F}_e^M$.*

**Example 4.6.** As an example, we can readily characterize the cycle/path counting power of various GNNs. Denote by $C_n/P_n$ the simple cycle/path of $n$ vertices. Let $\{u,v\} \in E_{C_n}$ be any edge in $C_n$, and $\{w,x\} \in E_{P_n}$ be any edge in $P_n$ where $w$ is an endpoint of $P_n$. The following table lists exactly all cycles/paths each model can count at graph/node/edge-level.

| Structure | Cycle | | | Path | | |
|---|---|---|---|---|---|---|
| Model | $C_n$ | $C_n^u$ | $C_n^{uv}$ | $P_n$ | $P_n^w$ | $P_n^{wx}$ |
| MPNN | None | None | None | $n \leq 3$ | $n \leq 3$ | $n \leq 3$ |
| Subgraph GNN | $n \leq 7$ | $n \leq 4$ | $n \leq 4$ | $n \leq 7$ | $n \leq 4$ | $n \leq 4$ |
| Local 2-GNN Local 2-FGNN 2-FGNN | | $n \leq 7$ | | | $n \leq 7$ | |

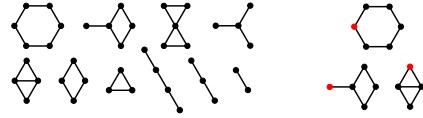

(a) $\mathsf{Spasm}^{\not\simeq}(C_6)$ has 10 graphs.  (b) Rooted $C_6$

Figure 4: Illustration of homomorphic images of the 6-cycle and rooted 6-cycle.

**Discussions with prior work**. Our results significantly extend Huang et al. (2023) in several aspects. First, we show Subgraph GNN *can* count 6-cycle at graph-level by simply enumerating its spasm (see Figure 4(a)). However, it cannot count rooted 5/6-cycle at node-level because the homomorphic image can contain cycles that do not pass the marked vertex (see Figure 4(b)). This provides novel insights into Huang et al. (2023) and extends their results (albeit with a simpler analysis). Second, we reveal that Local 2-GNN can already count all cycles/paths that 2-FWL can count (even at edge-level). This identifies a new architecture with both efficiency and strong expressiveness in subgraph counting, considerably extending the finding in the concurrent work of Zhou et al. (2023b).

In Appendix G.2 (Tables 4 and 5), we summarize the statistics of all moderate-size patterns each model can count under homomorphisms/subgraphs, which enables quantitative expressivity comparisons of different models in a clear and exact manner. We also comprehensively list the counting ability of all moderate-size patterns in Table 6, which we believe can be helpful for future research.

### 4.3 POLYNOMIAL EXPRESSIVITY

As the third implication, homomorphism expressivity is closely related to the *polynomial expressivity* recently proposed in Puny et al. (2023). Concretely, given a model $M$, a graph $F$ is in $\mathcal{F}^M$ if $M$ can express the invariant graph polynomial $P_F$ (defined in Puny et al. (2023), Section 2.2), and a rooted graph $F^{uv}$ is in $\mathcal{F}_e^M$ if $M$ can express the equivariant graph polynomial $P_{F^{uv}}$. Based on this connection, our work introduces a novel toolbox for studying polynomial expressivity via the NED framework and offers new insights into which graph polynomials can be computed for a variety of practical GNNs. Moreover, we readily settle an open question in Puny et al. (2023), which *upper bounds* the polynomial expressivity for their proposed PPGN++:

**Corollary 4.7.** *PPGN++ is bounded by (and thus as expressive as) the Prototypical edge-based model defined in Puny et al. (2023) for computing equivariant graph polynomials.*

Due to space limit, please refer to Appendix H for proof and more discussions.

## 5 EXPERIMENTS

This section aims to verify our theory through a comprehensive set of experiments. In each experiment, we implement four types of GNN models listed in Section 2, i.e., MPNN, Subgraph GNN, Local 2-GNN, and Local 2-FGNN. Note that all of these models are much more efficient than 2-FWL. Our primary objective here is not to produce SOTA results, but rather to provide a unified and equitable empirical comparison among these models. To ensure fairness, we employ the same GIN-based design (Xu et al., 2019) for all models and control their model sizes and training budgets to be roughly the same on each task. Details of model configurations are given in Appendix I. Our code is available at `https://github.com/subgraph23/homomorphism-expressivity`.

**Synthetic task**. We first test whether these GNN models can easily *learn* homomorphism information from data as our theory predicts. We use the benchmark dataset from Zhao et al. (2022a) and comprehensively test the homomorphism expressivity at graph/node/edge-level by carefully selecting 8 substructures shown in Table 1. The reported performance is measured by the normalized

Table 1: Experimental results on homomorphism counting. Red/blue nodes indicate marked vertices.

| Task / Model | Graph-level | | | Node-level | | Edge-level | | |
|---|---|---|---|---|---|---|---|---|
| MPNN | .300 | .233 | .254 | .505 | .478 | - | - | - |
| Subgraph GNN | .011 | .015 | .012 | .004 | .058 | .003 | .058 | .048 |
| Local 2-GNN | .008 | .008 | .010 | .003 | .004 | .005 | .006 | .008 |
| Local 2-FGNN | .003 | .005 | .004 | .005 | .005 | .007 | .007 | .008 |

Table 2: Experimental results on ZINC and Alchemy datasets. See Appendix I.4 for comparisons of more GNN models in literature.

| Task / Model | ZINC | | Alchemy |
|---|---|---|---|
| | Subset | Full | |
| MPNN | $.138 \pm .006$ | $.030 \pm .002$ | $.122 \pm .002$ |
| Subgraph GNN | $.110 \pm .007$ | $.028 \pm .002$ | $.116 \pm .001$ |
| Local 2-GNN | $.069 \pm .001$ | $.024 \pm .002$ | $.114 \pm .001$ |
| Local 2-FGNN | $.064 \pm .002$ | $.023 \pm .001$ | $.111 \pm .001$ |

Table 3: Experimental results on the (Chordal) Cycle Counting task.

| Task / Model | Graph-level | | | | | | Node-level | | | | | | Edge-level | | | | | |
|---|---|---|---|---|---|---|---|---|---|---|---|---|---|---|---|---|---|---|
| MPNN | .358 | .208 | .188 | .146 | .261 | .205 | .600 | .413 | .300 | .207 | .318 | .237 | - | - | - | - | - | - |
| Subgraph GNN | .010 | .020 | .024 | .046 | .007 | .027 | .003 | .005 | .092 | .082 | .050 | .073 | .001 | .003 | .090 | .096 | .038 | .065 |
| Local 2-GNN | .008 | .011 | .017 | .034 | .007 | .016 | .002 | .005 | .010 | .023 | .004 | .015 | .001 | .005 | .010 | .019 | .005 | .014 |
| Local 2-FGNN | .003 | .004 | .010 | .020 | .003 | .010 | .004 | .006 | .012 | .021 | .004 | .014 | .003 | .006 | .012 | .022 | .005 | .012 |

Mean Absolute Error (MAE) on the test dataset. It can be seen that the model performance indeed correlates to our theoretical predictions: (i) MPNN cannot encode any substructure under homomorphism; (ii) Subgraph GNN cannot encode the 2th, 3rd, 5th, 7th, 8th substructures; (iii) Local 2-GNN cannot encode the 3rd and 8th substructures; (iv) Local 2-FGNN can encode all substructures.

**Cycle counting power**. Cycles are important structures in numerous graph learning tasks, yet encoding them is notoriously hard for GNNs. We next test the ability of different GNN models to subgraph-count (chordal) cycles at graph/node/edge-level. We follow the setting in Frasca et al. (2022); Zhang et al. (2023a); Huang et al. (2023) and present results in Table 3 (measured by the normalized test MAE). Remarkably, *despite the same computational cost and model size*, Local 2-(F)GNN performs significantly better than Subgraph GNN and achieves good performance for counting all 3/4/5/6-cycles as well as chordal 4/5-cycles (even at edge-level). These results match Example 4.6 and may suggest Local 2-(F)GNN as generic, *efficient*, yet powerful architectures in solving chemical and biological tasks where counting cycles is essential (e.g., benzene rings).

**Real-world tasks**. We finally test these GNN models on three real-world benchmarks: ZINC-subset, ZINC-full (Dwivedi et al., 2020), and Alchemy (Chen et al., 2019a). Following the standard configuration, all models obey a 500K parameter budget. The results are shown in Table 2. It can be seen that the performance continues to improve when a more expressive model is used. In particular, Local 2-FGNN achieves the best performance on all tasks, suggesting that its theoretical expressivity guarantee can translate to practical performance in real-world settings.

## 6 CONCLUSION

In this paper, we present a new framework for systematically and quantitatively studying the expressive power of various GNN architectures. Through the lens of homomorphism expressivity, we give exact descriptions of the graph family each model can encode in terms of homomorphism counting. Our framework stands as a valuable toolbox to unify the landscape between different subareas in the GNN community, providing deep insights into a number of prior works and answering their open problems. In particular, one can establish a complete expressiveness hierarchy between models, determine the subgraph counting capabilities of GNNs at graph/node/edge-level, and understand their polynomial expressivity. On the theoretical side, our results establish deep connections with a series of fundamental topics in graph theory (see Appendix A.2); On the practical side, these results closely correlate with the empirical performance of GNN models, as demonstrated through extensive experiments. Finally, Appendix B outlines several open directions for further exploration, and we believe that the homomorphism expressivity framework paves a fresh way for future study of more expressive GNNs.

**Acknowledgement** Bohang Zhang would like to thank Shengjie Luo for helpful discussions. We sincerely appreciate all reviewers for the valuable suggestions. This work is supported by National Science and Technology Major Project (2022ZD0114902) and National Science Foundation of China (NSFC62276005, NSFC62376007).

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

# Appendix

## Table of Contents

# A  MORE RELATED WORK

## A.1  EXPRESSIVE GRAPH NEURAL NETWORKS

Since Morris et al. (2019); Xu et al. (2019) discovered the limited expressive power of MPNNs in distinguishing non-isomorphic graphs, a large amount of work has been devoted to developing GNNs with better expressiveness. Below, we briefly review representative approaches in this area. For a comprehensive survey on expressive GNNs, we refer readers to Morris et al. (2023).

**Higher-order GNNs**. Inspired by the relation between MPNNs and the 1-WL test, a natural approach to designing provably more expressive GNNs is to mimic the higher-order WL tests. This gives rise to two fundamental types of higher-order GNNs. One type of GNNs mimics the $k$-WL test (Grohe, 2017), and representative architectures include $k$-GNN (Morris et al., 2019) and $k$-IGN (Maron et al., 2019b;c); the other type of GNNs mimics the $k$-FWL (Folklore WL) test (Cai et al., 1992) and is referred to as the the $k$-FGNN (Maron et al., 2019a). Azizian & Lelarge (2021); Geerts & Reutter (2022) proved that each of these architectures is exactly as expressive as the corresponding higher-order WL/FWL test. Therefore, the expressiveness grows strictly as the order $k$ increases; when $k$ approaches infinity, they can universally approximate any continuous graph functions (Chen et al., 2019b; Keriven & Peyré, 2019). However, due to the inherent computation/memory complexity, these architectures are generally not practical in real-world applications.

**Local GNNs**. To improve computational efficiency, a subsequent line of work seeks to develop more scalable and practical GNN architectures by taking into account the local/sparse nature of graphs. Locality/sparsity is also an important inductive bias for graphs but is not well-exploited in higher-order GNNs, since their layer aggregation is inherently *global* and the graph adjacency information is only encoded in initial node features. To address these shortcomings, Morris et al. (2020) proposed the Local $k$-GNN (and several variants) as a replacement of $k$-GNN, which directly incorporates graph adjacency into network layers and only aggregates neighboring information instead of the global one. The authors further proved that Local $k$-GNN is strictly more expressive than $k$-GNN. Building upon Local $k$-GNN, Morris et al. (2022) proposed the $(k, s)$-SpeqNet that further reduces the computational cost by considering a subset of $k$-tuples whose vertices can be grouped into no more than $s$ connected components. A similar idea appeared in Zhao et al. (2022b), in which the authors proposed the $(k, s)$-SetGNN by considering $k$-sets instead of $k$-tuples. Besides Local $k$-GNN, recent architectures proposed in Frasca et al. (2022) and Zhang et al. (2023a) can be analogously understood as Local $k$-IGN and Local $k$-FGNN, respectively. Very recently, Feng et al. (2023); Zhou et al. (2023b) generalized the Local $k$-FGNN to a broad class of Folklore-type GNNs and achieved good performance on several benchmark datasets.

**Subgraph GNNs**. Graphs that are indistinguishable by WL tests typically possess a high degree of symmetry. In light of this observation, Subgraph GNNs have recently emerged as a compelling approach to designing expressive GNNs. The basic idea is to break symmetry by transforming the original graph into a collection of slightly modified subgraphs and feeding these subgraphs into a GNN model. The earliest forms of Subgraph GNNs may track back to Cotta et al. (2021); Papp et al. (2021) (albeit with a different motivation), where the authors proposed to feed node-deleted subgraphs into an MPNN. Papp & Wattenhofer (2022) later argued to use node marking instead of node deletion for better expressive power, resulting in the standard Subgraph GNN studied in this paper. Zhang & Li (2021); You et al. (2021) proposed the Nested GNN and Identity-aware GNN, both of which can be treated as variants of Subgraph GNNs that use *ego networks* as subgraphs. In particular, the heterogeneous message passing proposed in You et al. (2021) can also be seen as a form of node marking. We note that the model proposed in Vignac et al. (2020) can also be interpreted as a Subgraph GNN. Qian et al. (2022) proposed the higher-order Subgraph GNN by marking $k$ nodes per subgraph, resulting in $n^k$ different subgraphs when the original graph has $n$ vertices. We call this architecture Subgraph $k$-GNN in this paper. The authors proved that Subgraph $k$-GNN is strictly bounded by $(k + 1)$-FWL and is incomparable to $k$-FWL when $k > 1$. Zhou et al. (2023a) further generalized Subgraph $k$-GNN to $(l, k)$-GNN by using $l$-GNN instead of MPNN to process each subgraph. It was proved that $(l, k)$-GNN is bounded by $(k + l)$-GNN for $l \geq 2$.

Recently, Subgraph GNNs have been greatly extended by further allowing interactions *between* subgraphs. This is achieved by designing cross-subgraph aggregation layers (rather than feeding each subgraph independently into a GNN). Bevilacqua et al. (2022) developed the Equivariant Subgraph Aggregation Network that introduces a global aggregation between subgraphs. A similar design is

proposed in the concurrent work of Zhao et al. (2022a). These architectures were later proved to strictly improve the expressivity of the original Subgraph GNNs (Zhang et al., 2023b;a). Frasca et al. (2022) built a general design space of Subgraph GNNs that unifies prior work and showed that all models in this space are bounded by a variant of 2-IGN (dubbed the Local 2-IGN in this paper), which is then bounded by 2-FWL. Zhang et al. (2023a) later proved that Local 2-IGN is as expressive as Local 2-GNN and strictly less expressive than 2-FWL (2-FGNN). In this paper, we still use the term "Subgraph GNN" to refer to the original architecture in the previous paragraph, while using "Local 2-GNN" to refer to the general architecture in Frasca et al. (2022); Zhang et al. (2023a).

**Substructure-based GNNs**. Another line of work sought to develop expressive GNNs from practical considerations. In particular, Chen et al. (2020) pointed out that the ability of GNNs to detect/count graph substructures like path, cycle, and clique is crucial in numerous applications. Yet, MPNNs cannot subgraph-count any cycles/cliques. While higher-order WL tests can be more powerful in counting cycles (Fürer, 2017; Arvind et al., 2020), they suffer from substantial computational cost. As such, several works proposed to directly incorporate substructure/homomorphism counting into the node features as a preprocessing step to boost the expressiveness of MPNNs (Bouritsas et al., 2022; Barceló et al., 2021) or serve as an expressive graph kernel (Grohe, 2020; Nguyen & Maehara, 2020; Welke et al., 2023). Going beyond node features, Bodnar et al. (2021b;a); Giusti et al. (2023) further proposed a message-passing framework that enables interaction between nodes, edges, and higher-order substructures like cycles and cliques. We note that the Autobahn, TOGL, and Cy2C-GNN proposed in Thiede et al. (2021); Horn et al. (2022); Choi et al. (2022) can also be viewed as Substructure-based GNNs. However, most of the above approaches consider a fixed, predefined set of substructures rather than designing generic architectures that can *learn* substructures in an end-to-end fashion. Recently, Tahmasebi et al. (2023) proposed the RNP-GNN, an architecture that can count any substructure by recursively splitting a graph into a collection of vertex-marked subgraphs. We note that this design shares interesting similarities to higher-order subgraph GNNs and also the SpeqNet (Morris et al., 2022). Huang et al. (2023) proposed a generic model called $I^2$-GNN based on a variant of Subgraph 2-GNN, which can count 6-cycle at node-level. In this paper, we show the Local 2-GNN can already count all cycles/paths within 7 nodes even at edge-level while being more efficient than $I^2$-GNN (when using a similar ego network design). Finally, we remark that the polynomial expressivity proposed in Puny et al. (2023) can also be seen as a generalization of substructure counting, which further takes into account the real-valued node/edge features.

**Distance-based GNNs**. Besides structural information, distance serves as another fundamental attribute of a graph, which, again, is not captured by MPNNs and the 1-WL test. Li et al. (2020) first proposed to improve the expressive power of GNNs by augmenting node features with Distance Encoding (DE). Related to DE, another approach to injecting distance information is the $k$-hop MPNN, which aggregates $k$-hop neighbors in a message-passing layer (Feng et al., 2022; Abboud et al., 2022; Wang et al., 2023). Feng et al. (2022); Zhang et al. (2023a) proved that the expressive power of $k$-hop MPNN is strictly bounded by 2-FWL. Distance can also be naturally incorporated in Graph Transformers through *relative positional encoding*, yielding the Graphormer architecture that has achieved remarkable performance across various benchmarks (Ying et al., 2021). Recently, Zhang et al. (2023b) built an interesting connection between distance and *biconnectivity* properties, showing that distance-enhanced GNNs can detect cut vertices and cut edges of a graph. This provides insights into the practical superiority of these models as biconnectivity is closely linked to real applications in chemistry and social network analysis. Zhang et al. (2023a) proved that Local 2-GNN can provably encode the distance (and thus biconnectivity) of a graph.

**Spectral-based GNNs**. Graph spectra are also a class of fundamental properties and have long been used to design GNN models (Bruna et al., 2014; Defferrard et al., 2016). Balcilar et al. (2021b;a) showed that designing GNNs in the spectral domain can easily break the 1-WL expressivity. For Graph Transformers, Kreuzer et al. (2021); Dwivedi & Bresson (2020); Dwivedi et al. (2022) proposed to incorporate the spectra of the graph Laplacian matrix to boost the expressive power beyond the 1-WL test. Lim et al. (2023) further designed a principled equivariant architecture that takes the Laplacian eigenvalues and eigenvectors as inputs, which generalizes prior work.

**Other approaches**. Murphy et al. (2019); Chen et al. (2020) proposed Relational Pooling as a general approach to designing expressive GNN architectures, whose basic idea is to implement a permutation-invariant GNN by symmetrizing permutation-sensitive base models. Wijesinghe & Wang (2022b) proposed the GraphSNN, which improves the expressive power of MPNNs by using more distinguishing edge features. Specifically, each edge feature encodes the structure of the *over-*

*lap subgraph* of two 1-hop ego networks centered on the two endpoints of the edge. Very recently, Dimitrov et al. (2023) designed a GNN model that can distinguish all planar graphs, thus achieving a strong expressivity in chemical applications since molecular graphs are often planar.

## A.2 BROADER IMPACTS AND ADDITIONAL DISCUSSIONS

**Broader impact in graph theory**. Due to the fundamental nature of GNN architectures studied in this paper, our theoretical results may potentially have broader impacts on the graph theory community. Specifically, we study the color refinement (CR) algorithms corresponding to four types of (higher-order) GNNs: Subgraph $(k-1)$-GNN, Local $k$-GNN, Local $k$-FGNN, and $k$-FGNN. All algorithms can be seen as natural extensions of the 1-WL test since *they all reduce to 1-WL when $k = 1$*. In the graph theory community, Subgraph GNN has another name known as the *vertex-individualized* CR algorithm, which appears widely in literature (Babai, 2016; Rattan & Seppelt, 2023; Neuen & Schweitzer, 2018) and has become part of the core algorithm for fast graph isomorphism testing software (e.g., McKay & Piperno, 2014). On the other hand, Local $k$-GNN and Local $k$-FGNN are surprisingly related to the *guarded logic* (Barceló et al., 2020; De Rijke, 2000; Baader, 2003), since the aggregations are purely local (guarded by the edge). From this perspective, these CR algorithms can be seen as natural extensions of guarded logic in higher-order scenarios.

Besides these CR algorithms, our new extensions of NED may also have implications in graph theory. In particular, the strong NED (as well as its higher-order version) is elegant and may serve as a descriptive tool to characterize certain graph families. In addition, we establish intrinsic connections between NED and tree decomposition, which may have value in understanding other graph topics related to tree decomposition.

Finally, to our knowledge, the node/edge-level homomorphism and the corresponding subgraph counting abilities of different CR algorithms do not seem to have been systematically investigated before. Whereas in this paper, all graph/node/edge-level expressivity is studied in a unified manner. To achieve this, we introduce additional proof techniques which we believe may facilitate future study in related areas. For example, the original technique for constructing counterexample graphs satisfying Definition 3.1(b) does not apply to node/edge-level settings, since they are no longer counterexample graphs satisfying Definition 3.6(b) (no matter which vertices $u \in V_G, v \in V_H$ are marked). To address the problem, we propose clique-augmented Fürer graphs, a novel class of counterexample graphs that extend several prior works (e.g., Cai et al., 1992; Fürer, 2001), and conduct a fine-grained analysis of their automorphism property (see Appendix D.2). We believe this new technique can be used to generalize other results from graph-level to node/edge-level settings.

**Discussions with Barceló et al. (2021)**. In the GNN community, Barceló et al. (2021) first proposed to incorporate the homomorphism count of predefined substructures into node features as an approach to enhancing the expressivity of MPNNs. They systematically investigated the questions of what substructures are useful and how the homomorphism information of these substructures can boost the model expressivity to even break out $k$-FWL. Yet, they only gave a *partial* (incomplete) characterization of the substructures that can be counted by the *specific* $\mathcal{F}$-MPNN architecture and did not answer what substructures *cannot* be encoded, whereas our paper fully addresses both questions for a variety of popular GNN models. Note that these aspects are crucial to ensure that homomorphism expressivity is well-defined. As such, our paper first identifies that homomorphism expressivity is a complete, quantitative expressivity measure to compare different GNN models.

**Discussions with the concurrent work of Neuen (2023)**. After the initial submission, we became aware of a concurrent work (Neuen, 2023), which proved that $k$-FWL cannot count any graph with treewidth larger than $k$ under homomorphism. In our context, this result exactly shows that Definition 3.1(b) is satisfied, and thus the homomorphism expressivity of $k$-FWL is well-defined. Notably, their construction of counterexample graphs is also based on Fürer graphs. Nevertheless, the proof technique between the two works is quite different: the proof in Neuen (2023) is built upon a key concept called *oddomorphism*, while our proof is based on the relation between tree decomposition and the simplified pebble game developed in Zhang et al. (2023a). It is essential to underscore that our results and proof technique are more general and go beyond the standard $k$-FWL, in that (i) it applies to a broad range of color refinement algorithms related to practical GNN architectures and (ii) it further extends to node/edge-level homomorphism expressivity. Our theoretical results thus strictly incorporate the results in Neuen (2023). The approach in Neuen (2023) (based on oddomorphism) cannot be easily generalized to these settings.

## B    LIMITATIONS AND OPEN DIRECTIONS

There are still several open questions that are not fully explored in this paper. We list them below as promising directions for future study.

**Existence of homomorphism expressivity for refinement-based GNN architectures**. In this paper, we prove that homomorphism expressivity exists for a wide range of architectures defined in Section 2. On the other hand, we also note that it may not be well-defined for certain pathological GNNs, as illustrated in Appendix F.1. Given this observation, a fundamental question is: what conditions can guarantee that the homomorphism expressivity of a GNN exists? Here, we hypothesize that a very mild condition can suffice. Specifically, we conjecture that as long as a GNN architecture is defined following a general form of color refinement procedure that outputs stable color mappings, its homomorphism expressivity always exists. We leave this conjecture as an important open problem for future study.

**Regarding higher-order Local FGNN**. This paper characterizes the homomorphism expressivity for three classes of higher-order GNNs: Subgraph $k$-GNN, Local $k$-GNN, and $k$-FGNN. In particular, we introduce the $k$-order ear and $k$-order strong NED as a way to describe the homomorphism expressivity of Local $k$-GNN. However, it remains unclear how to give a simple description of the homomorphism expressivity for Local $k$-FGNN that can generalize the concept of almost-strong NED for $k = 2$. As such, our current expressiveness hierarchy (Figure 3) does not support Local $k$-FGNN yet. We leave this as an open problem and make the following conjecture below. We note that a similar open question has been informally raised in Zhang et al. (2023a).

**Conjecture B.1.** *For all $k \geq 2$, Local $k$-FGNN is strictly more expressive than Local $k$-GNN and strictly less expressive than $k$-FGNN.*

**Expressivity gap between Local 2-(F)GNN and 2-FGNN in practical aspects**. We have proved that 2-FGNN is strictly more expressive than Local 2-(F)GNN. However, from a practical perspective, we surprisingly find that the subgraph counting ability of Local 2-(F)GNN *matches* that of 2-FGNN for all structures within a moderate size (see Table 5), although the former is much more efficient. This leads to the intriguing question of what fundamental gaps exist between the two models in practical aspects, or is the efficiency gain free?

**Other architectures**. In this paper, we comprehensively study a variety of popular GNN architectures ranging from Subgraph GNNs and Local GNNs to higher-order GNNs, and further link these architectures to Substructure-based GNNs (see Appendix A.1). Yet, we still do not cover all popular GNN architectures, such as the GSWL-based Subgraph GNN (Zhang et al., 2023a; Bevilacqua et al., 2022), SpeqNet (Morris et al., 2022), and I²-GNN (Huang et al., 2023). Moreover, the classes of Distance-based GNNs and Spectral-based GNNs (Appendix A.1) are also widely used in practice, which deserve future study. We would like to raise the question of characterizing the homomorphism expressivity of Distance-based GNNs and Spectral-based GNNs as an important open question. In this way, one can gain deep insights into these models' true expressivity and enable quantitative comparisons between all mainstream architectures. Moreover, it will become clear to what extent other GNN models can encode distance and spectral information about a graph.

## C    PROOF OF THEOREM 3.4

This section gives the proof of the main theorem. For ease of reading, we first restate Theorem 3.4:

**Theorem 3.4.** *For all GNN models $M$ defined in Section 2, the graph family $\mathcal{F}^M$ satisfying Definition 3.1 exists (and is unique). Moreover, each $\mathcal{F}^M$ can be separately described below:*

- *Subgraph GNN: $\mathcal{F}^{\mathsf{Sub}} = \{F : F \text{ has an endpoint-shared NED}\}$;*

- *Local 2-GNN: $\mathcal{F}^{\mathsf{L}} = \{F : F \text{ has a strong NED}\}$;*

- *Local 2-FGNN: $\mathcal{F}^{\mathsf{LF}} = \{F : F \text{ has an almost-strong NED}\}$;*

- *2-FGNN: $\mathcal{F}^{\mathsf{F}} = \{F : F \text{ has a NED}\}$.*

For MPNN, since it is a special case of Subgraph $k$-GNN, the proof can be found in Appendix E.3.

## C.1 PRELIMINARY

**Additional notations and concepts**. Besides the notations defined in Section 2, we further define the following notations. We use the symbol $\mathcal{G}$ to denote the set of all finite, simple, undirected, labeled graphs. Let $G = (V_G, E_G, \ell_G)$ be a graph in $\mathcal{G}$. When the label is the same for all vertices, we can omit the term $\ell_G$ and write $G = (V_G, E_G)$. Given vertex $u \in V_G$, denote the degree of $u$ as $\deg_G(u) = |N_G(u)|$, and denote the closed neighborhood of $u$ as $N_G[u] = N_G(u) \cup \{u\}$. The shortest path distance between vertices $u$ and $v$ is denoted by $\mathsf{dis}_G(u, v)$.

Given a vertex set $S \subset V_G$, the *induced subgraph* of $G$ over $S$, denoted as $G[S]$, is the subgraph of $G$ with vertex set $S$ and edge set $\{\{u, v\} \in E_G : u, v \in S\}$. Similarly, without abuse of notation, given an edge set $R \subset E_G$, the *induced subgraph* of $G$ over $R$, denoted as $G[R]$, is the subgraph of $G$ with vertex set $\bigcup_{\{u,v\} \in R} \{u, v\}$ and edge set $R$. Given a vertex tuple $\boldsymbol{u} = (u_1, \cdots, u_k)$, denote by $G^{\boldsymbol{u}} = G^{u_1, \cdots, u_k}$ the rooted graph obtained from $G$ by marking vertices $u_1, \cdots, u_k$. The atomic type of $G$ over $\boldsymbol{u}$, denoted by $\mathsf{atp}_G(\boldsymbol{u})$, is a $k \times k$ matrix where the element at position $(i, j)$ is the tuple $(\mathbb{I}[u_i = u_j], \mathbb{I}[\{u_i, u_j\} \in E_G])$. Given two graphs $G, H$, the graph union $G \cup H$ is defined to be the graph $(V_G \cup V_H, E_G \cup E_H, \ell_{G \cup H})$, where $\ell_{G \cup H}(u) := \ell_G(u)$ for all $u \in V_G$ and $\ell_{G \cup H}(u) := \ell_H(u)$ for all $u \in V_H$. It is well-defined iff $\ell_G(u) = \ell_H(u)$ for all $u \in V_G \cap V_H$. Finally, we use the notation $G \simeq H$ to denote that $G$ and $H$ are isomorphic graphs.

A graph $T = (V_T, E_T, \ell_T)$ is called a tree if it does not contain cycles. Let $T^r$ be a rooted tree where $r$ is the root vertex. For each vertex $t \in V_T$, define its depth $\mathsf{dep}_{T^r}(t) := \mathsf{dis}_T(t, r)$ to be the distance to the root, and denote by $\mathsf{Desc}_{T^r}(t)$ the set of descendants of $t$, namely, $s \in \mathsf{Desc}_{T^r}(t)$ iff $\mathsf{dep}_{T^r}(s) = \mathsf{dep}_{T^r}(t) + \mathsf{dis}_T(t, s)$. For each $t \in V_T \backslash \{r\}$, denote by $\mathsf{pa}_{T^r}(t)$ the parent vertex of $t$, i.e., the unique vertex $s \in N_T(t)$ satisfying $\mathsf{dep}_{T^r}(t) = \mathsf{dep}_{T^r}(s) + 1$. Define the subtree of $T^r$ rooted at node $t$ by $T^r[t]$, which is exactly the induced subgraph $T[\mathsf{Desc}_{T^r}(t)]^t$ with root $t$.

**Tree decomposition**. Our proof is based on a central concept in graph theory, called tree decomposition. It can be formally defined below:

**Definition C.1** (Tree decomposition). Given a graph $G = (V_G, E_G, \ell_G)$, its tree decomposition is a tree $T = (V_T, E_T, \beta_T)$, where the label function $\beta_T : V_T \to 2^{V_G}$ satisfies the following conditions:

a) Each tree node $t \in V_T$ is associated to a non-empty subset of vertices $\beta_T(t) \subset V_G$ in $G$, called a *bag*. We say tree node $t$ *contains* vertex $u$ if $u \in \beta_T(t)$;

b) For each edge $\{u, v\} \in V_G$, there exists at least one tree node $t \in V_T$ that contains the edge, i.e., $\{u, v\} \subset \beta_T(t)$;

c) For each vertex $u \in V_G$, all tree nodes $t$ containing $u$ form a (non-empty) *connected* subtree. Formally, denoting $B_T(u) = \{t \in V_T : u \in \beta_T(t)\}$, then $T[B_T(u)]$ is connected.

If $T$ is a tree decomposition of $G$, we call the pair $(G, T)$ a tree-decomposed graph.

We remark that given a graph $G$, there are multiple ways to decompose it and thus its tree decomposition is not unique. Several examples of tree decomposition are given in Figure 5.

**Definition C.2** (Treewidth). The width of a tree decomposition is defined as one less than the maximum bag size, i.e., $\max_{t \in T} |\beta_T(t)| - 1$. The *treewidth* of a graph $G$, denoted as $\mathsf{tw}(G)$, is the minimum positive integer $k$ such that there exists a tree decomposition of width $k$.

Some important facts about treewidth are listed below:

**Fact C.3.** *For any graph $G$, the following hold:*

- *The treewidth of $G$ is at most $|V_G| - 1$, i.e., a trivial tree decomposition that only has one node $t$ and $\beta_T(t) = V_G$.*
- $\mathsf{tw}(G) = |V_G| - 1$ *iff $G$ is a clique.*
- $\mathsf{tw}(G) = 1$ *iff $G$ is a forest.*

The above definition of tree decomposition is quite flexible without constraints on the structure of the tree or the size of each bag. Below, we define several *restricted* variants of tree decomposition, which (we will later see) are closely related to the GNN architectures studied in this paper. To begin with, we first define a general concept that slight modifies the original definition (Definition C.1) such that the tree becomes *rooted* and each bag is a *multiset* of vertices rather than a set.

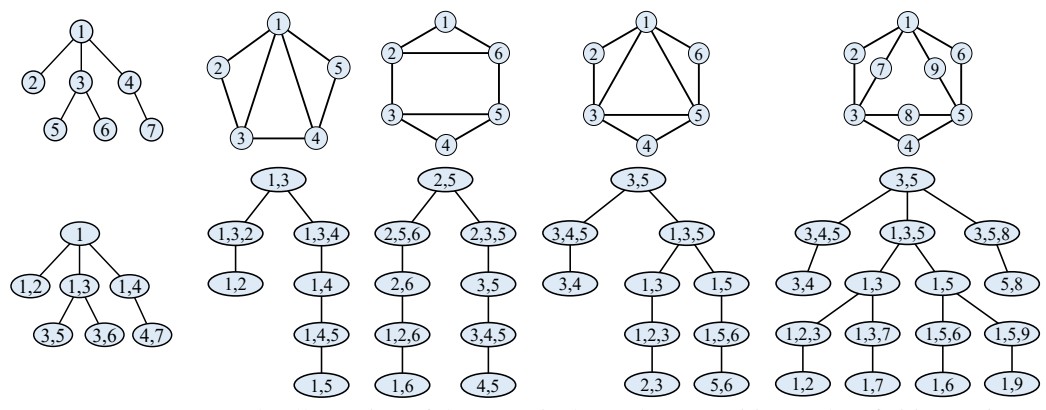

(a) The treewidth of a tree is 1.

(b) Illustration of the canonical tree decomposition and Definition C.6. The four tree-decomposed graphs are in $\mathcal{S}^{\mathsf{Sub}}$, $\mathcal{S}^{\mathsf{L}}$, $\mathcal{S}^{\mathsf{LF}}$, $\mathcal{S}^{\mathsf{F}}$, respectively.

Figure 5: Illustration of tree decomposition.

**Definition C.4** (Canonical tree decomposition). Given a graph $G = (V_G, E_G, \ell_G)$, a canonical tree decomposition of width $k$ is a rooted tree $T^r = (V_T, E_T, \beta_T)$ satisfying the following conditions:

a) The depth of $T$ is even, i.e. $\max_{t \in V_T} \mathsf{dep}_{T^r}(t)$ is even;

b) Each tree node $t \in V_T$ is associated to a multiset of vertices $\beta_T(t) \subset V_G$, called a *bag*. Moreover, $|\beta_T(t)| = k$ if $\mathsf{dep}_{T^r}(t)$ is *even* and $|\beta_T(t)| = k+1$ if $\mathsf{dep}_{T^r}(t)$ is *odd*;

c) For all tree edges $\{s, t\} \in E_T$ where $\mathsf{dep}_{T^r}(s)$ is even and $\mathsf{dep}_{T^r}(t)$ is odd, $\beta_T(s) \subset \beta_T(t)$ (where "$\subset$" denotes the multiset inclusion relation);

d) The conditions (b) and (c) in Definition C.1 are satisfied.

As examples, one can check that the tree decomposition of all graphs in Figure 5(b) is canonical, but the tree decomposition in Figure 5(a) is not. An important observation about canonical tree decomposition is shown below:

**Proposition C.5.** *Let $(F, T^r)$ be any tree-decomposed graph where $T^r$ is a canonical tree decomposition of $F$. For any vertices $u, v \in V_F$, either of the following holds:*

- *$u$ and $v$ are in the same bag of $T^r$, i.e., there is a node $t \in V_T$ such that $\{\!\{u, v\}\!\} \subset \beta_T(t)$;*

- *The induced subgraph $T[B_T(u) \cup B_T(v)]$ is disconnected.*

*Proof.* Assume that $u$ and $v$ are not in the same bag, i.e., $B_T(u) \cap B_T(v) = \emptyset$. Pick $s \in B_T(u)$ and $t \in B_T(v)$ such that $\mathsf{dep}_{T^r}(s)$ and $\mathsf{dep}_{T^r}(t)$ are minimal, respectively. Without loss of generality, assume that $\mathsf{dep}_{T^r}(s) \leq \mathsf{dep}_{T^r}(t)$. Then, $t \neq r$ is not the root node and thus we can pick its parent $\mathsf{pa}_{T^r}(t)$. It follows that $\mathsf{dep}_{T^r}(t)$ is odd and $\beta_T(\mathsf{pa}_{T^r}(t)) \subset \beta_T(t)$ by definition of canonical tree decomposition. Therefore, $u \notin \beta_T(\mathsf{pa}_{T^r}(t))$. Moreover, by the assumption that $\mathsf{dep}_{T^r}(s) \leq \mathsf{dep}_{T^r}(t)$, any node in $B_T(u)$ is not a descendent of $t$. We thus conclude that there does not exist a tree edge such that the two endpoints contain $u$ and $v$, respectively. □

Now we are ready to define several restricted variants of canonical tree decomposition:

**Definition C.6.** Define four families of tree-decomposed graphs $\mathcal{S}^{\mathsf{Sub}}$, $\mathcal{S}^{\mathsf{L}}$, $\mathcal{S}^{\mathsf{LF}}$, and $\mathcal{S}^{\mathsf{F}}$ as follows:

a) $(F, T^r) \in \mathcal{S}^{\mathsf{F}}$ iff $(F, T^r)$ satisfies Definition C.4 with width $k = 2$;

b) $(F, T^r) \in \mathcal{S}^{\mathsf{LF}}$ iff $(F, T^r)$ satisfies Definition C.4 with width $k = 2$, and for any tree node $t$ of odd depth, it has only one child if $w \notin \{v : v \in N_G[u], u \in \beta_T(s)\}$ where $s$ is the parent node of $t$ and $w$ is the unique vertex in $\beta_T(t) \backslash \beta_T(s)$;

c) $(F, T^r) \in \mathcal{S}^{\mathsf{L}}$ iff $(F, T^r)$ satisfies Definition C.4 with width $k = 2$, and any tree node $t$ of odd depth has only one child;

d) $(F, T^r) \in \mathcal{S}^{\mathsf{Sub}}$ iff $(F, T^r)$ satisfies Definition C.4 with width $k = 2$, and there exists a vertex $u \in V_G$ such that $u \in \beta_T(t)$ for all $t \in V_T$.

Examples of tree-decomposed graphs in the four families are illustrated in Figure 5(b).

Before closing this subsection, we define several notations for tree-decomposed graphs:

**Definition C.7.** Given canonical tree-decomposed graph $(G, T^r)$ and node $t \in V_T$, denote by $G[T^r[t]]$ the subgraph of $G$ induced by the vertex set $\{u : u \in \beta_{T^r}(s), s \in \mathsf{Desc}_{T^r}(t)\}$.

**Definition C.8.** Given two canonical tree-decomposed graphs $(G, T^r)$ and $(\tilde{G}, \tilde{T}^s)$, a pair of mappings $(\rho, \tau)$ is called an isomorphism from $(G, T^r)$ to $(\tilde{G}, \tilde{T}^s)$, denoted by $(G, T^r) \simeq (\tilde{G}, \tilde{T}^s)$, if the following hold:

a) $\rho$ is an isomorphism from $G$ to $\tilde{G}$;

b) $\tau$ is an isomorphism from $T^r$ to $\tilde{T}^s$ (ignoring labels $\beta$);

c) For any $t \in T^r$, $\rho(\beta_T(t)) = \beta_{\tilde{T}}(\tau(t))$.

**Equivalent formulation of GNN architectures**. Below, we give equivalent definitions for several GNN architectures presented in Section 2, which will be used in subsequent analysis. Let $G$ be a graph and $u, v \in V_G$. For all models $M$ including Subgraph GNN, Local 2-GNN, Local 2-FGNN, and 2-FGNN, we define the initial color $\tilde{\chi}_G^{M,(0)}(u, v)$ to be the isomorphism type of vertex pair $(u, v)$ (i.e., $\mathsf{atp}_G(u, v)$ plus labels of each vertex). Note that this matches the original definition except for Subgraph GNN. Then in each iteration $t$, the color is updated according to the following formula. Here, for clarity, we denote $\tilde{\chi}_G^{M,(t)}(u, S) = \{\!\{\tilde{\chi}_G^{M,(t)}(u, v) : v \in S\}\!\}$ and $\tilde{\chi}_G^{M,(t)}(S, v) = \{\!\{\tilde{\chi}_G^{M,(t)}(u, v) : u \in S\}\!\}$ for any model $M$ and set $S$.

- Subgraph GNN:

$$\tilde{\chi}_G^{\mathsf{Sub},(t+1)}(u, v) = \mathsf{hash}\left(\tilde{\chi}_G^{\mathsf{Sub},(t)}(u, v), \tilde{\chi}_G^{\mathsf{Sub},(t)}(u, N_G(v)), \tilde{\chi}_G^{\mathsf{Sub},(t)}(u, V_G)\right). \quad (7)$$

- Local 2-GNN:

$$\tilde{\chi}_G^{\mathsf{L},(t+1)}(u, v) = \mathsf{hash}\left(\tilde{\chi}_G^{\mathsf{L},(t)}(u, v), \tilde{\chi}_G^{\mathsf{L},(t)}(u, N_G(v)), \tilde{\chi}_G^{\mathsf{L},(t)}(N_G(u), v),\right.$$
$$\left. \tilde{\chi}_G^{\mathsf{L},(t)}(u, V_G), \tilde{\chi}_G^{\mathsf{L},(t)}(V_G, v)\right). \quad (8)$$

- Local 2-FGNN:

$$\tilde{\chi}_G^{\mathsf{LF},(t+1)}(u, v) = \mathsf{hash}\left(\tilde{\chi}_G^{\mathsf{LF},(t)}(u, v), \{\!\{(\tilde{\chi}_G^{\mathsf{LF},(t)}(w, v), \tilde{\chi}_G^{\mathsf{LF},(t)}(u, w)) : w \in N_G[u] \cup N_G[v]\}\!\},\right.$$
$$\left. \tilde{\chi}_G^{\mathsf{LF},(t)}(u, V_G), \tilde{\chi}_G^{\mathsf{LF},(t)}(V_G, v)\right). \quad (9)$$

- 2-FGNN:

$$\tilde{\chi}_G^{\mathsf{F},(t+1)}(u, v) = \mathsf{hash}\left(\tilde{\chi}_G^{\mathsf{F},(t)}(u, v), \{\!\{(\tilde{\chi}_G^{\mathsf{F},(t)}(w, v), \tilde{\chi}_G^{\mathsf{F},(t)}(u, w)) : w \in V_G\}\!\},\right.$$
$$\left. \tilde{\chi}_G^{\mathsf{F},(t)}(u, V_G), \tilde{\chi}_G^{\mathsf{F},(t)}(V_G, v)\right). \quad (10)$$

It can be seen that we additionally add global aggregations for these architectures. Moreover, in Local 2-FGNN we replace the neighbors by closed neighbors $N_G[u] \cup N_G[v]$. The stable color of $(u, v)$ for different models is denoted by $\tilde{\chi}_G^{\mathsf{Sub}}(u, v)$, $\tilde{\chi}_G^{\mathsf{L}}(u, v)$, $\tilde{\chi}_G^{\mathsf{LF}}(u, v)$, $\tilde{\chi}_G^{\mathsf{F}}(u, v)$, respectively. We have the following result:

**Proposition C.9.** *Let $M \in \{\mathsf{Sub}, \mathsf{L}, \mathsf{LF}, \mathsf{F}\}$ be any model. For any graphs $G, H$, $\tilde{\chi}_G^M(G) = \tilde{\chi}_H^M(H)$ iff $\chi_G^M(G) = \chi_H^M(H)$. Furthermore, if $G$ and $H$ are connected, then for any vertices $u, v \in V_G$ and $x, y \in V_H$, $\tilde{\chi}_G^M(u, v) = \tilde{\chi}_H^M(x, y)$ iff $\chi_G^M(u, v) = \chi_H^M(x, y)$. In other words, the color mapping $\tilde{\chi}^M$ is as fine as the original one $\chi^M$.*

*Proof.* The proof simply follows from Zhang et al. (2023a, Proposition 4.2 and Theorem 4.4), because (i) node marking in the initial color of Subgraph GNN is as expressive as using the isomorphism type, (ii) the global aggregation does not improve the expressivity when the corresponding local aggregation is presented, (iii) the single-point aggregation does not improve the expressivity in Local 2-FGNN. □

## C.2 PART 1: TREE DECOMPOSITION

We first define the *unfolding tree* of different CR algorithms, which is a standard tool in analyzing GNN expressivity.

**Definition C.10** (Unfolding tree of Subgraph GNN). Given a graph $G$, vertices $u, v \in V_G$, and a non-negative integer $D$, the depth-$2D$ Subgraph GNN unfolding tree of graph $G$ at $(u, v)$, denoted as $\left( F_G^{\mathsf{Sub},(D)}(u, v), T_G^{\mathsf{Sub},(D)}(u, v) \right)$, is a tree-decomposed graph $(F, T^r) \in \mathcal{S}^{\mathsf{Sub}}$ constructed as follows:

1. **Initialization**. At the beginning, $F = G[\{u, v\}]$ (if $u = v$, $F$ only has one vertex), and $T$ only has a root node $r$ with $\beta_T(r) = \{\!\{u, v\}\!\}$. Define a mapping $\pi : V_F \to V_G$ as $\pi(u) = u$ and $\pi(v) = v$.

2. **Loop for $D$ rounds**. For each leaf node $t$ in $T^r$, do the following procedure:
   Let $\beta_T(t) = \{\!\{u, x\}\!\}$. For each $w \in V_G$, add a fresh child node $t_w$ to $T^r$ and designate $t$ as its parent. Then, consider the following three cases:

   a) If $w \neq \pi(u)$ and $w \neq \pi(x)$, then add a fresh vertex $z$ to $F$ and extend $\pi$ with $\pi(z) = w$. The label of $z$ in $F$ is set by $\ell_F(z) = \ell_G(w)$. Define $\beta_T(t_w) = \beta_T(t) \cup \{\!\{z\}\!\}$. Then, we add edges between $z$ and $\beta_T(t)$, so that $\pi$ is an isomorphism from $F[\beta_T(t_w)]$ to $G[\pi(\beta_T(t_w))]$.
   b) If $w = \pi(u)$, then we simply set $\beta_T(t_w) = \beta_T(t) \cup \{\!\{u\}\!\}$ without modifying graph $F$.
   c) If $w = \pi(x)$, then we simply set $\beta_T(t_w) = \beta_T(t) \cup \{\!\{x\}\!\}$ without modifying graph $F$.

   Finally, add a fresh child node $t'_w$ to $T^r$, designate $t_w$ as its parent, and set $\beta_T(t'_w)$ based on the following three cases:

   a) If $w \neq \pi(u)$ and $w \neq \pi(x)$, then $\beta_T(t'_w) = \{\!\{u, z\}\!\}$.
   b) If $w = \pi(u)$, then $\beta_T(t'_w) = \{\!\{u, u\}\!\}$.
   c) If $w = \pi(x)$, then $\beta_T(t'_w) = \{\!\{u, x\}\!\}$.

It is easy to see that the depth of tree $T^r$ increases by 2 after each round, $T^r$ is always a canonical tree decomposition of $F$, and $(F, T^r) \in \mathcal{S}^{\mathsf{Sub}}$. An illustration of the construction of unfolding tree is given in Figure 6(a).

We next define the unfolding tree of Local 2-GNN, which differs in the loop part such that the bags $\beta_T(t_w)$ and $\beta_T(t'_w)$ now do not necessarily contain $u$.

**Definition C.11** (Unfolding tree of Local 2-GNN). Given a graph $G$, vertices $u, v \in V_G$, and a non-negative integer $D$, the depth-$2D$ Local 2-GNN unfolding tree of graph $G$ at $(u, v)$, denoted as $\left( F_G^{\mathsf{L},(D)}(u, v), T_G^{\mathsf{L},(D)}(u, v) \right)$, is a tree-decomposed graph $(F, T^r) \in \mathcal{S}^{\mathsf{L}}$ constructed as follows:

1. **Initialization**. The procedure is exactly the same as Subgraph GNN (Definition C.10).

2. **Loop for $D$ rounds**. For each leaf node $t$ in $T^r$, do the following procedure:
   Let $\beta_T(t) = \{\!\{x, y\}\!\}$. For each $w \in V_G$, add a fresh child node $t_w$ to $T$ and designate $t$ as its parent. Then, consider the following three cases:

   a) If $w \neq \pi(x)$ and $w \neq \pi(y)$, then add a fresh vertex $z$ to $F$ and extend $\pi$ with $\pi(z) = w$. The label of $z$ in $F$ is set by $\ell_F(z) = \ell_G(w)$. Define $\beta_T(t_w) = \beta_T(t) \cup \{\!\{z\}\!\}$. Then, we add edges between $z$ and $\beta_T(t)$, so that $\pi$ is an isomorphism from $F[\beta_T(t_w)]$ to $G[\pi(\beta_T(t_w))]$.
   b) If $w = \pi(x)$, then we simply set $\beta_T(t_w) = \beta_T(t) \cup \{\!\{x\}\!\}$ without modifying graph $F$.
   c) If $w = \pi(y)$, then we simply set $\beta_T(t_w) = \beta_T(t) \cup \{\!\{y\}\!\}$ without modifying graph $F$.

   Next, add a fresh child node $t'_w$ in $T^r$, designate $t_w$ as its parent, and set $\beta_T(t'_w)$ based on the following three cases:

   a) If $w \neq \pi(x)$ and $w \neq \pi(y)$, then $\beta_T(t'_w) = \{\!\{x, z\}\!\}$.
   b) If $w = \pi(x)$, then $\beta_T(t'_w) = \{\!\{x, x\}\!\}$.
   c) If $w = \pi(y)$, then $\beta_T(t'_w) = \{\!\{x, y\}\!\}$.

   Finally, we repeat the above procedure (point 2) once more, but this time the bag $\beta_T(t'_w)$ is replaced by the following three cases (changing $x$ to $y$):

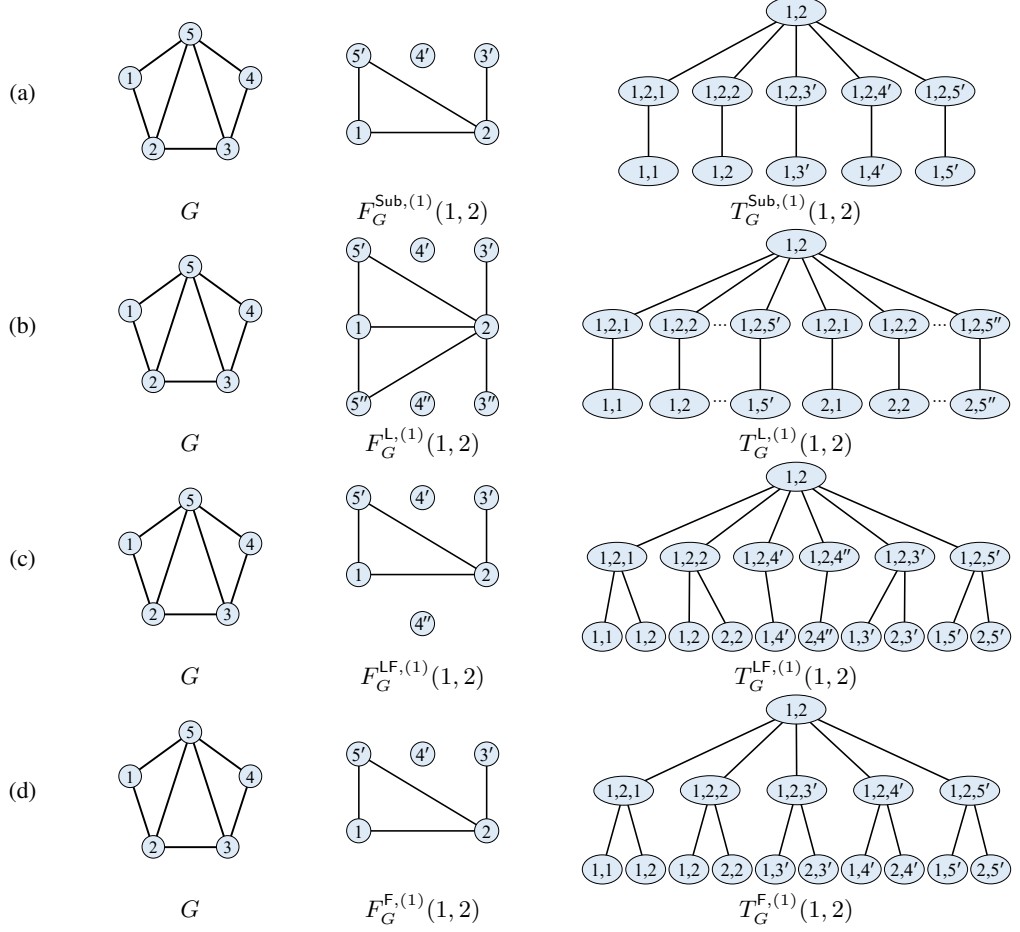

Figure 6: The depth-2 unfolding tree of graph $G$ at vertex pair (1,2) for Subgraph GNN, Local 2-GNN, Local 2-FGNN, and 2-FGNN, respectively.

    a) If $w \neq \pi(x)$ and $w \neq \pi(y)$, then $\beta_T(t'_w) = \{\!\{y, z\}\!\}$.
    b) If $w = \pi(y)$, then $\beta_T(t'_w) = \{\!\{y, y\}\!\}$.
    c) If $w = \pi(x)$, then $\beta_T(t'_w) = \{\!\{x, y\}\!\}$.

An illustration of the construction of unfolding tree is given in Figure 6(b).

We next define the unfolding tree of Local 2-FGNN, which differs in the loop part such that the node $t_w$ can have two children under certain conditions.

**Definition C.12** (Unfolding tree of Local 2-FGNN). Given a graph $G$, vertices $u, v \in V_G$, and a non-negative integer $D$, the depth-$2D$ Local 2-FGNN unfolding tree of graph $G$ at $(u, v)$, denoted as $\left( F_G^{\mathsf{LF},(D)}(u, v), T_G^{\mathsf{LF},(D)}(u, v) \right)$, is a tree-decomposed graph $(F, T^r) \in \mathcal{S}^{\mathsf{LF}}$ constructed as follows:

1. **Initialization**. The procedure is exactly the same as Subgraph GNN (Definition C.10).

2. **Loop for $D$ rounds**. For each leaf node $t$ in $T^r$, do the following procedure:
   Let $\beta_T(t) = \{\!\{x, y\}\!\}$. For each $w \in N_G[\pi(x)] \cup N_G[\pi(y)]$, add a fresh child node $t_w$ to $T$ and designate $t$ as its parent. Then, consider the following three cases:

       a) If $w \neq \pi(x)$ and $w \neq \pi(y)$, then add a fresh vertex $z$ to $F$ and extend $\pi$ with $\pi(z) = w$. The label of $z$ in $F$ is set by $\ell_F(z) = \ell_G(w)$. Define $\beta_T(t_w) = \beta_T(t) \cup \{\!\{z\}\!\}$. Then, we add edges between $z$ and $\beta_T(t)$, so that $\pi$ is an isomorphism from $F[\beta_T(t_w)]$ to $G[\pi(\beta_T(t_w))]$.
       b) If $w = \pi(x)$, then we simply set $\beta_T(t_w) = \beta_T(t) \cup \{\!\{x\}\!\}$ without modifying graph $F$.
       c) If $w = \pi(y)$, then we simply set $\beta_T(t_w) = \beta_T(t) \cup \{\!\{y\}\!\}$ without modifying graph $F$.

Next, add two fresh children $t'_w$ and $t''_w$ in $T^r$, designate $t_w$ as their parent, and set $\beta_T(t'_w)$ and $\beta_T(t''_w)$ based on the following three cases:

  a) If $w \neq \pi(x)$ and $w \neq \pi(y)$, then $\beta_T(t'_w) = \{\!\{x, z\}\!\}$ and $\beta_T(t''_w) = \{\!\{y, z\}\!\}$.
  b) If $w = \pi(x)$, then $\beta_T(t'_w) = \{\!\{x, x\}\!\}$ and $\beta_T(t''_w) = \{\!\{x, y\}\!\}$.
  c) If $w = \pi(y)$, then $\beta_T(t'_w) = \{\!\{x, y\}\!\}$ and $\beta_T(t''_w) = \{\!\{y, y\}\!\}$.

For each $w \notin N_G[\pi(x)] \cup N_G[\pi(y)]$, follow the the same procedure as Local 2-GNN (Definition C.11).

An illustration of the construction of unfolding tree is given in Figure 6(c).

We finally define the unfolding tree of 2-FGNN, which differs in the loop part such that all nodes $t_w$ have two children.

**Definition C.13** (Unfolding tree of 2-FGNN). Given a graph $G$, vertices $u, v \in V_G$, and a non-negative integer $D$, the depth-$2D$ 2-FGNN unfolding tree of graph $G$ at $(u, v)$, denoted as $\left( F_G^{\mathsf{Sub},(D)}(u, v), T_G^{\mathsf{Sub},(D)}(u, v) \right)$, is a tree-decomposed graph $(F, T^r) \in \mathcal{S}^\mathsf{F}$ constructed as follows:

  1. **Initialization**. The procedure is exactly the same as Local 2-FGNN (Definition C.12).

  2. **Loop for $D$ rounds**. The procedure is similar to Local 2-FGNN (Definition C.12) except that the condition $w \in N_G[\pi(x)] \cup N_G[\pi(y)]$ is relaxed to all vertices.

An illustration of the construction of unfolding tree is given in Figure 6(d).

We are now ready to present the first core result:

**Lemma C.14.** *Let $M \in \{\mathsf{Sub}, \mathsf{L}, \mathsf{LF}, \mathsf{F}\}$ be any model. For any two graphs $G, H$, any vertices $u, v \in V_G$, $x, y \in V_H$, and any non-negative integer $D$, $\tilde{\chi}_G^{M,(D)}(u, v) = \tilde{\chi}_H^{M,(D)}(x, y)$ iff there exists an isomorphism $(\rho, \tau)$ from $\left( F_G^{M,(D)}(u, v), T_G^{M,(D)}(u, v) \right)$ to $\left( F_H^{M,(D)}(x, y), T_H^{M,(D)}(x, y) \right)$ such that $\rho(u) = x, \rho(v) = y$.*

*Proof.* Here, we only give the proof for Local 2-GNN, and the proofs for Subgraph GNN, Local 2-FGNN and 2-FGNN are almost the same so we omit them for clarity.

*Proof for Local 2-GNN.* The proof is based on induction over $D$. When $D = 0$, the theorem obviously holds. Now assume that the theorem holds for $D \leq d$, and consider $D = d + 1$. Below, we omit $\mathsf{L}$ in the corner mark for clarity.

1. We first prove that $\tilde{\chi}_G^{(d+1)}(u, v) = \tilde{\chi}_H^{(d+1)}(x, y)$ implies that there exists an isomorphism $(\rho, \tau)$ from $\left( F_G^{(d+1)}(u, v), T_G^{(d+1)}(u, v) \right)$ to $\left( F_H^{(d+1)}(x, y), T_H^{(d+1)}(x, y) \right)$ such that $\rho(u) = x, \rho(v) = y$. If $\tilde{\chi}_G^{(d+1)}(u, v) = \tilde{\chi}_H^{(d+1)}(x, y)$, then

$$\{\!\{(\tilde{\chi}_G^{(d)}(u, w), \mathsf{atp}_G(u, v, w)) : w \in V_G\}\!\} = \{\!\{(\tilde{\chi}_H^{(d)}(x, z), \mathsf{atp}_H(x, y, z)) : z \in V_H\}\!\}, \quad (11)$$

$$\{\!\{(\tilde{\chi}_G^{(d)}(w, v), \mathsf{atp}_G(u, v, w)) : w \in V_G\}\!\} = \{\!\{(\tilde{\chi}_H^{(d)}(z, y), \mathsf{atp}_H(x, y, z)) : z \in V_H\}\!\}. \quad (12)$$

Let $n = |V_G| = |V_H|$. Thus, we can denote $V_G = \{w_1, \cdots, w_n\} = \{w'_1, \cdots, w'_n\}$ and $V_H = \{z_1, \cdots, z_n\} = \{z'_1, \cdots, z'_n\}$ such that

  - $(\tilde{\chi}_G^{(d)}(u, w_i), \mathsf{atp}_G(u, v, w_i)) = (\tilde{\chi}_H^{(d)}(x, z_i), \mathsf{atp}_H(x, y, z_i))$ for all $i \in [n]$;
  - $(\tilde{\chi}_G^{(d)}(w'_i, v), \mathsf{atp}_G(u, v, w'_i)) = (\tilde{\chi}_H^{(d)}(z'_i, y), \mathsf{atp}_H(x, y, z'_i))$ for all $i \in [n]$.

On the other hand, by definition of tree unfolding, we have

$$F_G^{(d+1)}(u, v) = \left( \bigcup_{w_i} F_G^{(d)}(u, w_i) \right) \cup \left( \bigcup_{w'_i} F_G^{(d)}(w'_i, v) \right) \cup F_G^{(1)}(u, v),$$

$$F_H^{(d+1)}(x, y) = \left( \bigcup_{z_i} F_H^{(d)}(x, z_i) \right) \cup \left( \bigcup_{z'_i} F_H^{(d)}(z'_i, y) \right) \cup F_H^{(1)}(x, y),$$

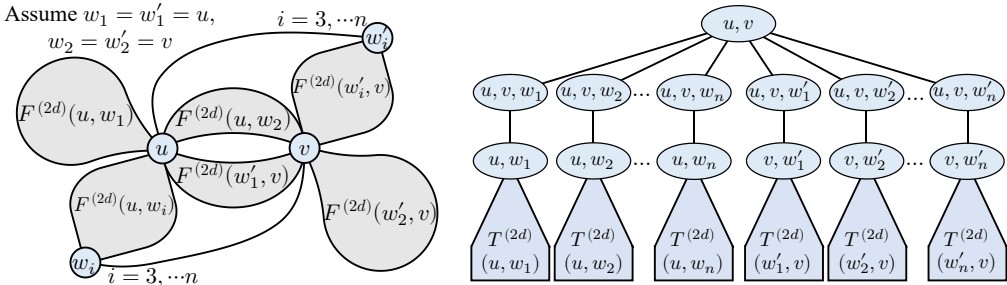

Figure 7: Illustration of the proof of Lemma C.14.

where $\cup$ represents the graph union. Here, all $w_i, w_j' \notin \{u, v\}$ in different graphs are treated as different vertices when taking the union, while $u, v$ in different graphs are shared. See Figure 7 for an illustration of the above equations.

By induction, there exists an isomorphism $(\rho_i, \tau_i)$ from $\left(F_G^{(d)}(u, w_i), T_G^{(d)}(u, w_i)\right)$ to $\left(F_H^{(d)}(x, z_i), T_H^{(d)}(x, z_i)\right)$ such that $\rho_i(u) = x$, $\rho_i(w_i) = z_i$ ($i \in [n]$), and there exist an isomorphism $\rho_i'$ from $\left(F_G^{(d)}(w_i', v), T_G^{(d)}(w_i', v)\right)$ to $\left(F_H^{(d)}(z_i', y), T_H^{(d)}(z_i', y)\right)$ such that $\rho_i'(v) = y$, $\rho_i'(w_i') = z_i'$ ($i \in [n]$). Moreover, we have $\mathsf{atp}_G(u, v, w_i) = \mathsf{atp}_H(x, y, z_i)$ and $\mathsf{atp}_G(u, v, w_i') = \mathsf{atp}_H(x, y, z_i')$ for $i \in [n]$, which implies that $F_G^{(1)}(u, v)$ is isomorphic to $F_H^{(1)}(x, y)$. Therefore, if we construct $\tilde{\rho}$ by merging all $\rho_i$ and $\rho_i'$ ($i \in [n]$), and construct $\tilde{\tau}$ by merging all $\tau_i$ and $\tau_i'$ and further specifying an appropriate mapping between tree nodes of depth no more than 1 in $T_G^{(d+1)}(u, v)$ and $T_H^{(d+1)}(x, y)$, then it is straightforward to see that $(\tilde{\rho}, \tilde{\tau})$ is well-defined and is an isomorphism from $\left(F_G^{(d+1)}(u, v), T_G^{(d+1)}(u, v)\right)$ to $\left(F_H^{(d+1)}(x, y), T_H^{(d+1)}(x, y)\right)$ such that $\tilde{\rho}(u) = x$, $\tilde{\rho}(v) = y$.

2. We next prove that if there exists an isomorphism $(\rho, \tau)$ from the tree-decomposed graph $\left(F_G^{(d+1)}(u, v), T_G^{(d+1)}(u, v)\right)$ to $\left(F_H^{(d+1)}(x, y), T_H^{(d+1)}(x, y)\right)$ such that $\rho(u) = x, \rho(v) = y$, then $\tilde{\chi}_G^{(d+1)}(u, v) = \tilde{\chi}_H^{(d+1)}(x, y)$.
Without loss of generality, assume $u \neq v$ and $x \neq y$. since $\tau$ is an isomorphism from $T_G^{(d+1)}(u, v)$ to $T_H^{(d+1)}(x, y)$, $\tau$ maps all tree nodes of depth 1 in $T_G^{(d+1)}(u, v)$ to all tree nodes of depth 1 in $T_H^{(d+1)}(x, y)$. Let $s_1, \cdots, s_n$ be all nodes of depth 2 in $T_G^{(d+1)}(u, v)$ such that $u \in \beta_{T_G^{(d+1)}(u, v)}(s_i)$ (it follows that $n = |V_G|$), and let $s_i'$ be the parent of $s_i$. Similarly, let $t_1, \cdots, t_n$ be all nodes of depth 2 in $T_H^{(d+1)}(x, y)$ such that $x \in \beta_{T_H^{(d+1)}(x, y)}(t_i)$, and let $t_i'$ be the parent of $t_i$. Moreover, we can arrange the order so that the following are satisfied (for each $i \in [n]$):

   a) $\tau$ is an isomorphism from the subtree $T_G^{(d+1)}(u, v)[s_i]$ to the subtree $T_H^{(d+1)}(x, y)[t_i]$.
   b) For all $s \in \mathsf{Desc}_{T_G^{(d+1)}(u, v)}(s_i)$, $\rho(\beta_{T_G^{(d+1)}(u, v)}(s)) = \beta_{T_H^{(d+1)}(x, y)}(\tau(s))$.
   c) By definition of the unfolding tree, $\rho$ is an isomorphism from the induced subgraph $F_G^{(d+1)}(u, v)[T_G^{(d+1)}(u, v)[s_i]]$ to the induced subgraph $F_H^{(d+1)}(x, y)[T_H^{(d+1)}(x, y)[t_i]]$ (see Definition C.7).
   d) Let $\beta_{T_G^{(d+1)}(u, v)}(s_i') = \{\!\{u, v, \tilde{w}_i\}\!\}$ and $\beta_{T_H^{(d+1)}(x, y)}(t_i') = \{\!\{x, y, \tilde{z}_i\}\!\}$. Then, $\rho(\tilde{w}_i) = \tilde{z}_i$, and thus $\{v, \tilde{w}_i\} \in E_{F_G^{(d+1)}(u, v)}$ iff $\{y, \tilde{z}_i\} \in E_{F_H^{(d+1)}(x, y)}$.

By items (a) to (c), $\left(F_G^{(d+1)}(u, v)\left[T_G^{(d+1)}(u, v)[s_i]\right], T_G^{(d+1)}(u, v)[s_i]\right)$ is isomorphic to $\left(F_H^{(d+1)}(x, y)\left[T_H^{(d+1)}(x, y)[t_i]\right], T_H^{(d+1)}(x, y)[t_i]\right)$. On the other hand, by definition of the unfolding tree, $\left(F_G^{(d+1)}(u, v)\left[T_G^{(d+1)}(u, v)[s_i]\right], T_G^{(d+1)}(u, v)[s_i]\right)$ is isomorphic to

the depth-$2d$ unfolding tree $\left(F_G^{(d)}(u, w_i), T_G^{(d)}(u, w_i)\right)$ for some $w_i \in V_G$ satisfying that $\{w_i, v\} \in E_G$ iff $\{\tilde{w}_i, v\} \in E_{F_G^{(d+1)}(u,v)}$.

Similarly, $\left(F_H^{(d+1)}(x, y)\left[T_H^{(d+1)}(x, y)[t_i]\right], T_H^{(d+1)}(x, y)[t_i]\right)$ is isomorphic to $\left(F_H^{(d)}(x, z_i), T_H^{(d)}(x, z_i)\right)$ for some $z_i \in V_H$ satisfying that $\{z_i, y\} \in E_H$ iff $\{\tilde{z}_i, y\} \in E_{F_H^{(d+1)}(x,y)}$. Combining all the above equivalence yields that $\left(F_G^{(d)}(u, w_i), T_G^{(d)}(u, w_i)\right)$ is isomorphic to $\left(F_H^{(d)}(x, z_i), T_H^{(d)}(x, z_i)\right)$, and $\{w_i, v\} \in E_G$ iff $\{z_i, y\} \in E_H$.

By induction, we have $\tilde{\chi}_G^{(d)}(u, w_i) = \tilde{\chi}_H^{(d)}(x, z_i)$. Moreover, we clearly have that $\{u, w_i\} \in E_G$ iff $\{x, z_i\} \in E_H$, and $\{u, v\} \in E_G$ iff $\{x, y\} \in E_H$. Therefore,

$$(\tilde{\chi}_G^{(d)}(u, w_i), \mathsf{atp}_G(u, v, w_i)) = (\tilde{\chi}_H^{(d)}(x, z_i), \mathsf{atp}_H(x, y, z_i)). \tag{13}$$

Next, note that $\tilde{w}_i$ are different from each other for $i \in [n]$ by definition of unfolding tree. Thus, $w_i$ are also different from each other. It follows that

$$\{\!\{(\tilde{\chi}_G^{(d)}(u, w), \mathsf{atp}_G(u, v, w)) : w \in V_G\}\!\} = \{\!\{(\tilde{\chi}_H^{(d)}(x, z), \mathsf{atp}_H(x, y, z)) : z \in V_H\}\!\}. \tag{14}$$

Again using the same analysis as before, we obtain

$$\{\!\{(\tilde{\chi}_G^{(d)}(w, v), \mathsf{atp}_G(u, v, w)) : w \in V_G\}\!\} = \{\!\{(\tilde{\chi}_H^{(d)}(z, y), \mathsf{atp}_H(x, y, z)) : z \in V_H\}\!\}. \tag{15}$$

It remains to prove that $\tilde{\chi}_G^{(d)}(u, v) = \tilde{\chi}_H^{(d)}(x, y)$. To prove this, note that Equation (14) implies that

$$\{\!\{(\tilde{\chi}_G^{(d')}(u, w), \mathsf{atp}_G(u, v, w)) : w \in V_G\}\!\} = \{\!\{(\tilde{\chi}_H^{(d')}(x, z), \mathsf{atp}_H(x, y, z)) : z \in V_H\}\!\} \tag{16}$$

holds for all $0 \le d' \le d$, and Equation (15) implies that

$$\{\!\{(\tilde{\chi}_G^{(d')}(w, v), \mathsf{atp}_G(u, v, w)) : w \in V_G\}\!\} = \{\!\{(\tilde{\chi}_H^{(d)}(z, y), \mathsf{atp}_H(x, y, z)) : z \in V_H\}\!\}. \tag{17}$$

holds for all $0 \le d' \le d$. Combined with Equations (16) and (17) and the fact that $\tilde{\chi}_G^{(0)}(u, v) = \tilde{\chi}_H^{(0)}(x, y)$, we can incrementally prove that $\tilde{\chi}_G^{(d')}(u, v) = \tilde{\chi}_H^{(d')}(x, y)$ for all $d' \le d + 1$.

We have thus concluded the proof. $\qquad\square$

**Definition C.15.** Let $M \in \{\mathsf{Sub}, \mathsf{L}, \mathsf{LF}, \mathsf{F}\}$ be any model. Given a graph $G$ and a tree-decomposed graph $(F, T^r)$, define

$$\mathsf{cnt}^M((F, T^r), G) := \left|\left\{(u, v) \in V_G^2 : \exists D \in \mathbb{N}_+ \text{ s.t. } \left(F_G^{M,(D)}(u, v), T_G^{M,(D)}(u, v)\right) \simeq (F, T^r)\right\}\right|,$$

where $\left(F_G^{M,(D)}(u, v), T_G^{M,(D)}(u, v)\right)$ is the depth-$2D$ unfolding tree of $G$ at $(u, v)$ for model $M$.

**Corollary C.16.** *Let $M \in \{\mathsf{Sub}, \mathsf{L}, \mathsf{LF}, \mathsf{F}\}$ be any model. For any graphs $G, H$, $\chi_G^M(G) = \chi_H^M(H)$ iff $\mathsf{cnt}^M((F, T^r), G) = \mathsf{cnt}^M((F, T^r), H)$ holds for all $(F, T^r) \in \mathcal{S}^M$.*

*Proof.* "$\Longrightarrow$". If $\chi_G^M(G) = \chi_H^M(H)$, then $\{\!\{\chi_G^M(u, v) : u, v \in V_G\}\!\} = \{\!\{\chi_H^M(x, y) : x, y \in V_H\}\!\}$. For each color $c$ in the above multiset, pick $u, v \in V_G$ with $\chi_G^M(u, v) = c$. It follows that if $(F, T^r) \simeq (F_G^{M,(D)}(u, v), T_G^{M,(D)}(u, v)) \in \mathcal{S}^M$ for some $D$, then $\mathsf{cnt}^M((F, T^r), G) = |\{(u, v) \in V_G^2 : \chi_G^M(u, v) = c\}| = |\{(x, y) \in V_H^2 : \chi_H^M(x, y) = c\}| = \mathsf{cnt}^M((F, T^r), H)$ by Lemma C.14. On the other hand, if $(F, T^r) \not\simeq (F_G^{M,(D)}(u, v), T_G^{M,(D)}(u, v))$ for all $u, v \in V_G$ and all $D$, then clearly $\mathsf{cnt}^M((F, T^r), G) = \mathsf{cnt}^M((F, T^r), H) = 0$.

"$\Longleftarrow$". If $\mathsf{cnt}^M((F, T^r), G) = \mathsf{cnt}^M((F, T^r), H)$ holds for all $(F, T^r) \in \mathcal{S}^M$, it clearly holds for all $(F_M^{(D)}(u, v), T_M^{(D)}(u, v))$ with $u, v \in V_G$ and a sufficiently large $D$. This guarantees that for all color $c$, $|\{(u, v) \in V_G^2 : \chi_G^M(u, v) = c\}| = |\{(x, y) \in V_H^2 : \chi_H^M(x, y) = c\}|$ by Lemma C.14. Therefore, $\{\!\{\chi_G^M(u, v) : u, v \in V_G\}\!\} = \{\!\{\chi_H^M(x, y) : x, y \in V_H\}\!\}$, concluding the proof. $\qquad\square$

We next define an important concept called *bag isomorphism* (Dell et al., 2018).

**Definition C.17.** Given a tree-decomposed graph $(F, T^r)$ and a graph $G$, a bag isomorphism from $(F, T^r)$ to $G$ (abbreviated as "bIso") is a homomorphism $f$ from $F$ to $G$ such that, for all $t \in V_T$, $f$ is an isomorphism from $F[\beta_T(t)]$ to $G[f(\beta_T(t))]$. Denote $\mathsf{BIso}((F, T^r), G)$ to be the set of all bag isomorphisms from $(F, T^r)$ to $G$, and denote $\mathsf{bIso}((F, T^r), G) = |\mathsf{BIso}((F, T^r), G)|$.

**Remark C.18.** To prove that a mapping $f$ from $(F, T^r)$ to $G$ is a bIso, it suffices to prove the following conditions:

1. For any two different vertices $u, v \in V_G$ in the same bag, $f(u) \neq f(v)$;

2. For any two vertices $u, v \in V_G$ in the same bag, $\{u, v\} \in E_F$ iff $\{f(u), f(v)\} \in E_G$;

3. For any $u \in V_G$, $\ell_F(u) = \ell_G(f(u))$.

The following fact is straightforward from the construction of the unfolding tree:

**Fact C.19.** Let $M \in \{\mathsf{Sub}, \mathsf{L}, \mathsf{LF}, \mathsf{F}\}$ be any model considered above. For any graph $G$, any vertex pair $(u, v) \in V_G^2$, and any non-negative integer $D$, there is a bIso $\pi$ from $\left( F_G^{M,(D)}(u, v), T_G^{M,(D)}(u, v) \right)$ to $G$.

Similarly, we need the following concept to describe the relation between two tree-decomposed graphs. We note that these technical concepts also appeared in Dell et al. (2018).

**Definition C.20.** Given two tree-decomposed graphs $(F, T^r)$ and $(\tilde{F}, \tilde{T}^s)$, a pair of mappings $(\rho, \tau)$ is called homomorphism from $(F, T^r)$ to $(\tilde{F}, \tilde{T}^s)$ if it satisfies the following conditions:

a) $\tau$ is a homomorphism from $T$ to $\tilde{T}$ (ignoring labels $\beta$) and is depth-preserving, i.e., $\mathsf{dep}_{T^r}(t) = \mathsf{dep}_{\tilde{T}^s}(\tau(t))$ for all $t \in V_T$;

b) For all $t \in V_T$, $\rho$ is a homomorphism from $F[\beta_T(t)]$ to $\tilde{F}[\beta_{\tilde{T}}(\tau(t))]$. Note that this implies that $\rho$ is a homomorphism from $F$ to $\tilde{F}$.

c) The depth of $T^r$ is equal to the depth of $\tilde{T}^s$.

**Definition C.21.** Under Definition C.20, $(\rho, \tau)$ is further called a *bag-isomorphism homomorphism* (abbreviated as "bIsoHom") from $(F, T^r)$ to $(\tilde{F}, \tilde{T}^s)$ if it a homomorphism satisfying that, for all $t \in V_T$, $\rho$ is an isomorphism from $F[\beta_T(t)]$ to $\tilde{F}[\beta_{\tilde{T}}(\tau(t))]$. Furthermore, $(\rho, \tau)$ is called a bIsoSurj if $\tau$ is surjective; and $(\rho, \tau)$ is called a bIsoInj if $\tau$ is injective. We use $\mathsf{BIsoHom}\left((F, T^r), (\tilde{F}, \tilde{T}^s)\right)$ to denote the set of bIsoHoms from $(F, T^r)$ to $(\tilde{F}, \tilde{T}^s)$, and let $\mathsf{bIsoHom}\left((F, T^r), (\tilde{F}, \tilde{T}^s)\right) = \left| \mathsf{BIsoHom}\left((F, T^r), (\tilde{F}, \tilde{T}^s)\right) \right|$. The notations $\mathsf{BIsoSurj}\left((F, T^r), (\tilde{F}, \tilde{T}^s)\right)$, $\mathsf{bIsoSurj}\left((F, T^r), (\tilde{F}, \tilde{T}^s)\right)$, $\mathsf{BIsoInj}\left((F, T^r), (\tilde{F}, \tilde{T}^s)\right)$, and $\mathsf{bIsoInj}\left((F, T^r), (\tilde{F}, \tilde{T}^s)\right)$ are defined accordingly.

**Remark C.22.** In the above definition, the depth of a tree $T^r$ is the maximal depth among all tree nodes in $T^r$. Note that we do not require that all leaf nodes have the same depth in $T^r$.

We are now ready to present the second core result:

**Lemma C.23.** Let $M \in \{\mathsf{Sub}, \mathsf{L}, \mathsf{LF}, \mathsf{F}\}$ be any model. For any graph $G$ and tree-decomposed graph $(F, T^r) \in \mathcal{S}^M$,

$$\mathsf{bIso}\left((F, T^r), G\right) = \sum_{(\tilde{F}, \tilde{T}^s) \in \mathcal{S}^M} \mathsf{bIsoHom}\left((F, T^r), \left(\tilde{F}, \tilde{T}^s\right)\right) \cdot \mathsf{cnt}^M \left(\left(\tilde{F}, \tilde{T}^s\right), G\right). \quad (18)$$

*Here, the summation ranges over all non-isomorphic (tree-decomposed) graphs in $\mathcal{S}^M$ and is well-defined as there are only a finite number of graphs making the value in the summation non-zero.*

*Proof.* Here, we only give the proof for Local 2-GNN, and the proofs for Subgraph GNN, Local 2-FGNN and 2-FGNN are almost the same so we omit them for clarity.

*Proof for Local 2-GNN.* We assume that the root bag of $(F, T^r)$ is $\{\!\{u, v\}\!\}$, and the depth of $(F, T^r)$ is $2d$. Let $y, z \in V_G$ be any vertices in $G$, and denote $(F_G^{(d)}(y, z), T_G^{(d)}(y, z))$ as the depth-$2d$ Local 2-GNN unfolding tree at $(y, z)$. Define the following two sets:

$$S_1(y, z) = \{g : g \in \mathsf{BIso}\left((F, T^r), G\right), g(u) = y, g(v) = z\},$$

$$S_2(y, z) = \left\{(\rho, \tau) : (\rho, \tau) \in \mathsf{BIsoHom}\left((F, T^r), (F_G^{(d)}(y, z), T_G^{(d)}(y, z))\right), \rho(u) = y, \rho(v) = z\right\}.$$

Then, Lemma C.23 is equivalent to the following equation:

$$\sum_{y, z \in V_G} |S_1(y, z)| = \sum_{y, z \in V_G} |S_2(y, z)|.$$

We will prove that $|S_1(y, z)| = |S_2(y, z)|$ for all $y, z \in V_G$.

Given $y, z \in V_G$, according to Fact C.19, there exists a bIso $\pi$ from $(F_G^{(d)}(y, z), T_G^{(d)}(y, z))$ to graph $G$. Define a mapping $\sigma$ such that $\sigma(\rho, \tau) = \pi \circ \rho$ for all $(\rho, \tau) \in S_2(y, z)$. It suffices to prove that $\sigma$ is a bijection from $S_2(y, z)$ to $S_1(y, z)$.

1. We first prove that $\sigma$ is a mapping from $S_2(y, z)$ to $S_1(y, z)$, i.e., $\pi \circ \rho \in S_1(y, z)$ for all $(\rho, \tau) \in S_2(y, z)$. First, we clearly have $(\pi \circ \rho)(u) = \pi(y) = y$, $(\pi \circ \rho)(v) = \pi(z) = z$. We next prove that $\pi \circ \rho \in \mathsf{BIso}\left((F, T^r), G\right)$. The proof is based on Remark C.18.

   a) Let $w, x \in V_F$, $w \neq x$ be any vertices in the same bag of $T^r$. Since $(\rho, \tau)$ is a bIsoHom, $\rho(w) \neq \rho(x)$ and $\rho(w)$ and $\rho(x)$ are in the same bag of $T_G^{(d)}(y, z)$. Again, since $\pi$ is a bIso, we have $\pi(\rho(w)) \neq \pi(\rho(x))$.

   b) Let $w, x \in V_F$ be any vertices in the same bag of $T^r$. Since $(\rho, \tau)$ is a bIso-Hom, $\rho(w)$ and $\rho(x)$ are in the same bag of $T_G^{(d)}(y, z)$, and $\{w, x\} \in E_F$ iff $\{\rho(w), \rho(x)\} \in E_{F_G^{(d)}(y, z)}$. Again, since $\pi$ is a bIso, $\{\rho(w), \rho(x)\} \in E_{F_G^{(d)}(y, z)}$ iff $\{\pi(\rho(w)), \pi(\rho(x))\} \in E_G$. Therefore, $\{w, x\} \in E_F$ iff $\{\pi(\rho(w)), \pi(\rho(x))\} \in E_G$.

   c) We clearly have $\ell_F(w) = \ell_{F_G^{(d)}(y, z)}(\rho(w)) = \ell_G(\pi(\rho(w)))$.

   We have proved that $\pi \circ \rho \in \mathsf{BIso}\left((F, T^r), G\right)$.

2. We then prove that $\sigma$ is a surjection. For all $g \in S_1(y, z)$, we define a mapping $(\rho, \tau)$ from $(F, T^r)$ to $(F_G^{(d)}(y, z), T_G^{(d)}(y, z))$ as follows. First define $\rho(u) = y$, $\rho(v) = z$, and set $\tau(r)$ to be the root of $(F_G^{(d)}(y, z), T_G^{(d)}(y, z))$. Let $w_1, \cdots, w_m \in V_F$ and $w'_1, \cdots, w'_{m'} \in V_F$ be vertices such that all $\{u, w_i\}$ and $\{w'_i, v\}$ correspond to bags of $T^r$ associated to all tree nodes of depth 2. Similarly, by definition of the Local 2-GNN unfolding tree, let $x_1, \cdots, x_n \in V_{F_G^{(d)}(y, z)}$ be different vertices and $x'_1, \cdots, x'_n \in V_{F_G^{(d)}(y, z)}$ be different vertices such that all $\{y, x_i\}$ and $\{x'_i, z\}$ correspond to bags of $T_G^{(d)}(y, z)$ associated to all tree nodes of depth 2. Since $g$ and $\pi$ are bIsos, we have:

   - For every $w_i$ ($i \in [m]$), there exists $x_j$ ($j \in [n]$), such that $g(w_i) = \pi(x_j) = \tilde{x}_j$ for some $\tilde{x}_j \in V_G$ and $F[\{\!\{u, v, w_i\}\!\}] \simeq F_G^{(d)}(y, z)[\{\!\{y, z, x_j\}\!\}] \simeq G[\{\!\{y, z, \tilde{x}_j\}\!\}]$;
   - For every $w'_i$ ($i \in [m']$), there exists $x'_j$ ($j \in [n]$), such that $g(w'_i) = \pi(x'_j) = \tilde{x}'_j$ for some $\tilde{x}'_j \in V_G$ and $F[\{\!\{u, v, w'_i\}\!\}] \simeq F_G^{(d)}(y, z)[\{\!\{y, z, x'_j\}\!\}] \simeq G[\{\!\{y, z, \tilde{x}'_j\}\!\}]$.

   We then define $\rho(w_i) = x_j$ for each $i \in [m]$ and $\rho(w'_i) = x'_j$ for each $i \in [m']$. Based on the above two items, one can easily define $\tau$ such that each node $s$ in $T^r$ of depth 1 or 2 is mapped by $\tau$ to a node $t$ in $T_G^{(d)}(y, z)$ of the same depth, $\rho(\beta_T(s)) = \beta_{T_G^{(d)}}(t)$, and $\rho$ is an isomorphism from $F[\beta_T(s)]$ to $F_G^{(d)}(y, z)[\beta_{T_G^{(d)}(y, z)}(t)]$.

   Next, we can recursively define $\rho$'s image on $F[T^r[s]]$ for each tree node $s$ of depth 2 following the same construction above. This is because $g$ is still a bIso from $(F[T^r[s]], T^r[s])$ to $G$, $\pi$ is still a bIso from $\left(F_G^{(d)}(y, z)[T_G^{(d)}(y, z)[\tau(s)]], T_G^{(d)}(y, z)[\tau(s)]\right)$ to $G$, and $g(\beta_T(s)) = \pi(\beta_{T_G^{(d)}}(\tau(s)))$. Recursively applying this procedure, we can construct $(\rho, \tau)$ such that it is a

bIsoHom from $(F, T^r)$ to $\left(F_G^{(d)}(y,z), T_G^{(d)}(y,z))\right)$. That is to say, we have proved that for all $g \in S_1(u,v)$, there is a preimage $(\rho, \tau) \in S_2(y,z)$ such that $\sigma(\rho, \tau) = g$.

3. We finally prove that $\sigma$ is an injection. Let $(\rho_1, \tau_1), (\rho_2, \tau_2) \in S_2(y,z)$ such that $\pi \circ \rho_1 = \pi \circ \rho_2$. Let $w_1, \cdots, w_m \in V_F$, $w_1', \cdots, w_{m'}' \in V_F$, $x_1, \cdots, x_n \in V_{F_G^{(d)}(y,z)}$, and $x_1', \cdots, x_n' \in V_{F_G^{(d)}(y,z)}$ be defined as in the previous item. For each $i \in [m]$, let $j_1(i)$ and $j_2(i)$ be indices satisfying $\rho_1(w_i) = x_{j_1(i)}$ and $\rho_2(w_i) = x_{j_2(i)}$. It follows that $\pi(x_{j_1(i)}) = \pi(x_{j_2(i)})$. By definition of the Local 2-GNN unfolding tree, we must have $x_{j_1(i)} = x_{j_2(i)}$, and thus $\rho_1(w_i) = \rho_2(w_i)$. Using a similar approach, we can prove that $\rho_1(w_i') = \rho_2(w_i')$ for each $i \in [m']$. Next, we can recursively apply the above procedure to the subtree $T^r[s]$ for each tree node of depth 2 following the previous item, and finally prove that $\rho_1 = \rho_2$. Therefore, $\sigma$ is an injection.

Combining the above three items completes the proof. □

**Proposition C.24.** *Under Definition C.21,* (i) *if $(\rho, \tau)$ is a bIsoSurj, then $\rho$ is a surjection from $F$ to $\tilde{F}$ on both vertices and edges;* (ii) *if $(\rho, \tau)$ is a bIsoInj, then $\rho$ is an injection from $F$ to $\tilde{F}$ on both vertices and edges.*

*Proof.* We first prove that $\rho$ is surjective if $(\rho, \tau)$ is a bIsoSurj. We will only prove that $\rho$ is surjective on edges, as proving that $\rho$ is surjective on vertices is almost the same. For any $\{x,y\} \in E_{\tilde{F}}$, by Definition C.1(b) we can pick $\tilde{t} \in V_{\tilde{T}}$ such that $\{x,y\} \in \beta_{\tilde{T}}(\tilde{t})$. Since $\tau$ is surjective, there exists $t \in V_T$ such that $\tau(t) = \tilde{t}$. By definition of bag isomorphism, there exists $u, v \in \beta_T(t)$ such that $\rho(u) = x, \rho(v) = y$, and $F[\{u,v\}] \simeq \tilde{F}[\{x,y\}]$. Therefore, $\{u,v\} \in E_F$.

We next prove that $\rho$ is injective if $(\rho, \tau)$ is a bIsoInj. Pick any $u \in V_F$. It suffices to prove that $\rho(u) = \rho(v)$ iff $u = v$ for any $v \in V_F$. If the result does not hold, consider two cases:

- There exists $v \in V_F$, $v \neq u$ such that $\rho(u) = \rho(v)$ and $\{u, v\}$ are in the same bag of $F$. This contradicts the definition of bag isomorphism.

- For all $v \in V_F$ such that $v \neq u$ and $\rho(u) = \rho(v)$, $\{u, v\}$ are not in the same bag of $F$. By Proposition C.5, $T\left[\bigcup_{v \in V_F : \rho(u) = \rho(v)} B_T(v)\right]$ is disconnected. We can thus pick a path $P$ in $T$ such that the endpoints $t_1$ and $t_2$ are in different connected components of $T\left[\bigcup_{v \in V_F : \rho(u) = \rho(v)} B_T(v)\right]$. This implies that there is tree node $t_3$ in $P$ such that $\beta_T(t_3) \cap \{v \in V_F : \rho(u) = \rho(v)\} = \emptyset$. Consequently, $\rho(u) \in \rho(\beta_T(t_1)) = \beta_{\tilde{T}}(\tau(t_1))$, $\rho(u) \in \rho(\beta_T(t_2)) = \beta_{\tilde{T}}(\tau(t_2))$, but $\rho(u) \notin \beta_{\tilde{T}}(\tau(t_3))$. On the other hand, since $\tau$ is injective, $\tilde{T}[\tau(V_P)]$ is also a path and $\tau(t_3)$ is on the path between $\tau(t_1)$ and $\tau(t_2)$ in $\tilde{T}$. This contradicts the definition of tree decomposition (Definition C.1(c)).

Combining the two cases concludes the proof. □

**Lemma C.25.** *Let $M \in \{\mathsf{Sub}, \mathsf{L}, \mathsf{LF}, \mathsf{F}\}$ be any model. For any tree-decomposed graphs $(F, T^r), (\tilde{F}, \tilde{T}^s) \in \mathcal{S}^M$,*

$$\mathsf{bIsoHom}((F,T^r), (\tilde{F}, \tilde{T}^s)) = \sum_{(\widehat{F}, \widehat{T}^t) \in \mathcal{S}^M} \frac{\mathsf{bIsoSurj}\left((F,T^r), (\widehat{F}, \widehat{T}^t)\right) \cdot \mathsf{bIsoInj}\left((\widehat{F}, \widehat{T}^t), (\tilde{F}, \tilde{T}^s)\right)}{\mathsf{aut}(\widehat{F}, \widehat{T}^t)},$$

*where $\mathsf{aut}(\widehat{F}, \widehat{T}^t)$ denotes the number of automorphisms of $(\widehat{F}, \widehat{T}^t)$. Here, the summation ranges over all non-isomorphic (tree-decomposed) graphs in $\mathcal{S}^M$ and is well-defined as there are only a finite number of graphs making the value in the summation non-zero.*

*Proof.* We define the following set of three-tuples:

$$S = \left\{ \left((\widehat{F}, \widehat{T}^t), (\rho^{\mathsf{S}}, \tau^{\mathsf{S}}), (\rho^{\mathsf{I}}, \tau^{\mathsf{I}})\right) : (\widehat{F}, \widehat{T}^t) \in \mathcal{S}^M, \right.$$
$$\left. (\rho^{\mathsf{S}}, \tau^{\mathsf{S}}) \in \mathsf{BIsoSurj}((F,T^r), (\widehat{F}, \widehat{T}^t)), (\rho^{\mathsf{I}}, \tau^{\mathsf{I}}) \in \mathsf{BIsoInj}((\widehat{F}, \widehat{T}^t), (\tilde{F}, \tilde{T}^s)) \right\}.$$

Define a mappings $\sigma$ such that

$$\sigma\left((\widehat{F}, \widehat{T}^t), (\rho^{\mathsf{S}}, \tau^{\mathsf{S}}), (\rho^{\mathsf{I}}, \tau^{\mathsf{I}})\right) = (\rho^{\mathsf{I}} \circ \rho^{\mathsf{S}}, \tau^{\mathsf{I}} \circ \tau^{\mathsf{S}})$$

for all $\left((\widehat{F}, \widehat{T}^t), (\rho^{\mathsf{S}}, \tau^{\mathsf{S}}), (\rho^{\mathsf{I}}, \tau^{\mathsf{I}})\right) \in S$. It suffices to prove the following three statements:

1. $\sigma$ is a mapping from $S$ to $\mathsf{BIsoHom}((F, T^r), (\tilde{F}, \tilde{T}^s))$;

2. $\sigma$ is surjective;

3. $\sigma\left((\widehat{F}_1, \widehat{T}_1^{t_1}), (\rho_1^{\mathsf{S}}, \tau_1^{\mathsf{S}}), (\rho_1^{\mathsf{I}}, \tau_1^{\mathsf{I}})\right) = \sigma\left((\widehat{F}_2, \widehat{T}_2^{t_2}), (\rho_2^{\mathsf{S}}, \tau_2^{\mathsf{S}}), (\rho_2^{\mathsf{I}}, \tau_2^{\mathsf{I}})\right)$ iff there exists an isomorphism $(\widehat{\rho}, \widehat{\tau})$ from $(\widehat{F}_1, \widehat{T}_1^{t_1})$ to $(\widehat{F}_2, \widehat{T}_2^{t_2})$ such that $\widehat{\rho} \circ \rho_1^{\mathsf{S}} = \rho_2^{\mathsf{S}}, \widehat{\tau} \circ \tau_1^{\mathsf{S}} = \tau_2^{\mathsf{S}}, \rho_1^{\mathsf{I}} = \rho_2^{\mathsf{I}} \circ \widehat{\rho}, \tau_1^{\mathsf{I}} = \tau_2^{\mathsf{I}} \circ \widehat{\tau}$.

We will prove these statements one by one.

1. We first prove that $\sigma$ is a mapping from $S$ to $\mathsf{BIsoHom}((F, T^r), (\tilde{F}, \tilde{T}^s))$. This simply follows from the fact that both bIsoSurj and bIsoInj are bIsoHom, and the composition of two bIsoHoms are still a bIsoHom.

2. We next prove that $\sigma$ is surjective. Given $(\rho^{\mathsf{H}}, \tau^{\mathsf{H}}) \in \mathsf{BIsoHom}((F, T^r), (\tilde{F}, \tilde{T}^s))$, we define $(\widehat{F}, \widehat{T}^t), (\rho^{\mathsf{S}}, \tau^{\mathsf{S}})$, and $(\rho^{\mathsf{I}}, \tau^{\mathsf{I}})$ as follows:

    a) Let $\widehat{F} = \tilde{F}[\rho^{\mathsf{H}}(V_F)]$ and $\widehat{T}^t = \tilde{T}^s[\tau^{\mathsf{H}}(V_T)]$. We clearly have $(\widehat{F}, \widehat{T}^t) \in \mathcal{S}^M$.
    b) Let $\rho^{\mathsf{S}} = \rho^{\mathsf{H}}$ and $\tau^{\mathsf{S}} = \tau^{\mathsf{H}}$. Obviously, $(\rho^{\mathsf{S}}, \tau^{\mathsf{S}})$ is a bIsoSurj from $(F, T^r)$ to $(\widehat{F}, \widehat{T}^t)$.
    c) Define identity mappings $\rho^{\mathsf{I}}(u) = u$ for all $u \in V_{\widehat{F}}$ and $\tau^{\mathsf{I}}(t) = t$ for all $t \in V_{\widehat{T}}$. Obviously, $(\rho^{\mathsf{I}}, \tau^{\mathsf{I}})$ is a bIsoInj from $(\widehat{F}, \widehat{T}^t)$ to $(\tilde{F}, \tilde{T}^s)$.

    We clearly have $\rho^{\mathsf{H}} = \rho^{\mathsf{I}} \circ \rho^{\mathsf{S}}$ and $\tau^{\mathsf{H}} = \tau^{\mathsf{I}} \circ \tau^{\mathsf{S}}$. Thus, $\sigma$ is a surjection.

3. We finally prove the aforementioned item 3. It suffices to prove only one direction, namely, $\sigma\left((\widehat{F}_1, \widehat{T}_1^{t_1}), (\rho_1^{\mathsf{S}}, \tau_1^{\mathsf{S}}), (\rho_1^{\mathsf{I}}, \tau_1^{\mathsf{I}})\right) = \sigma\left((\widehat{F}_2, \widehat{T}_2^{t_2}), (\rho_2^{\mathsf{S}}, \tau_2^{\mathsf{S}}), (\rho_2^{\mathsf{I}}, \tau_2^{\mathsf{I}})\right)$ implies that there exists an isomorphism $(\widehat{\rho}, \widehat{\tau})$ from $(\widehat{F}_1, \widehat{T}_1^{t_1})$ to $(\widehat{F}_2, \widehat{T}_2^{t_2})$ such that $\widehat{\rho} \circ \rho_1^{\mathsf{S}} = \rho_2^{\mathsf{S}}, \widehat{\tau} \circ \tau_1^{\mathsf{S}} = \tau_2^{\mathsf{S}}, \rho_1^{\mathsf{I}} = \rho_2^{\mathsf{I}} \circ \widehat{\rho}, \tau_1^{\mathsf{I}} = \tau_2^{\mathsf{I}} \circ \widehat{\tau}$.

    a) We first prove that $\widehat{F}_1 \simeq \widehat{F}_2$ and $\widehat{T}_1^{t_1} \simeq \widehat{T}_2^{t_2}$.
    For any $u, v \in V_F$, if $\rho_1^{\mathsf{S}}(u) \neq \rho_1^{\mathsf{S}}(v)$, then $\rho_1^{\mathsf{I}}(\rho_1^{\mathsf{S}}(u)) \neq \rho_1^{\mathsf{I}}(\rho_1^{\mathsf{S}}(v))$ since $\rho_1^{\mathsf{I}}$ is an injection (by Proposition C.24). Therefore, $\rho_2^{\mathsf{I}}(\rho_2^{\mathsf{S}}(u)) \neq \rho_2^{\mathsf{I}}(\rho_2^{\mathsf{S}}(v))$, and thus $\rho_2^{\mathsf{S}}(u) \neq \rho_2^{\mathsf{S}}(v)$. By symmetry, we also have that $\rho_2^{\mathsf{S}}(u) \neq \rho_2^{\mathsf{S}}(v)$ implies $\rho_1^{\mathsf{S}}(u) \neq \rho_1^{\mathsf{S}}(v)$. This proves that $\rho_1^{\mathsf{S}}(u) = \rho_1^{\mathsf{S}}(v)$ iff $\rho_2^{\mathsf{S}}(u) = \rho_2^{\mathsf{S}}(v)$.
    For any $u, v \in V_F$, if $\{\rho_1^{\mathsf{S}}(u), \rho_1^{\mathsf{S}}(v)\} \in E_{\widehat{F}_1}$, then $\{\rho_1^{\mathsf{I}}(\rho_1^{\mathsf{S}}(u)), \rho_1^{\mathsf{I}}(\rho_1^{\mathsf{S}}(v))\} \in E_{\tilde{F}}$ since $\rho_1^{\mathsf{I}}$ is a homomorphism. Therefore, $\{\rho_2^{\mathsf{I}}(\rho_2^{\mathsf{S}}(u)), \rho_2^{\mathsf{I}}(\rho_2^{\mathsf{S}}(v))\} \in E_{\tilde{F}}$. This implies that $\rho_2^{\mathsf{I}}(\rho_2^{\mathsf{S}}(u))$ and $\rho_2^{\mathsf{I}}(\rho_2^{\mathsf{S}}(v))$ are in the same bag of $\tilde{T}^s$. Therefore, there are vertices $x, y \in V_{\widehat{F}_2}$ in the same bag of $\widehat{T}_2^{s_2}$ such that $\rho_2^{\mathsf{I}}(x) = \rho_2^{\mathsf{I}}(\rho_2^{\mathsf{S}}(u))$ and $\rho_2^{\mathsf{I}}(y) = \rho_2^{\mathsf{I}}(\rho_2^{\mathsf{S}}(v))$, and by definition of bag isomorphism we have $\{x, y\} \in E_{\widehat{F}_2}$. Since $\rho_2$ is injective, $x = \rho_2^{\mathsf{S}}(u)$ and $y = \rho_2^{\mathsf{S}}(v)$, namely, $\{\rho_2^{\mathsf{S}}(u), \rho_2^{\mathsf{S}}(v)\} \in E_{\widehat{F}_2}$. By symmetry, we can prove that $\{\rho_1^{\mathsf{S}}(u), \rho_1^{\mathsf{S}}(v)\} \in E_{\widehat{F}_1}$ iff $\{\rho_2^{\mathsf{S}}(u), \rho_2^{\mathsf{S}}(v)\} \in E_{\widehat{F}_2}$.
    Finally, noting that $\rho_1^{\mathsf{S}}$ and $\rho_2^{\mathsf{S}}$ are surjective (by Proposition C.24) and $\ell_{\widehat{F}_1}(\rho_1^{\mathsf{S}}(u)) = \ell_F(u) = \ell_{\widehat{F}_2}(\rho_2^{\mathsf{S}}(u))$ for all $u \in V_F$, we obtain that $\widehat{F}_1 \simeq \widehat{F}_2$. Following the same procedure, we can prove that $\widehat{T}_1^{t_1} \simeq \widehat{T}_2^{t_2}$.
    b) Consequently, there exist isomorphisms $\widehat{\rho}$ and $\widehat{\tau}$ such that $\widehat{\rho} \circ \rho_1^{\mathsf{S}} = \rho_2^{\mathsf{S}}, \widehat{\tau} \circ \tau_1^{\mathsf{S}} = \tau_2^{\mathsf{S}}$. For any node $q \in V_T$,

    $$\widehat{\rho}(\beta_{\widehat{T}_1}(\tau_1^{\mathsf{S}}(q))) = \widehat{\rho}(\rho_1^{\mathsf{S}}(\beta_T(q))) = \rho_2^{\mathsf{S}}(\beta_T(q)) = \beta_{\widehat{T}_2}(\tau_2^{\mathsf{S}}(q)) = \beta_{\widehat{T}_2}(\widehat{\tau}(\tau_1^{\mathsf{S}}(q))).$$

    Since $\tau_1^{\mathsf{S}}$ is surjective, $\tau_1^{\mathsf{S}}(q)$ ranges over all nodes in $\widehat{T}_1^{s_1}$ when $q$ ranges over $V_T$. We thus conclude that $(\rho, \tau)$ is an isomorphism from $(\widehat{F}_1, \widehat{T}_1^{t_1})$ to $(\widehat{F}_2, \widehat{T}_2^{t_2})$ (see Definition C.8).

c) We finally prove that $\rho_1^l = \rho_2^l \circ \hat{\rho}$ and $\tau_1^l = \tau_2^l \circ \hat{\tau}$. Pick any $u \in V_F$, we have $\rho_2^l(\hat{\rho}(\rho_1^S(u))) = \rho_2^l(\rho_2^S(u)) = \rho_1^l(\rho_1^S(u))$. Since $\rho_1^S$ is surjective, $\rho_1^S(u)$ ranges over all nodes in $\hat{F}_1$ when $u$ ranges over $V_F$. This proves that $\rho_1^l = \rho_2^l \circ \hat{\rho}$. Following the same procedure, we can prove that $\tau_1^l = \tau_2^l \circ \hat{\tau}$.

Combining the above three items concludes the proof. $\qquad \square$

In the following, we further define two technical concepts that will be used to describe the next result. We note that these technical concepts also appeared in Dell et al. (2018).

**Definition C.26.** A tree-decomposed graph $(\tilde{F}, \tilde{T}^s)$ is called a *bag extension* of another tree-decomposed graph $(F, T^r)$ if there is a graph $H$ and a mapping $(\rho, \tau)$ such that $F$ is a subgraph of $H$ and $(\rho, \tau)$ is an isomorphism from the tree-decomposed graph $(H, T^r)$ to $(\tilde{F}, \tilde{T}^s)$. Define $\mathsf{BExt}\left((F, T^r), (\tilde{F}, \tilde{T}^s)\right)$ to be the set of all mappings $(\rho, \tau)$ that satisfies the above conditions, and define $\mathsf{bExt}\left((F, T^r), (\tilde{F}, \tilde{T}^s)\right) = \left|\mathsf{BExt}\left((F, T^r), (\tilde{F}, \tilde{T}^s)\right)\right|$.

**Remark C.27.** In other words, a bag extension of a tree-decomposed graph $(F, T^r)$ can be obtained by adding an arbitrary number of edges to $F$ while ensuring that each added edge is contained in a tree node in $T^r$. A immediate fact is that $(\rho, \tau)$ is a homomorphism from $(F, T^r)$ to $(\tilde{F}, \tilde{T}^s)$.

**Definition C.28.** Given a tree-decomposed graph $(F, T^r)$ and a graph $G$, a bag-strong homomorphism from $(F, T^r)$ to $G$ (abbreviated as "bStrHom") is a homomorphism $f$ from $F$ to $G$ such that, for all $t \in V_T$, $f$ is a strong homomorphism from $F[\beta_T(t)]$ to $G[f(\beta_T(t))]$, i.e., $\{u, v\} \in E_{F[\beta_T(t)]}$ iff $\{f(u), f(v)\} \in E_{G[f(\beta_T(t))]}$. Denote $\mathsf{BStrHom}((F, T^r), G)$ to be the set of all bStrHom from $(F, T^r)$ to $G$, and denote $\mathsf{bStrHom}((F, T^r), G) = |\mathsf{BStrHom}((F, T^r), G)|$.

The following equation is straightforward:

**Lemma C.29.** *Let $M \in \{\mathsf{Sub}, \mathsf{L}, \mathsf{LF}, \mathsf{F}\}$ be any model. For any graph $G$ and tree-decomposed graph $(F, T^r) \in \mathcal{S}^M$,*

$$\mathsf{hom}(F, G) = \sum_{(\tilde{F}, \tilde{T}^s) \in \mathcal{S}^M} \frac{\mathsf{bExt}\left((F, T^r), (\tilde{F}, \tilde{T}^s)\right) \cdot \mathsf{bStrHom}\left((\tilde{F}, \tilde{T}^s), G\right)}{\mathsf{aut}(\tilde{F}, \tilde{T}^s)},$$

*where $\mathsf{aut}(\tilde{F}, \tilde{T}^s)$ denotes the number of automorphisms of $(\tilde{F}, \tilde{T}^s)$. Here, the summation ranges over all non-isomorphic (tree-decomposed) graphs in $\mathcal{S}^M$ and is well-defined as there are only a finite number of graphs making the value in the summation non-zero.*

*Proof.* The proof has a similar structure to the previous lemma. We define the following set of three-tuples:

$$S = \left\{\left((\tilde{F}, \tilde{T}^s), (\rho, \tau), g\right) : (\tilde{F}, \tilde{T}^s) \in \mathcal{S}^M, (\rho, \tau) \in \mathsf{BExt}\left((F, T^r), (\tilde{F}, \tilde{T}^s)\right), g \in \mathsf{BStrHom}((\tilde{F}, \tilde{T}^s), G)\right\}.$$

Define a mappings $\sigma$ such that $\sigma\left((\tilde{F}, \tilde{T}^s), (\rho, \tau), g\right) = g \circ \rho$ for all $\left((\tilde{F}, \tilde{T}^s), (\rho, \tau), g\right) \in S$. It is straightforward to see that $\sigma$ is a mapping from $S$ to $\mathsf{BIsoHom}((F, T^r), (\tilde{F}, \tilde{T}^s))$, namely, $g \circ \rho$ is a homomorphism from $F$ to $G$ for all $\left((\tilde{F}, \tilde{T}^s), (\rho, \tau), g\right) \in S$.

We then prove that $\sigma$ is surjective. Given $h \in \mathsf{Hom}(F, G)$, define $(\tilde{F}, \tilde{T}^s)$, $(\rho, \tau)$, and $g$ as follows:

a) Define $\tilde{F}$ be the graph obtained from $F$ by adding edges $\{\{u, v\} : \exists t \in V_T$ s.t. $u, v \in \beta_T(t), \{h(u), h(v)\} \in E_G\}$, and let $\tilde{T}^s = T^r$. Clearly, $(\tilde{F}, \tilde{T}^s)$ is a bag extension of $(F, T^r)$.

b) Define identity mappings $\rho(u) = u$ for all $u \in V_F$ and $\tau(t) = t$ for all $t \in V_T$. Clearly, $(\rho, \tau) \in \mathsf{BExt}((F, T^r), (\tilde{F}, \tilde{T}^s))$.

c) Let $g = h$. It is easy to see that $g$ is a strong homomorphism from $\tilde{F}[\beta_{\tilde{T}}(t)]$ to $G[g(\beta_{\tilde{T}}(t))]$ for each $t \in V_{\tilde{T}}$. Thus, $g \in \mathsf{BStrHom}((\tilde{F}, \tilde{T}^s), G)$.

Noting that $h = g = g \circ \rho$, we have proved that $\sigma$ is a surjection.

We finally prove that $\sigma\left((\tilde{F}_1, \tilde{T}_1^{s_1}), (\rho_1, \tau_1), g_1\right) = \sigma\left((\tilde{F}_2, \tilde{T}_2^{s_2}), (\rho_2, \tau_2), g_2\right)$ implies that there exists an isomorphism $(\tilde{\rho}, \tilde{\tau})$ from $(\tilde{F}_1, \tilde{T}_1^{s_1})$ to $(\tilde{F}_2, \tilde{T}_2^{s_2})$ such that $\tilde{\rho} \circ \rho_1 = \rho_2$, $\tilde{\tau} \circ \tau_1 = \tau_2$, $g_1 = g_2 \circ \tilde{\rho}$. We first prove that $\tilde{F}_1 \simeq \tilde{F}_2$ and $\tilde{T}_1^{s_1} \simeq \tilde{T}_2^{s_2}$. (i) For any $u, v \in V_F$, we obviously have $\rho_1(u) = \rho_1(v)$ iff $u = v$ iff $\rho_2(u) = \rho_2(v)$. (ii) Let $i \in \{1, 2\}$. For any $u, v \in V_F$, $\{\rho_i(u), \rho_i(v)\} \in E_{\tilde{F}_i}$ iff $\{u, v\} \in E_F$ or $u, v$ are in the same bag of $T^r$ and $\{g_i(\rho_i(u)), g(\rho_i(v))\} \in E_G$. Since $g_1 \circ \rho_1 = g_2 \circ \rho_2$, we have $\{\rho_1(u), \rho_1(v)\} \in E_{\tilde{F}_1}$ iff $\{\rho_2(u), \rho_2(v)\} \in E_{\tilde{F}_2}$. (iii) Finally, noting that $\rho_1$ and $\rho_2$ are bijective and $\ell_{\tilde{F}_1}(\rho_1(u)) = \ell_F(u) = \ell_{\tilde{F}_2}(\rho_2(u))$ for all $u \in V_F$, we obtain that $\tilde{F}_1 \simeq \tilde{F}_2$. On the other hand, $\tilde{T}_1^{s_1} \simeq \tilde{T}_2^{s_2}$ trivially holds. The remaining procedure is almost the same as in the previous lemma. $\square$

We next show that the mapping bStrHom can be further decomposed as shown in Lemma C.31. We need an auxiliary concept:

**Definition C.30.** Given two tree-decomposed graphs $(F, T^r)$ and $(\tilde{F}, \tilde{T}^s)$, a homomorphism $(\rho, \tau)$ from $(F, T^r)$ to $(\tilde{F}, \tilde{T}^s)$ is called bag-strong surjective (abbreviated as "bStrSurj") if $\rho$ is a bag-strong homomorphism from $(F, T^r)$ to $\tilde{F}$ and is surjective on both vertices and edges, and $\tau$ is an isomorphism from $T^r$ to $\tilde{T}^s$. Denote $\mathsf{BStrSurj}((F, T^r), (\tilde{F}, \tilde{T}^s))$ to be the set of all bStrSurj from $(F, T^r)$ to $(\tilde{F}, \tilde{T}^s)$, and denote $\mathsf{bStrSurj}((F, T^r), (\tilde{F}, \tilde{T}^s)) = |\mathsf{BStrSurj}((F, T^r), (\tilde{F}, \tilde{T}^s))|$.

**Lemma C.31.** *Let $M \in \{\mathsf{Sub}, \mathsf{L}, \mathsf{LF}, \mathsf{F}\}$ be any model. For any graph $G$ and tree-decomposed graph $(F, T^r) \in \mathcal{S}^M$,*

$$\mathsf{bStrHom}((F, T^r), G) = \sum_{(\tilde{F}, \tilde{T}^s) \in \mathcal{S}^M} \frac{\mathsf{bStrSurj}\left((F, T^r), (\tilde{F}, \tilde{T}^s)\right) \cdot \mathsf{bIso}\left((\tilde{F}, \tilde{T}^s), G\right)}{\mathsf{aut}(\tilde{F}, \tilde{T}^s)},$$

*where $\mathsf{aut}(\tilde{F}, \tilde{T}^s)$ denotes the number of automorphisms of $(\tilde{F}, \tilde{T}^s)$. Here, the summation ranges over all non-isomorphic (tree-decomposed) graphs in $\mathcal{S}^M$ and is well-defined as there are only a finite number of graphs making the value in the summation non-zero.*

*Proof.* The proof has a similar structure to the previous lemma. We define the following set of three-tuples:

$$S = \left\{\left((\tilde{F}, \tilde{T}^s), (\rho, \tau), g\right) : (\tilde{F}, \tilde{T}^s) \in \mathcal{S}^M, (\rho, \tau) \in \mathsf{BStrSurj}\left((F, T^r), (\tilde{F}, \tilde{T}^s)\right), g \in \mathsf{BIso}((\tilde{F}, \tilde{T}^s), G)\right\}.$$

Define a mappings $\sigma$ such that $\sigma\left((\tilde{F}, \tilde{T}^s), (\rho, \tau), g\right) = g \circ \rho$ for all $\left((\tilde{F}, \tilde{T}^s), (\rho, \tau), g\right) \in S$. It suffices to prove the following three statements:

1. $\sigma$ is a mapping from $S$ to $\mathsf{BStrHom}((F, T^r), G)$;

2. $\sigma$ is surjective;

3. $\sigma\left((\tilde{F}_1, \tilde{T}_1^{s_1}), (\rho_1, \tau_1), g_1\right) = \sigma\left((\tilde{F}_2, \tilde{T}_2^{s_2}), (\rho_2, \tau_2), g_2\right)$ iff there exists an isomorphism $(\tilde{\rho}, \tilde{\tau})$ from $(\tilde{F}_1, \tilde{T}_1^{s_1})$ to $(\tilde{F}_2, \tilde{T}_2^{s_2})$ such that $\tilde{\rho} \circ \rho_1 = \rho_2$, $\tilde{\tau} \circ \tau_1 = \tau_2$, $g_1 = g_2 \circ \tilde{\rho}$.

We will prove these statements one by one.

1. We first prove that $\sigma$ is a mapping from $S$ to $\mathsf{BStrHom}((F, T^r), G)$. Pick any $\left((\tilde{F}, \tilde{T}^s), (\rho, \tau), g\right) \in S$. Pick any $t \in V_T$ and $u, v \in \beta_T(t)$. Then, $\{u, v\} \in E_F$ iff $\{\rho(u), \rho(v)\} \in E_{\tilde{F}}$ (since $\rho$ is a strong homomorphism from $F[\beta_T(t)]$ to $\tilde{F}[\rho(\beta_T(t))]$). Also, $\rho(u), \rho(v) \in \beta_{\tilde{T}}(\tau(t))$ are in the same bag. Similarly, $\{\rho(u), \rho(v)\} \in E_{\tilde{F}}$ iff $\{g(\rho(u)), g(\rho(v))\} \in E_G$ (since $g$ is a bIso). Thus, $g \circ \rho$ is a bag-strong homomorphism.

2. We next prove that $\sigma$ is surjective. Given $h \in \mathsf{BStrHom}((F, T^r), G)$, define $(\tilde{F}, \tilde{T}^s)$, $(\rho, \tau)$, and $g$ as follows. First define a relation $\sim$ on set $V_F$ such that $u \sim v$ iff the following hold:

   a) $h(u) = h(v)$;

b) There exists a path $P$ in $T^r$ with endpoints $t_1, t_2 \in V_T$ such that $u \in \beta_T(t_1)$, $v \in \beta_T(t_2)$, and all node $t$ on path $P$ satisfies that $h(u) \in h(\beta_T(t))$.

It is easy to see that $\sim$ is an equivalence relation on $V_F$. We can thus define a mapping $\rho$ that respects the relation, i.e., $\rho(u) = \rho(v)$ iff $u \sim v$ for all $u, v \in V_F$. Moreover, for any edge $\{u, v\} \in E_F$, $\rho(u) \neq \rho(v)$ (since $h$ is a homomorphism and $h(u) \neq h(v)$). This implies that we can define $\tilde{F}$ to be the homomorphic image of $F$ such that $\rho$ is the surjective homomorphism on both vertices and edges.

We then define the mapping $g : V_{\tilde{F}} \to V_G$ such that $g(\rho(u)) = h(u)$ for all $u \in V_F$. Note that $g$ is well-defined since $\rho(u) = \rho(v)$ implies $h(u) = h(v)$ for all $u, v \in V_F$, and $\rho : V_F \to V_{\tilde{F}}$ is surjective. It follows that $h = g \circ \rho$. To prove that $g$ is a homomorphism, note that for all $\{x, y\} \in E_{\tilde{F}}$, there exists an edge $\{u, v\} \in E_F$ with $\rho(u) = x$, $\rho(v) = y$, which implies that $\{h(u), h(v)\} \in E_G$ (since $h$ is a homomorphism), namely, $\{g(x), g(y)\} \in E_G$.

We next define tree $\tilde{T}^s = (V_T, E_T, \beta_{\tilde{T}})$, $s = t$, and identity mapping $\tau$ so that $\tau$ is an isomorphism from $T_r$ to $T_s$ (ignoring the labels). Set $\beta_{\tilde{T}}(t) = \rho(\beta_T(t))$ for all $t \in V_T$. We will prove that $(\tilde{F}, \tilde{T}^s) \in \mathcal{S}^M$ is a valid tree decomposition. It suffices to prove that Definition C.1(c) holds. Pick any vertex $x \in V_{\tilde{F}}$ and tree node $t_1, t_2 \in B_{\tilde{T}}(x)$. Then, there exists $u \in \beta_T(t_1), v \in \beta_T(t_2)$ such that $\rho(u) = x$, $\rho(v) = x$. Therefore, $u \sim v$. As such, there exists a path $P$ in $T^r$ such that all node $t$ on $P$ satisfies that there exists $w \in \beta_T(t)$ with $h(w) = h(u)$, namely, $w \sim u$. Consequently, $\rho(u) \in \beta_{\tilde{T}}(t)$, implying that $\tilde{T}^s[B_{\tilde{T}}(x)]$ is connected. This proves that $(\tilde{F}, \tilde{T}^s) \in \mathcal{S}^M$. Also, $(\rho, \tau)$ is clearly a homomorphism from $(F, T^r)$ to $(\tilde{F}, \tilde{T}^s)$ according to Definition C.20.

It remains to prove that $\rho$ is a bag-strong homomorphism and $g$ is a bIso. Pick any $t \in V_T$ and $u, v \in \beta_T(t)$. If $\{u, v\} \notin E_F$, then $\{h(u), h(v)\} \notin E_G$ (since $h$ is a bag-strong homomorphism). Therefore, $\{\rho(u), \rho(v)\} \notin E_{\tilde{F}}$ (since $g$ is a homomorphism), namely, $\rho$ is a bag-strong homomorphism. Since $\rho$ is surjective, $\{\rho(u), \rho(v)\}$ ranges over all vertices in the same bag of $\tilde{T}^s$ when $t \in V_T$ and $u, v \in \beta_T(t)$ are arbitrary. Therefore, $g$ is a bIso because $\{\rho(u), \rho(v)\} \notin E_{\tilde{F}}$ iff $\{h(u), h(v)\} \notin E_G$.

3. We finally prove that $\sigma\left((\tilde{F}_1, \tilde{T}_1^{s_1}), (\rho_1, \tau_1), g_1\right) = \sigma\left((\tilde{F}_2, \tilde{T}_2^{s_2}), (\rho_2, \tau_2), g_2\right)$ implies there exists an isomorphism $(\tilde{\rho}, \tilde{\tau})$ from $(\tilde{F}_1, \tilde{T}_1^{s_1})$ to $(\tilde{F}_2, \tilde{T}_2^{s_2})$ such that $\tilde{\rho} \circ \rho_1 = \rho_2$, $\tilde{\tau} \circ \tau_1 = \tau_2$, $g_1 = g_2 \circ \tilde{\rho}$. Let $h = g_1 \circ \rho_1 = g_2 \circ \rho_2$. Here, we will only prove that $\tilde{F}_1 \simeq \tilde{F}_2$ since the remaining procedure is almost the same as previous proofs. Since both $\tilde{F}_1$ and $\tilde{F}_2$ are homomorphic images of $F$, it suffices to prove that, for all $u, v \in V_F$, $\rho_1(u) = \rho_1(v)$ iff the following hold:

a) $h(u) = h(v)$;

b) There exists a path $P$ in $T^r$ with endpoints $t_1, t_2 \in V_T$ such that $u \in \beta_T(t_1)$, $v \in \beta_T(t_2)$, and all node $t$ on path $P$ satisfies that $h(u) \in h(\beta_T(t))$.

On one hand, if $\rho_1(u) = \rho_1(v)$, we clearly have $h(u) = h(v)$ and there exists $t_1 \in B_T(u), t_2 \in B_T(v)$ such that $t_1, t_2 \in B_{\tilde{T}_1}(\rho_1(u))$. Since $\tilde{T}_1[B_{\tilde{T}_1}(\rho_1(u))]$ is connected, there is a path $P$ with endpoints $t_1, t_2$ such that all node $t$ on $P$ satisfies $\rho_1(u) \in \rho_1(\beta_T(t))$ and thus $h(u) \in h(\beta_T(t))$.

On the other hand, if $\rho_1(u) \neq \rho_1(v)$ but the above items (a) and (b) hold, consider two cases:

- $u$ and $v$ are in the same bag of $T$. Then, $\rho_1(u)$ and $\rho_1(v)$ are in the same bag of $\tilde{T}_1$. Since $g_1$ is a bIso, $g_1(\rho_1(u)) \neq g_1(\rho_1(v))$, which contradicts item (a).
- $u$ and $v$ are not in the same bag of $T$. Then, there exist two adjacent nodes $t_1, t_2$ on $P$ such that $\rho_1(u) \in \beta_{\tilde{T}_1}(t_1)$, $\rho_1(u) \notin \beta_{\tilde{T}_1}(t_2)$. By Definition C.4(c), $\beta_{\tilde{T}_1}(t_2) \subset \beta_{\tilde{T}_1}(t_1)$. Also, item (b) implies that there exists $w \in \beta_T(t_2)$ such that $h(w) = h(u)$. Therefore, $\rho_1(u)$ and $\rho_1(w)$ are two different nodes in $\beta_{\tilde{T}_1}(t_1)$ with $g_1(\rho_1(u)) = h(u) = h(w) = g_1(\rho_1(w))$. This contradicts the condition that $g_1$ is a bIso.

This yields the desired result that $\tilde{F}_1 \simeq \tilde{F}_2$.

Combining the above three items concludes the proof. $\qquad \square$

Let $M \in \{\mathsf{Sub}, \mathsf{L}, \mathsf{LF}, \mathsf{F}\}$ be any model. We can list all *non-isomorphic* tree-decomposed (labeled) graphs in $\mathcal{S}^M$ into an infinite sequence $(F_1, T_1^{r_1}), (F_2, T_2^{r_2}), \cdots$. Consider two types of ordering:

- Ordered by the size of the graphs $F_i$. We denote this ordering as 1st. This ordering requires that for any $i < j$, either $|V_{F_i}| < |V_{F_j}|$ or $|V_{F_i}| = |V_{F_j}|$ and $|E_{F_i}| \geq |E_{F_j}|$. Note that when two graphs have an equal number of vertices, we place the graph with more edges to the front. When the number of edges is also the same, they can be arranged in any fixed order.
- Ordered by the size of the trees $T_i$. We denote this ordering as 2nd. This ordering requires that $|V_{T_i}| \leq |V_{T_j}|$ for any $i < j$. When the number of tree nodes is the same, they can be arranged in any fixed order.

Without loss of generality, we assume that the labels of each graph $F_i$ are integers in the range of $[1, |V_{F_i}|]$. Note that using integer labels of a bounded range is already sufficient to represent all non-isomorphic labeled graphs up to a bijective label transformation. This ensures that the index $i$ is countable.

For the 1st ordering, define the following notations:

a) Let $f : \mathcal{S}^M \times \mathcal{S}^M \to \mathbb{N}$ be any mapping. Define the associated (infinite) matrix $\boldsymbol{A}^{f,M,\mathsf{1st}} \in \mathbb{N}^{\mathbb{N}_+ \times \mathbb{N}_+}$ such that $A_{i,j}^{f,M,\mathsf{1st}} = f((F_i, T_i^{r_i}), (F_j, T_j^{r_j}))$.

b) Let $g : \mathcal{S}^M \times \mathcal{G} \to \mathbb{N}$ be any mapping. Given a graph $G \in \mathcal{G}$, define the (infinite) vector $\boldsymbol{p}_G^{g,M,\mathsf{1st}} \in \mathbb{N}^{\mathbb{N}_+}$ such that $p_{G,i}^{f,M,\mathsf{1st}} = g((F_i, T_i^{r_i}), G)$.

c) Let $h : \mathcal{G} \times \mathcal{G} \to \mathbb{N}$ be any mapping. Given a graph $G \in \mathcal{G}$, define the (infinite) vector $\boldsymbol{p}_G^{h,M,\mathsf{1st}} \in \mathbb{N}^{\mathbb{N}_+}$ such that $p_{G,i}^{h,M,\mathsf{1st}} = h(F_i, G)$.

We can similarly define $\boldsymbol{A}^{f,M,\mathsf{2nd}}, \boldsymbol{p}_G^{g,M,\mathsf{2nd}}, \boldsymbol{p}_G^{h,M,\mathsf{2nd}}$ for the ordering 2nd.

**Corollary C.32.** *Let* $M \in \{\mathsf{Sub}, \mathsf{L}, \mathsf{LF}, \mathsf{F}\}$ *be any model and* $G, H$ *be two graphs. Then,* $\mathsf{hom}(F, G) = \mathsf{hom}(F, H)$ *for all* $(F, T^r) \in \mathcal{S}^M$ *iff* $\mathsf{blso}((F, T^r), G) = \mathsf{blso}((F, T^r), H)$ *for all* $(F, T^r) \in \mathcal{S}^M$.

*Proof.* We separately consider each direction.

1. We first prove that if $\mathsf{blso}((F, T^r), G) = \mathsf{blso}((F, T^r), H)$ for all $(F, T^r) \in \mathcal{S}^M$, then $\mathsf{hom}(F, G) = \mathsf{hom}(F, H)$ for all $(F, T^r) \in \mathcal{S}^M$. According to Lemmas C.29 and C.31, we can rewrite the corresponding equations into matrix forms for any $F \in \mathcal{G}$:

$$\boldsymbol{p}_F^{\mathsf{hom},M,\mathsf{1st}} = \boldsymbol{A}^{\mathsf{bExt},M,\mathsf{1st}} (\boldsymbol{A}^{\mathsf{aut},M,\mathsf{1st}})^{-1} \boldsymbol{p}_F^{\mathsf{bStrHom},M,\mathsf{1st}} \tag{19}$$

$$\boldsymbol{p}_F^{\mathsf{bStrHom},M,\mathsf{1st}} = \boldsymbol{A}^{\mathsf{bStrSurj},M,\mathsf{1st}} (\boldsymbol{A}^{\mathsf{aut},M,\mathsf{1st}})^{-1} \boldsymbol{p}_F^{\mathsf{blso},M,\mathsf{1st}}. \tag{20}$$

From the above equations, we immediately obtain that $\boldsymbol{p}_G^{\mathsf{blso},M,\mathsf{1st}} = \boldsymbol{p}_H^{\mathsf{blso},M,\mathsf{1st}}$ implies $\boldsymbol{p}_G^{\mathsf{hom},M,\mathsf{1st}} = \boldsymbol{p}_H^{\mathsf{hom},M,\mathsf{1st}}$.

2. We next prove that if $\mathsf{hom}(F, G) = \mathsf{hom}(F, H)$ for all $(F, T^r) \in \mathcal{S}^M$, then $\mathsf{blso}((F, T^r), G) = \mathsf{blso}((F, T^r), H)$ for all $(F, T^r) \in \mathcal{S}^M$. This can be seen from the following facts:

   a) $\boldsymbol{A}^{\mathsf{aut},M,\mathsf{1st}}$ is a diagonal matrix and all diagonal elements are *positive* integers.

   b) $\boldsymbol{A}^{\mathsf{bExt},M,\mathsf{1st}}$ is a lower triangular matrix and all diagonal elements are *positive* integers. This is because for any two tree-composed graphs $(F_i, T_i^{r_i})$ and $(F_j, T_j^{r_j})$, $\mathsf{bExt}((F_i, T_i^{r_i}), (F_j, T_j^{r_j})) > 0$ only if $|V_{F_i}| = |V_{F_j}|$ and $|E_{F_i}| \leq |E_{F_j}|$.

   c) $\boldsymbol{A}^{\mathsf{bStrSurj},M,\mathsf{1st}}$ is also a lower triangular matrix and all diagonal elements are *positive* integers. This is because for any two tree-composed graphs $(F_i, T_i^{r_i})$ and $(F_j, T_j^{r_j})$, $\mathsf{bStrSurj}((F_i, T_i^{r_i}), (F_j, T_j^{r_j})) > 0$ only if $|V_{F_i}| > |V_{F_j}|$ or $(|V_{F_i}| = |V_{F_j}|$ and $|E_{F_i}| = |E_{F_j}|)$.

   Therefore, the composition $\boldsymbol{A}^{\mathsf{bExt},M,\mathsf{1st}} (\boldsymbol{A}^{\mathsf{aut},M,\mathsf{1st}})^{-1} \boldsymbol{A}^{\mathsf{bStrSurj},M,\mathsf{1st}} (\boldsymbol{A}^{\mathsf{aut},M,\mathsf{1st}})^{-1}$ is lower triangular and is invertible (although is it an infinite matrix). We thus arrive at the desired conclusion that $\boldsymbol{p}_G^{\mathsf{hom},M,\mathsf{1st}} = \boldsymbol{p}_H^{\mathsf{hom},M,\mathsf{1st}}$ implies $\boldsymbol{p}_G^{\mathsf{blso},M,\mathsf{1st}} = \boldsymbol{p}_H^{\mathsf{blso},M,\mathsf{1st}}$.

$\square$

**Corollary C.33.** *Let* $M \in \{\mathsf{Sub}, \mathsf{L}, \mathsf{LF}, \mathsf{F}\}$ *be any model and* $G, H$ *be two graphs. Then,* $\mathsf{cnt}^M((F, T^r), G) = \mathsf{cnt}^M((F, T^r), H)$ *for all* $(F, T^r) \in \mathcal{S}^M$ *iff* $\mathsf{blso}((F, T^r), G) = \mathsf{blso}((F, T^r), H)$ *for all* $(F, T^r) \in \mathcal{S}^M$.

*Proof.* We separately consider each direction.

1. We first prove that if $\mathsf{cnt}^M((F, T^r), G) = \mathsf{cnt}^M((F, T^r), H)$ for all $(F, T^r) \in \mathcal{S}^M$, then $\mathsf{blso}((F, T^r), G) = \mathsf{blso}((F, T^r), H)$ for all $(F, T^r) \in \mathcal{S}^M$. According to Lemma C.23, we can rewrite Equation (18) into the matrix form for all $F \in \mathcal{G}$:

$$\boldsymbol{p}_F^{\mathsf{blso},M,\mathsf{2nd}} = \boldsymbol{A}^{\mathsf{blsoHom},M,\mathsf{2nd}} \boldsymbol{p}_F^{\mathsf{cnt},M,\mathsf{2nd}}. \tag{21}$$

   This immediately obtains that $\boldsymbol{p}_G^{\mathsf{cnt},M,\mathsf{2nd}} = \boldsymbol{p}_H^{\mathsf{cnt},M,\mathsf{2nd}}$ implies $\boldsymbol{p}_G^{\mathsf{blso},M,\mathsf{2nd}} = \boldsymbol{p}_H^{\mathsf{blso},M,\mathsf{2nd}}$.

2. We next prove that $\boldsymbol{p}_G^{\mathsf{blso},M,\mathsf{2nd}} = \boldsymbol{p}_H^{\mathsf{blso},M,\mathsf{2nd}}$ implies $\boldsymbol{p}_G^{\mathsf{cnt},M,\mathsf{2nd}} = \boldsymbol{p}_H^{\mathsf{cnt},M,\mathsf{2nd}}$. According to Lemma C.25, we have

$$\boldsymbol{A}^{\mathsf{blsoHom},M,\mathsf{2nd}} = \boldsymbol{A}^{\mathsf{blsoSurj},M,\mathsf{2nd}} (\boldsymbol{A}^{\mathsf{aut},M,\mathsf{2nd}})^{-1} \boldsymbol{A}^{\mathsf{blsoInj},M,\mathsf{2nd}}. \tag{22}$$

   Moreover, we have the following facts:

   a) $\boldsymbol{A}^{\mathsf{aut},M,\mathsf{2nd}}$ is a diagonal matrix and all diagonal elements are positive integers.
   b) $\boldsymbol{A}^{\mathsf{blsoInj},M,\mathsf{2nd}}$ is an upper triangular matrix and all diagonal elements are positive integers. This is because for any two tree-composed graphs $(F_i, T_i^{r_i})$ and $(F_j, T_j^{r_j})$, $\mathsf{blsoInj}((F_i, T_i^{r_i}), (F_j, T_j^{r_j})) > 0$ only if $|V_{T_i}| \leq |V_{T_j}|$.
   c) Similarly, $\boldsymbol{A}^{\mathsf{blsoSurj},M,\mathsf{2nd}}$ is a lower triangular matrix and all diagonal elements are positive integers.

   Unfortunately, since $\boldsymbol{A}^{\mathsf{blsoInj},M,\mathsf{2nd}}$ is an *infinite* upper triangular matrix, the inverse matrix is not well-defined. Nevertheless, we can use a special property of bIsoHom to complete our proof, namely, $\mathsf{blsoHom}((F_i, T_i^{r_i}), (F_j, T_j^{r_j})) > 0$ implies that the depth of $T_i^{r_i}$ is equal to the depth of $T_j^{r_j}$. Therefore, denoting by $\boldsymbol{A}^{\mathsf{dep},d}$ the diagonal matrix where $A_{ii}^{\mathsf{dep},d} = \mathbb{I}[\text{the depth of } T_i^{r_i} \text{ is } d]$, we have

$$\boldsymbol{A}^{\mathsf{dep},d} \boldsymbol{p}_F^{\mathsf{blso},M,\mathsf{2nd}} = \boldsymbol{A}^{\mathsf{blsoSurj},M,\mathsf{2nd}} (\boldsymbol{A}^{\mathsf{aut},M,\mathsf{2nd}})^{-1} \boldsymbol{A}^{\mathsf{blsoInj},M,\mathsf{2nd}} \boldsymbol{A}^{\mathsf{dep},d} \boldsymbol{p}_F^{\mathsf{cnt},M,\mathsf{2nd}} \tag{23}$$

   for all $F \in \mathcal{G}$. Fix any integer $d \geq 0$, and assume that $\boldsymbol{A}^{\mathsf{dep},d} \boldsymbol{p}_G^{\mathsf{blso},M,\mathsf{2nd}} = \boldsymbol{A}^{\mathsf{dep},d} \boldsymbol{p}_H^{\mathsf{blso},M,\mathsf{2nd}}$. We will prove that $\boldsymbol{A}^{\mathsf{dep},d} \boldsymbol{p}_G^{\mathsf{cnt},M,\mathsf{2nd}} = \boldsymbol{A}^{\mathsf{dep},d} \boldsymbol{p}_H^{\mathsf{cnt},M,\mathsf{2nd}}$. Since $\boldsymbol{A}^{\mathsf{blsoSurj},M,\mathsf{2nd}}$ is lower triangular with positive diagonal elements, it is invertible and thus

$$\boldsymbol{A}^{\mathsf{blsoInj},M,\mathsf{2nd}} \boldsymbol{A}^{\mathsf{dep},d} \boldsymbol{p}_G^{\mathsf{cnt},M,\mathsf{2nd}} = \boldsymbol{A}^{\mathsf{blsoInj},M,\mathsf{2nd}} \boldsymbol{A}^{\mathsf{dep},d} \boldsymbol{p}_H^{\mathsf{cnt},M,\mathsf{2nd}}.$$

   Moreover, by definition of unfolding tree, there are only finite non-zero elements in both $\boldsymbol{A}^{\mathsf{dep},d} \boldsymbol{p}_G^{\mathsf{cnt},M,\mathsf{2nd}}$ and $\boldsymbol{A}^{\mathsf{dep},d} \boldsymbol{p}_H^{\mathsf{cnt},M,\mathsf{2nd}}$, and the corresponding non-zero indices can only be in a fixed (finite) set. In this case, the upper triangular matrix $\boldsymbol{A}^{\mathsf{blsoInj},M,\mathsf{2nd}}$ is reduced to a finite-dimensional matrix and thus $\boldsymbol{A}^{\mathsf{dep},d} \boldsymbol{p}_G^{\mathsf{cnt},M,\mathsf{2nd}} = \boldsymbol{A}^{\mathsf{dep},d} \boldsymbol{p}_H^{\mathsf{cnt},M,\mathsf{2nd}}$. By enumerating all $d \geq 0$, we obtain the desired result.

   $\square$

Combined with all previous results, we have arrived at the concluding corollary:

**Corollary C.34.** *Let* $M \in \{\mathsf{Sub}, \mathsf{L}, \mathsf{LF}, \mathsf{F}\}$ *be any model. For any two graphs* $G, H$, $\chi_G^M(G) = \chi_H^M(H)$ *iff* $\mathsf{hom}(F, G) = \mathsf{hom}(F, H)$ *for all* $(F, T^r) \in \mathcal{S}^M$.

*Proof.* According to Corollary C.16, for any two graphs $G, H$, we have $\chi_G^M(G) = \chi_H^M(H)$ iff $\mathsf{cnt}^M((F, T^r), G) = \mathsf{cnt}^M((F, T^r), H)$ for all $(F, T^r) \in \mathcal{S}^M$. Then, Corollary C.33 implies that $\mathsf{cnt}^M((F, T^r), G) = \mathsf{cnt}^M((F, T^r), H)$ for all $(F, T^r) \in \mathcal{S}^M$ iff $\mathsf{blso}((F, T^r), G) = \mathsf{blso}((F, T^r), H)$ for all $(F, T^r) \in \mathcal{S}^M$. Finally, Corollary C.32 implies that $\mathsf{blso}((F, T^r), G) = \mathsf{blso}((F, T^r), H)$ for all $(F, T^r) \in \mathcal{S}^M$ iff $\mathsf{hom}(F, G) = \mathsf{hom}(F, H)$ for all $(F, T^r) \in \mathcal{S}^M$. We thus conclude the proof. $\square$

### C.3 Part 2: nested ear decomposition

In this part, we give equivalent formulations of the set $\mathcal{S}^M$ for any model $M \in \{\mathsf{Sub}, \mathsf{L}, \mathsf{LF}, \mathsf{F}\}$ using the concept of NED defined in Definition 3.3.

We first present some technical lemmas that will be used to deal with the tree decomposition of disconnected graphs.

**Lemma C.35.** *Let $M \in \{\mathsf{Sub}, \mathsf{L}, \mathsf{LF}, \mathsf{F}\}$ be any model, and let $(F, T^r) \in \mathcal{S}^M$ be a tree-decomposed graph with $\beta_T(t) = \{\!\{u, v\}\!\}$ for some $u, v \in V_F$. Assume that $F$ has two connected components $S_1, S_2 \subset V_F$ and $u \in S_1$ and $v \in S_2$ are in different components. Then, there exist tree decompositions $(F[S_1], T_1^{r_1}), (F[S_2], T_2^{r_2}) \in \mathcal{S}^M$ such that $\beta_{T_1}(r_1) = \{\!\{u, u\}\!\}$ and $\beta_{T_2}(r_2) = \{\!\{v, v\}\!\}$.*

*Proof.* We can simply define $T_1 = (V_{T_1}, E_{T_1}, \beta_{T_1})$ with $V_{T_1} = V_T$, $E_{T_1} = E_T$, $r_1 = r$, and $\beta_{T_1}(t) = \{\!\{\phi_1(w) : w \in \beta_T(t)\}\!\}$ with

$$\phi_1(w) = \left\{ \begin{array}{ll} w & \text{if } w \in S_1, \\ u & \text{if } w \in S_2. \end{array} \right.$$

In other words, $T_1$ has the same structure as $T$ but with different bags such that all vertices in the same connected component as $v$ are replaced by $u$. It is easy to see that $T_1^{r_1}$ is a canonical tree decomposition of $F[S_1]$ following Definition C.1. Moreover, we clearly have that for all $M \in \{\mathsf{Sub}, \mathsf{L}, \mathsf{LF}, \mathsf{F}\}$, $(F[S_1], T_1^{r_1}) \in \mathcal{S}^M$. We can construct $T_2 = (V_{T_2}, E_{T_2}, \beta_{T_2})$ by symmetry, which concludes the proof. $\quad\square$

The above lemma can be immediately generalized into the following one:

**Corollary C.36.** *Let $M \in \{\mathsf{Sub}, \mathsf{L}, \mathsf{LF}, \mathsf{F}\}$ be any model, and let $(F, T^r) \in \mathcal{S}^M$ be a tree-decomposed graph. For each connected component $S \in V_F$ of $F$, pick any $t \in V_T$ with the minimum depth such that $S \cap \beta_T(t) \neq \emptyset$. Then, there exists a tree decomposition $\tilde{T}^s$ of $F[S]$ satisfying that $(F[S], \tilde{T}^s) \in \mathcal{S}^M$ and $S \cap \beta_T(t) \subset \beta_{\tilde{T}}(s)$.*

**Lemma C.37.** *Let $M \in \{\mathsf{Sub}, \mathsf{L}, \mathsf{LF}, \mathsf{F}\}$ be any model, and let $(F, T^r) \in \mathcal{S}^M$ be a tree-decomposed graph such that $F$ is connected. Then, for each child node $t$ of $r$ in $T^r$ and any vertex $x$ in $F[T^r[t]]$, there is a path in $F[T^r[t]]$ from $x$ to some vertex in $\beta_T(r)$. Consequently, the number of connected components of $F[T^r[t]]$ is bounded by the number of different elements in $\beta_T(r)$.*

*Proof.* Assume the above statement does not hold, and let $S \subset V_{F[T^r[t]]}$ be the connected component of $F[T^r[t]]$ such that $S \cap \beta_T(r) = \emptyset$. Note that $F[S] = F[T^r[t]][S]$. Since $F$ is connected, there exists an edge $\{v, w\} \in E_F$ such that $v \in S$ and $w \notin S$. Since $S$ is a connected component of $F[T^r[t]]$, we have $w \notin V_{F[T^r[t]]}$, and thus both $v, w \notin \beta_T(r)$. This yields a contradiction since $\{v, w\}$ should be contained in a bag in $T^r$ but $B_T(v) \cap B_T(w) = \emptyset$. $\quad\square$

To reduce duplication, below we only give proofs for Local 2-GNN and Local 2-FGNN. One can easily write a proof for Subgrapph GNN based on the proof of Local 2-GNN, and write a proof for 2-FGNN based on the proof of Local 2-FGNN.

**Lemma C.38.** *For any tree-decomposed graph $(F, T^r) \in \mathcal{S}^{\mathsf{L}}$, $F$ has a strong NED.*

*Proof.* Based on Corollary C.36, we can assume that $F$ is connected without loss of generality. We will prove the following stronger result: for any connected $(F, T^r) \in \mathcal{S}^{\mathsf{L}}$ with $\beta_T(r) = \{\!\{u, v\}\!\}$, $F$ has a strong NED where $u, v$ are endpoints of the first ear. (For the case of $u = v$, the other endpoint can be arbitrary.)

The proof is based on induction over the number the vertices in $T^r$. The above statement obviously holds for the base case of $|V_T| = 1$. Now assume that the statement holds when $|V_T| \leq m$, and consider the case of $|V_T| = m + 1$. Note that for any two different children $t, t'$ of $r$, $F[T^r[t]]$ and $F[T^r[t']]$ can only share vertices $u, v$. For each child node $t$ of $r$, denote its unique child node as $\tilde{t}$. It is easy to see that $T^r[\tilde{t}]$ is a canonical tree decomposition of $F[T^r[\tilde{t}]]$ and $(F[T^r[\tilde{t}]], T^r[\tilde{t}]) \in \mathcal{S}^{\mathsf{L}}$. However, one needs to be cautious as $F[T^r[\tilde{t}]]$ may not be connected (unlike the original graph $F$). Below, we separately consider the following cases:

1. First consider the case when $u = v$. In this case, Lemma C.37 impies that $F[T^r[\tilde{t}]]$ is connected. According to the induction hypothesis, $F[T^r[\tilde{t}]]$ has a strong NED (denoted as $\mathcal{P}_{\tilde{t}}$) such that $u$ is an endpoint of the first ear. We then merge the ear decompositions $\mathcal{P}_{\tilde{t}}$ for all $\tilde{t}$ into a whole $\mathcal{P}$, specify a root ear $P_1$ in any $\mathcal{P}_{\tilde{t}}$, and let the first ear of other $\mathcal{P}_{\tilde{t}}$ nest on $P_1$ (with empty nested interval). It is easy to see that $\mathcal{P}$ is a strong NED of $F$ and $u$ is an endpoint of the first ear.

2. Next consider the case when $u \neq v$. In this case, without loss of generality, assume $\beta_T(\tilde{t}) = \{\!\{u, w\}\!\}$ for some $w \in V_F$.

   - Subcase 1: $F[T^r[\tilde{t}]]$ is connected. Then, $F[T^r[\tilde{t}]]$ has a strong NED $\mathcal{P}_{\tilde{t}}$ such that $u$ and $w$ are endpoints of the first ear.
     a) If $w = v$, then $F[T^r[t]] = F[T^r[\tilde{t}]]$ and $F[T^r[t]]$ clearly has a strong NED.
     b) If $w \notin N_F[v]$ or $w = u$, there are two additional subcases depending on whether $\{u, v\} \in E_F$. If $\{u, v\} \in E_F$, then $F[T^r[t]]$ has a strong NED (which can be constructed from $\mathcal{P}_{\tilde{t}}$ by adding an ear $\{\{u, v\}\}$ and letting the first ear in $\mathcal{P}_{\tilde{t}}$ nest on $\{\{u, v\}\}$ (with empty nested interval). Otherwise, $F[T^r[t]]\backslash\{v\}$ has a strong NED and $v$ is an isolated vertex in $F[T^r[t]]$.
     c) If $w \in N_F(v)$ and $w \neq u$, then $F[T^r[t]]$ also has a strong NED, which can be constructed from $\mathcal{P}_{\tilde{t}}$ by extending the first ear to incorporate the edge $\{w, v\}$ (we still need an additional ear $\{\{u, v\}\}$ if $\{u, v\} \in E_F$).
   - Subcase 2: $F[T^r[\tilde{t}]]$ is disconnected. In this subcase, Lemma C.37 implies that $F[T^r[\tilde{t}]]$ has exactly two connected components, and $u$ and $w$ are in different connected components (which can be easily proved by noting that $v$ can only link to $u$ or $w$ in $F[T^r[\tilde{t}]]$). We can thus invoke Lemma C.35, which shows that both connected components of $F[T^r[\tilde{t}]]$, denoted as $\widehat{F}_1$ and $\widehat{F}_2$, admits tree decompositions $\widehat{T}_1^{s_1}$ and $\widehat{T}_2^{s_2}$ such that $(\widehat{F}_1, \widehat{T}_1^{s_1}), (\widehat{F}_2, \widehat{T}_2^{s_2}) \in \mathcal{S}^{\mathsf{L}}$, $\beta_{\widehat{T}_1}(s_1) = \{\!\{u, u\}\!\}$ and $\beta_{\widehat{T}_2}(s_2) = \{\!\{w, w\}\!\}$. Moreover, $|V_{\widehat{T}_1}| = |V_{\widehat{T}_2}| = |V_{T^r[\tilde{t}]}| < m + 1$. Therefore, according to the induction hypothesis, $\widehat{F}_1$ has a strong NED $\mathcal{P}_{\tilde{t},1}$ with $u$ as an endpoint of the first ear, and $\widehat{F}_2$ has a strong NED $\mathcal{P}_{\tilde{t},2}$ with $w$ as an endpoint of the first ear. If $\{w, v\} \in E_F$, we can extend the first ear in $\mathcal{P}_{\tilde{t},2}$ to include the edge $\{w, v\}$. Finally, if $\{u, v\} \in E_F$, by setting the first ear to be $\{\{u, v\}\}$, we can merge the two NEDs $\mathcal{P}_{\tilde{t},1}$ and $\mathcal{P}_{\tilde{t},2}$ to obtain the NED of $F[T^r[t]]$ such that $u, v$ are endpoints of the first ear.

   Overall, we always have that:
   a) If $F[T^r[t]]$ is connected, then it admits a strong NED such that $u, v$ are endpoints of the first ear.
   b) If $F[T^r[t]]$ is disconnected, then it has two connected components each admitting a strong NED such that $u, v$ belong to an endpoint of the first ear for each of the two NEDs, respectively.

   Finally, noting that $F = \bigcup_{t, \mathsf{pa}_{T^r}(t)=r} F[T^r[t]]$, we can merge all NEDs of $F[T^r[t]]$ to obtain a strong NED of $F$ satisfying that $u, v$ are endpoints of the first ear.

We thus conclude the proof of the induction step. $\qquad\square$

**Lemma C.39.** *For any graph $F$, if $F$ admits a strong NED, then there exists a tree decomposition $T^r$ of $F$ such that $(F, T^r) \in \mathcal{S}^{\mathsf{L}}$.*

*Proof.* We can assume that $F$ is connected without loss of generality, as it is easy to merge the tree decompositions of different connected components to form a single tree which is a tree decomposition of the whole graph. We will prove the following stronger result: for any connected graph $F$, if $F$ admits a strong NED where $u$ and $v$ are two different endpoints of the first ear, then there exists a tree decomposition $T^r$ of $F$ such that $(F, T^r) \in \mathcal{S}^{\mathsf{L}}$ and $\beta_T(r) = \{\!\{u, v\}\!\}$.

The proof is based on induction over the number the edges in $F$. The above statement obviously holds for the base case of $|E_F| = 1$. Now assume that the statement holds when $|E_F| \leq m$, and consider the case of $|E_F| = m + 1$. Denote the $i$-th ear as $P_i$ and denote $P_1 = \{\{w_0, w_1\}, \cdots, \{w_{l-1}, w_l\}\}$ where $w_0 = u$, $w_l = v$. Consider the following three cases:

1. Either $u$ or $v$ is a cut vertex in $F$. Without loss of generality, assume that $u$ is a cut vertex. In this case, there must exist a child ear $P_i$ nested on $P_1$ such that $u$ is an endpoint of $P_i$ and $I(P_i) = \emptyset$. We can thus split all ears into two parts $\mathcal{P}_1$ and $\mathcal{P}_2$, where $\mathcal{P}_1$ contains the ear $P_i$ and its descendant ears in the NED tree, and $\mathcal{P}_2$ contains other ears. Denote by $F[\mathcal{P}_k]$ the connected graph induced by all edges in $\mathcal{P}_k$ ($k \in \{1, 2\}$). Then, $\mathcal{P}_k$ is a strong NED of the graph $F[\mathcal{P}_k]$. Since both $\mathcal{P}_1$ and $\mathcal{P}_2$ contain no more than $m$ edges, according to the induction hypothesis, there are tree decompositions $T_k^{r_k}$ of $F[\mathcal{P}_k]$ ($k \in \{1, 2\}$) such that $\beta_{T_1}(r_1) = \{\!\{u, w\}\!\}$ for some $w \in V_F$, $\beta_{T_2}(r_2) = \{\!\{u, v\}\!\}$, and $(F[\mathcal{P}_k], T_k^{r_k}) \in \mathcal{S}^{\mathsf{L}}$ ($k \in \{1, 2\}$). We can glue the tree $T_1^{r_1}$ and $T_2^{r_2}$ into a larger tree $T^{r_2}$ by adding a new node $t$ with $\beta_T(t) = \{\!\{u, v, w\}\!\}$, setting $\mathsf{pa}_{T^r}(t) = r_2$ and $\mathsf{pa}_{T^r}(r_1) = t$. Clearly, $T^{r_2}$ is a valid tree decomposition of $F$, $\beta_T(r_2) = \{\!\{u, v\}\!\}$, and $(F, T^{r_2}) \in \mathcal{S}^{\mathsf{L}}$.

2. There is an ear $P_i$ nested on $P_1$ with $I(P_i) = P_1$. In this case, we can split all ears into two parts $\mathcal{P}_1$ and $\mathcal{P}_2$, where $\mathcal{P}_1$ contains the ear $P_i$ and its descendant ears in the NED tree, and $\mathcal{P}_2$ contains other ears. Then, $\mathcal{P}_k$ is a strong NED of the graph $F[\mathcal{P}_k]$. Since both $\mathcal{P}_1$ and $\mathcal{P}_2$ contain no more than $m$ edges, according to the induction hypothesis, there is a tree decomposition $T_k^{r_k}$ of $F[\mathcal{P}_k]$ such that $\beta_{T_k}(r_k) = \{\!\{u, v\}\!\}$ and $(F[\mathcal{P}_k], T_k^{r_k}) \in \mathcal{S}^{\mathsf{L}}$. By merging the root node, we can glue the tree $T_1^{r_1}$ and $T_2^{r_2}$ into a larger tree $T^r$. Clearly, $T^r$ is a valid tree decomposition of $F$, $\beta_T(r) = \{\!\{u, v\}\!\}$, and $(F, T^r) \in \mathcal{S}^{\mathsf{L}}$.

3. Neither $u$ nor $v$ is a cut vertex in $F$ and all ears $P_i$ nested on $P_1$ satisfies $I(P_i) \subsetneq P_1$. In this case, we have either $\{w_0, w_1\} \notin I(P_i)$ for all ear $P_i$ nested on $P_1$ or $\{w_{l-1}, w_l\} \notin I(P_i)$ for all ear $P_i$ nested on $P_1$ (otherwise, it would contradict the definition of strong NED). Without loss of generality, assume $\{w_0, w_1\} \notin I(P_i)$ for all ears $P_i$ nested on $P_1$. Then, it is clear that $l > 1$, and the subgraph $F \backslash \{u\}$ is connected and also admits a strong NED where $w_1$ and $v$ are two endpoints of the first ear. Therefore, according to the induction hypothesis, there is a tree decomposition $\tilde{T}^s$ of $F \backslash \{u\}$ satisfying $(F \backslash \{u\}, \tilde{T}^s) \in \mathcal{S}^{\mathsf{L}}$ and $\beta_{\tilde{T}}(s) = \{\!\{w_1, v\}\!\}$. We can then construct a tree $T^r$ from $\tilde{T}^s$ by adding two fresh nodes $r$ and $r'$ and setting $\mathsf{pa}_{T^r}(s) = r'$ and $\mathsf{pa}_{T^r}(r') = r$. Set $\beta_T(r) = \{\!\{u, v\}\!\}$, $\beta_T(r') = \{\!\{u, v, w_1\}\!\}$, and $\beta_T(t) = \beta_{\tilde{T}}(t)$ for all $t \in V_{\tilde{T}}$. It is easy to see that the constructed $T^r$ is a tree decomposition of $F$ and $(F, T^r) \in \mathcal{S}^{\mathsf{L}}$.

Combining the above three cases concludes the induction step. $\qquad\square$

**Theorem C.40.** *For any graph $F$, $F$ has a strong NED iff there is a tree decomposition $T^r$ of $F$ such that $(F, T^r) \in \mathcal{S}^{\mathsf{L}}$.*

*Proof.* This is a direct consequence of Lemmas C.38 and C.39. $\qquad\square$

We next turn to Local 2-FGNN, where the proof has a similar structure as that of Local 2-GNN.

**Lemma C.41.** *For any tree-decomposed graph $(F, T^r) \in \mathcal{S}^{\mathsf{LF}}$, $F$ has an almost-strong NED.*

*Proof.* Based on Corollary C.36, we can assume that $F$ is connected without loss of generality. We will prove the following stronger result: for any connected $(F, T^r) \in \mathcal{S}^{\mathsf{LF}}$ with $\beta_T(r) = \{\!\{u, v\}\!\}$, $F$ has an almost-strong NED where $u, v$ are endpoints of the first ear. (For the case of $u = v$, the other endpoint can be arbitrary.)

Similar to the proof of Lemma C.38, assume that the statement holds when $|V_T| \le m$, and consider the case of $|V_T| = m+1$. For each child node $t$ of $r$, if $t$ only has one child, the proof exactly follows the one in Lemma C.38. Therefore, it suffices to consider the case where $t$ has two children $q_1$ and $q_2$. Denote $\beta_T(t) = \{\!\{u, v, w\}\!\}$, $\beta_T(q_1) = \{\!\{u, w\}\!\}$, and $\beta_T(q_2) = \{\!\{v, w\}\!\}$. Since $w \in N_F[u] \cup N_F[v]$ (by definition of $\mathcal{S}^{\mathsf{LF}}$), we can assume that $w \in N_F[v]$ without loss of generality. Recall that $(F[T^r[q_1]], T^r[q_1]), (F[T^r[q_2]], T^r[q_2]) \in \mathcal{S}^{\mathsf{LF}}$. Below, we separately consider the following cases:

1. First consider the case when $u = v$. In this case, Lemma C.37 implies that both $F[T^r[q_1]]$ and $F[T^r[q_2]]$ are connected (since either $u = w$ or $\{u, w\} \in E_F$). According to the induction hypothesis, both $F[T^r[q_1]]$ and $F[T^r[q_2]]$ admit an almost-strong NED such that $u$ and $w$ are endpoints of the first ear. If $w = u$, we can merge the two almost-strong NEDs so that the first ear in one NED is nested on the first ear of the other (with an empty nested interval). Otherwise, the first ears in the two almost-strong NEDs share both endpoints and we can clearly merge them (in case of $\{u, w\} \in E_F$, there is a common ear $\{\!\{u, w\}\!\}$ in both NEDs,

which is taken only once). In both subcases, we obtain an almost-strong NED of $F[T^r[t]]$ such that $u$ is an endpoint of the first ear.

2. Next consider the case when $u \neq v$. In this case, $F[T^r[q_1]]$ and $F[T^r[q_2]]$ share only one vertex $w$. Note that $F[T^r[q_2]]$ is connected by Lemma C.37. We separately consider two subcases:

   - Subcase 1: $F[T^r[q_1]]$ is connected. Then, according to the induction hypothesis, $F[T^r[q_1]]$ admits an almost-strong NED $\mathcal{P}_1$ such that $u$ and $w$ are endpoints of the first ear, and $F[T^r[q_2]]$ admits an almost-strong NED $\mathcal{P}_2$ such that $w$ and $v$ are endpoints of the first ear. (i) If $w = v$, we can merge the two almost-strong NEDs so that the first ear of $\mathcal{P}_2$ is nested on the first ear of $\mathcal{P}_1$ (with empty nested interval). (ii) If $w = u$, we can merge the two almost-strong NEDs so that the first ear of $\mathcal{P}_1$ is nested on the first ear of $\mathcal{P}_2$ (with empty nested interval). (iii) If $w \in N_F(v)$ and $w \neq u$, the first ear of $\mathcal{P}_2$ can be chosen as $\{\{w, v\}\}$. Then, we can merge the two almost-strong NEDs by gluing the first year in $\mathcal{P}_1$ with the ear $\{\{w, v\}\}$ in $\mathcal{P}_2$. One can see that the resulting NED is almost-strong. Overall, we always have that $F[T^r[t]]$ admits an almost-strong NED such that $u, v$ are endpoints of the first ear.
   - Subcase 2: $F[T^r[q_1]]$ is disconnected. In this subcase, $w \neq u$. Similar to the proof of Lemma C.38, we obtain that $F[T^r[q_1]]$ has exactly two connected components, and $u$ and $w$ are in different connected components. We can thus invoke Lemma C.35, which shows that both connected components of $F[T^r[q_1]]$, denoted as $\widehat{F}_1$ and $\widehat{F}_2$, admit almost-strong NEDs $\mathcal{P}_1$ and $\mathcal{P}_2$, respectively. Moreover, $u$ is an endpoint of the first ear in $\mathcal{P}_1$, and $w$ is an endpoint of the first ear in $\mathcal{P}_2$. Besides, $F[T^r[q_2]]$ admits an almost-strong NED $\mathcal{P}_3$ with $w, v$ as the endpoints of the first ear. By letting the first ear of $\mathcal{P}_2$ nest on the first ear of $\mathcal{P}_3$ (with an empty nested interval), we can merge $\mathcal{P}_2$ and $\mathcal{P}_3$. Then, we can merge $\mathcal{P}_1$ and $\mathcal{P}_2 \cup \mathcal{P}_3$ following the same procedure as Subcase 2 in the proof of Lemma C.38.

   Overall, we always have that:

   a) If $F[T^r[t]]$ is connected, then it admits an almost-strong NED such that $u, v$ are endpoints of the first ear.
   b) If $F[T^r[t]]$ is disconnected, then it has two connected components each admitting an almost-strong NED such that $u, v$ belong to an endpoint of the first ear for each of the two NEDs, respectively.

   In both subcases, it follows that $F[T^r[t]]$ admits an almost-strong NED with $u, v$ as two endpoints. Noting that $F = \bigcup_{t, \mathsf{pa}_{T^r}(t)=r} F[T^r[t]]$, we can merge all NEDs of $F[T^r[t]]$ to obtain an almost-strong NED of $F$ with $u, v$ as two endpoints.

We thus conclude the proof of the induction step. $\qquad \square$

**Lemma C.42.** *For any graph $F$, if $F$ admits an almost-strong NED, then there exists a tree decomposition $T^r$ of $F$ such that $(F, T^r) \in \mathcal{S}^{\mathsf{LF}}$.*

*Proof.* We can assume that $F$ is connected without loss of generality. We will prove the following stronger result: for any graph $F$, if $F$ admits an almost-strong NED where $u$ and $v$ are two different endpoints of the first ear, then there exists a tree decomposition $T^r$ of $F$ such that $(F, T^r) \in \mathcal{S}^{\mathsf{LF}}$ and $\beta_T(r) = \{\!\{u, v\}\!\}$.

Similar to the proof of Lemma C.39, assume that the statement holds when $|E_F| \leq m$, and consider the case of $|E_F| = m+1$. Denote the first ear as $P_1 = \{\{w_0, w_1\}, \cdots, \{w_{l-1}, w_l\}\}$ where $w_0 = u$, $w_l = v$. Consider the following three cases:

1. Either $u$ or $v$ is a cut vertex in $F$. This case is exactly the same as in Lemma C.39.

2. There is an ear $P_i$ nested on $P_1$ with $I(P_i) = P_1$. This case is also the same as in Lemma C.39.

3. Otherwise, $l \geq 2$.

   - If there is an ear $P_i$ nested on $P_1$ with $\{w_0, w_1\}, \{w_1, w_2\} \in I(P_i)$, then $l \geq 3$. By definition of almost-strong NED, there does not exist an ear $P_j$ nested on $P_1$ with $\{w_{l-2}, w_{l-1}\}, \{w_{l-1}, w_l\} \in I(P_i)$. In this case, we can split $P_1$ into two parts:

$P_{1,u} = P_1 \backslash \{\{w_{l-1}, w_l\}\}$, and $P_{1,v} = \{\{w_{l-1}, w_l\}\}$. Then, we can rearrange any ear nested on $P_1$ so that it is nested on either $P_{1,u}$ or $P_{1,v}$. In this way, we can split all ears into two sets $\mathcal{P}_u$ and $\mathcal{P}_v$, one corresponding to $P_{1,u}$ and its descendant ears, and the other corresponding to $P_{1,v}$ and its descendant ears. Therefore, according to the induction hypothesis, there is a tree decomposition $\tilde{T}_u^s$ of $F[\mathcal{P}_u]$ satisfying $(F[\mathcal{P}_u], \tilde{T}_u^s) \in \mathcal{S}^{\mathsf{LF}}$ and $\beta_{\tilde{T}_u}(s) = \{\{u, w_{l-1}\}\}$, and similarly, there is a tree decomposition $\tilde{T}_v^{s'}$ of $F[\mathcal{P}_v]$ satisfying $(F[\mathcal{P}_v], \tilde{T}_v^{s'}) \in \mathcal{S}^{\mathsf{LF}}$ and $\beta_{\tilde{T}_v}(s) = \{\{w_{l-1}, v\}\}$. We can then construct a tree $T^r$ by merging $\tilde{T}_u^s$ and $\tilde{T}_v^{s'}$ and adding two fresh nodes $r, r'$, where $r$ is the root node, $\mathsf{pa}_{T^r}(r') = r$ and $\mathsf{pa}_{T^r}(s) = \mathsf{pa}_{T^r}(s') = r'$. Set $\beta_T(r) = \{\{u, v\}\}$ and $\beta_T(r') = \{\{u, v, w_{l-1}\}\}$. It is easy to see that the constructed $T^r$ is a tree decomposition of $F$ and $(F, T^r) \in \mathcal{S}^{\mathsf{LF}}$.

- If there does not exist an ear $P_i$ nested on $P_1$ with $\{w_0, w_1\}, \{w_1, w_2\} \in I(P_i)$, then we follow exactly the same analysis as in the previous item, except that now we split $P_1$ into two parts: $P_{1,u} = \{\{w_0, w_1\}\}$, and $P_{1,v} = P_1 \backslash \{\{w_0, w_1\}\}$. We can still construct a tree decomposition $T^r$ of $F$ such that $(F, T^r) \in \mathcal{S}^{\mathsf{LF}}$.

Combining the above three cases concludes the proof. $\square$

**Theorem C.43.** *For any graph $F$, $F$ has an almost-strong NED iff there is a tree decomposition $T^r$ of $F$ such that $(F, T^r) \in \mathcal{S}^{\mathsf{LF}}$.*

*Proof.* This is a direct consequence of Lemmas C.41 and C.42. $\square$

## C.4  PART 3: PEBBLE GAME

In this part, we prove that $\mathcal{S}^M$ is maximal (Definition 3.1(b)) for any $M \in \{\mathsf{Sub}, \mathsf{L}, \mathsf{LF}, \mathsf{F}\}$. To achieve this, we first introduce a general class of graphs which we call Fürer graphs (Fürer, 2001). Intuitions and illustrations of Fürer graphs can be found in Zhang et al. (2023a).

**Definition C.44** (Fürer graphs). Given any connected graph $F = (V_F, E_F, \ell_F)$, the Fürer graph $G(F) = (V_{G(F)}, E_{G(F)}, \ell_{G(F)})$ is constructed as follows:

$$V_{G(F)} = \{(x, X) : x \in V_F, X \subset N_F(x), |X| \bmod 2 = 0\},$$
$$E_{G(F)} = \{\{(x, X), (y, Y)\} \subset V_G : \{x, y\} \in E_F, (x \in Y \leftrightarrow y \in X)\},$$
$$\ell_{G(F)}(x, X) = \ell_F(x) \quad \forall (x, X) \in V_{G(F)}.$$

Here, $x \in Y \leftrightarrow y \in X$ holds when either ($x \in Y$ and $y \in X$) or ($x \notin Y$ and $y \notin X$) holds. For each $x \in V_F$, denote the set

$$\mathsf{Meta}_F(x) := \{(x, X) : X \subset N_F(x), |X| \bmod 2 = 0\}, \tag{24}$$

which is called the meta vertices of $G(F)$ associated to $x$. Note that $V_{G(F)} = \bigcup_{x \in V_F} \mathsf{Meta}_F(x)$.

We next define an operation called "twist":

**Definition C.45** (Twist). Let $G(F) = (V_{G(F)}, E_{G(F)}, \ell_{G(F)})$ be the Fürer graph of $F = (V_F, E_F, \ell_F)$, and let $\{x, y\} \in E_F$ be an edge of $F$. The *twisted* Fürer graph of $G(F)$ for edge $\{x, y\}$, is constructed as follows: $\mathsf{twist}(G(F), \{x, y\}) := (V_{G(F)}, E_{\mathsf{twist}(G(F), \{x,y\})}, \ell_{G(F)})$, where

$$E_{\mathsf{twist}(G(F), \{x,y\})} := E_{G(F)} \triangle \{\{\xi, \eta\} : \xi \in \mathsf{Meta}_F(x), \eta \in \mathsf{Meta}_F(y)\},$$

and $\triangle$ is the symmetric difference operator, i.e., $A \triangle B = (A \backslash B) \cup (B \backslash A)$. For an edge set $S = \{e_1, \cdots, e_k\} \subset E_F$, we further define

$$\mathsf{twist}(G(F), S) := \mathsf{twist}(\cdots \mathsf{twist}(G(F), e_1) \cdots, e_k). \tag{25}$$

Note that Equation (25) is well-defined as the resulting graph does not depend on the order of edges $e_1, \cdots, e_k$ for twisting.

The following result is well-known (see e.g., Zhang et al., 2023a, Corollary I.5 and Lemma I.7)):

**Theorem C.46.** *For any connected graph $F$ and any set $S_1, S_2 \subset E_F$, $\mathsf{twist}(G(F), S_1) \simeq \mathsf{twist}(G(F), S_2)$ iff $|S_1| \bmod 2 = |S_2| \bmod 2$.*

This result shows that the structure of twisted Fürer graph does not depend on which edge is twisted. Therefore, we can pick any edge $\{u, v\} \in E_F$ and denote $H(F) = \mathsf{twist}(G(F), \{u, v\})$. We now show that $G(F), H(F)$ can be distinguished via homomorphism information:

**Theorem C.47.** *For any graph $F$, let (twisted) Fürer graphs $G(F), H(F)$ be defined above. Then,* $\mathsf{hom}(F, G(F)) \neq \mathsf{hom}(F, H(F))$.

*Proof.* In the proof below, we use $g$ and $h$ to denote mappings where $g : V_F \to V_{G(F)}$ and $h : V_F \to V_{H(F)}$. The proof is divided into the following parts.

1. We first prove that there is a homomorphism $g \in \mathsf{Hom}(F, G(F))$ satisfying $g(w) \in \mathsf{Meta}_F(w)$ for all $w \in V_F$, but there is no homomorphism $h \in \mathsf{Hom}(F, H(F))$ satisfying $h(w) \in \mathsf{Meta}_F(w)$ for all $w \in V_F$.

   a) Define a mapping $g : V_F \to V_{G(F)}$ such that $g(w) = (w, \emptyset)$ for all $w \in V_F$. We clearly have $\{w, x\} \in E_F$ implies that $\{g(w), g(x)\} \in E_{G(F)}$. Moreover, $\ell_{G(F)}(g(w)) = \ell_F(w)$. Therefore, $g \in \mathsf{Hom}(F, G(F))$ is indeed a homomorphism.

   b) If we similarly define $h : V_F \to V_{H(F)}$ such that $h(w) = (w, \emptyset)$ for all $w \in V_F$, then for all $\{w, x\} \in E_F \backslash \{\{u, v\}\}$ we have $\{h(w), h(x)\} \in E_{H(F)}$, but $\{h(u), h(v)\} \notin E_{H(F)}$ since the edge $\{u, v\}$ is twisted.

   It remains to prove that for all $h : V_F \to V_{H(F)}$ of the form $h(w) = (w, U_w)$ (for each $w \in V_F$), $h \notin \mathsf{Hom}(F, H(F))$. It suffices to prove that there is an *odd* number of edges $\{w, x\} \in E_F$ such that $\{h(w), h(x)\} \notin E_{H(F)}$. Let $h, \tilde{h}$ be two such mappings that differ in only one vertex $z$, i.e., $h(w) = \tilde{h}(w)$ for all $w \neq z$ but $h(z) = (z, U_z) \neq (z, \tilde{U}_z) = \tilde{h}(z)$. Denote $D_z = \tilde{U}_z \triangle U_z$. Based on the definition of Fürer graph, it follows that

   - for all $\{w, x\} \in E_F$ with $w \neq z, x \neq z$, we have $\{h(w), h(x)\} \in E_{H(F)}$ iff $\{\tilde{h}(w), \tilde{h}(x)\} \in E_{H(F)}$;
   - for all $\{w, z\} \in E_F$ with $w \notin D_z$, we also have $\{h(w), h(z)\} \in E_{H(F)}$ iff $\{\tilde{h}(w), \tilde{h}(z)\} \in E_{H(F)}$;
   - for all $\{w, z\} \in E_F$ with $w \in D_z$, we have $\{h(w), h(z)\} \in E_{H(F)}$ iff $\{\tilde{h}(w), \tilde{h}(z)\} \notin E_{H(F)}$.

   Since $|U_z| \bmod 2 = 0$ and $|\tilde{U}_z| \bmod 2 = 0$, we have $|D_z| \bmod 2 = 0$ and thus the number of edges $\{w, x\} \in E_F$ such that $\{h(w), h(x)\} \notin E_{H(F)}$ has the same parity as the number of edges such that $\{\tilde{h}(w), \tilde{h}(x)\} \notin E_{H(F)}$. Finally, noting that all mappings $h$ can be obtained from the one in (b) by continually modifying $U_w$ for each $w \in V_F$ and the parity remains unchanged, we have concluded the proof of this part.

2. We next prove that for any permutation $\pi : V_F \to V_F$, there does not exist a homomorphism $h \in \mathsf{Hom}(F, H(F))$ satisfying $h(w) \in \mathsf{Meta}_F(\pi(w))$ for all $w \in V_F$. Assume the conclusion does not hold and pick any $h$ satisfying the above condition. Consider the following cases:

   a) Case 1: if $\pi$ is an automorphism of $F$, then it is easy to see that $h \circ \pi^{-1} \in \mathsf{Hom}(F, H(F))$, because $\{w, x\} \in E_F \implies \{\pi^{-1}(w), \pi^{-1}(x)\} \in E_F \implies \{h(\pi^{-1}(w)), h(\pi^{-1}(x))\} \in E_F$ for all $w, x \in V_F$, and $\ell_F(w) = \ell_F(\pi^{-1}(w)) = \ell_{G(F)}(h(\pi^{-1}(w)))$ for all $w \in V_F$. Moreover, $h \circ \pi^{-1}$ satisfies that $h(\pi^{-1}(w)) \in \mathsf{Meta}_F(w)$, yielding a contradiction to point 1.

   b) Case 2: if $\pi$ is not an automorphism of $F$, then there exists an edge $\{w, x\} \in E_F$ such that $\{\pi(w), \pi(x)\} \notin E_F$. In this case, we must have $\{h(w), h(x)\} \notin E_{H(F)}$ since by definition $h(w) \in \mathsf{Meta}_F(\pi(w))$, $h(x) \in \mathsf{Meta}_F(\pi(x))$, and $\{\pi(w), \pi(x)\}$ is not an edge of $F$.

   In both cases, $h$ is invalid and thus there is no homomorphism $h \in \mathsf{Hom}(F, H(F))$ satisfying $h(w) \in \mathsf{Meta}_F(\pi(w))$ for all $w \in V_F$.

3. We finally prove that the following two sets have equal size (i.e., $|S_G| = |S_H|$):

$$S_G := \{g \in \mathsf{Hom}(F, G(F)) : \exists w, x, y \in V_F \text{ s.t. } x \neq y, g(x), g(y) \in \mathsf{Meta}_F(w)\},$$
$$S_H := \{h \in \mathsf{Hom}(F, H(F)) : \exists w, x, y \in V_F \text{ s.t. } x \neq y, h(x), h(y) \in \mathsf{Meta}_F(w)\}.$$

It suffices to prove that, for any proper subset $U \subsetneq V_F$, we have $|S_G^U| = |S_H^U|$, where $S_G$ and $S_H$ are defined as follows:

$$S_G^U := \left\{ g \in \mathsf{Hom}(F, G(F)) : g(x) \in \bigcup_{w \in U} \mathsf{Meta}_F(w) \ \forall x \in V_F \right\},$$

$$S_H^U := \left\{ h \in \mathsf{Hom}(F, H(F)) : h(x) \in \bigcup_{w \in U} \mathsf{Meta}_F(w) \ \forall x \in V_F \right\}.$$

Fix $U \subsetneq V_F$ and pick $z \in V_F \backslash U$. Let $P$ be a simple path from $u$ to $z$ of the form $P = \{\{w_0, w_1\}, \cdots, \{w_{k-1}, w_k\}\} \subset E_F$ where $\{w_0, w_1\} = \{u, v\}$, $w_k = z$. Define a mapping $\sigma$ that takes $g \in S_G^U$ as input and outputs a mapping $h : V_F \to V_{H(F)}$:

$$h(x) = \begin{cases} g(x) & \text{if } [g(x)]_0 \notin \{w_1, \cdots, w_{k-1}\}, \\ ([g(x)]_0, [g(x)]_1 \triangle \{w_{i-1}, w_{i+1}\}) & \text{if } [g(x)]_0 = w_i, i \in [k-1], \end{cases}$$

where we write $g(x) = ([g(x)]_0, [g(x)]_1)$. We will prove that $h \in S_H^U$. Since we clearly have $h(w) \in U$ for all $w \in V_F$, it suffices to prove that $h \in \mathsf{Hom}(F, H(F))$. Let $\{x, y\} \in E_F$ be any edge in $F$.

- If $[g(x)]_0, [g(y)]_0 \notin \{w_1, \cdots, w_{k-1}\}$, then $\{h(x), h(y)\} = \{g(x), g(y)\} \in E_{G(F)}$. Also, since $\{[g(x)]_0, [g(y)]_0\} \neq \{u, v\}$, $\{[g(x)]_0, [g(y)]_0\}$ is not twisted and thus $\{h(x), h(y)\} \in E_{H(F)}$.
- If $[g(x)]_0 \notin \{w_1, \cdots, w_{k-1}\}$, $[g(y)]_0 = w_i$ for some $i \in [k-1]$, and $\{[g(x)]_0, [g(y)]_0\} \neq \{\{u, v\}, \{w_{k-1}, w_k\}\}$, then

$$\{h(x), h(y)\} = \{([g(x)]_0, [g(x)]_1), ([g(y)]_0, [g(y)]_1 \triangle \{w_{i-1}, w_{i+1}\})\}.$$

We have

$$\begin{aligned} \{g(x), g(y)\} \in E_{G(F)} &\iff ([g(x)]_0 \in [g(y)]_1) \leftrightarrow ([g(y)]_0 \in [g(x)]_1) \\ &\iff ([g(x)]_0 \in [g(y)]_1 \triangle \{w_{i-1}, w_{i+1}\}) \leftrightarrow ([g(y)]_0 \in [g(x)]_1) \\ &\iff \{h(x), h(y)\} \in E_{G(F)}. \end{aligned}$$

Also, since $\{[g(x)]_0, [g(y)]_0\}$ is not twisted, $\{h(x), h(y)\} \in E_{H(F)}$.
- If $[g(x)]_0, [g(y)]_0 \in \{w_1, \cdots, w_{k-1}\}$, the analysis is similar to the above one and we can still prove that $\{h(x), h(y)\} \in E_{G(F)}$ and thus $\{h(x), h(y)\} \in E_{G(F)}$.
- If $\{[g(x)]_0, [g(y)]_0\} = \{u, v\}$, without loss of generality assume $w_0 = u = [g(x)]_0$ and $w_1 = v = [g(y)]_0$. We have

$$\begin{aligned} \{g(x), g(y)\} \in E_{G(F)} &\iff (u \in [g(y)]_1) \leftrightarrow (v \in [g(x)]_1) \\ &\iff (u \notin [g(y)]_1 \triangle \{u, w_2\}) \leftrightarrow (v \in [g(x)]_1) \\ &\iff \{h(x), h(y)\} \notin E_{G(F)}. \end{aligned}$$

However, since $\{u, v\}$ is twisted, we still have $\{h(x), h(y)\} \in E_{H(F)}$.
- Finally, we do not need to consider the case $\{[g(x)]_0, [g(y)]_0\} = \{w_{k-1}, w_k\}$ since $w_k = z \notin U$.

This proves that $h \in S_H^U$. Moreover, it is straightforward to see that the mapping $\sigma(g) = h$ is a bijection from $S_G^U$ to $S_H^U$. We have thus proved that $|S_G^U| = |S_H^U|$.

Combining the above three items, we obtain $\mathsf{Hom}(F, G(F)) > \mathsf{Hom}(F, H(F))$, concluding the proof. $\qquad\square$

In the subsequent analysis, we will prove that for all model $M$ considered in Theorem 3.4 and any connected graph $F$, $\chi_{G(F)}^M(G(F)) = \chi_{H(F)}^M(H(F))$ if $F \notin \mathcal{S}^M$. The proof is based on an important technique developed in Cai et al. (1992), called the pebble game. When restricting our analysis on Fürer graphs, the pebble game can be greatly simplified as shown in Fürer (2001); Zhang et al. (2023a). Below, we separately describe the corresponding pebble game for each model $M$.

We first define a key concept called the connected component.

**Definition C.48** (Connected components). Let $F = (V_F, E_F)$ be a connected graph and let $U \subset V_F$ be a vertex set, called separation vertices. We say two edges $\{u, v\}, \{x, y\} \in E_F$ are in the same connected component if there is a simple path $\{\{y_0, y_1\}, \cdots, \{y_{k-1}, y_k\}\}$ satisfying that $\{y_0, y_1\} = \{u, v\}, \{y_{k-1}, y_k\} = \{x, y\}$ and $y_i \notin U$ for all $i \in [k-1]$. It is easy to see that the above relation between edges forms an *equivalence relation*. Therefore, we can define a partition over the edge set, denoted by $\mathsf{CC}_F(U) = \{P_i : i \in [m]\}$ for some $m$, where each $P_i \subset E_F$ is called a connected component.

We are now ready to describe the game rule. There are two players (named Alice and Bob), a graph $F$, and several pebbles. At the beginning, all pebbles lie outside the graph. Through the game process, some pebbles will be placed on the vertices of $F$ and thus separate the edges $E_F$ into connected components according to Definition C.48. In each game round, Alice updates the location of pebbles, while Bob maintains a subset of connected components, ensuring that the number of selected components is *odd*. There are three major types of operations:

1. **Add a pebble** p. Alice places a pebble p (previously outside the graph) on some vertex of $F$. If introducing this new pebble does not change the connected components, then Bob does nothing. Otherwise, there must be a connected component $P$ separated by p into several components $P = \bigcup_{i \in [m]} P_i$ for some $m$. Bob will update his selected components as follows: if $P$ is selected, then remove $P$ and add a subset of connected components in $\{P_1, \cdots, P_m\}$ while ensuring that the number of selected components in total is odd.

2. **Remove a pebble** p. Alice removes a pebble p (previously on some vertex) outside the graph. If introducing this new pebble does not change the connected components, then Bob does nothing. Otherwise, there are multiple connected component $P_1, \cdots, P_m$ getting merged into a whole $P = \bigcup_{i \in [m]} P_i$. Bob will update his selected components by removing all $P_i, i \in [m]$ (if selected) and optionally adding $P$, while ensuring that the number of selected components in total is odd.

3. **Swap two pebbles** p **and** p′. Alice swaps the position of two pebbles p and p′. This operation does not change the connected components and thus Bob does nothing.

At any time, if there is an edge $\{x, y\}$ such that both endpoints hold pebbles and the connected component $\{\{x, y\}\}$ is selected by Bob, then Bob loses the game and Alice wins. If Alice cannot win through the game process, then Bob wins.

We now define the concrete pebble game for each model $M$ considered in this paper. In cases of Subgraph GNN, Local 2-GNN, Local 2-FGNN, and 2-FGNN, there are three pebbles $p_u$, $p_v$, $p_w$. As described before, all pebbles lie outside the graph at the beginning. Alice first adds the pebble $p_u$ (operation 1) and then adds the pebble $p_v$ (operation 1). Next, the game cyclically executes the following process:

- **Subgraph GNN**. Alice can choose either one of the following ways to play:
    - Remove the pebble $p_v$ (operation 2), and re-add the pebble $p_v$ (operation 1).
    - Add the pebble $p_w$ (operation 1) adjacent to the pebble $p_v$, swap pebble $p_v$ with $p_w$ (operation 3), and remove the pebble $p_w$ (operation 2).
- **Local 2-GNN**. Alice can choose either one of the following ways to play:
    - Remove the pebble $p_u$ (operation 2), and re-add the pebble $p_u$ (operation 1).
    - Remove the pebble $p_v$ (operation 2), and re-add the pebble $p_v$ (operation 1).
    - Add the pebble $p_w$ (operation 1) adjacent to the pebble $p_u$, swap pebble $p_u$ with $p_w$ (operation 3), and remove the pebble $p_w$ (operation 2).
    - Add the pebble $p_w$ (operation 1) adjacent to the pebble $p_v$, swap pebble $p_v$ with $p_w$ (operation 3), and remove the pebble $p_w$ (operation 2).
- **Local 2-FGNN**. Alice can choose either one of the following ways to play:
    - Remove the pebble $p_u$ (operation 2), and re-add the pebble $p_u$ (operation 1).
    - Remove the pebble $p_v$ (operation 2), and re-add the pebble $p_v$ (operation 1).
    - Add the pebble $p_w$ (operation 1) adjacent to either pebble $p_u$ or pebble $p_v$, swap pebble $p_w$ with any adjacent pebble (operation 3), and remove the pebble $p_w$ (operation 2).
- **2-FGNN**. Alice adds the pebble $p_w$ (operation 1), swap the pebbles $p_w$ with either $p_u$ or $p_v$ (operation 3), and remove the pebble $p_w$ (operation 2).

**Proposition C.49.** *For any model $M \in \{\mathsf{Sub}, \mathsf{L}, \mathsf{LF}, \mathsf{F}\}$ and any graph $F$, if Alice cannot win the pebble game associated to model $M$ on graph $F$, then $\chi_{G(F)}^M(G(F)) = \chi_{H(F)}^M(H(F))$.*

*Proof.* Note that the pebble game does not depend on the vertex labels $\ell_F$ in $F$. We first assume that the vertex labels are all different, i.e., $\ell_F(x) \neq \ell_F(y)$ for all $x, y \in V_F$. Then, the labels of vertices $\xi \in V_{G(F)}, \eta \in V_{H(F)}$ are the same iff $\xi$ and $\eta$ belong to the same meta vertex, i.e., $\xi, \eta \in \mathsf{Meta}_F(x)$ for some $x \in V_F$. Based on Proposition C.9, $\chi_{G(F)}^M(G(F)) = \chi_{H(F)}^M(H(F))$ iff $\tilde{\chi}_{G(F)}^M(G(F)) = \tilde{\chi}_{H(F)}^M(H(F))$ where $\tilde{\chi}^M$ is defined in Equations (7) to (10). In this case, the pebble games above exactly correspond to the aggregation formulas in Equations (7) to (10), and we can invoke the results in Zhang et al. (2023a, Theorem I.17) to show that $\tilde{\chi}_{G(F)}^M(G(F)) = \tilde{\chi}_{H(F)}^M(H(F))$ (note that we do not need to consider the augmented Fürer graphs defined in their paper based on our label assignment). Therefore, $\chi_{G(F)}^M(G(F)) = \chi_{H(F)}^M(H(F))$.

We next consider the general case when multiple vertices can have the same label in $F$. However, this can only make it harder to distinguish between $G(F)$ and $H(F)$, and we clearly have $\chi_{G(F)}^M(G(F)) = \chi_{H(F)}^M(H(F))$. $\qquad\square$

Based on the above result, in the subsequent proof we will analyze the players' strategy in the pebble game. Surprisingly, it turns out that given a graph $F$, if Alice can win the game, then her strategy can be described using the tree decomposition of $F$ defined in Definition C.6.

To illustrate this point, we need the concept of game state graph. Given a graph $F$, a game state is a three-tuple $(u, v, Q)$ where $u, v \in V_F \cup \{\emptyset\}$ and $Q \subset \mathsf{CC}_F(\{u, v\})$, denoting the vertex that holds pebble $\mathsf{p}_u$, the vertex that holds pebble $\mathsf{p}_v$, and a subset of connected component selected by Bob, respectively. Here, the symbol $\emptyset$ means that a pebble is left outside the graph. One can see that after any round, whether Alice can win the remaining game purely depends on this tuple. Now fix Alice's strategy. In each round, each game state will be transited to a finite number of states depending on how Bob plays, and all states and transitions form a directed graph, which we call the game state graph. The state $(u, v, Q)$ is called a terminal state if $\min_{P \in Q} |P| = 1$. It is straightforward to see that Alice wins the game at any terminal state, as stated in the following result:

**Proposition C.50.** *Given graph $F$ and model $M \in \{\mathsf{Sub}, \mathsf{L}, \mathsf{LF}, \mathsf{F}\}$, let $G^{\mathsf{S}}$ be the game state graph defined above corresponding to an Alice's strategy. Then, Alice can win the game if there is an integer $t$ such that any path in $G^{\mathsf{S}}$ of length $t$ starting from the initial state $(\emptyset, \emptyset, \{E_F\})$ goes through a terminal state.*

*Proof.* It suffices to prove that, at any terminal state, Alice can win the game. Let $(u, v, Q)$ be a terminal state with $\{x, y\} \in Q$. If $\{u, v\} = \{x, y\}$, Alice already wins. If $u \notin \{x, y\}$ and $v \in \{x, y\}$, Alice also wins in the next round since she can add the pebble $\mathsf{p}_w$ on some vertex $w$ adjacent to $\mathsf{p}_v$ such that $\{v, w\} = \{x, y\}$. This is a valid game rule for all model $M$. Next, if $u \in \{x, y\}$ and $v \notin \{x, y\}$, Alice also wins in the next round since she can first remove the pebble $\mathsf{p}_v$ and then place it on the unique vertex in $\{x, y\} \backslash \{u\}$. This is also a valid game rule for all model $M$. Note the removing $\mathsf{p}_v$ does not merge the connected component $\{\{x, y\}\}$ since $v \notin \{x, y\}$. Finally, if both $u \notin \{x, y\}$ and $v \notin \{x, y\}$, $F$ only has one edge and Alice can clearly win. Combining these cases, we conclude that Alice can always win. $\qquad\square$

Based on the above proposition, a game state $(u, v, Q)$ is called *unreachable* if any path starting from the initial state $(\emptyset, \emptyset, \{E_F\})$ and ending at $(u, v, Q)$ goes through some terminal state. We do not need to consider unreachable states since Alice always wins before reaching it. We next introduce an important technical concept:

**Definition C.51.** *Given a game state graph $G^{\mathsf{S}}$, a state $(u, v, \{P\})$ is termed as "contracted" if for any transition $((u, v, \{P\}), (u', v', \{P'\})) \in E_{G^{\mathsf{S}}}$, $P' \subset P$. It is called strictly contracted if for any transition $((u, v, \{P\}), (u', v', \{P'\})) \in E_{G^{\mathsf{S}}}$, $P' \subsetneq P$.*

We have the following result:

**Lemma C.52.** *For any model $M \in \{\mathsf{Sub}, \mathsf{L}, \mathsf{LF}, \mathsf{F}\}$ and any graph $F$, if Alice can win the pebble game associated to model $M$ on graph $F$, then there exists a game state graph $G^{\mathsf{S}}$ corresponding to a winning strategy of Alice such that any reachable and non-terminal state is strictly contracted.*

*Proof.* We first prove that there is a strategy for Alice such that any reachable and non-terminal state is contracted. We will assume that for any reachable and non-terminal state $(u, v, \{P\})$, if there is only one transition of the form $((u, v, \{P\}), (u', v', \{P'\}))$, then $P \not\subset P'$, since making the connected component larger should be a bad strategy. (Otherwise, it is straightforward to construct a new strategy for Alice satisfying the assumption.) Now suppose that there exists a reachable state $(u, v, \{P\})$ that is not contracted. Note that the game state graph induced by all reachable states is a Directed Acyclic Graph (DAG), so we can choose the state $(u, v, \{P\})$ such that any path from the initial state $(\emptyset, \emptyset, \{E_F\})$ to $(u, v, \{P\})$ does not pass any intermediate state that is not contracted. Below, we will construct a new strategy that removes the state $(u, v, \{P\})$ (making it unreachable).

Note that we clearly have $u \neq \emptyset$ and $v \neq \emptyset$. Without loss of generality, assume that $((u, v, \{P\}), (u, v', \{P'\}))$ is a transition such that $P' \not\subset P$ (the case of transition $((u, v, \{P\}), (u', v, \{P'\}))$ is the same by symmetry). Moreover, we can assume that $P \notin \mathsf{CC}_F(\{u\})$ and $P \notin \mathsf{CC}_F(\{v\})$ (in other words, both $u$ and $v$ are at the boundary of the connected component $P$). It would be easier to analyze the case where either $u$ or $v$ are not at the boundary. We separately consider the following cases:

1. The transition $((u, v, \{P\}), (u, v', \{P'\}))$ corresponds to Alice removing pebble $\mathsf{p}_v$ and placing it on $v'$. Let $(u_0, v_0, Q_0), \cdots, (u_T, v_T, Q_T)$ be any path from the initial state $(\emptyset, \emptyset, \{E_F\})$ to $(u, v, \{P\})$, and let $t \leq T$ be the maximal number such that $v_t \neq v$. We will construct a new strategy for Alice as follows.

   - At state $(u_t, v_t, Q_t)$, she removes pebble $\mathsf{p}_v$ and places it on $v'$, yielding (possibly multiple) states of the form $(u_t, v', \tilde{Q}_t)$.

   - For all $t < \tilde{t} < T$, we have $v_t = v$. For each $\tilde{t}$ and possible $\tilde{Q}_{\tilde{t}}$, she will apply the original strategy at $(u_{\tilde{t}}, v_{\tilde{t}}, Q_{\tilde{t}})$ to the state $(u_{\tilde{t}}, v', \tilde{Q}_{\tilde{t}})$, namely, placing pebble $\mathsf{p}_w$ on vertex $u_{\tilde{t}+1}$, swapping pebble $\mathsf{p}_u$ with $\mathsf{p}_w$, and leaving $\mathsf{p}_w$ outside the graph. (Note that she cannot remove pebble $\mathsf{p}_u$ first; otherwise the state $(u_{\tilde{t}}, v_{\tilde{t}}, Q_{\tilde{t}})$ will not be contracted since $u$ is at the boundary of $Q_{\tilde{t}}$.)

   It follows that $((u_{\tilde{t}}, v', \tilde{Q}_{\tilde{t}}), (u_{\tilde{t}+1}, v', \tilde{Q}_{\tilde{t}+1}))$ is a transition of the new strategy, $(u_{\tilde{t}}, v', \tilde{Q}_{\tilde{t}})$ is contracted, and all states of the form $(u, v', \tilde{Q}_{\tilde{T}})$ is reachable in the original strategy.

   We can repeat the above procedure for all paths from the initial state $(\emptyset, \emptyset, \{E_F\})$ to state $(u, v, \{P\})$. Then, in the new strategy $(u, v, \{P\})$ will be unreachable. However, the state $(u_t, v_t, Q_t)$ now may violate the condition in Lemma C.52. Note that the assumption in the first paragraph is still satisfied for state $(u_t, v_t, Q_t)$. In this case, we can recursively apply the above procedure for state $(u_t, v_t, Q_t)$. Note that the procedure will only repeat a finite number of times, as the length of the path from the initial state to the state $(u_t, v_t, Q_t)$ is strictly less than the length of the path from the initial state to the state $(u, v, \{P\})$.

2. The transition $((u, v, \{P\}), (u, v', \{P'\}))$ corresponds to Alice placing pebble $\mathsf{p}_w$ on $v'$, swapping $\mathsf{p}_v$ and $\mathsf{p}_w$, and removing pebble $\mathsf{p}_w$. In this case, we must have $P \notin \mathsf{CC}_F(\{u, v, v'\})$ (i.e., $v'$ further splits the component $P$). Otherwise, the state $(u, v, \{P\})$ must have only one transition and $P \subset P'$, a contradiction to the first paragraph. Since $P \notin \mathsf{CC}_F(\{u, v, v'\})$, it is thus easy to see that, if Alice changes her strategy by just removing pebble $\mathsf{p}_v$ and placing it on vertex $v'$, all transitions starting from $(u, v, \{P\})$ does not change. Therefore, we can just invoke the previous item to construct a desired strategy.

Combining the two cases, we conclude that there is a strategy for Alice such that any reachable and non-terminal state is contracted. We next prove that any reachable and non-terminal state can be *strictly* contracted. Assume the result does not hold and the state $(u, v, \{P\})$ is reachable, non-terminal, but not strictly contracted. Then, there is a transition $((u, v, \{P\}), (u', v', \{P\})) \in E_{G^s}$. Consider the following two cases:

1. $P \notin \mathsf{CC}_F(\{u\})$ and $P \notin \mathsf{CC}_F(\{v\})$ (i.e., both $u$ and $v$ are at the boundary of the connected component $P$). It follows that $u' = u$ and $v' = v$. This implies that the game state graph is not acyclic, a contradiction.

2. $P \in \mathsf{CC}_F(\{u\})$ or $P \in \mathsf{CC}_F(\{v\})$. Without loss of generality, assume that $P \in \mathsf{CC}_F(\{u\})$. Since $(u, v, \{P\})$ is non-terminal and not strictly contracted, there is a reachable and non-terminal state $(\tilde{u}, \tilde{v}, \{P\})$ with either $\tilde{u} = u$ or $\tilde{v} = u$ such that there is a path from $(u, v, \{P\})$

to $(\tilde{u}, \tilde{v}, \{P\})$ and $(\tilde{u}, \tilde{v}, \{P\})$ is strictly contracted. We can then change the strategy at state $(u, v, \{P\})$ to make it strictly contracted. Concretely, Alice can remove pebble $\mathsf{p}_v$ (which does not merge connected components selected by Bob since $P \in \mathsf{CC}_F(\{u\})$), and place pebble $\mathsf{p}_v$ on the vertex that corresponds to the strategy at state $(\tilde{u}, \tilde{v}, \{P\})$ (possibly with a difference that $u$ and $v$ become exchanged). This makes the state $(u, v, \{P\})$ strictly contracted.

Combining the two cases concludes the proof. $\qquad\square$

We are now ready to state the main theorem:

**Theorem C.53.** *Let $M \in \{\mathsf{Sub}, \mathsf{L}, \mathsf{LF}, \mathsf{F}\}$ be any model. Given any connected graph $F$, if Alice can win the pebble game associated with model $M$ on graph $F$, then there is a tree decomposition $T^r$ of $F$ such that $(F, T^r) \in \mathcal{S}^M$, where $\mathcal{S}^M$ is defined in Definition C.6.*

*Proof.* Let $G^{\mathsf{S}}$ be the game state graph satisfying Lemma C.52. For each game state $s$, denote by $\mathsf{next}_{G^{\mathsf{S}}}(s)$ the set of states $s'$ such that $(s, s')$ is a transition in $G^{\mathsf{S}}$ and $s'$ contains only a single connected component, i.e., $s'$ has the form $(u, v, \{P\})$. For a terminal state $s$ of the form $(u, v, \{\{x, y\}\})$, define $\mathsf{next}_{G^{\mathsf{S}}}(s) = \{(x, y, \{\{x, y\}\})\}$. By definition, $\mathsf{next}_{G^{\mathsf{S}}}(\emptyset, \emptyset, \{E_F\}) = \{(u, \emptyset, Q_1), \cdots, (u, \emptyset, Q_m)\}$ for some $u \in V_F$ and $Q_1, \cdots, Q_m$ is the finest partition of $\mathsf{CC}_F(\{u\})$.

The tree $T^r$ will be recursively constructed as follows. First create the tree root $r$ with $\beta_T(r) = \{\!\{u, u\}\!\}$. As we will see later, the root node will be associated with the set of states $S(r) := \mathsf{next}_{G^{\mathsf{S}}}(\emptyset, \emptyset, \{E_F\})$. We then do the following procedure:

Let $t$ be a leaf node in the current tree associated with a non-empty set of game states $S(t)$ such that $|\bigcup_{(u,v,\{P\}) \in S(t)} P| > 1$. For each state $(u, v, \{P\}) \in S(t)$, create a new node $\tilde{t}$ and set its parent to be $t$. Pick any state $(u', v', \{P'\}) \in \mathsf{next}_{G^{\mathsf{S}}}(u, v, \{P\})$. Then, there must be a unique vertex $w \in \{\!\{u', v'\}\!\} \backslash \{\!\{u, v\}\!\}$ and $w$ does not depend on which $(u', v', \{P'\})$ is picked (by definition of the game rule). Set $\beta_T(\tilde{t}) = \beta_T(t) \cup \{\!\{w\}\!\}$. Then, do the following constructions:

- If there is a state of the form $(u, w, \{P'\}) \in \mathsf{next}_{G^{\mathsf{S}}}(u, v, \{P\})$, then create a new node $t'$ and connect it to the parent $\tilde{t}$. Set $\beta_T(t') = \{\!\{u, w\}\!\}$, and the node $t'$ will be associated to the set of states $S(t') = \{(u, w, \{P'\}) : (u, w, \{P'\}) \in \mathsf{next}_{G^{\mathsf{S}}}(u, v, \{P\})\}$.

- If there is a state of the form $(w, v, \{P''\}) \in \mathsf{next}_{G^{\mathsf{S}}}(u, v, \{P\})$, then create a new node $t''$ and connect it to the parent $\tilde{t}$. Set $\beta_T(t'') = \{\!\{w, v\}\!\}$, and the node $t''$ will be associated to the set of states $S(t'') = \{(w, v, \{P''\}) : (w, v, \{P''\}) \in \mathsf{next}_{G^{\mathsf{S}}}(u, v, \{P\})\}$.

Note that (i) either $w \neq u$ or $w \neq v$ (since the game state graph is a DAG); (ii) both items can be used for one state $(u, v, \{P\})$, which happens for FWL-type GNNs. The construction of the tree completes when each leaf node is associated with only one game state of the form $(x, y, \{\{x, y\}\})$. It is easy to see that the above procedure terminates after adding a finite number of tree nodes.

We now prove that $T^r$ is a canonical tree decomposition of $F$ and $(F, T^r) \in \mathcal{S}^M$.

1. We first prove that any edge in $F$ is contained in some bag of $T^r$. Pick any non-leaf tree node $t$ of even depth and any of its child $\tilde{t}$, denote $(u, v, P) \in S(t)$ be the state associated with $\tilde{t}$ in the above construction, and denote $\beta_T(\tilde{t}) = \{\!\{u, v, w\}\!\}$. We have

$$P = \left( \bigcup_{\substack{t', \mathsf{pa}_{T^r}(t') = \tilde{t}, \\ (x, y, \{P'\}) \in S(t')}} P' \right) \cup \{\{u, w\} \in E_F\} \cup \{\{v, w\} \in E_F\}.$$

Here, $P' \subset P$ is a consequence of Lemma C.52, and the remaining edges $\{u, w\}$ or $\{v, w\}$ are in $P$ because $(u, v, P)$ is strictly contracted and $w$ must be in the interior of $P$. Therefore,

$$P \cup \{\{u, v\} \in E_F\} = \left( \bigcup_{\substack{t', \mathsf{pa}_{T^r}(t') = \tilde{t}, \\ (x, y, \{P'\}) \in S(t')}} (P' \cup \{\{x, y\} \in E_F\}) \right) \cup \{\{x, y\} \in E_F : x, y \in \beta_T(\tilde{t})\}.$$

Recursively applying the above equation yields the desired result

$$E_F = \bigcup_{\tilde{t}, \text{dep}_T(\tilde{t}) \text{ is odd}} \{\{x, y\} \in E_F : x, y \in \beta_T(\tilde{t})\},$$

because (i) $\bigcup_{(u, \emptyset, P) \in S(r)} P = E_F$, and (ii) all leaf node $t$ with $S(t) = (x, y, \{\{x, y\}\})$ contains an edge $\{x, y\}$ in its bag.

2. We next prove that $T^r$ satisfies the condition of Definition C.1(c). Fix any vertex $w \in V_F$, and let $t$ be the tree node with minimal depth that contains $w$. Without loss of generality, assume that $t$ is not the root. In this case, the depth of $t$ is odd and $\beta_T(t) = \{\{u, v, w\}\}$ for some $u \neq w$, $v \neq w$. Let $t'$ be a child node of $t$ and we have $w \in \beta_T(t')$. It remains to prove that for any descendent $\tilde{t} \in \text{Desc}_T(t')$, if $w \in \beta_T(\tilde{t})$, then $w \in \beta_T(\hat{t})$ for all $\hat{t}$ on the path between $t'$ and $\tilde{t}$. This is actually a direct consequence of Lemma C.52, because when a pebble originally placed on $w$ is removed, all edges linked to $w$ will not be selected by Bob and thus any pebble can never be placed on $w$ again.

3. $T^r$ is canonical as Definition C.4 is clearly satisfied.

4. Finally, it is also easy to see that $(F, T^r)$ satisfies Definition C.6.

We thus conclude the proof. $\square$

**Corollary C.54.** *Let $M \in \{\text{Sub}, \text{L}, \text{LF}, \text{F}\}$ be any model. For any connected graph $F \notin \mathcal{F}^M$, let $G(F)$ and $H(F)$ be the Fürer graph and twisted Fürer graph with respect to $F$. Then, $\text{hom}(F, G(F)) \neq \text{hom}(F, H(F))$ and $\chi^M_{G(F)}(G(F)) = \chi^M_{H(F)}(H(F))$.*

*Proof.* The proof directly follows from applying Theorems C.47 and C.53 and Proposition C.49. $\square$

Finally, we remark that our construction can be easily generalized for disconnected graphs $F \notin \mathcal{F}^M$. Let $F$ be the disjoint union of graphs $\{F_i : i \in [m]\}$ where each $F_i$ is the graph corresponding to a connected component of $F$. Assume that $F_1$ is the connected component with the most number of edges (in case of a tie, pick the graph with the most number of vertices). Define $\tilde{G}(F)$ be the disjoint union of $G(F_1), F_2, \cdots, F_m$ and $\tilde{H}(F)$ to be the disjoint union of $H(F_1), F_2, \cdots, F_m$. It follows that $\chi_{\tilde{G}(F)}\tilde{G}(F) = \chi_{\tilde{H}(F)}\tilde{H}(F)$ and

$$\text{hom}(F, \tilde{G}(F)) = \prod_{i \in [m]} \text{hom}(F_i, \tilde{G}(F)) = \prod_{i \in [m]} (\text{hom}(F_i, G(F_1)) + \text{hom}(F_i, F_2 \cup \cdots \cup F_m))$$

$$> \prod_{i \in [m]} (\text{hom}(F_i, H(F_1)) + \text{hom}(F_i, F_2 \cup \cdots \cup F_m)) = \text{hom}(F, \tilde{H}(F)),$$

where we use the fact that $\text{hom}(F_i, G(F_1)) = \text{hom}(F_i, H(F_1))$ when $F_1 \notin \text{Spasm}(F_i)$ (which can be easily proved following Theorem C.47). This concludes the proof of the general case.

## D  NODE/EDGE-LEVEL EXPRESSIVITY

This section aims to prove Theorem 3.7. For the ease of reading, we first restate it below:

**Theorem 3.7** *For all model $M$ defined in Section 2, $\mathcal{F}^M_n$ and $\mathcal{F}^M_e$ (except MPNN) exist. Moreover,*

- **MPNN:** $\mathcal{F}^{\text{MP}}_n = \{F^w : F \text{ is a tree}\}$;

- **Subgraph GNN:**
  $\mathcal{F}^{\text{Sub}}_n = \{F^w : F \text{ has a NED with shared endpoint } w\} = \{F^w : F \backslash \{w\} \text{ is a forest}\}$,
  $\mathcal{F}^{\text{Sub}}_e = \{F^{wx} : F \text{ has a NED with shared endpoint } w\} = \{F^{wx} : F \backslash \{w\} \text{ is a forest}\}$;

- **2-FGNN:** $\mathcal{F}^{\text{F}}_n = \{F^w : F \text{ has a NED where } w \text{ is an endpoint of the first ear}\}$,
  $\mathcal{F}^{\text{F}}_e = \{F^{wx} : F \text{ has a NED where } w \text{ and } x \text{ are endpoints of the first ear}\}$.

*The cases of Local 2-GNN and Local 2-FGNN are similar to 2-FGNN by replacing "NED" with "strong NED" and "almost-strong NED", respectively.*

We remark that the definition of node/edge-level homomorphism expressivity involves only connected graphs for sake of simplicity. Similar to the proof of the graph-level expressivity, the proof of Theorem 3.7 consists of three parts. Among them, the proof related to tree decomposition and ear decomposition is quite similar to that of graph-level expressivity, so we only illustrate the proof sketch for clarity (Appendix D.1). However, the proof related to pebble game and counterexample graphs will require additional techniques, which is detailed in Appendix D.2.

### D.1 RELATED TO TREE DECOMPOSITION AND EAR DECOMPOSITION

We first extend several notations that are used in Appendix C.2.

**Definition D.1** (Tree decomposition for rooted graphs). Given a rooted graph $G^{\boldsymbol{u}}$ and tree $T^r$, we say $T^r$ is a tree decomposition of $G^{\boldsymbol{u}}$ if $T^r$ is a tree decomposition of $G$ and all elements in $\boldsymbol{u}$ belongs to the root bag, i.e., $u_i \in \beta_T(r)$ for all $i$.

Based on this definition, given model $M$, we can define $\mathcal{S}_{\mathsf{n}}^M$ to be the family of tree-decomposed graphs $(F^u, T^r)$ such that $(F, T^r) \in \mathcal{S}^M$; similarly, we define $\mathcal{S}_{\mathsf{e}}^M$ to be the family of tree-decomposed graphs $(F^{uv}, T^r)$ such that $(F, T^r) \in \mathcal{S}^M$.

**Definition D.2** (Bag isomorphism for rooted graphs). Given tree-decomposed graph $(F^{\boldsymbol{u}}, T^r)$ and rooted graph $G^{\boldsymbol{v}}$ where $\boldsymbol{u}$ and $\boldsymbol{v}$ have equal length, a bag isomorphism from $(F^{\boldsymbol{u}}, T^r)$ to $G^{\boldsymbol{v}}$ is a homomorphism $f$ from $F^{\boldsymbol{u}}$ to $G^{\boldsymbol{v}}$ such that $f$ is a bag isomorphism from $(F, T^r)$ to $G$.

We can similarly define bag-isomorphism homomorphism (bIsoHom, bIsoSurj, bIsoInj), bag extension (bExt), and bag-strong surjective (bStrSurj) from tree-decomposed graph $(F^{\boldsymbol{u}}, T^r)$ to tree-decomposed graph $(\tilde{F}^{\boldsymbol{v}}, \tilde{T}^s)$.

**Definition D.3** (Generalization of Definition C.15 for rooted graphs). Let $M \in \{\mathsf{Sub}, \mathsf{L}, \mathsf{LF}, \mathsf{F}\}$ be any model. Given a rooted graph $G^u$ and a tree-decomposed graph $(F^w, T^r)$, define

$$\mathsf{cnt}^M\left((F^w, T^r), G^u\right) := \left|\left\{v \in V_G : \exists D \in \mathbb{N}_+ \text{ s.t. } \left(\left[F_G^{M,(D)}(u,v)\right]^u, T_G^{M,(D)}(u,v)\right) \simeq (F^w, T^r)\right\}\right|.$$

Given a rooted graph $G^{uv}$ and a tree-decomposed graph $(F^{wx}, T^r)$, define

$$\mathsf{cnt}^M\left((F^{wx}, T^r), G^{uv}\right) := \mathbb{I}\left[\exists D \in \mathbb{N}_+ \text{ s.t. } \left(\left[F_G^{M,(D)}(u,v)\right]^{uv}, T_G^{M,(D)}(u,v)\right) \simeq (F^{wx}, T^r)\right].$$

Here, $\left(F_G^{M,(D)}(u,v), T_G^{M,(D)}(u,v)\right)$ is the depth-$2D$ unfolding tree of $G$ at $(u,v)$ for model $M$.

All the following lemmas are straightforward extensions of those in Appendix C.2.

**Lemma D.4.** Let $M \in \{\mathsf{Sub}, \mathsf{L}, \mathsf{LF}, \mathsf{F}\}$ be any model. For any graph $G^v$ and tree-decomposed graph $(F^u, T^r) \in \mathcal{S}_{\mathsf{n}}^M$, we have

$$\mathsf{bIso}\left((F^u, T^r), G^v\right) = \sum_{(\tilde{F}^w, \tilde{T}^s) \in \mathcal{S}_{\mathsf{n}}^M} \mathsf{bIsoHom}\left((F^u, T^r), \left(\tilde{F}^w, \tilde{T}^s\right)\right) \cdot \mathsf{cnt}^M\left(\left(\tilde{F}^w, \tilde{T}^s\right), G^v\right).$$

*The edge-level result is similar.*

**Lemma D.5.** Let $M \in \{\mathsf{Sub}, \mathsf{L}, \mathsf{LF}, \mathsf{F}\}$ be any model. For any tree-decomposed graphs $(F^u, T^r), (\tilde{F}^v, \tilde{T}^s) \in \mathcal{S}_{\mathsf{n}}^M$,

$$\mathsf{bIsoHom}((F^u, T^r), (\tilde{F}^v, \tilde{T}^s)) = \sum_{(\widehat{F}^w, \widehat{T}^t) \in \mathcal{S}_{\mathsf{n}}^M} \frac{\mathsf{bIsoSurj}\left((F^u, T^r), (\widehat{F}^w, \widehat{T}^t)\right) \cdot \mathsf{bIsoInj}\left((\widehat{F}^w, \widehat{T}^t), (\tilde{F}^v, \tilde{T}^s)\right)}{\mathsf{aut}(\widehat{F}^w, \widehat{T}^t)}.$$

*The edge-level result is similar.*

**Lemma D.6.** Let $M \in \{\mathsf{Sub}, \mathsf{L}, \mathsf{LF}, \mathsf{F}\}$ be any model. For any graph $G^v$ and tree-decomposed graph $(F^u, T^r) \in \mathcal{S}_{\mathsf{n}}^M$,

$$\mathsf{hom}(F^u, G^v) = \sum_{(\tilde{F}^w, \tilde{T}^s) \in \mathcal{S}_{\mathsf{n}}^M} \frac{\mathsf{bExt}\left((F^u, T^r), (\tilde{F}^w, \tilde{T}^s)\right) \cdot \mathsf{bStrHom}\left((\tilde{F}^w, \tilde{T}^s), G^v\right)}{\mathsf{aut}(\tilde{F}^w, \tilde{T}^s)}.$$

*The edge-level result is similar.*

**Lemma D.7.** *Let $M \in \{\mathsf{Sub}, \mathsf{L}, \mathsf{LF}, \mathsf{F}\}$ be any model. For any graph $G^v$ and tree-decomposed graph $(F^u, T^r) \in \mathcal{S}_{\mathsf{n}}^M$,*

$$\mathsf{bStrHom}((F^u, T^r), G^v) = \sum_{(\tilde{F}^w, \tilde{T}^s) \in \mathcal{S}_{\mathsf{n}}^M} \frac{\mathsf{bStrSurj}\left((F^u, T^r), (\tilde{F}^w, \tilde{T}^s)\right) \cdot \mathsf{bIso}\left((\tilde{F}^w, \tilde{T}^s), G^v\right)}{\mathsf{aut}(\tilde{F}^w, \tilde{T}^s)},$$

*The edge-level result is similar.*

**Corollary D.8.** *Let $M \in \{\mathsf{Sub}, \mathsf{L}, \mathsf{LF}, \mathsf{F}\}$ be any model. For any two graphs $G^u, H^v$, $\chi_G^M(u) = \chi_H^M(v)$ iff $\mathsf{hom}(F^w, G^u) = \mathsf{hom}(F^w, H^v)$ for all $(F^w, T^r) \in \mathcal{S}_{\mathsf{n}}^M$. For any two graphs $G^{uv}, H^{wx}$, $\chi_G^M(u, v) = \chi_H^M(w, x)$ iff $\mathsf{hom}(F^{yz}, G^{uv}) = \mathsf{hom}(F^{yz}, H^{wx})$ for all $(F^{yz}, T^r) \in \mathcal{S}_{\mathsf{e}}^M$.*

**Lemma D.9.** *Let $M \in \{\mathsf{Sub}, \mathsf{L}, \mathsf{LF}, \mathsf{F}\}$ be any model.*

- *For any rooted connected graph $F^u$, $F^u \in \mathcal{F}_{\mathsf{n}}^M$ iff there is a tree decomposition $T^r$ of $F^u$ such that $(F^u, T^r) \in \mathcal{S}_{\mathsf{n}}^M$.*

- *For any rooted connected graph $F^{uv}$, $F^{uv} \in \mathcal{F}_{\mathsf{e}}^M$ iff there is a tree decomposition $T^r$ of $F^{uv}$ such that $(F^{uv}, T^r) \in \mathcal{S}_{\mathsf{e}}^M$.*

By combining the proof of previous lemmas and theorems, we can prove that both $\mathcal{F}_{\mathsf{n}}^M$ and $\mathcal{F}_{\mathsf{e}}^M$ satisfy Definition 3.6(a) for all $M \in \{\mathsf{Sub}, \mathsf{L}, \mathsf{LF}, \mathsf{F}\}$.

## D.2 COUNTEREXAMPLES

Let $F^{\boldsymbol{w}}$ be a rooted graph that marks the special vertices $w_1, \cdots, w_m$. Assume that $F^{\boldsymbol{w}} \notin \mathcal{F}_{\mathsf{n}}^M$ ($m = 1$) or $F^{\boldsymbol{w}} \notin \mathcal{F}_{\mathsf{e}}^M$ ($m = 2$). In this subsection, we will construct a pair of graphs $G^{\boldsymbol{u}}$ and $H^{\boldsymbol{v}}$ such that $\chi_G^M(\boldsymbol{u}) = \chi_H^M(\boldsymbol{v})$ and $\mathsf{hom}(F^{\boldsymbol{w}}, G^{\boldsymbol{u}}) \neq \mathsf{hom}(F^{\boldsymbol{w}}, H^{\boldsymbol{v}})$. However, it turns out that, if we naïvely follow the proof in Appendix C.4 by constructing the Fürer graph and twisted Fürer graph with respect to $F$ without considering the marked vertices $\boldsymbol{w}$, then the graphs may no longer be counterexamples here. For example, for the edge-level expressivity, $F^{w_1, w_2} \notin \mathcal{F}_{\mathsf{e}}^M$ does not imply that $F \notin \mathcal{F}^M$ where $M$ can be any model studied in this paper.

To address the problem, we instead introduce a novel construction of counterexample graphs defined as follows:

**Definition D.10** (Clique-augmented Fürer graphs). Let $F$ be any connected graph and $w_1, \cdots, w_m \in V_F$ be a sequence of vertices. Given an integer $k \geq m$, the $k$-clique-augmented Fürer graph with respect to $F^{\boldsymbol{w}}$, denoted by $G_k(F^{\boldsymbol{w}})$, is the Fürer graph of $\tilde{F}$ where $\tilde{F}$ is the union of graph $F$ and a $k$-clique that contains $w_1, \cdots, w_m$ (and does not contain other vertices in $F$). The twisted Fürer graph of $G_k(F^{\boldsymbol{w}})$ is denoted as $H_k(F^{\boldsymbol{w}})$.

We have the following main result:

**Theorem D.11.** *Let $M \in \{\mathsf{Sub}, \mathsf{L}, \mathsf{LF}, \mathsf{F}\}$ be any model defined in Section 2.*

- *For any rooted graph $F^w$ marking vertex $w$, if $F^w \notin \mathcal{F}_{\mathsf{n}}^M$, then there is a vertex $(w, U) \in \mathsf{Meta}_{\tilde{F}}(w)$ such that $U \subset V_F$ and $\chi_{G_4(F^w)}^M((w, \emptyset)) = \chi_{H_4(F^w)}^M((w, U))$;*

- *For any rooted graph $F^{\boldsymbol{w}}$ marking two vertices $w_1, w_2$, if $F^{\boldsymbol{w}} \notin \mathcal{F}_{\mathsf{e}}^M$, then there are two vertices $(w_1, U_1) \in \mathsf{Meta}_{\tilde{F}}(w_1)$ and $(w_2, U_2) \in \mathsf{Meta}_{\tilde{F}}(w_2)$ such that $U_1, U_2 \subset V_F$ and $\chi_{G_4(F^{\boldsymbol{w}})}^M((w_1, \emptyset), (w_2, \emptyset)) = \chi_{H_4(F^{\boldsymbol{w}})}^M((w_1, U_1), (w_2, U_2))$.*

*Proof.* Below, we only give a proof of the second item. Let $\tilde{F}$ be the union of $F$ and the 4-clique according to Definition D.10. Similar to the graph-level expressivity, we will extend the pebble game defined in Appendix C.4 to edge-level. For the edge-level pebble game, the two pebbles $\mathsf{p}_u, \mathsf{p}_v$ are initially placed on two *predefined* vertices $w_1, w_2$, and Bob initially selects a *predefined* set of connected components $Q \subset \mathsf{CC}_{\tilde{F}}(\{w_1, w_2\})$ (of odd size). Then, the game executes the same process as in the text above Proposition C.49. The winning criterion for the two players is also the same. We denote the above edge-level pebble game as $\mathsf{PG}_{\tilde{F}}(w_1, w_2, Q)$.

Based on the isomorphism property of Fürer graphs established in Zhang et al. (2023a, Theorem I.9 and and Theorem I.17), it is straightforward to see that, for any vertices $\xi, \xi' \in \mathsf{Meta}_{\tilde{F}}(w_1)$ and

$\eta, \eta' \in \mathsf{Meta}_{\tilde{F}}(w_2)$, $\chi^M_{G_4(F^w)}(\xi, \eta) = \chi^M_{H_4(F^w)}(\xi', \eta')$ *iff* Alice cannot win the edge-level pebble game $\mathsf{PG}_{\tilde{F}}(w_1, w_2, Q)$ for some $Q \subset \mathsf{CC}_{\tilde{F}}(\{w_1, w_2\})$ (depending on $\xi, \xi', \eta, \eta'$). Moreover, let $\xi' = (w_1, \emptyset)$, $\eta' = (w_2, \emptyset)$ and let $\xi_2$ and $\eta_2$ range over all even-size subset of $N_F(w_1)$ and $N_F(w_2)$, respectively, that is to say, $\xi_2$ and $\eta_2$ does not contain vertices that are only in the 4-clique. Then, the equivalent pebble game ranges over all $Q$ of odd size such that every connected component in $Q$ contains only vertices in $F$, i.e., $Q \subset \mathsf{CC}_F(\{w_1, w_2\})$. Therefore, if Alice cannot win the edge-level pebble game $\mathsf{PG}_{\tilde{F}}(w_1, w_2, Q)$ for some $Q \subset \mathsf{CC}_F(\{w_1, w_2\})$, then there are two vertices $\xi \in \mathsf{Meta}_{\tilde{F}}(w_1)$ and $\eta \in \mathsf{Meta}_{\tilde{F}}(w_2)$ such that $\xi_2, \eta_2 \subset V_F$ and $\chi^M_{G_4(F^w)}((w_1, \emptyset), (w_2, \emptyset)) = \chi^M_{H_4(F^w)}(\xi, \eta)$.

It thus remains to prove that, if Alice can win the edge-level pebble game $\mathsf{PG}_{\tilde{F}}(w_1, w_2, Q)$ for all $Q \subset \mathsf{CC}_F(\{w_1, w_2\})$, then $F^w \in \mathcal{F}^M_{\mathsf{e}}$. To prove this result, we similarly define the game state graph as in Appendix C.4, with the only difference that the initial state is now $(w_1, w_2, Q)$. The proof is divided into the following steps:

1. Assume that $Q$ contains a single connected component, i.e., $Q = \{P\}$. We will prove that, if Alice can win the game $\mathsf{PG}_{\tilde{F}}(w_1, w_2, \{P\})$, then there is a game strategy for Alice such that for any reachable and non-terminal state $(x, y, \{P'\})$, $P' \subset P$. If it is not the case, pick any state $(x, y, \{P'\})$ such that $P' \not\subset P$ and any path from $(w_1, w_2, \{P\})$ to $(x, y, \{P'\})$ does not pass any intermediate state that is not contracted. It follows that either $w_1$ or $w_2$ does not hold any pebble, and all edges connected to this vertex is in $P'$. Therefore, $P'$ contains all vertices in the 4-clique. In this case, Alice cannot win as Bob can always keep the connected component containing all vertices in the 4-clique in subsequent rounds, i.e., Bob always selects a component containing a vertex in the 4-clique that does not hold a pebble.

2. Based on the above result, in the game $\mathsf{PG}_{\tilde{F}}(w_1, w_2, \{P\})$, the connected components selected by Bob are always in $F$, and thus the game process is the same as $\mathsf{PG}_F(w_1, w_2, \{P\})$. We can then prove the same result as Lemma C.52. Concretely, if Alice can win the edge-level pebble game $\mathsf{PG}_{\tilde{F}}(w_1, w_2, \{P\})$, then there exists a game state graph $G^{\mathsf{S}}$ corresponding to a winning strategy, such that for any transition $((x, y, \{P'\}), (x', y', \{P''\}))$ where $(x, y, \{P'\})$ is a reachable and non-terminal state, we have $P'' \subsetneq P'$.

3. Next, we can follow the same procedure in the proof of Theorem C.53 to construct a tree decomposition $T^r$ for the subgraph $F[P]$ containing all edges in $P$, such that $(F[P], T^r) \in \mathcal{S}^M_{\mathsf{e}}$ and $\beta_T(r) = \{\!\{w_1, w_2\}\!\}$.

4. Finally, let $P$ ranges over all connected component in $\mathsf{CC}_F(\{w_1, w_2\})$, for each $P$ we have a tree-decomposed graph. We can glue these tree-decomposed graphs by merging the root to obtain the tree decomposition of $F$, because the root bags of all $T^r$ are the same and any vertex $x \notin \{w_1, w_2\}$ appears in the bag of only one $T^r$. Moreover, the glued tree $\tilde{T}^s$ clearly satisfies that $(F, \tilde{T}^s) \in \mathcal{S}^M_{\mathsf{e}}$.

Combining these items shows that $F^w \in \mathcal{F}^M_{\mathsf{e}}$ and concludes the proof. $\qquad\square$

**Corollary D.12.** *Let $F^w$ any rooted graph marking $m$ vertices such that $F^w \notin \mathcal{F}^M_{\mathsf{n}}$ ($m = 1$) or $F^w \notin \mathcal{F}^M_{\mathsf{e}}$ ($m = 2$), and let $G_k(F^w)$ and $H_k(F^w)$ be clique-augmented Fürer graphs defined above. Denote by $\boldsymbol{\xi}, \boldsymbol{\eta} \in V^m_{G_k(F^w)}$ two vertex tuples of length $m$ where $\xi_i = (w_i, \emptyset)$ and $\eta_i \in \mathsf{Meta}_{\tilde{F}}(w_i)$ with $\eta_{i,2} \subset V_F$ for $i \in [m]$. Then, $\mathsf{hom}(F^w, [G_k(F^w)]^{\boldsymbol{\xi}}) \neq \mathsf{hom}(F^w, [H_k(F^w)]^{\boldsymbol{\eta}}))$.*

*Proof.* The proof exactly parallels that of Theorem C.47 by separately considering three cases. We omit the proof for clarity. $\qquad\square$

**Corollary D.13.** *Let $M \in \{\mathsf{Sub}, \mathsf{L}, \mathsf{LF}, \mathsf{F}\}$ be any model.*

- *For any graph $F^w \notin \mathcal{F}^M_{\mathsf{n}}$, let $G_4(F^w)$ and $H_4(F^w)$ be the clique-augmented Fürer graph and the corresponding twisted Fürer graph. Then, there exists two vertices $\xi, \xi'$ such that $\mathsf{hom}(F^w, [G_4(F^w)]^{\xi}) \neq \mathsf{hom}(F^w, [H_4(F^w)]^{\xi'})$ and $\chi^M_{G_4(F^w)}(\xi) = \chi^M_{H_4(F^w)}(\xi')$.*

- *For any graph $F^{wx} \notin \mathcal{F}^M_{\mathsf{e}}$, let $G_4(F^{wx})$ and $H_4(F^{wx})$ be the clique-augmented Fürer graph and the corresponding twisted Fürer graph defined in Definition D.10. Then, there exists*

*four vertices $\xi, \eta, \xi', \eta'$ such that* $\mathsf{hom}(F^{wx}, [G_4(F^{wx})]^{\xi\eta}) \neq \mathsf{hom}(F^{wx}, [H_4(F^{wx})]^{\xi'\eta'})$ *and* $\chi^M_{G_4(F^{wx})}(\xi, \eta) = \chi^M_{H_4(F^{wx})}(\xi', \eta').$

*Proof.* The proof directly follows from Corollary D.12 and Theorem D.11. □

# E HIGHER-ORDER GNNS

## E.1 DEFINITION OF HIGHER-ORDER GNNS

In this subsection, we give formal definitions of the CR algorithms for various higher-order GNNs, each of which generalizes a model in Section 2.

- **Subgraph $k$-GNN.** In a Subgraph $k$-GNN, a graph $G$ is treated as a set of subgraphs $\{\!\{G^{\boldsymbol{u}} : \boldsymbol{u} \in V_G^k\}\!\}$, where each subgraph $G^{\boldsymbol{u}}$ is obtained from $G$ by marking $k$ special vertices $\boldsymbol{u} \in V_G^k$, and thus there are $n^k$ subgraphs when $G$ has $n$ vertices. Subgraph GNN maintains a color $\chi_G^{\mathsf{Sub}(k)}(\boldsymbol{u}, v)$ for each vertex $v$ in graph $G^{\boldsymbol{u}}$. Initially, $\chi_G^{\mathsf{Sub}(k),(0)}(\boldsymbol{u}, v) = (\ell_G(v), \mathsf{atp}_G(\boldsymbol{u}), \mathbb{I}[u_1 = v], \cdots, \mathbb{I}[u_k = v])$. It then runs MPNNs independently on each graph $G^{\boldsymbol{u}}$:

$$\chi_G^{\mathsf{Sub}(k),(t+1)}(\boldsymbol{u}, v) = \mathsf{hash}\left(\chi_G^{\mathsf{Sub}(k),(t)}(\boldsymbol{u}, v), \{\!\{\chi_G^{\mathsf{Sub}(k),(t)}(\boldsymbol{u}, w) : w \in N_G(v)\}\!\}\right). \quad (26)$$

Denote the stable color as $\chi_G^{\mathsf{Sub}(k)}(\boldsymbol{u}, v)$. Define $\chi_G^{\mathsf{Sub}(k)}(\boldsymbol{u}) := \{\!\{\chi_G^{\mathsf{Sub}(k)}(\boldsymbol{u}, v) : v \in V_G\}\!\}$. Then, the graph representation is defined as $\chi_G^{\mathsf{Sub}(k)}(G) = \{\!\{\chi_G^{\mathsf{Sub}(k)}(\boldsymbol{u}) : \boldsymbol{u} \in V_G\}\!\}$. We remark that Subgraph $k$-GNN is precisely the $k$-VSAN proposed in Qian et al. (2022).

- **Local $k$-GNN.** Local $k$-GNN maintains a color $\chi_G^{\mathsf{L}(k)}(\boldsymbol{u})$ for each vertex $k$-tuple $\boldsymbol{u} \in V_G^k$. Initially, $\chi_G^{\mathsf{L}(k),(0)}(\boldsymbol{u}) = (\ell_G(u_1), \cdots, \ell_G(u_k), \mathsf{atp}_G(\boldsymbol{u}))$, called the isomorphism type of vertex $k$-tuple $\boldsymbol{u}$, where $\mathsf{atp}_G(\boldsymbol{u})$ is the *atomic type* of $\boldsymbol{u}$. Then, in each iteration $t + 1$,

$$\chi_G^{\mathsf{L}(k),(t+1)}(\boldsymbol{u}) = \mathsf{hash}\left(\chi_G^{\mathsf{L}(k),(t)}(\boldsymbol{u}), \{\!\{\chi_G^{\mathsf{L}(k),(t)}(\boldsymbol{v}) : \boldsymbol{v} \in N_G^{(1)}(\boldsymbol{u})\}\!\}, \cdots, \right.$$
$$\left. \{\!\{\chi_G^{\mathsf{L}(k),(t)}(\boldsymbol{v}) : \boldsymbol{v} \in N_G^{(k)}(\boldsymbol{u})\}\!\}\right), \quad (27)$$

where $N_G^{(j)}(\boldsymbol{u}) = \{(u_1, \cdots, u_{j-1}, w, u_{j+1}, \cdots, u_k) : w \in N_G(u_j)\}$. Denote the stable color as $\chi_G^{\mathsf{L}(k)}(\boldsymbol{u})$. The graph representation is defined as $\chi_G^{\mathsf{L}(k)}(G) := \{\!\{\chi_G^{\mathsf{L}(k)}(\boldsymbol{u}) : \boldsymbol{u} \in V_G\}\!\}$. We remark that Local $k$-GNN is precisely the $\delta$-$k$-LWL proposed in Morris et al. (2020).

- **Local $k$-FGNN.** Local $k$-FGNN is almost the same as Local $k$-GNN except that the update formula is replaced by the following one:

$$\chi_G^{\mathsf{LF}(k),(t+1)}(\boldsymbol{u}) = \mathsf{hash}\left(\chi_G^{\mathsf{LF}(k),(t)}(\boldsymbol{u}), \{\!\{(\chi_G^{\mathsf{LF}(k),(t)}(w, u_2, \cdots, u_k), \chi_G^{\mathsf{LF}(k),(t)}(u_1, w, u_3, \cdots, u_k), \right.$$
$$\left. \cdots, \chi_G^{\mathsf{LF}(k),(t)}(u_1, \cdots, u_{k-1}, w)) : w \in N_G(u_1) \cup \cdots \cup N_G(u_k)\}\!\}\right). \quad (28)$$

We remark that Local $k$-FGNN is precisely the $\mathsf{SLFWL}(k)$ proposed in Zhang et al. (2023a).

- **$k$-FGNN.** It is just the standard $k$-FWL (Cai et al., 1992). Compared with Local $k$-FGNN, the update formula is now global:

$$\chi_G^{\mathsf{F}(k),(t+1)}(\boldsymbol{u}) = \mathsf{hash}\left(\chi_G^{\mathsf{F}(k),(t)}(\boldsymbol{u}), \{\!\{(\chi_G^{\mathsf{F}(k),(t)}(w, u_2, \cdots, u_k), \chi_G^{\mathsf{F}(k),(t)}(u_1, w, u_3, \cdots, u_k), \right.$$
$$\left. \cdots, \chi_G^{\mathsf{F}(k),(t)}(u_1, \cdots, u_{k-1}, w)) : w \in V_G\}\!\}\right). \quad (29)$$

We note that the computational complexity of Subgraph $(k-1)$-GNN, Local $k$-GNN, and Local $k$-FGNN is the same, i.e., $\Theta(n^{k-1}m)$ for a graph of $n$ vertices and $m$ edges. Moreover, it is easy to see that Subgraph 0-GNN, Local 1-GNN, and Local 1-FGNN all reduce to MPNN. For $k$-FGNN, the computational complexity is $\Theta(n^{k+1})$, which is strictly higher than Subgraph $(k-1)$-GNN, Local $k$-GNN, and Local $k$-FGNN.

### E.2 HIGHER-ORDER STRONG NED

In this subsection, we generalize the strong NED into higher-order versions. We first define the concept of higher-order ear:

**Definition E.1** ($k$-order ear). Given integer $k \geq 1$, a $k$-order ear is a graph $G$ formed by the union of $k$ paths $P_1, \cdots, P_k$ (possibly with zero length) plus an edge set $Q$ satisfying the following conditions:

- For each path $P_i$, denote its two endpoints as $u_i, v_i$, called the outer endpoint and inner endpoint, respectively. Then, all edges in $Q$ are linked between inner endpoints, i.e., $Q \subset \{\{v_i, v_j\} : 1 \leq i, j \leq k, v_i \neq v_j\}$.

- Any different paths $P_i, P_j$ do not intersect except at the inner endpoint (when $v_i = v_j$).

- $G$ is a connected graph.

The endpoints of the $k$-order ear is defined to be all outer endpoints $u_1, \cdots, u_k$.

It is easy to see that a 2-order ear is precisely a simple path, since linking two different paths (possibly with an additional edge) still yields a path. Below, we denote by $\mathsf{inner}(G)$ and $\mathsf{outer}(G)$ the set of inner endpoints and outer endpoints in ear $G$, respectively. We also denote by $\mathsf{path}(G)$ the set of paths in ear $G$. It follows that $|\mathsf{inner}(G)| = |\mathsf{outer}(G)| = |\mathsf{path}(G)| = k$. Finally, given a path $P$ and two vertices $w_1, w_2$ in $P$, denote by $\mathsf{subpath}_P(w_1, w_2)$ the subpath in $P$ such that the two endpoints are $w_1, w_2$.

**Definition E.2** (Nested interval). Let $G$ and $H$ be two $k$-order ears with $\mathsf{inner}(G) = \{v_1, \cdots, v_k\}$, $\mathsf{outer}(G) = \{u_1, \cdots, u_k\}$, and $\mathsf{outer}(H) = \{w_1, \cdots, w_k\}$, where each $\{u_i, v_i\}$ corresponds to the endpoints of a path $P_i \in \mathsf{path}(G)$. We say $H$ is nested on $G$ if one or more endpoint $w_i$ of $H$ ($i \in [k]$) is in path $P_i$, and all other vertices in $H$ are not in $G$. The nested interval is defined to be the union of subpaths $\mathsf{subpath}_{P_i}(w_i, v_i)$ for all $i \in [k]$ satisfying that $w_i$ is in $P_i$.

We give an illustration of the nested interval of two 3-order ears in Figure 2. Equipped with the above definition, we are ready to introduce the higher-order strong NED:

**Definition E.3** ($k$-order strong NED). Given a graph $G$, a $k$-order strong NED $\mathcal{P}$ is a partition of the edge set $E_G$ into a sequence of edge sets $Q_1, \cdots, Q_m$, which satisfies the following conditions:

- Each $Q_i$ is a $k$-order ear.

- Any two ears $Q_i$ and $Q_j$ with indices $1 \leq i < j \leq c$ do not intersect, where $c$ is the number of connected components of $G$.

- For each $Q_j$ with index $j > c$, it is nested on some $k$-order ear $Q_i$ with index $1 \leq i < j$. Moreover, except for the endpoints of $Q_j$ on $Q_i$, no other vertices in $Q_j$ are in any previous ear $Q_k$ for $1 \leq k < i$.

- Denote by $I(Q_j) \subset Q_i$ the *nested interval* of $Q_j$ in $Q_i$. For all $Q_j, Q_k$ with $c < j < k \leq m$, if $Q_j$ and $Q_k$ are nested on the same ear, then $I(Q_j) \subset I(Q_k)$.

### E.3 PROOFS IN SECTION 3.4

We first generalize the tree-decomposed graphs in Definition C.6 to higher-order versions:

**Definition E.4.** Define four families of tree-decomposed graphs $\mathcal{S}^{\mathsf{Sub}(k)}$, $\mathcal{S}^{\mathsf{L}(k)}$, $\mathcal{S}^{\mathsf{LF}(k)}$, and $\mathcal{S}^{\mathsf{F}(k)}$ as follows:

a) $(F, T^r) \in \mathcal{S}^{\mathsf{F}(k)}$ iff $(F, T^r)$ satisfies Definition C.4 with width $k$;

b) $(F, T^r) \in \mathcal{S}^{\mathsf{LF}(k)}$ iff $(F, T^r)$ satisfies Definition C.4 with width $k$, and for any tree node $t$ of odd depth, it has only one child if $w \notin \{v : v \in N_G(u), u \in \beta_T(s)\}$ where $s$ is the parent node of $t$ and $w$ is the unique vertex in $\beta_T(t) \backslash \beta_T(s)$;

c) $(F, T^r) \in \mathcal{S}^{\mathsf{L}(k)}$ iff $(F, T^r)$ satisfies Definition C.4 with width $k$, and any tree node $t$ of odd depth has only one child;

d) $(F, T^r) \in \mathcal{S}^{\mathsf{Sub}(k)}$ iff $(F, T^r)$ satisfies Definition C.4 with width $k$, and there exists a multiset $U \subset V_G$ of size $|U| = k$ such that $U \subset \beta_T(t)$ for all $t \in V_T$.

Then, we can analogously prove the following theorems. The proofs are almost the same as in Appendix C, so we omit them for clarity.

**Theorem E.5.** *Let $M \in \{\mathsf{Sub}(k), \mathsf{L}(k), \mathsf{LF}(k), \mathsf{F}(k)\}$. Then, any graphs $G$ and $H$ have the same representation under model $M$ (i.e., $\chi_G^M(G) = \chi_H^M(H)$) iff $\mathsf{hom}(F, G) = \mathsf{hom}(F, H)$ for all $(F, T^r) \in \mathcal{S}^M$.*

**Theorem E.6.** *Let $M \in \{\mathsf{Sub}(k), \mathsf{L}(k), \mathsf{LF}(k), \mathsf{F}(k)\}$ be any model, and let $F$ be any graph such that no tree decomposition $(F, T^r) \in \mathcal{S}^M$. Let $G(F)$ and $H(F)$ be the Fürer graph and twisted Fürer graph with respect to $F$. Then, $\mathsf{hom}(F, G(F)) \neq \mathsf{hom}(F, H(F))$ and $\chi_{G(F)}^M(G(F)) = \chi_{H(F)}^M(H(F))$.*

**Theorem E.7.** *For any graph $F$, there is a tree decomposition $T^r$ of $F$ such that $(F, T^r) \in \mathcal{S}^{\mathsf{Sub}(k)}$ iff there exists $U \subset V_F$ such that $|U| \leq k$ and $F \backslash U$ is a forest.*

**Theorem E.8.** *For any graph $F$, there is a tree decomposition $T^r$ of $F$ such that $(F, T^r) \in \mathcal{S}^{\mathsf{L}(k)}$ iff $F$ has a $k$-order strong NED.*

**Theorem E.9.** *For any graph $F$, there is a tree decomposition $T^r$ of $F$ such that $(F, T^r) \in \mathcal{S}^{\mathsf{F}(k)}$ iff $\mathsf{tw}(F) \leq k$.*

### E.4 EXPRESSIVITY GAP BETWEEN HIGHER-ORDER GNNs

In this subsection, we show how homomorphism expressivity can be used to build a complete expressiveness hierarchy for higher-order GNNs as shown in Corollary 4.2.

**Gap between Subgraph $k$-GNN and Local $(k+1)$-GNN ($k \geq 1$).** The counterexample graph is a $(k+1) \times (2k+2)$ grid consisting $(k+1) \times (2k+2)$ vertices. First, it is easy to see that the graph is not in $\mathcal{F}^{\mathsf{Sub}(k)}$, i.e., deleting $k$ vertices of the graph cannot yield a forest. To see this, note that the graph consists of $k \times (2k+1)$ "squares", and each vertex is related to at most four squares. Therefore, deleting $k$ vertices cannot eliminate all squares when $k \geq 2$ (because $4k < k \times (2k+1)$). For the case of $k = 1$, we clearly have that deleting one vertex cannot eliminate all squares.

We next show that the $(k+1) \times (2k+2)$ grid is in $\mathcal{F}^{\mathsf{L}(k+1)}$. This is also simple as shown in Figure 8, where each color represents a $k$-order ear. It can be seen that the $(k+1) \times (2k+2)$ grid has a $(k+1)$-order strong NED.

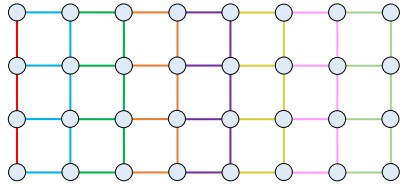

Figure 8: An 4-order strong NED of the $4 \times 8$ grid graph.

**Relation between Subgraph $k$-GNN and $k$-FGNN ($k \geq 2$).** To show that $\mathcal{F}^{\mathsf{Sub}(k)} \not\subset \mathcal{F}^{\mathsf{F}(k)}$, consider the $(k+2)$-clique. Clearly, deleting $k$ vertices from the $(k+2)$-clique yields a graph consisting of two vertices linked by an edge, which is a tree. On the other hand, the treewidth of a $(k+2)$-clique is $k+1$ (Fact C.3). So the $(k+2)$-clique is in $\mathcal{F}^{\mathsf{Sub}(k)}$ but not in $\mathcal{F}^{\mathsf{F}(k)}$.

To show that $\mathcal{F}^{\mathsf{F}(k)} \not\subset \mathcal{F}^{\mathsf{Sub}(k)}$, we can again use the grid graph, but this time consider the $k \times (2k+2)$ grid. On the one hand, a standard result in graph theory shows that the treewidth of a $a \times b$ grid graph is $\min(a, b)$. On the other hand, following the similar analysis above, we can prove that deleting $k$ vertices from the $k \times (2k+2)$ grid cannot eliminate all squares.

**Gap between Local $k$-GNN and $k$-FGNN ($k \geq 2$).** The counterexample graph $F$ is the union of the following graphs $K_0 \cup K_1 \cup \cdots \cup K_k$, where $K_0$ is a $(k+1)$-clique with vertex set $\{u_1, \cdots, u_{k+1}\}$, and for $i \in [k+1]$, $K_i$ is a $(k+1)$-clique with vertex set $\{u_1, \cdots, u_{i-1}, v_i, u_{i+1}, \cdots, u_{k+1}\}$ where $v_i$ is a new vertex (not in $K_0$) and $v_i \neq v_j$ for $i \neq j$. Namely, each $K_i$ has $k$ common vertices with $K_0$. It is easy to construct a tree decomposition $T^r$ of $F$ such that $(F, T^r) \in \mathcal{S}^{\mathsf{F}(k)}$. On the other hand, it is easy to see that the graph does not have a $k$-order strong NED (equivalently, one can easily check that $F$ does not admit a tree decomposition $T^r$ satisfying $(F, T^r) \in \mathcal{S}^{\mathsf{L}(k)}$).

**Gap between $k$-FGNN and Local $(k+1)$-GNN ($k \geq 2$).** The counterexample graph is again the $(k+1) \times (2k+2)$ grid. We have proved that the graph is in $\mathcal{F}^{\mathsf{L}(k+1)}$ but not in $\mathcal{F}^{\mathsf{L}(k)}$.

**Regarding Local $k$-IGN and Frasca et al. (2022).** Finally, we remark that based on the above results, we essentially proved an open question raised in Frasca et al. (2022) regarding the expressive power of Local $k$-IGN (i.e., the RelGN($k$) proposed in their paper). Following Zhang et al. (2023a),

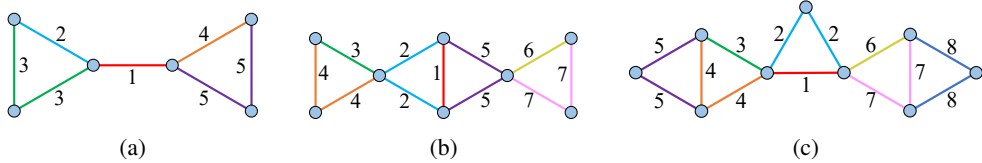

Figure 9: More illustration of NED. All NEDs in these example graphs are strong but not endpoint-shared. See Appendix F.2 for additional discussions.

it is straightforward to see that Local $k$-IGN is as expressive as Local $k$-GNN. Therefore, Local $k$-IGN is strictly more expressive than $(k-1)$-FGNN and strictly less expressive than $k$-FGNN.

# F  ADDITIONAL DISCUSSIONS

## F.1  REGARDING THE DEFINITION OF HOMOMORPHISM EXPRESSIVITY

In this paper, we have shown that homomorphism expressivity exists for a variety of popular GNNs. Unfortunately, due to the "iff" statement in Definition 3.1, homomorphism expressivity may not always be well-defined in the general case. We remark in this subsection that, there do exist pathological, intentionally designed GNNs such that the homomorphism expressivity is not well-defined.

Consider a simple GNN $M$ that outputs the representation of a graph $G$ as follows. If $G$ is a cycle of odd length, it outputs $(1, L)$ where $L$ is the length of the cycle. Otherwise, it outputs $(0, \chi_G^{\mathsf{MP}}(G))$, namely, running a MPNN on the graph. It follows that $M$ is strictly more powerful than MPNN, e.g., it can distinguish between the 9-cycle and three triangles, which MPNN fails to distinguish. As a result, $M$ can count all trees under homomorphism. Moreover, it cannot count other patterns under homomorphism, as the counterexample graphs for MPNN (i.e., Fürer graphs) are not cycles of odd length, so they are still counterexample graphs for model $M$. Therefore, the homomorphism expressivity of $M$ should be exactly the family of forests if it exists. However, the homomorphism information of forests cannot determine the representation of model $M$ since $M$ is strictly more powerful than MPNN. So we conclude that $\mathcal{F}^M$ does not exist.

Note that the above GNN construction is inherently unnatural and hardly appears in practice. Actually, when a GNN is defined via the message-passing paradigm, we suspect that its homomorphism expressivity is always well-defined. As stated in Appendix B, we conjecture that, for any GNN characterized by a color refinement algorithm that outputs stable colors, the homomorphism expressivity always exists.

## F.2  REGARDING THE DEFINITION OF NED

In the main text, we have illustrated several types of NED with simple example graphs. To gain a deeper understanding of Definition 3.3, in this section we will present a few more complex examples (see Figure 9). Notably, unlike graphs in Figure 1(b), for all graphs in Figure 9 their NED contains ears such that only one endpoint is in its nested ear (e.g., ears 2 and 4 in Figure 9(a)). In other words, these ears have empty nested intervals.

The presence of ears with empty nested interval stems from the fact that the corresponding graph is not *biconnected* (Zhang et al., 2023b). Indeed, one can check that all graphs in Definition 3.3(b) do not have cut vertices, while all graphs in Figure 9 have cut vertices. Moreover, in these examples, the number of ears with empty nested interval always equals to the number of biconnected components minus one. In particular, for biconnected graphs, it can proved that any NED does not contain an ear with empty nested interval.

# G HOMOMORPHISM AND SUBGRAPH COUNTING POWER

## G.1 PROOF OF THEOREM 4.5

Our proof draws inspiration from a recent technique developed in Seppelt (2023). We note that the original results in Seppelt (2023) are described for unlabeled graphs, but it is easy to extend these results to labeled graphs. To begin with, we define a concept called the graph categorical product.

**Definition G.1.** The categorical product of two graphs $G, H$, denoted as $G \times H$, is a graph where $V_{G \times H} = V_G \times V_H$, $\ell_{G \times H}(g, h) = (\ell_G(g), \ell_H(h))$ for all $g \in V_G$, $h \in V_H$, and $E_{G \times H} = \{\{(g, h), (g', h')\} : \{g, g'\} \in E_G, \{h, h'\} \in E_H\}$.

**Lemma G.2** (Seppelt (2023))**.** *For any graphs $F$, $G$, and $H$, $\mathsf{hom}(F, G \times H) = \mathsf{hom}(F, G) \cdot \mathsf{hom}(F, H)$.*

*Proof.* We define a mapping $\tau$ from $\mathsf{Hom}(F, G) \times \mathsf{Hom}(F, H)$ to $\mathsf{Hom}(F, G \times H)$ as follows. for all $\sigma_1 \in \mathsf{Hom}(F, G)$ and $\sigma_2 \in \mathsf{Hom}(F, H)$, define $\rho = \tau(\sigma_1, \sigma_2)$ where $\rho(f) = (\sigma_1(f), \sigma_2(f))$ for all $f \in V_F$. It is easy to see that $\tau$ is a bijective mapping from $\mathsf{Hom}(F, G) \times \mathsf{Hom}(F, H)$ to $\mathsf{Hom}(F, G \times H)$. □

**Definition G.3.** Given graphs $F$ and $G$, denote by $\mathsf{Surj}(F, G)$ the set of all homomorphisms from $F$ to $G$ that are surjective on both the vertices and edges of $G$, and define $\mathsf{surj}(F, G) = |\mathsf{Surj}(F, G)|$.

The following proposition is straightforward (similar to Lemma C.25):

**Proposition G.4.** $\mathsf{hom}(G, H) = \sum_F \mathsf{surj}(G, F) \cdot \mathsf{sub}(F, H)$, *where $F$ ranges over all non-isomorphic graphs.*

**Lemma G.5.** *Let $M$ be a GNN model such that its homomorphism expressivity $\mathcal{F}^M$ exists. Given a finite set of graphs $\mathcal{L}$ and a function $\alpha : \mathcal{L} \to \mathbb{R}/\{0\}$, if*

$$\chi_G^M(G) = \chi_H^M(H) \implies \sum_{L \in \mathcal{L}} \alpha(L)\mathsf{hom}(L, G) = \sum_{L \in \mathcal{L}} \alpha(L)\mathsf{hom}(L, H),$$

*holds for all graphs $G$ and $H$, then $\mathcal{L} \subset \mathcal{F}^M$.*

*Proof.* Let $n$ be the largest number of vertices for all graphs in $\mathcal{L}$, and let $\tilde{\mathcal{L}}$ be the set of all non-isomorphic graphs with no more than $n$ vertices. We can arrange all graphs in $\tilde{\mathcal{L}}$ into a sequence $L_1, L_2, \ldots, L_N$ satisfying the following property: $|V_{L_i}| \leq |V_{L_{i+1}}|$ or $(|V_{L_i}| = |V_{L_{i+1}}|$ and $|E_{L_i}| \leq |E_{L_{i+1}}|)$ for all $i = 1, 2, \ldots, N - 1$. We then define the matrices $\boldsymbol{A}^{\mathsf{hom}}$, $\boldsymbol{A}^{\mathsf{surj}}$, and $\boldsymbol{A}^{\mathsf{sub}}$, where the elements in the $i^{th}$ row and $j^{th}$ column are $\mathsf{hom}(L_i, L_j)$, $\mathsf{surj}(L_i, L_j)$, and $\mathsf{sub}(L_i, L_j)$, respectively. Proposition G.4 implies that $\boldsymbol{A}^{\mathsf{hom}} = \boldsymbol{A}^{\mathsf{surj}} \cdot \boldsymbol{A}^{\mathsf{sub}}$. Since $\boldsymbol{A}^{\mathsf{surj}}$ is a lower triangular matrix with non-zero diagonal elements and $\boldsymbol{A}^{\mathsf{sub}}$ is an upper triangular matrix with non-zero diagonal elements, the matrix $\boldsymbol{A}^{\mathsf{hom}}$ is invertible.

We next extend $\alpha$ to a function $\tilde{\alpha} : \tilde{\mathcal{L}} \to \mathbb{R}$ by setting $\tilde{\alpha}(L) = \alpha(L)$ for all $L \in \mathcal{L}$ and $\tilde{\alpha}(\tilde{L}) = 0$ for all $\tilde{L} \in \tilde{\mathcal{L}} \backslash \mathcal{L}$. Additionally, if $\chi_G^M(G) = \chi_H^M(H)$, then $\mathsf{hom}(F, G) = \mathsf{hom}(F, H)$ for all $F \in \mathcal{F}^M$ (by definition of homomorphism expressivity). Then, given any graph $K \in \tilde{\mathcal{L}}$, Lemma G.2 implies that $\mathsf{hom}(F, G \times K) = \mathsf{hom}(F, H \times K)$ for all $F \in \mathcal{F}^M$. This further implies that $\chi_{G \times K}^M(G \times K) = \chi_{H \times K}^M(H \times K)$ by definition of homomorphism expressivity. Therefore, Lemma G.2 implies that

$$\sum_{L \in \mathcal{L}} \alpha(L)\mathsf{hom}(L, G) \cdot \mathsf{hom}(L, K) = \sum_{L \in \mathcal{L}} \alpha(L)\mathsf{hom}(L, H) \cdot \mathsf{hom}(L, K). \tag{30}$$

Namely,

$$\sum_{L \in \tilde{\mathcal{L}}} \tilde{\alpha}(L)\mathsf{hom}(L, G) \cdot \mathsf{hom}(L, K) = \sum_{L \in \tilde{\mathcal{L}}} \tilde{\alpha}(L)\mathsf{hom}(L, H) \cdot \mathsf{hom}(L, K). \tag{31}$$

Now define vectors $\boldsymbol{p}_G$ and $\boldsymbol{p}_H$, where the $i^{th}$ element of vector $\boldsymbol{p}_G$ and $\boldsymbol{p}_H$ is $\tilde{\alpha}(L_i)\mathsf{hom}(L_i, G)$ and $\tilde{\alpha}(L_i)\mathsf{hom}(L_i, H)$, respectively. We then have the following equation:

$$\boldsymbol{A}^{\mathsf{hom}} \cdot \boldsymbol{p}_G = \boldsymbol{A}^{\mathsf{hom}} \cdot \boldsymbol{p}_H.$$

Since $\boldsymbol{A}^{\mathsf{hom}}$ is invertible, $\boldsymbol{p}_G^{\mathsf{hom}} = \boldsymbol{p}_H^{\mathsf{hom}}$. Therefore, $\tilde{\alpha}(L)\mathsf{hom}(L, G) = \tilde{\alpha}(L)\mathsf{hom}(L, H)$ for all $L \in \tilde{\mathcal{L}}$, namely, $\mathsf{hom}(L, G) = \mathsf{hom}(L, H)$ for all $L \in \mathcal{L}$. To sum up, we have prove that for all graph $G, H, \chi_G^M(G) = \chi_H^M(H)$ implies that $\mathsf{hom}(L, G) = \mathsf{hom}(L, H)$ for all $L \in \mathcal{L}$. By definition of homomorphism expressivity, we conclude that $\mathcal{L} \subset \mathcal{F}^M$. $\square$

**Theorem G.6.** *Let $M \in \{\mathsf{Sub}, \mathsf{L}, \mathsf{LF}, \mathsf{F}\}$ be any model. For any graph $F$, if $\mathsf{Spasm}(F) \backslash \mathcal{F}^M \neq \emptyset$, then there exists a pair of graphs $G, H$ such that $\chi_G^M(G) = \chi_H^M(H)$ and $\mathsf{sub}(F, G) \neq \mathsf{sub}(F, H)$.*

*Proof.* Recall that $\mathsf{sub}(F, G) = \sum_{\tilde{F} \in \mathsf{Spasm}^{\neq}(F)} \alpha(F, \tilde{F}) \cdot \mathsf{hom}(\tilde{F}, G)$ where $\alpha(F, \tilde{F}) \neq 0$ for all $\tilde{F} \in \mathsf{Spasm}^{\neq}(F)$. If the above theorem does not hold, then for all graphs $G, H$, $\chi_G^M(G) = \chi_H^M(H)$ implies that

$$\sum_{\tilde{F} \in \mathsf{Spasm}^{\neq}(F)} \alpha(F, \tilde{F}) \cdot \mathsf{hom}(\tilde{F}, G) = \sum_{\tilde{F} \in \mathsf{Spasm}^{\neq}(F)} \alpha(F, \tilde{F}) \cdot \mathsf{hom}(\tilde{F}, G).$$

Then, Lemma G.5 implies that $\mathsf{Spasm}(F) \subset \mathcal{F}^M$, yielding a contradiction. $\square$

We next extend our analysis to the node/edge-level subgraph counting. Since the proof techniques are almost the same as the graph-level setting, we only present key definitions and lemmas below while omitting the detailed proofs for clarity.

**Definition G.7.** Given two rooted graphs $G^{\boldsymbol{u}}$ and $H^{\boldsymbol{v}}$ with $\boldsymbol{u} \in V_G^m$ and $\boldsymbol{v} \in V_H^m$ for some $m \in \mathbb{N}_+$, the categorical product of $G^{\boldsymbol{u}}$ and $H^{\boldsymbol{v}}$, denoted as $(G \times H)^{(u_1, v_1), \cdots, (u_m, v_m)}$, is a rooted graph obtained from $G \times H$ by marking vertices $(u_1, v_1), \cdots, (u_m, v_m)$.

**Lemma G.8** (Extension of Lemma G.2). *For any rooted graphs $F^{\boldsymbol{w}}$, $G^{\boldsymbol{u}}$, $H^{\boldsymbol{v}}$ where $|\boldsymbol{u}| = |\boldsymbol{v}| = |\boldsymbol{w}| = m$,*

$$\mathsf{hom}(F^{\boldsymbol{w}}, (G \times H)^{(u_1, v_1), \cdots, (u_m, v_m)}) = \mathsf{hom}(F^{\boldsymbol{w}}, G^{\boldsymbol{u}}) \cdot \mathsf{hom}(F^{\boldsymbol{w}}, H^{\boldsymbol{v}}).$$

**Proposition G.9** (Extension of Proposition G.4). *For any rooted graphs $G^{\boldsymbol{u}}$, $H^{\boldsymbol{v}}$ where $|\boldsymbol{u}| = |\boldsymbol{v}| = m$, $\mathsf{hom}(G^{\boldsymbol{u}}, H^{\boldsymbol{v}}) = \sum_{F^{\boldsymbol{w}}} \mathsf{surj}(G^{\boldsymbol{u}}, F^{\boldsymbol{w}}) \cdot \mathsf{sub}(F^{\boldsymbol{w}}, H^{\boldsymbol{v}})$, where $F^{\boldsymbol{w}}$ ranges over all non-isomorphic rooted graphs marking $m$ vertices.*

We next present the main lemma for node-level subgraph counting. We omit the edge-level result for clarity.

**Lemma G.10** (Extension of Lemma G.5). *Let $M$ be a GNN model such that its node-level homomorphism expressivity $\mathcal{F}_{\mathsf{n}}^M$ exists. Given a finite set of rooted graphs $\mathcal{L}_{\mathsf{n}}$ and a function $\alpha : \mathcal{L}_{\mathsf{n}} \to \mathbb{R}/\{0\}$, if*

$$\chi_G^M(u) = \chi_H^M(v) \implies \sum_{L^w \in \mathcal{L}_{\mathsf{n}}} \alpha(L^w)\mathsf{hom}(L^w, G^u) = \sum_{L^w \in \mathcal{L}_{\mathsf{n}}} \alpha(L^w)\mathsf{hom}(L^w, H^v),$$

*holds for all rooted graphs $G^u$ and $H^v$, then $\mathcal{L}_{\mathsf{n}} \subset \mathcal{F}_{\mathsf{n}}^M$.*

**Theorem G.11.** *Let $M \in \{\mathsf{Sub}, \mathsf{L}, \mathsf{LF}, \mathsf{F}\}$ be any model. For any rooted graph $F^w$, if $\mathsf{Spasm}(F^w) \backslash \mathcal{F}_{\mathsf{n}}^M \neq \emptyset$, then there exist a pair of graphs $G, H$ and vertices $u \in V_G$, $v \in V_H$, such that $\chi_G^M(u) = \chi_H^M(v)$ and $\mathsf{sub}(F^w, G^u) \neq \mathsf{sub}(F^w, H^v)$.*

## G.2    GRAPH STATISTICS AND EXAMPLES

In this section, we list the statistics of all moderate-size graphs that can/cannot be counted for each model in Section 2 at graph/node/edge-level. Table 4 presents the statistics under homomorphism count, while Table 5 presents the statistics under subgraph count. These tables offer a clear picture into how large the expressivity gaps are between different models. Several important findings are discussed below:

- For homomorphism counting, graphs of 8 edges suffices to reveal the expressivity gaps between each pair of architectures (at edge-level).
- However, for subgraph counting, moderate-size graphs cannot reveal the gap between Local 2-GNN, Local 2-FGNN, and 2-FGNN. Moreover, even Subgraph GNN already matches the power of 2-FWL in counting small subgraphs *at graph-level*, but it is quite weak in counting subgraphs at node/edge-level.

- Subgraph counting is much more challenging than homomorphism counting. Intuitively, this is because the homomorphism image of a graph usually has 4-cliques, and any graph that contains a 4 clique as subgraph cannot be counted under homomorphism.

In Table 6, we list all subgraphs at moderate size (no more than 6 vertices or 8 edges) that can/cannot be counted by each GNN model. Here, we only list the graph-level expressivity as the node/edge-level expressivity involves too many non-isomorphic rooted graphs and cannot be fully presented (see Tables 4 and 5). We believe these results is comprehensive enough to cover most substructures of interest in the GNN community.

Table 4: The number of (connected) graphs (or rooted graphs) of $n$ vertices or $m$ edges that can be counted under homomorphism by different models. These statistics can be viewed as a quantitative expressivity comparison between models.

| | | Number of vertices $n$ | | | | | Number of edges $m$ | | | | | | | |
|---|---|---|---|---|---|---|---|---|---|---|---|---|---|---|
| | | 2 | 3 | 4 | 5 | 6 | 1 | 2 | 3 | 4 | 5 | 6 | 7 | 8 |
| Graph level | MPNN | 1 | 1 | 2 | 3 | 6 | 1 | 1 | 2 | 3 | 6 | 11 | 23 | 47 |
| | Subgraph GNN | 1 | 2 | 5 | 15 | 51 | 1 | 1 | 3 | 5 | 12 | 29 | 76 | 210 |
| | Local 2-GNN | 1 | 2 | 5 | 15 | 55 | 1 | 1 | 3 | 5 | 12 | 29 | 77 | 216 |
| | Local 2-FGNN | 1 | 2 | 5 | 15 | 56 | 1 | 1 | 3 | 5 | 12 | 29 | 77 | 216 |
| | 2-FGNN | 1 | 2 | 5 | 15 | 56 | 1 | 1 | 3 | 5 | 12 | 29 | 77 | 216 |
| | All | 1 | 2 | 6 | 21 | 112 | 1 | 1 | 3 | 5 | 12 | 30 | 79 | 227 |
| Node level | MPNN | 1 | 2 | 4 | 9 | 20 | 1 | 2 | 4 | 9 | 20 | 48 | 115 | 286 |
| | Subgraph GNN | 1 | 3 | 8 | 27 | 88 | 1 | 2 | 5 | 12 | 31 | 83 | 228 | 640 |
| | Local 2-GNN | 1 | 3 | 10 | 44 | 215 | 1 | 2 | 5 | 13 | 37 | 113 | 361 | 1210 |
| | Local 2-FGNN | 1 | 3 | 10 | 44 | 217 | 1 | 2 | 5 | 13 | 37 | 113 | 361 | 1210 |
| | 2-FGNN | 1 | 3 | 10 | 44 | 217 | 1 | 2 | 5 | 13 | 37 | 113 | 361 | 1210 |
| | All | 1 | 3 | 11 | 58 | 407 | 1 | 2 | 5 | 13 | 37 | 114 | 367 | 1248 |
| Edge level | Subgraph GNN | 1 | 4 | 18 | 77 | 340 | 1 | 3 | 10 | 33 | 107 | 347 | 1126 | 3664 |
| | Local 2-GNN | 1 | 4 | 21 | 116 | 693 | 1 | 3 | 10 | 35 | 124 | 450 | 1665 | 6267 |
| | Local 2-FGNN | 1 | 4 | 21 | 118 | 735 | 1 | 3 | 10 | 35 | 124 | 451 | 1678 | 6373 |
| | 2-FGNN | 1 | 4 | 21 | 118 | 735 | 1 | 3 | 10 | 35 | 124 | 451 | 1678 | 6374 |
| | All | 1 | 4 | 23 | 162 | 1549 | 1 | 3 | 10 | 35 | 125 | 460 | 1747 | 6830 |

Table 5: The number of (connected) graphs (or rooted graphs) of $n$ vertices or $m$ edges that can be subgraph-counted by different models. These statistics can be viewed as a quantitative expressivity comparison between models.

| | | Number of vertices $n$ | | | | | Number of edges $m$ | | | | | | | |
|---|---|---|---|---|---|---|---|---|---|---|---|---|---|---|
| | | 2 | 3 | 4 | 5 | 6 | 1 | 2 | 3 | 4 | 5 | 6 | 7 | 8 |
| Graph level | MPNN | 1 | 1 | 1 | 1 | 1 | 1 | 1 | 1 | 1 | 1 | 1 | 1 | 1 |
| | Subgraph GNN | 1 | 2 | 5 | 13 | 24 | 1 | 1 | 3 | 5 | 12 | 20 | 21 | 22 |
| | Local 2-GNN | 1 | 2 | 5 | 13 | 24 | 1 | 1 | 3 | 5 | 12 | 20 | 21 | 22 |
| | Local 2-FGNN | 1 | 2 | 5 | 13 | 24 | 1 | 1 | 3 | 5 | 12 | 20 | 21 | 22 |
| | 2-FGNN | 1 | 2 | 5 | 13 | 24 | 1 | 1 | 3 | 5 | 12 | 20 | 21 | 22 |
| | All | 1 | 2 | 6 | 21 | 112 | 1 | 1 | 3 | 5 | 12 | 30 | 79 | 227 |
| Node level | MPNN | 1 | 2 | 2 | 2 | 2 | 1 | 2 | 2 | 2 | 2 | 2 | 2 | 2 |
| | Subgraph GNN | 1 | 3 | 8 | 18 | 27 | 1 | 2 | 5 | 10 | 16 | 23 | 30 | 38 |
| | Local 2-GNN | 1 | 3 | 10 | 37 | 84 | 1 | 2 | 5 | 13 | 37 | 72 | 75 | 86 |
| | Local 2-FGNN | 1 | 3 | 10 | 37 | 84 | 1 | 2 | 5 | 13 | 37 | 72 | 75 | 86 |
| | 2-FGNN | 1 | 3 | 10 | 37 | 84 | 1 | 2 | 5 | 13 | 37 | 72 | 75 | 86 |
| | All | 1 | 3 | 11 | 58 | 407 | 1 | 2 | 5 | 13 | 37 | 114 | 367 | 1248 |
| Edge level | Subgraph GNN | 1 | 4 | 18 | 47 | 81 | 1 | 3 | 10 | 25 | 46 | 69 | 95 | 124 |
| | Local 2-GNN | 1 | 4 | 21 | 92 | 208 | 1 | 3 | 10 | 35 | 105 | 171 | 179 | 216 |
| | Local 2-FGNN | 1 | 4 | 21 | 92 | 208 | 1 | 3 | 10 | 35 | 105 | 171 | 179 | 216 |
| | 2-FGNN | 1 | 4 | 21 | 92 | 208 | 1 | 3 | 10 | 35 | 105 | 171 | 179 | 216 |
| | All | 1 | 4 | 23 | 162 | 1549 | 1 | 3 | 10 | 35 | 125 | 460 | 1747 | 6830 |

Table 6: The ability of GNNs to homomorphism-count and subgraph-count different graphs $F$ within a bounded size of $n \leq 6$ vertices or $m \leq 8$ edges. When one or more GNNs fail to subgraph-count $F$, this table also gives a homomorphic image $\tilde{F} \in \mathsf{Spasm}(F)$ that can be used to construct counterexample graphs (see Section 4.2).

| $n\ m$ | $F$ | Homomorphism MP Sub L LF F | Subgraph MP Sub L LF F | $\tilde{F}$ | $n\ m$ | $F$ | Homomorphism MP Sub L LF F | Subgraph MP Sub L LF F | $\tilde{F}$ |
|---|---|---|---|---|---|---|---|---|---|
| 2 1 | [graph] | ✓ ✓ ✓ ✓ ✓ | ✓ ✓ ✓ ✓ ✓ | - | 5 7 | [graph] | ✗ ✓ ✓ ✓ ✓ | ✗ ✓ ✓ ✓ ✓ | [graph] |
| 3 2 | [graph] | ✓ ✓ ✓ ✓ ✓ | ✓ ✓ ✓ ✓ ✓ | - | 5 8 | [graph] | ✗ ✗ ✗ ✗ ✗ | ✗ ✗ ✗ ✗ ✗ | [graph] |
| 3 3 | [graph] | ✗ ✓ ✓ ✓ ✓ | ✗ ✓ ✓ ✓ ✓ | [graph] | 5 8 | [graph] | ✗ ✗ ✗ ✗ ✗ | ✗ ✗ ✗ ✗ ✗ | [graph] |
| 4 3 | [graph] | ✓ ✓ ✓ ✓ ✓ | ✗ ✓ ✓ ✓ ✓ | [graph] | 5 9 | [graph] | ✗ ✗ ✗ ✗ ✗ | ✗ ✗ ✗ ✗ ✗ | [graph] |
| 4 3 | [graph] | ✓ ✓ ✓ ✓ ✓ | ✓ ✓ ✓ ✓ ✓ | - | 5 10 | [graph] | ✗ ✗ ✗ ✗ ✗ | ✗ ✗ ✗ ✗ ✗ | [graph] |
| 4 4 | [graph] | ✗ ✓ ✓ ✓ ✓ | ✗ ✓ ✓ ✓ ✓ | [graph] | 6 5 | [graph] | ✓ ✓ ✓ ✓ ✓ | ✗ ✓ ✓ ✓ ✓ | [graph] |
| 4 4 | [graph] | ✗ ✓ ✓ ✓ ✓ | ✗ ✓ ✓ ✓ ✓ | [graph] | 6 5 | [graph] | ✓ ✓ ✓ ✓ ✓ | ✗ ✓ ✓ ✓ ✓ | [graph] |
| 4 5 | [graph] | ✗ ✓ ✓ ✓ ✓ | ✗ ✓ ✓ ✓ ✓ | [graph] | 6 5 | [graph] | ✓ ✓ ✓ ✓ ✓ | ✓ ✓ ✓ ✓ ✓ | - |
| 4 6 | [graph] | ✗ ✗ ✗ ✗ ✗ | ✗ ✗ ✗ ✗ ✗ | [graph] | 6 5 | [graph] | ✓ ✓ ✓ ✓ ✓ | ✗ ✓ ✓ ✓ ✓ | [graph] |
| 5 4 | [graph] | ✓ ✓ ✓ ✓ ✓ | ✗ ✓ ✓ ✓ ✓ | [graph] | 6 5 | [graph] | ✓ ✓ ✓ ✓ ✓ | ✗ ✓ ✓ ✓ ✓ | [graph] |
| 5 4 | [graph] | ✓ ✓ ✓ ✓ ✓ | ✗ ✓ ✓ ✓ ✓ | [graph] | 6 5 | [graph] | ✓ ✓ ✓ ✓ ✓ | ✗ ✓ ✓ ✓ ✓ | [graph] |
| 5 4 | [graph] | ✓ ✓ ✓ ✓ ✓ | ✓ ✓ ✓ ✓ ✓ | - | 6 6 | [graph] | ✗ ✓ ✓ ✓ ✓ | ✗ ✓ ✓ ✓ ✓ | [graph] |
| 5 5 | [graph] | ✗ ✓ ✓ ✓ ✓ | ✗ ✓ ✓ ✓ ✓ | [graph] | 6 6 | [graph] | ✗ ✓ ✓ ✓ ✓ | ✗ ✓ ✓ ✓ ✓ | [graph] |
| 5 5 | [graph] | ✗ ✓ ✓ ✓ ✓ | ✗ ✓ ✓ ✓ ✓ | [graph] | 6 6 | [graph] | ✗ ✓ ✓ ✓ ✓ | ✗ ✓ ✓ ✓ ✓ | [graph] |
| 5 5 | [graph] | ✗ ✓ ✓ ✓ ✓ | ✗ ✓ ✓ ✓ ✓ | [graph] | 6 6 | [graph] | ✗ ✓ ✓ ✓ ✓ | ✗ ✗ ✗ ✗ ✗ | [graph] |
| 5 5 | [graph] | ✗ ✓ ✓ ✓ ✓ | ✗ ✓ ✓ ✓ ✓ | [graph] | 6 6 | [graph] | ✗ ✓ ✓ ✓ ✓ | ✗ ✓ ✓ ✓ ✓ | [graph] |
| 5 5 | [graph] | ✗ ✓ ✓ ✓ ✓ | ✗ ✓ ✓ ✓ ✓ | [graph] | 6 6 | [graph] | ✗ ✓ ✓ ✓ ✓ | ✗ ✓ ✓ ✓ ✓ | [graph] |
| 5 6 | [graph] | ✗ ✓ ✓ ✓ ✓ | ✗ ✗ ✗ ✗ ✗ | [graph] | 6 6 | [graph] | ✗ ✓ ✓ ✓ ✓ | ✗ ✗ ✗ ✗ ✗ | [graph] |
| 5 6 | [graph] | ✗ ✓ ✓ ✓ ✓ | ✗ ✓ ✓ ✓ ✓ | [graph] | 6 6 | [graph] | ✗ ✓ ✓ ✓ ✓ | ✗ ✓ ✓ ✓ ✓ | [graph] |
| 5 6 | [graph] | ✗ ✓ ✓ ✓ ✓ | ✗ ✓ ✓ ✓ ✓ | [graph] | 6 6 | [graph] | ✗ ✓ ✓ ✓ ✓ | ✗ ✗ ✗ ✗ ✗ | [graph] |
| 5 6 | [graph] | ✗ ✓ ✓ ✓ ✓ | ✗ ✓ ✓ ✓ ✓ | [graph] | 6 6 | [graph] | ✗ ✓ ✓ ✓ ✓ | ✗ ✓ ✓ ✓ ✓ | [graph] |
| 5 6 | [graph] | ✗ ✓ ✓ ✓ ✓ | ✗ ✓ ✓ ✓ ✓ | [graph] | 6 6 | [graph] | ✗ ✓ ✓ ✓ ✓ | ✗ ✓ ✓ ✓ ✓ | [graph] |
| 5 7 | [graph] | ✗ ✗ ✗ ✗ ✗ | ✗ ✗ ✗ ✗ ✗ | [graph] | 6 6 | [graph] | ✗ ✓ ✓ ✓ ✓ | ✗ ✗ ✗ ✗ ✗ | [graph] |
| 5 7 | [graph] | ✗ ✗ ✗ ✗ ✗ | ✗ ✗ ✗ ✗ ✗ | [graph] | 6 7 | [graph] | ✗ ✓ ✓ ✓ ✓ | ✗ ✓ ✓ ✓ ✓ | [graph] |
| 5 7 | [graph] | ✗ ✓ ✓ ✓ ✓ | ✗ ✗ ✗ ✗ ✗ | [graph] | | | | | |

(see the next page)

| n m | F | Homomorphism MP | Sub | L | LF | F | Subgraph MP | Sub | L | LF | F | F̃ | n m | F | Homomorphism MP | Sub | L | LF | F | Subgraph MP | Sub | L | LF | F | F̃ |
|---|---|---|---|---|---|---|---|---|---|---|---|---|---|---|---|---|---|---|---|---|---|---|---|---|---|
| 6 7 | | ✗ | ✓ | ✓ | ✓ | ✓ | ✗ | ✗ | ✗ | ✗ | ✗ | | 6 8 | | ✗ | ✓ | ✓ | ✓ | ✓ | ✗ | ✗ | ✗ | ✗ | ✗ | |
| 6 7 | | ✗ | ✓ | ✓ | ✓ | ✓ | ✗ | ✓ | ✓ | ✓ | ✓ | | 6 8 | | ✗ | ✓ | ✓ | ✓ | ✓ | ✗ | ✗ | ✗ | ✗ | ✗ | |
| 6 7 | | ✗ | ✓ | ✓ | ✓ | ✓ | ✗ | ✗ | ✗ | ✗ | ✗ | | 6 8 | | ✗ | ✓ | ✓ | ✓ | ✓ | ✗ | ✗ | ✗ | ✗ | ✗ | |
| 6 7 | | ✗ | ✓ | ✓ | ✓ | ✓ | ✗ | ✗ | ✗ | ✗ | ✗ | | 6 8 | | ✗ | ✗ | ✗ | ✗ | ✗ | ✗ | ✗ | ✗ | ✗ | ✗ | |
| 6 7 | | ✗ | ✓ | ✓ | ✓ | ✓ | ✗ | ✗ | ✗ | ✗ | ✗ | | 6 8 | | ✗ | ✗ | ✗ | ✗ | ✗ | ✗ | ✗ | ✗ | ✗ | ✗ | |
| 6 7 | | ✗ | ✓ | ✓ | ✓ | ✓ | ✗ | ✗ | ✗ | ✗ | ✗ | | 6 8 | | ✗ | ✓ | ✓ | ✓ | ✓ | ✗ | ✓ | ✓ | ✓ | ✓ | |
| 6 7 | | ✗ | ✓ | ✓ | ✓ | ✓ | ✗ | ✗ | ✗ | ✗ | ✗ | | 6 8 | | ✗ | ✗ | ✗ | ✗ | ✗ | ✗ | ✗ | ✗ | ✗ | ✗ | |
| 6 7 | | ✗ | ✓ | ✓ | ✓ | ✓ | ✗ | ✓ | ✓ | ✓ | ✓ | | 6 8 | | ✗ | ✓ | ✓ | ✓ | ✓ | ✗ | ✗ | ✗ | ✗ | ✗ | |
| 6 7 | | ✗ | ✓ | ✓ | ✓ | ✓ | ✗ | ✓ | ✓ | ✓ | ✓ | | 6 8 | | ✗ | ✓ | ✓ | ✓ | ✓ | ✗ | ✗ | ✗ | ✗ | ✗ | |
| 6 7 | | ✗ | ✓ | ✓ | ✓ | ✓ | ✗ | ✓ | ✓ | ✓ | ✓ | | 6 8 | | ✗ | ✗ | ✗ | ✗ | ✗ | ✗ | ✗ | ✗ | ✗ | ✗ | |
| 6 7 | | ✗ | ✓ | ✓ | ✓ | ✓ | ✗ | ✓ | ✓ | ✓ | ✓ | | 6 8 | | ✗ | ✗ | ✗ | ✗ | ✗ | ✗ | ✗ | ✗ | ✗ | ✗ | |
| 6 7 | | ✗ | ✓ | ✓ | ✓ | ✓ | ✗ | ✗ | ✗ | ✗ | ✗ | | 6 8 | | ✗ | ✓ | ✓ | ✓ | ✓ | ✗ | ✗ | ✗ | ✗ | ✗ | |
| 6 7 | | ✗ | ✓ | ✓ | ✓ | ✓ | ✗ | ✗ | ✗ | ✗ | ✗ | | 6 8 | | ✗ | ✓ | ✓ | ✓ | ✓ | ✗ | ✓ | ✓ | ✓ | ✓ | |
| 6 7 | | ✗ | ✓ | ✓ | ✓ | ✓ | ✗ | ✗ | ✗ | ✗ | ✗ | | 6 8 | | ✗ | ✓ | ✓ | ✓ | ✓ | ✗ | ✗ | ✗ | ✗ | ✗ | |
| 6 7 | | ✗ | ✓ | ✓ | ✓ | ✓ | ✗ | ✗ | ✗ | ✗ | ✗ | | 6 9 | | ✗ | ✗ | ✗ | ✗ | ✗ | ✗ | ✗ | ✗ | ✗ | ✗ | |
| 6 7 | | ✗ | ✓ | ✓ | ✓ | ✓ | ✗ | ✗ | ✗ | ✗ | ✗ | | 6 9 | | ✗ | ✗ | ✗ | ✗ | ✗ | ✗ | ✗ | ✗ | ✗ | ✗ | |
| 6 7 | | ✗ | ✓ | ✓ | ✓ | ✓ | ✗ | ✗ | ✗ | ✗ | ✗ | | 6 9 | | ✗ | ✗ | ✓ | ✓ | ✓ | ✗ | ✗ | ✗ | ✗ | ✗ | |
| 6 7 | | ✗ | ✗ | ✓ | ✓ | ✓ | ✗ | ✗ | ✗ | ✗ | ✗ | | 6 9 | | ✗ | ✓ | ✓ | ✓ | ✓ | ✗ | ✗ | ✗ | ✗ | ✗ | |
| 6 8 | | ✗ | ✗ | ✗ | ✗ | ✗ | ✗ | ✗ | ✗ | ✗ | ✗ | | 6 9 | | ✗ | ✓ | ✓ | ✓ | ✓ | ✗ | ✗ | ✗ | ✗ | ✗ | |
| 6 8 | | ✗ | ✓ | ✓ | ✓ | ✓ | ✗ | ✗ | ✗ | ✗ | ✗ | | 6 9 | | ✗ | ✗ | ✗ | ✗ | ✗ | ✗ | ✗ | ✗ | ✗ | ✗ | |
| 6 8 | | ✗ | ✗ | ✗ | ✗ | ✗ | ✗ | ✗ | ✗ | ✗ | ✗ | | 6 9 | | ✗ | ✗ | ✗ | ✗ | ✗ | ✗ | ✗ | ✗ | ✗ | ✗ | |
| 6 8 | | ✗ | ✗ | ✗ | ✗ | ✗ | ✗ | ✗ | ✗ | ✗ | ✗ | | 6 9 | | ✗ | ✗ | ✗ | ✗ | ✗ | ✗ | ✗ | ✗ | ✗ | ✗ | |
| 6 8 | | ✗ | ✓ | ✓ | ✓ | ✓ | ✗ | ✗ | ✗ | ✗ | ✗ | | 6 9 | | ✗ | ✗ | ✗ | ✗ | ✗ | ✗ | ✗ | ✗ | ✗ | ✗ | |
| 6 8 | | ✗ | ✗ | ✓ | ✓ | ✓ | ✗ | ✗ | ✗ | ✗ | ✗ | | 6 9 | | ✗ | ✗ | ✗ | ✗ | ✗ | ✗ | ✗ | ✗ | ✗ | ✗ | |
| 6 8 | | ✗ | ✗ | ✗ | ✗ | ✗ | ✗ | ✗ | ✗ | ✗ | ✗ | | 6 9 | | ✗ | ✗ | ✗ | ✗ | ✗ | ✗ | ✗ | ✗ | ✗ | ✗ | |
| 6 8 | | ✗ | ✗ | ✓ | ✓ | ✓ | ✗ | ✗ | ✗ | ✗ | ✗ | | 6 9 | | ✗ | ✓ | ✓ | ✓ | ✓ | ✗ | ✓ | ✓ | ✓ | ✓ | |

(see the next page)

| n | m | F | Homomorphism MP | Sub | L | LF | F | Subgraph MP | Sub | L | LF | F | F̃ | n | m | F | Homomorphism MP | Sub | L | LF | F | Subgraph MP | Sub | L | LF | F | F̃ |
|---|---|---|---|---|---|---|---|---|---|---|---|---|---|---|---|---|---|---|---|---|---|---|---|---|---|---|---|
| 6 | 9 | [graph] | ✗ | ✗ | ✗ | ✗ | ✗ | ✗ | ✗ | ✗ | ✗ | ✗ | [graph] | 6 | 11 | [graph] | ✗ | ✗ | ✗ | ✗ | ✗ | ✗ | ✗ | ✗ | ✗ | ✗ | [graph] |
| 6 | 9 | [graph] | ✗ | ✗ | ✗ | ✗ | ✗ | ✗ | ✗ | ✗ | ✗ | ✗ | [graph] | 6 | 11 | [graph] | ✗ | ✗ | ✗ | ✗ | ✗ | ✗ | ✗ | ✗ | ✗ | ✗ | [graph] |
| 6 | 9 | [graph] | ✗ | ✗ | ✗ | ✓ | ✓ | ✗ | ✗ | ✗ | ✗ | ✗ | [graph] | 6 | 11 | [graph] | ✗ | ✗ | ✗ | ✗ | ✗ | ✗ | ✗ | ✗ | ✗ | ✗ | [graph] |
| 6 | 9 | [graph] | ✗ | ✗ | ✗ | ✗ | ✗ | ✗ | ✗ | ✗ | ✗ | ✗ | [graph] | 6 | 11 | [graph] | ✗ | ✗ | ✗ | ✗ | ✗ | ✗ | ✗ | ✗ | ✗ | ✗ | [graph] |
| 6 | 9 | [graph] | ✗ | ✗ | ✗ | ✗ | ✗ | ✗ | ✗ | ✗ | ✗ | ✗ | [graph] | 6 | 11 | [graph] | ✗ | ✗ | ✗ | ✗ | ✗ | ✗ | ✗ | ✗ | ✗ | ✗ | [graph] |
| 6 | 9 | [graph] | ✗ | ✗ | ✗ | ✗ | ✗ | ✗ | ✗ | ✗ | ✗ | ✗ | [graph] | 6 | 12 | [graph] | ✗ | ✗ | ✗ | ✗ | ✗ | ✗ | ✗ | ✗ | ✗ | ✗ | [graph] |
| 6 | 9 | [graph] | ✗ | ✗ | ✗ | ✗ | ✗ | ✗ | ✗ | ✗ | ✗ | ✗ | [graph] | 6 | 12 | [graph] | ✗ | ✗ | ✗ | ✗ | ✗ | ✗ | ✗ | ✗ | ✗ | ✗ | [graph] |
| 6 | 9 | [graph] | ✗ | ✗ | ✗ | ✗ | ✗ | ✗ | ✗ | ✗ | ✗ | ✗ | [graph] | 6 | 12 | [graph] | ✗ | ✗ | ✗ | ✗ | ✗ | ✗ | ✗ | ✗ | ✗ | ✗ | [graph] |
| 6 | 10 | [graph] | ✗ | ✗ | ✗ | ✗ | ✗ | ✗ | ✗ | ✗ | ✗ | ✗ | [graph] | 6 | 12 | [graph] | ✗ | ✗ | ✗ | ✗ | ✗ | ✗ | ✗ | ✗ | ✗ | ✗ | [graph] |
| 6 | 10 | [graph] | ✗ | ✗ | ✗ | ✗ | ✗ | ✗ | ✗ | ✗ | ✗ | ✗ | [graph] | 6 | 12 | [graph] | ✗ | ✗ | ✗ | ✗ | ✗ | ✗ | ✗ | ✗ | ✗ | ✗ | [graph] |
| 6 | 10 | [graph] | ✗ | ✗ | ✗ | ✗ | ✗ | ✗ | ✗ | ✗ | ✗ | ✗ | [graph] | 6 | 13 | [graph] | ✗ | ✗ | ✗ | ✗ | ✗ | ✗ | ✗ | ✗ | ✗ | ✗ | [graph] |
| 6 | 10 | [graph] | ✗ | ✗ | ✗ | ✗ | ✗ | ✗ | ✗ | ✗ | ✗ | ✗ | [graph] | 6 | 13 | [graph] | ✗ | ✗ | ✗ | ✗ | ✗ | ✗ | ✗ | ✗ | ✗ | ✗ | [graph] |
| 6 | 10 | [graph] | ✗ | ✗ | ✗ | ✗ | ✗ | ✗ | ✗ | ✗ | ✗ | ✗ | [graph] | 6 | 14 | [graph] | ✗ | ✗ | ✗ | ✗ | ✗ | ✗ | ✗ | ✗ | ✗ | ✗ | [graph] |
| 6 | 10 | [graph] | ✗ | ✗ | ✗ | ✗ | ✗ | ✗ | ✗ | ✗ | ✗ | ✗ | [graph] | 6 | 15 | [graph] | ✗ | ✗ | ✗ | ✗ | ✗ | ✗ | ✗ | ✗ | ✗ | ✗ | [graph] |
| 6 | 10 | [graph] | ✗ | ✗ | ✗ | ✗ | ✗ | ✗ | ✗ | ✗ | ✗ | ✗ | [graph] | 7 | 6 | [graph] | ✓ | ✓ | ✓ | ✓ | ✓ | ✗ | ✗ | ✗ | ✗ | ✗ | [graph] |
| 6 | 10 | [graph] | ✗ | ✗ | ✗ | ✗ | ✗ | ✗ | ✗ | ✗ | ✗ | ✗ | [graph] | 7 | 6 | [graph] | ✓ | ✓ | ✓ | ✓ | ✓ | ✓ | ✓ | ✓ | ✓ | ✓ | - |
| 6 | 10 | [graph] | ✗ | ✗ | ✗ | ✗ | ✗ | ✗ | ✗ | ✗ | ✗ | ✗ | [graph] | 7 | 6 | [graph] | ✓ | ✓ | ✓ | ✓ | ✓ | ✗ | ✓ | ✓ | ✓ | ✓ | [graph] |
| 6 | 10 | [graph] | ✗ | ✗ | ✗ | ✗ | ✗ | ✗ | ✗ | ✗ | ✗ | ✗ | [graph] | 7 | 6 | [graph] | ✓ | ✓ | ✓ | ✓ | ✓ | ✗ | ✓ | ✓ | ✓ | ✓ | [graph] |
| 6 | 10 | [graph] | ✗ | ✗ | ✗ | ✗ | ✗ | ✗ | ✗ | ✗ | ✗ | ✗ | [graph] | 7 | 6 | [graph] | ✓ | ✓ | ✓ | ✓ | ✓ | ✗ | ✓ | ✓ | ✓ | ✓ | [graph] |
| 6 | 10 | [graph] | ✗ | ✗ | ✗ | ✗ | ✗ | ✗ | ✗ | ✗ | ✗ | ✗ | [graph] | 7 | 6 | [graph] | ✓ | ✓ | ✓ | ✓ | ✓ | ✗ | ✓ | ✓ | ✓ | ✓ | [graph] |
| 6 | 10 | [graph] | ✗ | ✗ | ✗ | ✗ | ✗ | ✗ | ✗ | ✗ | ✗ | ✗ | [graph] | 7 | 6 | [graph] | ✓ | ✓ | ✓ | ✓ | ✓ | ✗ | ✗ | ✗ | ✗ | ✗ | [graph] |
| 6 | 10 | [graph] | ✗ | ✗ | ✗ | ✗ | ✗ | ✗ | ✗ | ✗ | ✗ | ✗ | [graph] | 7 | 6 | [graph] | ✓ | ✓ | ✓ | ✓ | ✓ | ✗ | ✗ | ✗ | ✗ | ✗ | [graph] |
| 6 | 11 | [graph] | ✗ | ✗ | ✗ | ✗ | ✗ | ✗ | ✗ | ✗ | ✗ | ✗ | [graph] | 7 | 6 | [graph] | ✓ | ✓ | ✓ | ✓ | ✓ | ✗ | ✗ | ✗ | ✗ | ✗ | [graph] |
| 6 | 11 | [graph] | ✗ | ✗ | ✗ | ✗ | ✗ | ✗ | ✗ | ✗ | ✗ | ✗ | [graph] | 7 | 6 | [graph] | ✓ | ✓ | ✓ | ✓ | ✓ | ✗ | ✓ | ✓ | ✓ | ✓ | [graph] |
| 6 | 11 | [graph] | ✗ | ✗ | ✗ | ✗ | ✗ | ✗ | ✗ | ✗ | ✗ | ✗ | [graph] | 7 | 6 | [graph] | ✓ | ✓ | ✓ | ✓ | ✓ | ✗ | ✓ | ✓ | ✓ | ✓ | [graph] |
| 6 | 11 | [graph] | ✗ | ✗ | ✗ | ✗ | ✗ | ✗ | ✗ | ✗ | ✗ | ✗ | [graph] | 7 | 7 | [graph] | ✗ | ✓ | ✓ | ✓ | ✓ | ✗ | ✗ | ✗ | ✗ | ✗ | [graph] |

(see the next page)

| n | m | F | Hom MP | Hom Sub | Hom L | Hom LF | Hom F | Sub MP | Sub Sub | Sub L | Sub LF | Sub F | $\tilde{F}$ |
|---|---|---|---|---|---|---|---|---|---|---|---|---|---|
| 7 | 7 | (graph) | ✗ | ✓ | ✓ | ✓ | ✓ | ✗ | ✗ | ✗ | ✗ | | (graph) |
| 7 | 7 | (graph) | ✗ | ✓ | ✓ | ✓ | ✓ | ✗ | ✗ | ✗ | ✗ | | (graph) |
| 7 | 7 | (graph) | ✗ | ✓ | ✓ | ✓ | ✓ | ✗ | ✗ | ✗ | ✗ | | (graph) |
| 7 | 7 | (graph) | ✗ | ✓ | ✓ | ✓ | ✓ | ✗ | ✗ | ✗ | ✗ | | (graph) |
| 7 | 7 | (graph) | ✗ | ✓ | ✓ | ✓ | ✓ | ✗ | ✓ | ✓ | ✓ | ✓ | (graph) |
| 7 | 7 | (graph) | ✗ | ✓ | ✓ | ✓ | ✓ | ✗ | ✗ | ✗ | ✗ | | (graph) |
| 7 | 7 | (graph) | ✗ | ✓ | ✓ | ✓ | ✓ | ✗ | ✓ | ✓ | ✓ | ✓ | (graph) |
| 7 | 7 | (graph) | ✗ | ✓ | ✓ | ✓ | ✓ | ✗ | ✗ | ✗ | ✗ | | (graph) |
| 7 | 7 | (graph) | ✗ | ✓ | ✓ | ✓ | ✓ | ✗ | ✗ | ✗ | ✗ | | (graph) |
| 7 | 7 | (graph) | ✗ | ✓ | ✓ | ✓ | ✓ | ✗ | ✗ | ✗ | ✗ | | (graph) |
| 7 | 7 | (graph) | ✗ | ✓ | ✓ | ✓ | ✓ | ✗ | ✗ | ✗ | ✗ | | (graph) |
| 7 | 7 | (graph) | ✗ | ✓ | ✓ | ✓ | ✓ | ✗ | ✓ | ✓ | ✓ | ✓ | (graph) |
| 7 | 7 | (graph) | ✗ | ✓ | ✓ | ✓ | ✓ | ✗ | ✓ | ✓ | ✓ | ✓ | (graph) |
| 7 | 7 | (graph) | ✗ | ✓ | ✓ | ✓ | ✓ | ✗ | ✗ | ✗ | ✗ | | (graph) |
| 7 | 7 | (graph) | ✗ | ✓ | ✓ | ✓ | ✓ | ✗ | ✗ | ✗ | ✗ | | (graph) |
| 7 | 7 | (graph) | ✗ | ✓ | ✓ | ✓ | ✓ | ✗ | ✗ | ✗ | ✗ | | (graph) |
| 7 | 7 | (graph) | ✗ | ✓ | ✓ | ✓ | ✓ | ✗ | ✗ | ✗ | ✗ | | (graph) |
| 7 | 7 | (graph) | ✗ | ✓ | ✓ | ✓ | ✓ | ✗ | ✗ | ✗ | ✗ | | (graph) |
| 7 | 7 | (graph) | ✗ | ✓ | ✓ | ✓ | ✓ | ✗ | ✗ | ✗ | ✗ | | (graph) |
| 7 | 7 | (graph) | ✗ | ✓ | ✓ | ✓ | ✓ | ✗ | ✗ | ✗ | ✗ | | (graph) |
| 7 | 7 | (graph) | ✗ | ✓ | ✓ | ✓ | ✓ | ✗ | ✓ | ✓ | ✓ | ✓ | (graph) |
| 7 | 7 | (graph) | ✗ | ✓ | ✓ | ✓ | ✓ | ✗ | ✗ | ✗ | ✗ | | (graph) |
| 7 | 7 | (graph) | ✗ | ✓ | ✓ | ✓ | ✓ | ✗ | ✓ | ✓ | ✓ | ✓ | (graph) |
| 7 | 7 | (graph) | ✗ | ✓ | ✓ | ✓ | ✓ | ✗ | ✗ | ✗ | ✗ | | (graph) |
| 7 | 7 | (graph) | ✗ | ✓ | ✓ | ✓ | ✓ | ✗ | ✗ | ✗ | ✗ | | (graph) |
| 7 | 7 | (graph) | ✗ | ✓ | ✓ | ✓ | ✓ | ✗ | ✗ | ✗ | ✗ | | (graph) |

| n | m | F | Hom MP | Hom Sub | Hom L | Hom LF | Hom F | Sub MP | Sub Sub | Sub L | Sub LF | Sub F | $\tilde{F}$ |
|---|---|---|---|---|---|---|---|---|---|---|---|---|---|
| 7 | 7 | (graph) | ✗ | ✓ | ✓ | ✓ | ✓ | ✗ | ✗ | ✗ | ✗ | | (graph) |
| 7 | 7 | (graph) | ✗ | ✓ | ✓ | ✓ | ✓ | ✗ | ✗ | ✗ | ✗ | | (graph) |
| 7 | 7 | (graph) | ✗ | ✓ | ✓ | ✓ | ✓ | ✗ | ✓ | ✓ | ✓ | ✓ | (graph) |
| 7 | 7 | (graph) | ✗ | ✓ | ✓ | ✓ | ✓ | ✗ | ✗ | ✗ | ✗ | | (graph) |
| 7 | 7 | (graph) | ✗ | ✓ | ✓ | ✓ | ✓ | ✗ | ✗ | ✗ | ✗ | | (graph) |
| 7 | 7 | (graph) | ✗ | ✓ | ✓ | ✓ | ✓ | ✗ | ✗ | ✗ | ✗ | | (graph) |
| 7 | 8 | (graph) | ✗ | ✗ | ✓ | ✓ | ✓ | ✗ | ✗ | ✗ | ✗ | | (graph) |
| 7 | 8 | (graph) | ✗ | ✓ | ✓ | ✓ | ✓ | ✗ | ✗ | ✗ | ✗ | | (graph) |
| 7 | 8 | (graph) | ✗ | ✓ | ✓ | ✓ | ✓ | ✗ | ✗ | ✗ | ✗ | | (graph) |
| 7 | 8 | (graph) | ✗ | ✓ | ✓ | ✓ | ✓ | ✗ | ✗ | ✗ | ✗ | | (graph) |
| 7 | 8 | (graph) | ✗ | ✓ | ✓ | ✓ | ✓ | ✗ | ✗ | ✗ | ✗ | | (graph) |
| 7 | 8 | (graph) | ✗ | ✓ | ✓ | ✓ | ✓ | ✗ | ✗ | ✗ | ✗ | | (graph) |
| 7 | 8 | (graph) | ✗ | ✓ | ✓ | ✓ | ✓ | ✗ | ✗ | ✗ | ✗ | | (graph) |
| 7 | 8 | (graph) | ✗ | ✓ | ✓ | ✓ | ✓ | ✗ | ✓ | ✓ | ✓ | ✓ | (graph) |
| 7 | 8 | (graph) | ✗ | ✓ | ✓ | ✓ | ✓ | ✗ | ✗ | ✗ | ✗ | | (graph) |
| 7 | 8 | (graph) | ✗ | ✓ | ✓ | ✓ | ✓ | ✗ | ✗ | ✗ | ✗ | | (graph) |
| 7 | 8 | (graph) | ✗ | ✓ | ✓ | ✓ | ✓ | ✗ | ✗ | ✗ | ✗ | | (graph) |
| 7 | 8 | (graph) | ✗ | ✗ | ✓ | ✓ | ✓ | ✗ | ✗ | ✗ | ✗ | | (graph) |
| 7 | 8 | (graph) | ✗ | ✓ | ✓ | ✓ | ✓ | ✗ | ✗ | ✗ | ✗ | | (graph) |
| 7 | 8 | (graph) | ✗ | ✓ | ✓ | ✓ | ✓ | ✗ | ✗ | ✗ | ✗ | | (graph) |
| 7 | 8 | (graph) | ✗ | ✓ | ✓ | ✓ | ✓ | ✗ | ✗ | ✗ | ✗ | | (graph) |
| 7 | 8 | (graph) | ✗ | ✓ | ✓ | ✓ | ✓ | ✗ | ✓ | ✓ | ✓ | ✓ | (graph) |
| 7 | 8 | (graph) | ✗ | ✓ | ✓ | ✓ | ✓ | ✗ | ✗ | ✗ | ✗ | | (graph) |
| 7 | 8 | (graph) | ✗ | ✓ | ✓ | ✓ | ✓ | ✗ | ✗ | ✗ | ✗ | | (graph) |
| 7 | 8 | (graph) | ✗ | ✓ | ✓ | ✓ | ✓ | ✗ | ✗ | ✗ | ✗ | | (graph) |

(see the next page)

| n m | F | Homomorphism MP Sub L LF F | Subgraph MP Sub L LF F | F̃ | n m | F | Homomorphism MP Sub L LF F | Subgraph MP Sub L LF F | F̃ |
|---|---|---|---|---|---|---|---|---|---|
| 7 8 | (graph) | ✗ ✓ ✓ ✓ ✓ | ✗ ✓ ✓ ✓ ✓ | (graph) | 7 8 | (graph) | ✗ ✓ ✓ ✓ ✓ | ✗ ✗ ✗ ✗ | (graph) |
| 7 8 | (graph) | ✗ ✓ ✓ ✓ ✓ | ✗ ✗ ✗ ✗ | (graph) | 7 8 | (graph) | ✗ ✓ ✓ ✓ ✓ | ✗ ✗ ✗ ✗ | (graph) |
| 7 8 | (graph) | ✗ ✓ ✓ ✓ ✓ | ✗ ✗ ✗ ✗ | (graph) | 7 8 | (graph) | ✗ ✓ ✓ ✓ ✓ | ✗ ✗ ✗ ✗ | (graph) |
| 7 8 | (graph) | ✗ ✓ ✓ ✓ ✓ | ✗ ✗ ✗ ✗ | (graph) | 7 8 | (graph) | ✗ ✓ ✓ ✓ ✓ | ✗ ✗ ✗ ✗ | (graph) |
| 7 8 | (graph) | ✗ ✓ ✓ ✓ ✓ | ✗ ✗ ✗ ✗ | (graph) | 7 8 | (graph) | ✗ ✓ ✓ ✓ ✓ | ✗ ✗ ✗ ✗ | (graph) |
| 7 8 | (graph) | ✗ ✓ ✓ ✓ ✓ | ✗ ✗ ✗ ✗ | (graph) | 7 8 | (graph) | ✗ ✓ ✓ ✓ ✓ | ✗ ✗ ✗ ✗ | (graph) |
| 7 8 | (graph) | ✗ ✓ ✓ ✓ ✓ | ✗ ✗ ✗ ✗ | (graph) | 7 8 | (graph) | ✗ ✓ ✓ ✓ ✓ | ✗ ✓ ✓ ✓ ✓ | (graph) |
| 7 8 | (graph) | ✗ ✓ ✓ ✓ ✓ | ✗ ✗ ✗ ✗ | (graph) | 7 8 | (graph) | ✗ ✓ ✓ ✓ ✓ | ✗ ✗ ✗ ✗ | (graph) |
| 7 8 | (graph) | ✗ ✓ ✓ ✓ ✓ | ✗ ✗ ✗ ✗ | (graph) | 7 8 | (graph) | ✗ ✓ ✓ ✓ ✓ | ✗ ✗ ✗ ✗ | (graph) |
| 7 8 | (graph) | ✗ ✓ ✓ ✓ ✓ | ✗ ✗ ✗ ✗ | (graph) | 7 8 | (graph) | ✗ ✓ ✓ ✓ ✓ | ✗ ✗ ✗ ✗ | (graph) |
| 7 8 | (graph) | ✗ ✓ ✓ ✓ ✓ | ✗ ✗ ✗ ✗ | (graph) | 7 8 | (graph) | ✗ ✓ ✓ ✓ ✓ | ✗ ✗ ✗ ✗ | (graph) |
| 7 8 | (graph) | ✗ ✓ ✓ ✓ ✓ | ✗ ✗ ✗ ✗ | (graph) | 7 8 | (graph) | ✗ ✓ ✓ ✓ ✓ | ✗ ✗ ✗ ✗ | (graph) |
| 7 8 | (graph) | ✗ ✓ ✓ ✓ ✓ | ✗ ✗ ✗ ✗ | (graph) | 7 8 | (graph) | ✗ ✓ ✓ ✓ ✓ | ✗ ✗ ✗ ✗ | (graph) |
| 7 8 | (graph) | ✗ ✓ ✓ ✓ ✓ | ✗ ✗ ✗ ✗ | (graph) | 7 8 | (graph) | ✗ ✓ ✓ ✓ ✓ | ✗ ✗ ✗ ✗ | (graph) |
| 7 8 | (graph) | ✗ ✓ ✓ ✓ ✓ | ✗ ✗ ✗ ✗ | (graph) | 7 8 | (graph) | ✗ ✓ ✓ ✓ ✓ | ✗ ✗ ✗ ✗ | (graph) |
| 7 8 | (graph) | ✗ ✓ ✓ ✓ ✓ | ✗ ✗ ✗ ✗ | (graph) | 7 8 | (graph) | ✗ ✓ ✓ ✓ ✓ | ✗ ✗ ✗ ✗ | (graph) |
| 7 8 | (graph) | ✗ ✗ ✓ ✓ ✓ | ✗ ✗ ✗ ✗ | (graph) | 7 8 | (graph) | ✗ ✓ ✓ ✓ ✓ | ✗ ✗ ✗ ✗ | (graph) |
| 7 8 | (graph) | ✗ ✓ ✓ ✓ ✓ | ✗ ✗ ✗ ✗ | (graph) | 7 8 | (graph) | ✗ ✓ ✓ ✓ ✓ | ✗ ✗ ✗ ✗ | (graph) |
| 7 8 | (graph) | ✗ ✓ ✓ ✓ ✓ | ✗ ✗ ✗ ✗ | (graph) | 7 8 | (graph) | ✗ ✓ ✓ ✓ ✓ | ✗ ✓ ✓ ✓ ✓ | (graph) |
| 7 8 | (graph) | ✗ ✓ ✓ ✓ ✓ | ✗ ✗ ✗ ✗ | (graph) | 7 8 | (graph) | ✗ ✗ ✓ ✓ ✓ | ✗ ✗ ✗ ✗ | (graph) |
| 7 8 | (graph) | ✗ ✓ ✓ ✓ ✓ | ✗ ✗ ✗ ✗ | (graph) | 7 8 | (graph) | ✗ ✓ ✓ ✓ ✓ | ✗ ✗ ✗ ✗ | (graph) |
| 7 8 | (graph) | ✗ ✓ ✓ ✓ ✓ | ✗ ✗ ✗ ✗ | (graph) | 8 7 | (graph) | ✓ ✓ ✓ ✓ ✓ | ✗ ✓ ✓ ✓ ✓ | (graph) |
| 7 8 | (graph) | ✗ ✓ ✓ ✓ ✓ | ✗ ✗ ✗ ✗ | (graph) | 8 7 | (graph) | ✓ ✓ ✓ ✓ ✓ | ✗ ✗ ✗ ✗ | (graph) |
| 7 8 | (graph) | ✗ ✓ ✓ ✓ ✓ | ✗ ✗ ✗ ✗ | (graph) | 8 7 | (graph) | ✓ ✓ ✓ ✓ ✓ | ✗ ✗ ✗ ✗ | (graph) |
| 7 8 | (graph) | ✗ ✓ ✓ ✓ ✓ | ✗ ✗ ✗ ✗ | (graph) | 8 7 | (graph) | ✓ ✓ ✓ ✓ ✓ | ✗ ✗ ✗ ✗ | (graph) |
| 7 8 | (graph) | ✗ ✓ ✓ ✓ ✓ | ✗ ✗ ✗ ✗ | (graph) | 8 7 | (graph) | ✓ ✓ ✓ ✓ ✓ | ✗ ✗ ✗ ✗ | (graph) |

(see the next page)

| n m | F | Homomorphism MP Sub L LF F | Subgraph MP Sub L LF F | F̃ | n m | F | Homomorphism MP Sub L LF F | Subgraph MP Sub L LF F | F̃ |
|---|---|---|---|---|---|---|---|---|---|
| 8 7 | (graph) | ✓ ✓ ✓ ✓ ✓ | ✗ ✓ ✓ ✓ ✓ | (graph) | 8 8 | (graph) | ✗ ✓ ✓ ✓ ✓ | ✗ ✗ ✗ ✗ ✗ | (graph) |
| 8 7 | (graph) | ✓ ✓ ✓ ✓ ✓ | ✗ ✓ ✓ ✓ ✓ | (graph) | 8 8 | (graph) | ✗ ✓ ✓ ✓ ✓ | ✗ ✗ ✗ ✗ ✗ | (graph) |
| 8 7 | (graph) | ✓ ✓ ✓ ✓ ✓ | ✗ ✗ ✗ ✗ ✗ | (graph) | 8 8 | (graph) | ✗ ✓ ✓ ✓ ✓ | ✗ ✗ ✗ ✗ ✗ | (graph) |
| 8 7 | (graph) | ✓ ✓ ✓ ✓ ✓ | ✗ ✗ ✗ ✗ ✗ | (graph) | 8 8 | (graph) | ✗ ✓ ✓ ✓ ✓ | ✗ ✗ ✗ ✗ ✗ | (graph) |
| 8 7 | (graph) | ✓ ✓ ✓ ✓ ✓ | ✗ ✗ ✗ ✗ ✗ | (graph) | 8 8 | (graph) | ✗ ✓ ✓ ✓ ✓ | ✗ ✗ ✗ ✗ ✗ | (graph) |
| 8 7 | (graph) | ✓ ✓ ✓ ✓ ✓ | ✗ ✗ ✗ ✗ ✗ | (graph) | 8 8 | (graph) | ✗ ✓ ✓ ✓ ✓ | ✗ ✗ ✗ ✗ ✗ | (graph) |
| 8 7 | (graph) | ✓ ✓ ✓ ✓ ✓ | ✓ ✓ ✓ ✓ ✓ | - | 8 8 | (graph) | ✗ ✓ ✓ ✓ ✓ | ✗ ✗ ✗ ✗ ✗ | (graph) |
| 8 7 | (graph) | ✓ ✓ ✓ ✓ ✓ | ✗ ✗ ✗ ✗ ✗ | (graph) | 8 8 | (graph) | ✗ ✓ ✓ ✓ ✓ | ✗ ✗ ✗ ✗ ✗ | (graph) |
| 8 7 | (graph) | ✓ ✓ ✓ ✓ ✓ | ✗ ✗ ✗ ✗ ✗ | (graph) | 8 8 | (graph) | ✗ ✓ ✓ ✓ ✓ | ✗ ✗ ✗ ✗ ✗ | (graph) |
| 8 7 | (graph) | ✓ ✓ ✓ ✓ ✓ | ✗ ✓ ✓ ✓ ✓ | (graph) | 8 8 | (graph) | ✗ ✓ ✓ ✓ ✓ | ✗ ✓ ✓ ✓ ✓ | (graph) |
| 8 7 | (graph) | ✓ ✓ ✓ ✓ ✓ | ✗ ✓ ✓ ✓ ✓ | (graph) | 8 8 | (graph) | ✗ ✓ ✓ ✓ ✓ | ✗ ✗ ✗ ✗ ✗ | (graph) |
| 8 7 | (graph) | ✓ ✓ ✓ ✓ ✓ | ✗ ✓ ✓ ✓ ✓ | (graph) | 8 8 | (graph) | ✗ ✓ ✓ ✓ ✓ | ✗ ✗ ✗ ✗ ✗ | (graph) |
| 8 7 | (graph) | ✓ ✓ ✓ ✓ ✓ | ✗ ✗ ✗ ✗ ✗ | (graph) | 8 8 | (graph) | ✗ ✓ ✓ ✓ ✓ | ✗ ✗ ✗ ✗ ✗ | (graph) |
| 8 7 | (graph) | ✓ ✓ ✓ ✓ ✓ | ✗ ✗ ✗ ✗ ✗ | (graph) | 8 8 | (graph) | ✗ ✓ ✓ ✓ ✓ | ✗ ✗ ✗ ✗ ✗ | (graph) |
| 8 7 | (graph) | ✓ ✓ ✓ ✓ ✓ | ✗ ✗ ✗ ✗ ✗ | (graph) | 8 8 | (graph) | ✗ ✓ ✓ ✓ ✓ | ✗ ✗ ✗ ✗ ✗ | (graph) |
| 8 7 | (graph) | ✓ ✓ ✓ ✓ ✓ | ✗ ✗ ✗ ✗ ✗ | (graph) | 8 8 | (graph) | ✗ ✓ ✓ ✓ ✓ | ✗ ✗ ✗ ✗ ✗ | (graph) |
| 8 7 | (graph) | ✓ ✓ ✓ ✓ ✓ | ✗ ✗ ✗ ✗ ✗ | (graph) | 8 8 | (graph) | ✗ ✓ ✓ ✓ ✓ | ✗ ✓ ✓ ✓ ✓ | (graph) |
| 8 7 | (graph) | ✓ ✓ ✓ ✓ ✓ | ✗ ✗ ✗ ✗ ✗ | (graph) | 8 8 | (graph) | ✗ ✓ ✓ ✓ ✓ | ✗ ✓ ✓ ✓ ✓ | (graph) |
| 8 8 | (graph) | ✗ ✓ ✓ ✓ ✓ | ✗ ✗ ✗ ✗ ✗ | (graph) | 8 8 | (graph) | ✗ ✓ ✓ ✓ ✓ | ✗ ✗ ✗ ✗ ✗ | (graph) |
| 8 8 | (graph) | ✗ ✓ ✓ ✓ ✓ | ✗ ✗ ✗ ✗ ✗ | (graph) | 8 8 | (graph) | ✗ ✓ ✓ ✓ ✓ | ✗ ✗ ✗ ✗ ✗ | (graph) |
| 8 8 | (graph) | ✗ ✓ ✓ ✓ ✓ | ✗ ✗ ✗ ✗ ✗ | (graph) | 8 8 | (graph) | ✗ ✓ ✓ ✓ ✓ | ✗ ✗ ✗ ✗ ✗ | (graph) |
| 8 8 | (graph) | ✗ ✓ ✓ ✓ ✓ | ✗ ✗ ✗ ✗ ✗ | (graph) | 8 8 | (graph) | ✗ ✓ ✓ ✓ ✓ | ✗ ✗ ✗ ✗ ✗ | (graph) |
| 8 8 | (graph) | ✗ ✓ ✓ ✓ ✓ | ✗ ✗ ✗ ✗ ✗ | (graph) | 8 8 | (graph) | ✗ ✓ ✓ ✓ ✓ | ✗ ✗ ✗ ✗ ✗ | (graph) |
| 8 8 | (graph) | ✗ ✓ ✓ ✓ ✓ | ✗ ✗ ✗ ✗ ✗ | (graph) | 8 8 | (graph) | ✗ ✓ ✓ ✓ ✓ | ✗ ✗ ✗ ✗ ✗ | (graph) |
| 8 8 | (graph) | ✗ ✓ ✓ ✓ ✓ | ✗ ✗ ✗ ✗ ✗ | (graph) | 8 8 | (graph) | ✗ ✓ ✓ ✓ ✓ | ✗ ✗ ✗ ✗ ✗ | (graph) |
| 8 8 | (graph) | ✗ ✓ ✓ ✓ ✓ | ✗ ✗ ✗ ✗ ✗ | (graph) | 8 8 | (graph) | ✗ ✓ ✓ ✓ ✓ | ✗ ✗ ✗ ✗ ✗ | (graph) |

(see the next page)

| n m | F | Homomorphism MP Sub L LF F | | | | | Subgraph MP Sub L LF F | | | | | $\tilde{F}$ | n m | F | Homomorphism MP Sub L LF F | | | | | Subgraph MP Sub L LF F | | | | | $\tilde{F}$ |
|---|---|---|---|---|---|---|---|---|---|---|---|---|---|---|---|---|---|---|---|---|---|---|---|---|---|
| 8 8 | | ✗ | ✓ | ✓ | ✓ | ✓ | ✗ | ✗ | ✗ | ✗ | ✗ | | 8 8 | | ✗ | ✓ | ✓ | ✓ | ✓ | ✗ | ✗ | ✗ | ✗ | ✗ | |
| 8 8 | | ✗ | ✓ | ✓ | ✓ | ✓ | ✗ | ✓ | ✓ | ✓ | ✓ | | 8 8 | | ✗ | ✓ | ✓ | ✓ | ✓ | ✗ | ✗ | ✗ | ✗ | ✗ | |
| 8 8 | | ✗ | ✓ | ✓ | ✓ | ✓ | ✗ | ✗ | ✗ | ✗ | ✗ | | 8 8 | | ✗ | ✓ | ✓ | ✓ | ✓ | ✗ | ✓ | ✓ | ✓ | ✓ | |
| 8 8 | | ✗ | ✓ | ✓ | ✓ | ✓ | ✗ | ✗ | ✗ | ✗ | ✗ | | 8 8 | | ✗ | ✓ | ✓ | ✓ | ✓ | ✗ | ✗ | ✗ | ✗ | ✗ | |
| 8 8 | | ✗ | ✓ | ✓ | ✓ | ✓ | ✗ | ✗ | ✗ | ✗ | ✗ | | 8 8 | | ✗ | ✓ | ✓ | ✓ | ✓ | ✗ | ✗ | ✗ | ✗ | ✗ | |
| 8 8 | | ✗ | ✓ | ✓ | ✓ | ✓ | ✗ | ✗ | ✗ | ✗ | ✗ | | 8 8 | | ✗ | ✓ | ✓ | ✓ | ✓ | ✗ | ✗ | ✗ | ✗ | ✗ | |
| 8 8 | | ✗ | ✓ | ✓ | ✓ | ✓ | ✗ | ✗ | ✗ | ✗ | ✗ | | 8 8 | | ✗ | ✓ | ✓ | ✓ | ✓ | ✗ | ✗ | ✗ | ✗ | ✗ | |
| 8 8 | | ✗ | ✓ | ✓ | ✓ | ✓ | ✗ | ✗ | ✗ | ✗ | ✗ | | 8 8 | | ✗ | ✓ | ✓ | ✓ | ✓ | ✗ | ✗ | ✗ | ✗ | ✗ | |
| 8 8 | | ✗ | ✓ | ✓ | ✓ | ✓ | ✗ | ✗ | ✗ | ✗ | ✗ | | 8 8 | | ✗ | ✓ | ✓ | ✓ | ✓ | ✗ | ✗ | ✗ | ✗ | ✗ | |
| 8 8 | | ✗ | ✓ | ✓ | ✓ | ✓ | ✗ | ✗ | ✗ | ✗ | ✗ | | 8 8 | | ✗ | ✓ | ✓ | ✓ | ✓ | ✗ | ✗ | ✗ | ✗ | ✗ | |
| 8 8 | | ✗ | ✓ | ✓ | ✓ | ✓ | ✗ | ✗ | ✗ | ✗ | ✗ | | 8 8 | | ✗ | ✓ | ✓ | ✓ | ✓ | ✗ | ✗ | ✗ | ✗ | ✗ | |
| 8 8 | | ✗ | ✓ | ✓ | ✓ | ✓ | ✗ | ✗ | ✗ | ✗ | ✗ | | 8 8 | | ✗ | ✓ | ✓ | ✓ | ✓ | ✗ | ✗ | ✗ | ✗ | ✗ | |
| 8 8 | | ✗ | ✓ | ✓ | ✓ | ✓ | ✗ | ✗ | ✗ | ✗ | ✗ | | 8 8 | | ✗ | ✓ | ✓ | ✓ | ✓ | ✗ | ✓ | ✓ | ✓ | ✓ | |
| 8 8 | | ✗ | ✓ | ✓ | ✓ | ✓ | ✗ | ✗ | ✗ | ✗ | ✗ | | 8 8 | | ✗ | ✓ | ✓ | ✓ | ✓ | ✗ | ✗ | ✗ | ✗ | ✗ | |
| 8 8 | | ✗ | ✓ | ✓ | ✓ | ✓ | ✗ | ✗ | ✗ | ✗ | ✗ | | 8 8 | | ✗ | ✓ | ✓ | ✓ | ✓ | ✗ | ✗ | ✗ | ✗ | ✗ | |
| 8 8 | | ✗ | ✓ | ✓ | ✓ | ✓ | ✗ | ✗ | ✗ | ✗ | ✗ | | 8 8 | | ✗ | ✓ | ✓ | ✓ | ✓ | ✗ | ✗ | ✗ | ✗ | ✗ | |
| 8 8 | | ✗ | ✓ | ✓ | ✓ | ✓ | ✗ | ✗ | ✗ | ✗ | ✗ | | 8 8 | | ✗ | ✓ | ✓ | ✓ | ✓ | ✗ | ✗ | ✗ | ✗ | ✗ | |
| 8 8 | | ✗ | ✓ | ✓ | ✓ | ✓ | ✗ | ✗ | ✗ | ✗ | ✗ | | 8 8 | | ✗ | ✓ | ✓ | ✓ | ✓ | ✗ | ✓ | ✓ | ✓ | ✓ | |
| 8 8 | | ✗ | ✓ | ✓ | ✓ | ✓ | ✗ | ✗ | ✗ | ✗ | ✗ | | 8 8 | | ✗ | ✓ | ✓ | ✓ | ✓ | ✗ | ✗ | ✗ | ✗ | ✗ | |
| 8 8 | | ✗ | ✓ | ✓ | ✓ | ✓ | ✗ | ✗ | ✗ | ✗ | ✗ | | 8 8 | | ✗ | ✓ | ✓ | ✓ | ✓ | ✗ | ✗ | ✗ | ✗ | ✗ | |
| 8 8 | | ✗ | ✓ | ✓ | ✓ | ✓ | ✗ | ✗ | ✗ | ✗ | ✗ | | 8 8 | | ✗ | ✓ | ✓ | ✓ | ✓ | ✗ | ✗ | ✗ | ✗ | ✗ | |
| 8 8 | | ✗ | ✓ | ✓ | ✓ | ✓ | ✗ | ✗ | ✗ | ✗ | ✗ | | 8 8 | | ✗ | ✓ | ✓ | ✓ | ✓ | ✗ | ✓ | ✓ | ✓ | ✓ | |
| 8 8 | | ✗ | ✓ | ✓ | ✓ | ✓ | ✗ | ✗ | ✗ | ✗ | ✗ | | 8 8 | | ✗ | ✓ | ✓ | ✓ | ✓ | ✗ | ✗ | ✗ | ✗ | ✗ | |
| 8 8 | | ✗ | ✓ | ✓ | ✓ | ✓ | ✗ | ✗ | ✗ | ✗ | ✗ | | 8 8 | | ✗ | ✓ | ✓ | ✓ | ✓ | ✗ | ✗ | ✗ | ✗ | ✗ | |
| 8 8 | | ✗ | ✓ | ✓ | ✓ | ✓ | ✗ | ✗ | ✗ | ✗ | ✗ | | 8 8 | | ✗ | ✓ | ✓ | ✓ | ✓ | ✗ | ✗ | ✗ | ✗ | ✗ | |

| n m | F | Homomorphism MP Sub L LF F | Subgraph MP Sub L LF F | F̃ | n m | F | Homomorphism MP Sub L LF F | Subgraph MP Sub L LF F | F̃ |
|---|---|---|---|---|---|---|---|---|---|
| 8 8 | (graph) | ✗ ✓ ✓ ✓ ✓ | ✗ ✗ ✗ ✗ | (graph) | 9 8 | (graph) | ✓ ✓ ✓ ✓ ✓ | ✗ ✗ ✗ ✗ | (graph) |
| 8 8 | (graph) | ✗ ✓ ✓ ✓ ✓ | ✗ ✗ ✗ ✗ | (graph) | 9 8 | (graph) | ✓ ✓ ✓ ✓ ✓ | ✗ ✗ ✗ ✗ | (graph) |
| 8 8 | (graph) | ✗ ✓ ✓ ✓ ✓ | ✗ ✗ ✗ ✗ | (graph) | 9 8 | (graph) | ✓ ✓ ✓ ✓ ✓ | ✗ ✗ ✗ ✗ | (graph) |
| 9 8 | (graph) | ✓ ✓ ✓ ✓ ✓ | ✗ ✗ ✗ ✗ | (graph) | 9 8 | (graph) | ✓ ✓ ✓ ✓ ✓ | ✗ ✗ ✗ ✗ | (graph) |
| 9 8 | (graph) | ✓ ✓ ✓ ✓ ✓ | ✗ ✓ ✓ ✓ ✓ | (graph) | 9 8 | (graph) | ✓ ✓ ✓ ✓ ✓ | ✗ ✓ ✓ ✓ ✓ | (graph) |
| 9 8 | (graph) | ✓ ✓ ✓ ✓ ✓ | ✗ ✗ ✗ ✗ | (graph) | 9 8 | (graph) | ✓ ✓ ✓ ✓ ✓ | ✗ ✗ ✗ ✗ | (graph) |
| 9 8 | (graph) | ✓ ✓ ✓ ✓ ✓ | ✗ ✗ ✗ ✗ | (graph) | 9 8 | (graph) | ✓ ✓ ✓ ✓ ✓ | ✗ ✗ ✗ ✗ | (graph) |
| 9 8 | (graph) | ✓ ✓ ✓ ✓ ✓ | ✗ ✗ ✗ ✗ | (graph) | 9 8 | (graph) | ✓ ✓ ✓ ✓ ✓ | ✗ ✗ ✗ ✗ | (graph) |
| 9 8 | (graph) | ✓ ✓ ✓ ✓ ✓ | ✗ ✗ ✗ ✗ | (graph) | 9 8 | (graph) | ✓ ✓ ✓ ✓ ✓ | ✗ ✗ ✗ ✗ | (graph) |
| 9 8 | (graph) | ✓ ✓ ✓ ✓ ✓ | ✗ ✗ ✗ ✗ | (graph) | 9 8 | (graph) | ✓ ✓ ✓ ✓ ✓ | ✗ ✗ ✗ ✗ | (graph) |
| 9 8 | (graph) | ✓ ✓ ✓ ✓ ✓ | ✗ ✓ ✓ ✓ ✓ | (graph) | 9 8 | (graph) | ✓ ✓ ✓ ✓ ✓ | ✗ ✗ ✗ ✗ | (graph) |
| 9 8 | (graph) | ✓ ✓ ✓ ✓ ✓ | ✓ ✓ ✓ ✓ ✓ | - | 9 8 | (graph) | ✓ ✓ ✓ ✓ ✓ | ✗ ✗ ✗ ✗ | (graph) |
| 9 8 | (graph) | ✓ ✓ ✓ ✓ ✓ | ✗ ✓ ✓ ✓ ✓ | (graph) | 9 8 | (graph) | ✓ ✓ ✓ ✓ ✓ | ✗ ✗ ✗ ✗ | (graph) |
| 9 8 | (graph) | ✓ ✓ ✓ ✓ ✓ | ✗ ✗ ✗ ✗ | (graph) | 9 8 | (graph) | ✓ ✓ ✓ ✓ ✓ | ✗ ✗ ✗ ✗ | (graph) |
| 9 8 | (graph) | ✓ ✓ ✓ ✓ ✓ | ✗ ✓ ✓ ✓ ✓ | (graph) | 9 8 | (graph) | ✓ ✓ ✓ ✓ ✓ | ✗ ✓ ✓ ✓ ✓ | (graph) |
| 9 8 | (graph) | ✓ ✓ ✓ ✓ ✓ | ✗ ✗ ✗ ✗ | (graph) | 9 8 | (graph) | ✓ ✓ ✓ ✓ ✓ | ✗ ✗ ✗ ✗ | (graph) |
| 9 8 | (graph) | ✓ ✓ ✓ ✓ ✓ | ✗ ✗ ✗ ✗ | (graph) | 9 8 | (graph) | ✓ ✓ ✓ ✓ ✓ | ✗ ✗ ✗ ✗ | (graph) |
| 9 8 | (graph) | ✓ ✓ ✓ ✓ ✓ | ✗ ✗ ✗ ✗ | (graph) | 9 8 | (graph) | ✓ ✓ ✓ ✓ ✓ | ✗ ✓ ✓ ✓ ✓ | (graph) |
| 9 8 | (graph) | ✓ ✓ ✓ ✓ ✓ | ✗ ✗ ✗ ✗ | (graph) | | | | | |
| 9 8 | (graph) | ✓ ✓ ✓ ✓ ✓ | ✗ ✗ ✗ ✗ | (graph) | | | | | |
| 9 8 | (graph) | ✓ ✓ ✓ ✓ ✓ | ✗ ✗ ✗ ✗ | (graph) | | | | | |
| 9 8 | (graph) | ✓ ✓ ✓ ✓ ✓ | ✗ ✗ ✗ ✗ | (graph) | | | | | |
| 9 8 | (graph) | ✓ ✓ ✓ ✓ ✓ | ✗ ✗ ✗ ✗ | (graph) | | | | | |
| 9 8 | (graph) | ✓ ✓ ✓ ✓ ✓ | ✗ ✗ ✗ ✗ | (graph) | | | | | |
| 9 8 | (graph) | ✓ ✓ ✓ ✓ ✓ | ✗ ✗ ✗ ✗ | (graph) | | | | | |
| 9 8 | (graph) | ✓ ✓ ✓ ✓ ✓ | ✗ ✗ ✗ ✗ | (graph) | | | | | |

## H  POLYNOMIAL EXPRESSIVITY

Puny et al. (2023) proposed the equivariant graph polynomials, which are polynomials $P$ that take real squared matrices $\boldsymbol{X} \in \mathbb{R}^{n \times n}$ as input and outputs $P(\boldsymbol{X}) \in \mathbb{R}^{n \times n}$, such that $P$ is equivariant under permutations, i.e, $P(\pi \cdot \boldsymbol{X}) = \pi \cdot P(\boldsymbol{X})$ for all permutation $\pi \in S_n$. The authors gave a concrete basis of equivariant polynomials, where each basis element $P_{F^{uv}}$ corresponds to a rooted multi-graph $F^{uv}$ marking two vertices $u, v$. Puny et al. (2023) showed that when $\boldsymbol{X}$ is restricted to be the adjacency matrix of an undirected simple graph $G$, each $F^{uv}$ will also reduce to an undirected simple graph, and $P_{F^{uv}}(\boldsymbol{X})$ precisely computes the (unlabeled) homomorphism count $\mathrm{hom}(F^{uv}, G^{wx})$ for all $w, x \in V_G$. Besides the original definition, Puny et al. (2023) also proposed the invariant graph polynomials and node-level equivariant graph polynomials, which are similarly related to the graph-level and node-level homomorphism count.

As a direct consequence, if a GNN model $M$ cannot count graph $F^{uv}/F^{u}/F$ under homomorphism, it then cannot compute the equivariant/invariant graph polynomial $P_{F^{uv}}/P_{F^{u}}/P_F$. Based on these connections, our results can be directly used to provide insights into which equivariant graph polynomials cannot be computed by model $M$. This recovers several results in their paper and answers an open problem shown below.

**Corollary H.1.** *MPNN and PPGN++ are bounded by the Prototypical node-based model and Prototypical edge-based model (defined in Puny et al. (2023)) for computing node-level and edge-level equivariant graph polynomials, respectively.*

*Proof.* Without loss of generality, we assume that the corresponding graphs of all equivariant polynomials are connected. According to Puny et al. (2023, Proposition H.2), the Prototypical node-based model can compute all $P_{F^{u}}$ where $F^{u}$ is a (rooted) tree and cannot compute other graph polynomials. If MPNN is not bounded by the Prototypical node-based model, then it can compute some $P_{\tilde{F}^{u}}$ where $\tilde{F}$ is not a tree. However, this is impossible since MPNN can only count forests under homomorphism according to Theorem 3.7. Note that the MPNN defined in their paper is equivalent to our definition when only considering *connected* graphs (the extra global aggregation $\mathbf{1}\mathbf{1}^{\top}\boldsymbol{X}$ in their definition (9) does not improve the homomorphism expressivity).

We next turn to PPGN++, and the proof is similar (but more involved). We first show that the Prototypical edge-based model can compute any $P_{F^{uv}}$ satisfying that the treewidth of the graph $\tilde{F} := (V_F, E_F \cup \{\{u, v\}\}, \ell_F)$ is no more than 2. If $\mathrm{tw}(\tilde{F}) \leq 2$, it is a partial 2-tree. Thus, there is an ordering $w_1, \cdots, w_n$ of the vertex set $V_F$ such that when deleting each vertex $w_i$ and all incident edges in turn, we only ever delete vertices of degree at most 2. Now we claim that we can always order the two vertices $u, v$ at the end, i.e., $w_{n-1} = u$ and $w_n = v$. Otherwise, there is a subset $U \subset V_F$ such that all vertices in the induced subgraph $\tilde{F}[U]$ are of degree at least 3 except $u, v$. It follows that the $\mathrm{tw}(\tilde{F}[U]) \geq \mathrm{tw}(H) \geq 3$ where graph $H$ is the graph obtained from $\tilde{F}[U]$ by contracting $u$ (or $v$) if $\deg_{\tilde{F}[U]}(u) \leq 2$ (or $\deg_{\tilde{F}[U]}(v) \leq 2$). This yields a contradiction and verifies the claim that we can always set $w_{n-1} = u$ and $w_n = v$.

Now, following the proof in Puny et al. (2023, Proposition H.3), the Prototypical edge-based model can contract $\tilde{F}^{uv}$ to a graph with only two vertices $u, v$ and thus can compute the edge-level polynomial $P_{\tilde{F}^{uv}}$. If PPGN++ is not bounded by the Prototypical edge-based model, then it can compute some $P_{\tilde{F}^{uv}}$ where $\mathrm{tw}(\tilde{F}) \geq 3$. Therefore, it can count the graph $\tilde{F}$ under homomorphism *at graph-level* (since it can already count $\tilde{F}$ at edge-level). This implies that PPGN++ is strictly more expressive than 2-FGNN (2-FWL) because we have proved that all graphs in $\mathcal{F}^{\mathsf{F}}$ have a treewidth no more than 2. This yields a contradiction since PPGN++ is still bounded by 2-FWL in distinguishing non-isomorphic graphs. □

We also provide insights into the following results in their paper:

**Corollary H.2.** *The Prototypical node-based model is not 3-node-polynomial-exact. The Prototypical edge-based model is not 6-node-polynomial-exact and not 5-edge-polynomial-exact.*

This is simply because the triangle is not a tree and the 4-clique does not have a NED (or equivalently, the treewidth of a 4-clique is 3). It is also clear why the degree of the edge-based polynomial is 6, which is one less than that of the node-based polynomial using the concept of NED.

## I EXPERIMENTAL DETAILS

In this section, we provide all the experimental details in Section 5.

### I.1 DATASETS

We conduct experiments on tive tasks: (i) graph homomorphism counting, (ii) subgraph counting, (iii) ZINC-subset (Dwivedi et al., 2020), (iii) ZINC-full (Dwivedi et al., 2020), and (iv) Alchemy (Chen et al., 2019a).

**Homomorphism/Subgraph Counting**. For both homomorphism and subgraph counting tasks, we use the standard synthetic graph dataset constructed in Zhao et al. (2022a) (which has been used in a number of papers, see e.g., Frasca et al., 2022; Huang et al., 2023; Zhang et al., 2023a). For homomorphism counting, we count the number of graph/node/edge-level homomorphisms for each pattern in Table 1 and normalize the value by the mean and variance across all graphs in the dataset. The evaluation metric of graph-level expressivity is chosen as the Mean Absolute Error (MAE). For node/edge-level expressivity, the error on each graph is defined to be the *sum* of absolute error over all vertices/edges. We then report the MAE across all graphs in the dataset. This ensures that graph/node/edge-level errors are roughly at the same scale (since $\mathrm{hom}(F, G) = \sum_{w \in V_G} \mathrm{hom}(F^u, G^w) = \sum_{w,x \in V_G} \mathrm{hom}(F^{uv}, G^{wx})$ for all $u, v \in V_F$). For subgraph counting, the data processing and evaluation metric is similar to homomorphism counting, but there is a slight difference in the node/edge-level setting: there are no marked vertices in the pattern graph $F$ (see Table 3). Instead, given a graph $G$ in the dataset and a vertex $w \in V_G$, we count the number of subgraphs containing $w$ that are isomorphic to $F$ and $w$ can be mapped to any vertex in $F$. Due to this difference, in the node/edge-level setting, the error on each graph is defined to be the *average* of absolute error over all vertices/edges.

**ZINC**. ZINC (Dwivedi et al., 2020) is a standard real-world dataset for benchmarking molecular property prediction. The dataset consists of 250K molecular graphs, and the task is to predict the constrained solubility of the given molecule. In addition to the full dataset (denoted as ZINC-full), ZINC-subset is a sampled dataset with 12k molecules from the ZINC-full dataset. We train and test our models on both datasets following the standard protocol from Dwivedi et al. (2020).

**Alchemy**. Alchemy (Chen et al., 2019a) is another real-world dataset with 12 graph-level quantum mechanical properties. We follow the sampling and training protocol from Lim et al. (2023); Puny et al. (2023), using 100K samples for training, 10K samples for testing, and 10K samples for validation.

### I.2 MODEL DETAILS

All models are implemented using the PyTorch (Paszke et al., 2019) framework and the PyTorch Geometric library (Fey & Lenssen, 2019). We consider four types of GNNs defined in Section 2: MPNN, Subgraph GNN, Local 2-GNN, and Local 2-FGNN. For each GNN model, the feature initialization, message-passing layers, and final pooling operation are separately defined below.

**Initialization**. On both ZINC and Alchemy datasets, each graph node is an atom. We maintain a learnable atom embedding for each type of atom and use it to initialize features in GNN models. For MPNN, the initial feature $h^{(0)}(u)$ of node $u$ is simply the atom embedding, denoted as $h^{(0)}(u) = e^{\mathsf{N}}_{\mathrm{atom}(u)}$. For other models, the initial feature $h^{(0)}(u, v)$ of node pair $(u, v)$ consists of two parts: the first part is the node embedding of $v$, and the second part is a distance encoding that embeds the shortest path distance between $u$ and $v$, as adopted in Zhang et al. (2023a). We note that while incorporating distance encoding does not increase the models' theoretical expressive power (see Zhang et al. (2023a)), it may add an inductive bias that can be helpful in real-world tasks. Formally, the initial feature can be written as $h^{(0)}(u, v) = [e^{\mathsf{N}}_{\mathrm{atom}(v)}, e^{\mathsf{D}}_{\mathrm{clip}(\mathrm{dis}(u,v))}]$. Here, we clip the distance to a predefined value `max_dis` so that there are a finite number of distance embeddings, and distances greater than the hyper-parameter `max_dis` (including the disconnected case) share the embedding.

**Propagation.** On both ZINC and Alchemy datasets, each edge in a graph corresponds to a chemical bond and has a bond type. We maintain a learnable edge embedding for each type of edges in each layer and denote the embedding of edge $\{u, v\}$ in layer $l$ as $g^{(l)}(u, v)$. For MPNN, we use the

standard GIN architecture proposed in Xu et al. (2019), which has the following form:

$$h^{(l+1)}(u) = \mathsf{ReLU}(\mathsf{BN}^{(l)}(f^{(l)}(u))), \tag{32}$$

$$f^{(l)}(u) = \mathsf{GIN}^{(l)}\left(h^{(l)}(u), \sum_{v \in N_G(u)} \mathsf{ReLU}\left(\mathsf{FC}^{(l)}(h^{(l)}(v)) + g^{(l)}(u,v)\right)\right), \tag{33}$$

where

$$\mathsf{GIN}^{(l)}(\boldsymbol{x}, \boldsymbol{y}) = \mathsf{MLP}^{(l)}\left((1 + \epsilon^{(l)})\boldsymbol{x} + \boldsymbol{y}\right). \tag{34}$$

Here, $\mathsf{FC}^{(l)}$ is a parameterized linear transformation, $\epsilon^{(l)}$ is a learnable parameter, $\mathsf{BN}^{(l)}$ is the batch normalization (Ioffe & Szegedy, 2015), and $\mathsf{MLP}^{(l)}$ is a two-layer feed-forward network with another batch normalization in the hidden layer.

For other architectures, the $l$-th GNN layer analogously has the following form:

$$h^{(l+1)}(u, v) = \mathsf{ReLU}(\mathsf{BN}^{(l)}(f^{(l)}(u, v))), \tag{35}$$

where the term $f^{(l)}(u, v)$ is defined separately for each model:

- **Subgraph GNN**:

$$f^{(l)}(u, v) = \mathsf{GIN}^{(l)}\left(h^{(l)}(u, v), \sum_{w \in N_G(v)} \mathsf{ReLU}\left(\mathsf{FC}^{(l)}(h^{(l)}(u, w)) + g^{(l)}(w, v)\right)\right). \tag{36}$$

- **Local 2-GNN**:

$$\begin{aligned}
f^{(l)}(u, v) = {} & \mathsf{GIN}^{(l,1)}\left(h^{(l)}(u, v), \sum_{w \in N_G(u)} \mathsf{ReLU}\left(\mathsf{FC}^{(l,1)}(h^{(l)}(w, v)) + g^{(l)}(u, w)\right)\right) \\
& + \mathsf{GIN}^{(l,2)}\left(h^{(l)}(u, v), \sum_{w \in N_G(v)} \mathsf{ReLU}\left(\mathsf{FC}^{(l,2)}(h^{(l)}(u, w)) + g^{(l)}(w, v)\right)\right).
\end{aligned} \tag{37}$$

- **Local 2-FGNN**:

$$\begin{aligned}
& f^{(l)}(u, v) \\
& = \mathsf{GIN}^{(l,1)}\left(h^{(l)}(u, v), \sum_{w \in N_G(u)} \mathsf{ReLU}\left(\mathsf{FC}^{(l,1)}(h^{(l)}(u, w)) + \mathsf{FC}^{(l,2)}(h^{(l)}(w, v)) + g^{(l)}(u, w)\right)\right) \\
& + \mathsf{GIN}^{(l,2)}\left(h^{(l)}(u, v), \sum_{w \in N_G(v)} \mathsf{ReLU}\left(\mathsf{FC}^{(l,1)}(h^{(l)}(w, v)) + \mathsf{FC}^{(l,2)}(h^{(l)}(u, w)) + g^{(l)}(w, v)\right)\right).
\end{aligned} \tag{38}$$

For all the above GNN architectures, it can be seen that each layer only aggregates the *local* neighborhood of vertices or vertex pairs. This design will have shortcomings for disconnected graphs since a vertex cannot aggregate information from other connected components no matter how deep the model is (Barceló et al., 2020). Note that there do exist disconnected graphs in real-world datasets like ZINC. Therefore, on real-world datasets like ZINC and Alchemy, we also incorporate a global aggregation with the following form for each layer (similar to the global aggregation in Frasca et al. (2022); Zhang et al. (2023a)):

$$\mathsf{GIN}^{(l,\mathsf{G})}\left(h^{(l)}(u, v), \sum_{w \in V_G} h^{(l)}(u, w)\right) \tag{39}$$

Note that the global aggregation does not increase model's theoretical expressive power according to Zhang et al. (2023a).

**Pooling.** Except for edge-level tasks, a final pooling layer is used to produce node-level features $h(u)$ for all nodes $u$. It is implemented as follows:

$$h(u) = \mathsf{MLP}\left(\sum_{v \in \mathcal{V}} h^{(L)}(u, v)\right), \tag{40}$$

where MLP is a 2-layer perceptron. For graph-level tasks, we further use a mean pooling layer to aggregate all $h(u)$ and obtain the graph representation.

### I.3 TRAINING DETAILS

All experiments are run on a single NVIDIA Tesla V100 GPU. For all tasks, we use the distance encoding hyper-parameter `max_dis` $= 5$. To enable a fair comparison between models, for each task we keep the same depth for different models while varying the hidden dimension so that the number of model parameters is roughly the same. Note that the dimensions are chosen such that all models roughly obey the 500K parameter budget in ZINC and Alchemy. The hidden dimension size and the number of model parameters are listed as follows. All models are trained using the Adam optimizer.

Table 7: Model size in different tasks.

| Task | Hidden dimension | | | # Parameters | | |
|---|---|---|---|---|---|---|
| | Counting | ZINC | Alchemy | Counting | ZINC | Alchemy |
| MPNN | 128 | 150 | 150 | 314,119 | 510,158 | 509,719 |
| Subgraph GNN | 128 | 120 | 120 | 314,759 | 503,774 | 503,425 |
| Local 2-GNN | 96 | 96 | 96 | 317,388 | 495,188 | 494,911 |
| Local 2-FGNN | 96 | 96 | 96 | 317,388 | 495,188 | 494,911 |

**Homomorphism/Subgraph Counting**. We use a model depth of $L = 5$ in all experiments. Following prior work (Huang et al., 2023), we remove all BN layers in all models. The initial learning rate is chosen as 0.001 and is decayed by a factor of 0.9 once the MAE on the validation set plateaus for 10 epochs. Each model is trained for 1200 epochs with a batch size of 512. We ran each experiment 4 times independently with different seeds and reported the average performance at the last epoch. We found that the standard deviation among different seeds is negligible.

**ZINC**. We use a model depth of $L = 6$ in all experiments for both ZINC-subset and ZINC-full. Following prior work (Zhang et al., 2023a; Frasca et al., 2022), The initial learning rate is chosen as 0.001 and is decayed by a factor of 0.5 once the MAE on the validation set plateaus for 20 epochs. Each model is trained for 400 epochs on ZINC-subset and 500 epochs on ZINC-full, both with a batch size of 128. We report the MAE for the model checkpoint with the best validation performance. We ran each experiment 10 times independently with different seeds and reported the average performance as well as the standard deviation.

**Alchemy**. We use a model depth of $L = 6$ in all experiments. Following prior work (Lim et al., 2023; Puny et al., 2023), The initial learning rate is chosen as 0.002 and is decayed by a factor of 0.5 once the MAE on the validation set plateaus for 20 epochs. Each model is trained for 500 epochs with a batch size of 128. We report the MAE for the model checkpoint with the best validation performance. We ran each experiment 10 times independently with different seeds and reported the average performance as well as the standard deviation.

### I.4 PERFORMANCE OF BASELINE MODELS IN LITERATURE

For completeness, in this subsection we give a comprehensive list of the performance of GNN models in the literature on ZINC and Alchemy datasets. The numbers in each table below are directly taken from the original papers.

Table 8: Performance of different GNN models on ZINC dataset reported in the literature.

| Method | Model | Reference | Test MAE |
|---|---|---|---|
| MPNN | GIN | (Xu et al., 2019) | 0.526±0.051 |
| | GraphSAGE | (Hamilton et al., 2017) | 0.398±0.002 |
| | GAT | (Veličković et al., 2018) | 0.384±0.007 |
| | GCN | (Kipf & Welling, 2017) | 0.367±0.011 |
| | MoNet | (Monti et al., 2017) | 0.292±0.006 |
| | GatedGCN-PE | (Bresson & Laurent, 2017) | 0.214±0.006 |
| | MPNN(sum) | (Gilmer et al., 2017) | 0.145±0.007 |
| | PNA | (Corso et al., 2020) | 0.142±0.010 |
| Higher-order GNN | RingGNN | (Chen et al., 2019b) | 0.353±0.019 |
| | PPGN | (Maron et al., 2019a) | 0.303±0.068 |
| | PPGN | (Puny et al., 2023) | 0.079±0.005 |
| | PPGN++ | (Puny et al., 2023) | 0.076±0.003 |
| Subgraph GNN | NGNN | (Zhang & Li, 2021) | 0.111±0.003 |
| | GNN-AK | (Zhao et al., 2022a) | 0.105±0.010 |
| | GNN-AK+ | (Zhao et al., 2022a) | 0.091±0.002 |
| | ESAN | (Bevilacqua et al., 2022) | 0.102±0.003 |
| | SUN | (Frasca et al., 2022) | 0.083±0.003 |
| | $I^2$-GNN | (Huang et al., 2023) | 0.083±0.001 |
| | ID-MPNN | (Zhou et al., 2023a) | 0.083±0.003 |
| Local (F)GNN | SetGNN | (Zhao et al., 2022b) | 0.075±0.003 |
| | GNN-SSWL | (Zhang et al., 2023a) | 0.082±0.010 |
| | GNN-SSWL+ | (Zhang et al., 2023a) | 0.070±0.005 |
| | $N^2$-GNN | (Feng et al., 2023) | 0.059±0.002 |
| Substructure-based GNN | GSN | (Bouritsas et al., 2022) | 0.101±0.010 |
| | CIN (Small) | (Bodnar et al., 2021a) | 0.094±0.004 |
| | CIN | (Bodnar et al., 2021a) | 0.079±0.006 |
| | CIN++ | (Giusti et al., 2023) | 0.077±0.004 |
| Graph Transformer | SAN | (Kreuzer et al., 2021) | 0.139±0.006 |
| | K-Subgraph SAT | (Chen et al., 2022) | 0.094±0.008 |
| | Graphormer | (Ying et al., 2021) | 0.122±0.006 |
| | URPE | (Luo et al., 2022) | 0.086±0.007 |
| | Graphormer-GD | (Zhang et al., 2023b) | 0.081±0.009 |
| | GPS | (Rampasek et al., 2022) | 0.070±0.004 |
| Other | PF-GNN | (Dupty et al., 2021) | 0.122±0.010 |
| | KP-GIN | (Feng et al., 2022) | 0.093±0.007 |
| | SignNet | (Lim et al., 2023) | 0.084±0.006 |
| | PathNN | (Michel et al., 2023) | 0.090±0.004 |
| | PPGN++(6) | (Puny et al., 2023) | 0.071±0.001 |
| | PlanE | (Dimitrov et al., 2023) | 0.076±0.003 |

Table 9: Performance of different GNN models on Alchemy dataset reported in the literature.

| Model | Reference | Test MAE |
|---|---|---|
| GIN | (Xu et al., 2019) | 0.180±0.006 |
| PF-GNN | (Dupty et al., 2021) | 0.111±0.010 |
| $\delta$-2-GNN | (Morris et al., 2020) | 0.118±0.001 |
| Recon-GNN | (Cotta et al., 2021) | 0.125±0.001 |
| SpeqNet | (Morris et al., 2022) | 0.115±0.001 |
| SignNet | (Lim et al., 2023) | 0.113±0.002 |
| PPGN | (Puny et al., 2023) | 0.113±0.001 |
| PPGN++ | (Puny et al., 2023) | 0.111±0.002 |
| PPGN++(6) | (Puny et al., 2023) | 0.109±0.001 |

