# OpenReview forum: "Beyond Weisfeiler-Lehman: A Quantitative Framework for GNN Expressiveness"
_ICLR.cc/2024/Conference — ICLR 2024 oral_

### Official Review · Reviewer_wf9E · 2023-10-19

**Soundness:** 4 excellent
**Presentation:** 3 good
**Contribution:** 3 good
**Rating:** 8
**Confidence:** 3

**Summary:**

The expressive power of GNNs is commonly measured via their correspondence to Weisfeiler-Lehman (WL) graph isomorphism tests. In efforts to address limitations of the WL framework, the current paper advocates quantifying GNN expressivity through *homomorphism expressivity* — their ability to count graph substructures under homomorphism. The main theoretical contribution is a characterization of homomorphism expressivity for popular classes of GNNs in the context of graph, node, and edge representation. Among others, this allows a more fine-grained comparison between architectures, resolving several open questions. Experiments on synthetic and a couple of real-world tasks corroborate the theory, showing that the performance of GNNs aligns with the proposed expressivity measure.

**Strengths:**

1. Homomorphism expressivity provides a refreshing perspective on the expressivity of GNNs, alleviating some of the shortcomings of the popular WL framework. In particular, equivalence to WL tests can be too coarse of a measure and proving that one architecture is superior to another is often based on individual exemplar graphs. In contrast, for several GNN architectures, the current paper completely characterizes the graph homomorphisms they can count, facilitating a more fine-grained comparison and analyzing their subgraph counting ability.

2. In general, the paper is well-written. The motivation and main results are clearly described, along with discussions on relevant literature and technical details. Though the paper is a bit dense. I would suggest allocating more space for providing examples and intuitions as to which types of graphs are included in the characterizations of Theorem 3.3 that are based on nested ear decompositions (NEDs). In my opinion, this can come at the expense of the extensions in Subsections 3.3 and 3.4, which can be deferred to an appendix.

3. The theoretical analysis establishes elegant connections to concepts from graph theory, e.g. nested ear decompositions, which may prove useful for future work.

I believe the technical contributions pose a promising step forward in formalizing the expressive power of GNNs, and therefore recommend acceptance. Yet, there are a few weaknesses (listed below). Most importantly, a technical issue with the definition of homomorphism expressivity that requires clarification.

**Weaknesses:**

1. There seems to be a technical issue with the definition of homomorphism expressivity (Definition 3.1). I believe it can be solved, but requires addressing or further clarification. Specifically, $\mathcal{F}^M$ is not necessarily defined for all models $M$ since it does not take into account graphs $F$ for which there exist graphs $G$ and $H$ with equal homomorphism counts with respect to $F$ but different representations according to $M$. Homomorphism expressivity of $M$ is defined only if no such $F$ exists since it cannot be within $\mathcal{F}^M$ and also cannot be outside $\mathcal{F}^M$. For example, homomorphism expressivity is not defined for a universal architecture as it can assign different representations to any two non-isomorphic graphs.
    1. Is it the case that for the GNNs considered $\mathcal{F}^M$ is always well-defined? This seems to be indicated by Theorem 3.3 and the subsequent discussion, however, it is not clear enough. I strongly recommend elaborating upon this point near the definition to avoid confusion.

    2. Homomorphism expressivity is referred throughout as a “complete” measure. Given that it is not defined for all architectures $M$, I believe this terminology can be misleading.

2. Homomorphism expressivity suffers from a limitation that also exists in the WL framework: it disregards features of vertices/edges. The proposed expressivity measure takes into account only the ability to capture graph structures, assuming each vertex is associated with a unique discrete label. While identifying graph structure is indeed important, in many cases the features of vertices play an equal, if not more important role (see, e.g., [1]). I do not believe this significantly harms the quality of the current paper, and is perhaps a consideration for future work, but to me referring to homomorphism expressivity as a “complete” measure may require some hedging.

[1] Liu, Renming, et al. "Towards a Taxonomy of Graph Learning Datasets." arXiv preprint arXiv:2110.14809 (2021).

**Questions:**

Aside from the question in weakness 1.1 above, modifying (a) in the definition of homomorphism expressivity such that, it only demands that the homomorphism counts to be equal if the representations are, makes it well-defined for all models $M$. Does this allow taking into account all possible models or is it incompatible with the results in the paper?

An additional (more minor) comment. In the paragraph at the bottom of page 5 it is stated:

> “First, we highlight a key insight that homomorphism can serve as a fundamental
expressivity measure, which is achieved by further proving a non-trivial result that FM is maximal
(Definition 3.1(b)).”

By definition $\mathcal{F}^M$ is maximal, hence, it is not clear the intention of this sentence is. Was it that the existence of a maximal $\mathcal{F}^M$ is proven?

---

> ### Author Response · Authors · 2023-11-20
> **Response to Reviewer wf9E (Part 1/2)**
>
> We sincerely thank Reviewer wf9E for the thoughtful reading, detailed comments, and positive feedback, and for raising a fantastic technical question. Below, we would like to give detailed responses to each of your comments and questions.
>
> **Regarding the definition of $\mathcal F^M$.** This is an excellent question that deserves detailed discussions. First, we would like to clarify that, for a specific graph $F\in\mathcal F^M$, our definition does **not** imply that two graphs $G, H$ must have the same representation if $\mathsf{hom}(F, G)=\mathsf{hom}(F, H)$. Instead, according to our definition, two graphs $G, H$ must have the same representation if $\mathsf{hom}(F, G)=\mathsf{hom}(F, H)$ holds for **all** graphs $F\in\mathcal F^M$. Therefore, the homomorphism expressivity corresponding to the universal architecture in your example is still well-defined (as will be shown below).
>
> Consider a universal architecture that assigns different representations to any two non-isomorphic graphs. We will show that the homomorphism expressivity $\mathcal F^M$ is the set of all graphs. The proof is quite straightforward:
> - First, it is clear that the universal architecture can count any graph under homomorphism.
> - It remains to prove that for any two graphs $G,H$, if $\mathsf{hom}(F,G)=\mathsf{hom}(F,H)$ holds for all graphs $F$, then $G$ is isomorphic to $H$. This is actually a celebrated result proved by Lovász (1967). We thus conclude the proof.
>
> Lovász, 1967. Operations with structures.
>
> We now consider the general case and discuss whether homomorphism expressivity is always well-defined. We highlight that, for all common GNN architectures such as the ones studied in this paper and also a variety of architectures discussed in Reviewer TDGD, **their homomorphism expressivity all exists**. Nevertheless, there do exist pathological, *intentionally designed* GNNs such that the homomorphism expressivity is not well-defined. We give an example below.
>
> Consider a GNN $M$ that outputs the representation of graph $G$ as follows. If $G$ is a cycle, it outputs $(1,l)$ where $l$ is the length of the cycle. Otherwise, it outputs $(0,\chi^\mathsf{MP}_G(G))$, namely, running a MPNN on the graph. It follows that the GNN $M$ is more powerful than MPNN and can thus count all trees under homomorphism. Moreover, it cannot count other patterns under homomorphism, as the counterexample graphs $G, H$ for MPNN are not cycles and still apply here. Therefore, the homomorphism expressivity of $M$ should be exactly the family of all trees if it exists. However, the homomorphism information of all trees cannot determine the representation of model $M$ since $M$ is strictly more powerful than MPNN (e.g., it can distinguish between 6-cycle and two triangles). Therefore, we conclude that the homomorphism expressivity of $M$ does not exist.
>
> Note that the above GNN construction is inherently *unnatural* and hardly appears in practice, as it tailors the representation of specific "corner-case" graphs while employing a different model for other graphs. When a GNN is described via a *unified* message-passing formula, we cannot find any counterexample GNN model. **We conjecture that, for any GNN characterized by a color refinement algorithm that outputs stable colors, the homomorphism expressivity always exists.**

---

> ### Author Response · Authors · 2023-11-20
> **Response to Reviewer wf9E (Part 2/2)**
>
> **Regarding the "completeness" of homomorphism expressivity.** Sorry for the confusion regarding "completeness". We say an expressivity measure is complete if it encodes all aspects of expressiveness. We regard homomorphism expressivity as a complete measure due to the following reasons: $(\mathrm{i})$ whenever a model $M_1$ is more expressive than another model $M_2$ in *certain* aspect, homomorphism can capture it via $\mathcal F^{M_1}\backslash\mathcal F^{M_2}$. $(\mathrm{ii})$ Conversely, $\mathcal F^{M_2}\subset\mathcal F^{M_1}$ will imply that model $M_1$ is more expressive than model $M_2$ in *any* aspect.
>
> To avoid confusion, it might be more desirable to change the word "complete" to "comprehensive" and will make this change if they are helpful.
>
> **Regarding features of vertices/edges.** We would like to clarify that our framework does not omit vertex features. Indeed, all results in this paper are described in the context of vertex-labeled graphs (see Section 2). As an application, our results can be used to analyze whether a GNN model can count the number of Guanines in a protein, which is a crucial structure in biological chemistry. Note that Guanines involves 5-cycles and 6-cycles with different atom types (vertex features). Our results can be used to show that MPNN and Subgraph GNN cannot count Guanines at node-level, while Local 2-(F)GNN can count them.
>
> > **Question.** "First, we highlight a key insight that homomorphism can serve as a fundamental expressivity measure, which is achieved by further proving a non-trivial result that $\mathcal F^M$ is maximal (Definition 3.1(b))." By definition $\mathcal F^M$ is maximal, hence, it is not clear what the intention of this sentence is. Was it that the existence of a maximal is proven?
>
> Sorry for the confusion. Yes, we prove that each graph family given in Theorem 3.3 is maximal and thus satisfies all conditions in Definition 3.1. The maximal property is not proved in prior work and is a key technical contribution of this paper.

---

> ### Comment · Reviewer_wf9E · 2023-11-20
>
> Thank you for the detailed response. I would like to keep my initial positive assessment.
>
> Though there is a question from my initial review that was not addressed in the response, and I would appreciate your input on it. Correct me if I missed anything, but the reason that there exist models $M$ for which homomorphism expressivity is not defined is due to the "if and only if" in requirement (a) of Definition 3.1. As asked in my original review, say that we modify (a) to only require that, if the representations of two graphs are equal, then their homomorphism counts are equal for any $F \in \mathcal{F}_M$ (and not the other direction whereby if their homomorphism counts are the same then their representations are equal). Does this allow taking into account all possible models or is it incompatible with the results in the paper? This limitation of homomorphism expressivity that makes it not well-defined for all architectures seems somewhat artificial to me. In a sense, it penalizes models that are more expressive (since it is not defined for them).
>
> Note: Regarding features of vertices. As mentioned in my review, the limitation I see in homomorphism expressivity (and WL analyses as well) is that it disregards vertex features **beyond simple discrete labels**. In practice, vertices are often associated with real-valued feature vectors. These features play in many cases an equal, if not more important role than the graph structure.

---

> ### Author Response · Authors · 2023-11-21
> **Thank you for your prompt response**
>
> Thank you again for your insightful feedback and the question. Below, we would like to give a detailed response to this interesting question.
>
> Indeed, your observation is correct. By substituting "if and only if" in Definition 3.2(a) with "implies," the homomorphism expressivity becomes well-defined for all GNN models. However, while this modification results in a better definition from the perspective of *universality*, it comes at the cost of forfeiting a critical attribute of homomorphism expressivity—completeness. This is extremely important and distinguishes our work from previously proposed expressivity measures. The absence of completeness would render most results in Section 4, including expressivity comparisons between GNN models and subgraph counting power, unattainable. Details are as follows:
> - **Section 4.1 (Expressivity comparisons between GNN models)**
>
>   Suppose that we have two models $M_1$ and $M_2$ and $\mathcal F^{M_1}\subset \mathcal F^{M_2}$. We want to prove that $M_2$ is more powerful than $M_1$, namely, for any two graphs indistinguishable by $M_2$, they are also indistinguishable by $M_1$. The derivation is shown below.
>   - Pick two graphs $G,H$ such that $\chi^{M_2}_G(G)=\chi^{M_2}_H(H)$
>   - $\implies$ $\mathsf{hom}(F,G)=\mathsf{hom}(F,H)$ for all $F\in \mathcal F^{M_2}$
>   - $\implies$ $\mathsf{hom}(F,G)=\mathsf{hom}(F,H)$ for all $F\in \mathcal F^{M_1}$ (since $\mathcal F^{M_1}\subset \mathcal F^{M_2}$)
>   - $\implies$ $\chi^{M_1}_G(G)=\chi^{M_1}_H(H)$
>
>   Here, the last step uses of completeness of homomorphism expressivity. Without this property, making expressivity comparisons between GNN models becomes impossible. Similarly, $\mathcal F^{M_1}\subsetneq \mathcal F^{M_2}$ no longer implies that $M_2$ is strictly more powerful than $M_1$.
> - **Section 4.2 (Subgraph counting power)**
>
>   Our central theorem states that a model $M$ can subgraph-count $F$ if and only if $\mathsf{Spasm}(F)\subset\mathcal F^M$. The proof of this theorem crucially relies on the completeness of homomorphism expressivity (see the proof of Lemma G.5). Without completeness, one can easily construct a pathological GNN $M$ following our previous response, such that $\mathsf{Spasm}(F)\backslash\mathcal F^M\neq\emptyset$ but $M$ can subgraph-count $F$.
>
> Given the pivotal role of completeness in analyzing the expressivity and subgraph counting power and the practical satisfaction of this property by most GNN models, we believe that it would be beneficial to include it in the definition. On the other hand, we feel that it is necessary to add a remark to point out that completeness is not always satisfied and homomorphism expressivity may not exist for pathological GNN models, and we will definitely add these discussions in our paper.
>
> ---
>
> Finally, we fully acknowledge that real-valued feature vectors play an important role in practical scenarios and cannot be tackled in our framework. We view this as an exciting research direction worthy of future exploration.

---

> > ### Comment · Reviewer_wf9E · 2023-11-21
> >
> > Thank you for clarifying, the response has fully addressed my question.

---

### Official Review · Reviewer_TDGD · 2023-10-29

**Soundness:** 4 excellent
**Presentation:** 3 good
**Contribution:** 4 excellent
**Rating:** 10
**Confidence:** 4

**Summary:**

This paper proposes a new theoretical way to analyze the expressive power of several types of expressive GNNs (including MPNN, k-subgraph GNN, k-local GNN, and k-local FGNN) using the concept of homomorphism. Concretely, the authors prove that the set of the base graphs a GNN can identify under the homomorphism to the input graph can be a sufficient evaluation of the expressive power of GNN. Then, the authors systemically analyze several popular GNN variants using this metric through the novel concept of nested ear decomposition (NED) and give a complete expressive hierarchy. Moreover, the proposed metric also provides a new way of analyzing the subgraph counting power of GNN variants under moderate graph size. Finally, the authors conduct experiments to verify the theorems.

**Strengths:**

1. Although the paper may still be hard for general readers in the DL community to understand, I think the authors already did a great job of formalizing and presenting these extensive theoretical concepts and theorems.

2. The proposed metric and the theoretical proofs are comprehensive and elegant. It provides a new way to reveal the strict hierarchy and the expressive gap among several high-expressive GNNs. Moreover, the subgraph counting power analysis is also impressive.

**Weaknesses:**

I cannot see any major weakness but have one potential minor weakness for the authors:

Although the authors claim that the proposed framework can quantitatively analyze the expressive power of different GNN variants based on the NED, the authors didn’t characterize the exact number of NED that exists for each NED class (share-point/strong/near strong/general NED). Thus, it is still hard to see the quantitative expressive gap between each GNN variant. I understand the exact number could be hard to count but It would be excellent if the authors could at least give a rough scale for each NED class.

I also have a few questions/thoughts in mind when reading the paper. Please see the question section for the details.

**Questions:**

For the framework:
1. For the subgraph GNN, the authors seem to only discuss the variant that first pool nodes within each subgraph and then pool all subgraph, which corresponds to the SWL(VS) proposed in the [1]. However, [1] also characterizes other different subgraph GNN variants with different theoretical expressive power (form a strict hierarchy). I am wondering could the proposed framework be used to analyze other variants and reveal the same result as discussed in [1].

2. Can the proposed framework also be used to analyze other more general GNN frameworks like [2] or [3]?

3. The subgraph counting power is proved for only moderate graph size (nodes size $\leq$ 6 or edge size $\leq$ 8). I am wondering, what is the major gap or obstacle for proving a more general graph size? I am guessing it is related to the equation stated in the paper that characterizes the quantitative relationship between homomorphism count and subgraph count. For graphs of general size, is the proposed theorem at least an essential condition for successful counting?

For the broader impact:
1. So far, the paper only discusses the analysis result of the proposed framework on some existing GNN variants. I am wondering if the authors could discuss how the proposed framework can be used to analyze new GNN variants when it is proposed or how could other researchers design new powerful and efficient GNN variants based on this analysis framework.


I understand some questions may not be easy to answer and the answer will not negatively affect my score. I think the current paper is already worth a clear acceptance. However, it is hard for me to check all the proofs given this short period of time. But I will try my best to check it later.

References:

[1] Zhang et al., A complete expressiveness hierarchy for subgraph gnns via subgraph weisfeiler-lehman tests, ICML23.

[2] Zhou et al., From relational pooling to subgraph gnns: A universal framework for more expressive graph neural networks, ICML23.

[3] Feng et al., Towards arbitrarily expressive gnns in $O(n^2)$ space by rethinking folklore weisfeiler-lehman, NeurIPS23.

---

> ### Author Response · Authors · 2023-11-20
> **Response to Reviewer TDGD (Part 1/4)**
>
> We sincerely thank Reviewer TDGD for the thorough reading and appreciation of our work, and for raising a series of very insightful questions. Below, we would like to give detailed responses to each of your comments and questions.
>
> > **Weakness**. The authors didn’t characterize the exact number of NED that exists for each NED class (share-point/strong/near strong/general NED). Thus, it is still hard to see the quantitative expressive gap between each GNN variant. I understand the exact number could be hard to count but It would be excellent if the authors could at least give a rough scale for each NED class.
>
> Thank you for the suggestion. We are sorry that due to the space limit, we did not put these statistics in the main text but in the Appendix (Tables 4 and 5). Here, In Table 4 we list exactly the number of graphs of a certain size for each NED class at graph/node/edge-level, allowing for a clear and quantitative comparison between GNN models. In Table 5, we list the number of graphs of a certain size each GNN model can subgraph-count at graph/node/edge-level. We have also made additional discussions in Appendix G. We hope this can address your concern and will consider moving these tables to the main text once our paper is accepted (so that more space is allowed).

---

> ### Author Response · Authors · 2023-11-20
> **Response to Reviewer TDGD (Part 2/4)**
>
> > **Question 1.** For the subgraph GNN, the authors seem to only discuss the variant that first pool nodes within each subgraph and then pool all subgraphs, which corresponds to the SWL(VS) proposed in the [1]. However, [1] also characterizes other different subgraph GNN variants with different theoretical expressive power (form a strict hierarchy). I am wondering could the proposed framework be used to analyze other variants and reveal the same result as discussed in [1].
>
> Thank you for raising this fantastic question! Indeed, our analysis can be easily extended to other architectures in [1] by specifying new variants of NED. Below, we will take the architecture PSWL(VS) as an example. It turns out that its homomorphism expressivity can still be derived in a surprisingly elegant way, by which we are able to recover all theoretical results about PSWL(VS) in [1].
>
> Our results are described based on the latest definition of NED in the updated paper (please see the General Response on why we updated the definition of NED). Moreover, we need to define a concept called *block*:
> - Two ears $P_i, P_j$ are in the same block if one ear is nested on the other with a non-empty nested interval, denoted as $P_i\sim P_j$;
> - For any ears $P_i,P_j,P_k$, if $P_i\sim P_j$ and $P_j\sim P_k$, then $P_i\sim P_k$.
>
> It is easy to see that "block" is an equivalence relation between ears. We then define several restricted variants of NED as follows:
> - **Strongly endpoint-shared NED**: a NED is called strongly endpoint-shared if all ears with non-empty nested intervals share a common endpoint.
> - **Weakly endpoint-shared NED**: a NED is called weakly endpoint-shared if all ears in the same block share a common endpoint.
> - **Strong NED**: a NED is called strong if for any two children $P_j$, $P_k$ ($j<k$) nested on the same parent ear, we have $I(P_j)\subset I(P_k)$.
>
> While the above concept may be hard to understand at first sight, it can be easily understood via an illustration. In the updated paper, we provide two example graphs that admit weakly endpoint-shared NED (Figure 9(a,b)) and one example graph that does not admit weakly endpoint-shared NED (Figure 9(c)).
>
> > **Theorem**: The homomorphism expressivity of the above three architectures exists and can be described below:
> >   - **Subgraph GNN (SWL(VS)-GNN)**: $\mathcal F^\mathsf{Sub}=\\\{F:F\text{ has a strongly endpoint-shared NED}\\\}$;
> >   - **PSWL(VS)-GNN**: $\mathcal F^\mathsf{PSWL}=\\\{F:F\text{ has a weakly endpoint-shared NED}\\\}$;
> >   - **Local 2-GNN (SSWL-GNN)**: $\mathcal F^\mathsf{L}=\\\{F:F\text{ has a strong NED}\\\}$.
>
> This readily reveals the expressivity gap among the above three architectures:
>
> > **Corollary**: $\mathcal F^\mathsf{Sub}\subsetneq \mathcal F^\mathsf{PSWL}\subsetneq \mathcal F^\mathsf{L}$. Thus, the expressive power of the following three GNN models strictly increases in order: **SWL(VS)**, **PSWL(VS)**, and **SSWL**.
>
> **Proof**: It is clear that any strongly endpoint-shared NED is a weakly endpoint-shared NED and any weakly endpoint-shared NED is a strong NED. To prove strict separation result, observe that:
> - The graph corresponding to Figure 5 in [1] is exactly a counterexample graph that reveals the gap between SWL(VS) and PSWL(VS). It has a weakly endpoint-shared NED but is not strongly endpoint-shared.
> - The graph corresponding to Figure 6 in [1] is exactly a counterexample graph between PSWL(VS) and SSWL. It has a strong NED but is not weakly endpoint-shared.
>
> Moreover, as a significant implication, we have the following result:
> > **Corollary**: PSWL(VS) cannot count 6 cycles at node-level.
>
> **Proof**: in our paper, the bottom-right graph in Figure 4(b) is a homomorphic image of the rooted 6-cycle, which, unfortunately, does not have a weakly endpoint-shared NED if the root vertex is an endpoint of the first ear. Therefore, PSWL(VS) cannot count 6 cycles at node-level.
>
> We believe other architectures in [1] can be similarly analyzed using our framework in an elegant way. Due to the tight schedule, we cannot present all results, but we believe it is an exciting research direction to unify all these prior architectures using the perspective of homomorphism expressivity. Thank you again for raising this insightful question!

---

> ### Author Response · Authors · 2023-11-20
> **Response to Reviewer TDGD (Part 3/4)**
>
> > **Question 2.** Can the proposed framework also be used to analyze other more general GNN frameworks like [2] or [3]?
>
> Yes, our framework can indeed be used to analyze other GNN frameworks and give a precise characterization of their homomorphism expressivity and subgraph counting power. We will take [2] as an example and consider their proposed $k,l$-WL. Since the analysis is similar, we will only show the main result below:
> > **Theorem**: The homomorphism expressivity of $k,l$-WL exists and can be described below:
> >   - $1,l$-WL: $\mathcal F^{(1,l)}=\\\{F:\exists U\subset V_F\text{ s.t. }|U|\le l\text{ and }F\backslash U\text{ is a forest}\\\}$;
> >   - $k,l$-WL for $k\ge 2$: $\mathcal F^{(k,l)}=\\\{F:\exists U\subset V_F\text{ s.t. }|U|\le l\text{ and }\mathrm{treewidth}(F\backslash U)\le k-1\\\}$.
>
> As can be seen, the above theorem generalizes the homomorphism expressivity of both Subgraph $l$-GNN and $k$-FGNN, which coincides with the fact that $(k,l)$-WL is a general framework that unifies a series of architectures like Subgraph GNN and higher-order GNN.
>
> We believe the architectures in [3] can be similarly analyzed using our framework (although analyzing them seems to be harder than analyzing $k,l$-GNN). Due to the tight schedule, we may not have enough time to derive a result. Again, we consider the unification of all these prior architectures from the perspective of homomorphic expressivity to be a very exciting research direction for future work.
>
> > **Question 3.** The subgraph counting power is proved for only moderate graph size (nodes size $\le$ 6 or edge size $\le 8$). I am wondering, what is the major gap or obstacle for proving a more general graph size? I am guessing it is related to the equation stated in the paper that characterizes the quantitative relationship between homomorphism count and subgraph count. For graphs of general size, is the proposed theorem at least an essential condition for successful counting?
>
> Thank you for the question! We are thrilled to announce that we have fully settled this question in recent days. Below, we will demonstrate what is the main obstacle in our previous analysis, and how we resolve it using a new approach.
>
> As shown in Section 4.2, the subgraph counts of graph $F$ is linearly related to the homomorphism counts of all its homomorphic images. Therefore, if all homomorphic images of $F$ are in $\mathcal F^M$, then clearly the model $M$ can subgraph-count $F$. This is what Proposition 4.4 states. On the other hand, when exactly one homomorphic image of $F$ is not in $\mathcal F^M$, it is also clear that the model $M$ cannot subgraph-count $F$ (otherwise, the model will be able to count under homomorphism the homomorphic image by using the linear relation, a contradiction). However, things become complicated when two or more homomorphic images of $F$ are not in $\mathcal F^M$. We exhaustively verified all moderate-size patterns and found that the result still holds; However, a general proof is challenging as we cannot find universal rules or patterns about which homomorphic image can be used to construct counterexample graphs.
>
> We resolve this challenge by leveraging a recently proposed technique in the graph theory community (Seppelt, 2023). Intuitively speaking, when the homomorphic images of $F$ contain $N$ non-isomorphic graphs, the method in Seppelt (2023) gave an approach to construct $N$ pairs of counterexample graphs $(G_1, H_1),\cdots,(G_N, H_N)$ so that $\mathsf{sub}(F, G_i)=\mathsf{sub}(F, H_i)\  \forall i\in[N]$ iff $\mathsf{hom}(F, G_i)=\mathsf{hom}(F, H_i)\  \forall i\in[N]$. This approach addresses the above difficulty: as long as there is at least one pair of graphs $(G_i, H_i)$ with $\mathsf{hom}(F, G_i)\neq\mathsf{hom}(F, H_i)$, there will be at least one pair of graphs $(G_j, H_j)$ with $\mathsf{sub}(F, G_j)\neq\mathsf{sub}(F, H_j)$. By picking each $(G_i, H_i)$ the Furer graphs expanded by a homomorphic image of $F$, we can obtain the desired result.
>
> Seppelt, 2023. Logical equivalences, homomorphism indistinguishability, and forbidden minors.

---

> ### Author Response · Authors · 2023-11-20
> **Response to Reviewer TDGD (Part 4/4)**
>
> > **Question for broader impact.** So far, the paper only discusses the analysis result of the proposed framework on some existing GNN variants. I am wondering if the authors could discuss how the proposed framework can be used to analyze new GNN variants when it is proposed or how could other researchers design new powerful and efficient GNN variants based on this analysis framework.
>
> Thank you for raising this very insightful question. As discussed in the previous response, we truly believe that our proposed framework can universally apply to a broad range of popular GNN architectures proposed before and even in the future. Below, we give some tips and examples on $(\mathrm{i})$ how to derive the homomorphism expressivity when a new architecture is proposed and $(\mathrm{ii})$ how to guide the design of provably powerful architectures using our framework.
>
> - When a new GNN variant is proposed, one may first identify the order of the GNN, i.e., how the computational complexity grows w.r.t. the number of graph vertices. Typically, we will need the $k$-order ear to describe homomorphism expressivity if the GNN order is $k$ (i.e., a worst-case complexity of $O(n^{k+1})$ for a graph of $n$ vertices). Then, we need to specify how different ears can nest on each other. This paper gives a general analyzing procedure: we can gain insights into the structure of NED from the *unfolding tree* of a GNN model. We note that the unfolding tree is a standard tool in analyzing GNN expressivity and has been extensively used in prior work. In this paper, we show that for most architectures, the structure of the unfolding tree is deeply linked to a specific type of tree decomposition, which is then linked to the structure of NED. We believe this analysis framework is universal and should apply to future GNN models.
>
> - We next discuss how our framework can guide the design of provably powerful architectures. As an example, let's consider the task of designing a GNN that is provably powerful for encoding 8-cycles. Then, based on Theorem 4.5, a necessary and sufficient condition is that the homomorphism expressivity $\mathcal F^M$ must contain all homomorphic images of the 8-cycle. Notably, $\mathcal F^M$ must contain the 4-clique. This will imply that one should at least consider a 3-order GNN (since any 2-order NED cannot form a 4-clique). Interestingly, this justifies the architectures proposed in prior work such as the I$^2$-GNN (Huang et al. 2023) or N$^2$-GNN [3], in that I$^2$-GNN is a 3-order GNN and N$^2$-GNN is a 4-order GNN, both of which can encode 4-cliques as shown in their experiments. Actually, we believe that both I$^2$-GNN and N$^2$-GNN can count 8-cycles.
>
> Huang et al., 2023. Boosting the cycle counting power of graph neural networks with I$^2$-GNNs.
>
> We hope our response as well as the updated paper can address your concerns. It is really a great pleasure to study these interesting questions you raised, and we are more than willing to delve further into any aspect of the aforementioned response and provide additional details as needed.

---

> > ### Comment · Reviewer_TDGD · 2023-11-20
> >
> > Thanks for the detailed answers to all my questions. All of my questions are well-addressed and the new results are impressive. I believe the proposed framework can serve as a general and complete measurement of GNN expressive power in the future if authors can further extend it. In light of this, I would like to further increase my score towards a strong acceptance.
> >
> > One minor comment is that the current paper is still too dense and complicated for general researchers to understand and use. It would be great if the authors could further simplify it or provide documents for other researchers to use to analyze their models in the future.

---

> > > ### Author Response · Authors · 2023-11-21
> > > **Thank you**
> > >
> > > Thank you for your positive feedback and great suggestion. We are committed to achieving this goal in future work, making homomorphism a fundamental and simple tool for general researchers to analyze the expressive power of GNNs.

---

### Official Review · Reviewer_3wF7 · 2023-10-31

**Soundness:** 4 excellent
**Presentation:** 3 good
**Contribution:** 4 excellent
**Rating:** 8
**Confidence:** 4

**Summary:**

The paper deals with the expressive power of GNN-type architectures for graph-, node-, and edge-level representation learning. Specifically, the authors propose a framework based on graph homomorphisms to compare the expressive power of four classes of GNN architectures, namely, MPNNs, subgraph GNNs, local $k$-GNNs, folklore-WL GNNs, and their ability to count subgraphs.

To that, they extend the known results of Dell et al. connecting $k$-WL expressivity and homomorphism counts via a novel variant of nested ear decompositions. They show which kind of homomorphism counts the above three GNN types can distinguish, compare them, and shed some light on their cycle and path counting ability.

Finally, they probe their theoretical results empirically, showing that increased expressivity indeed translates into increased homomorphism
 and cycle-counting ability. Further, they show a similar effect on the ZINC and Alchemy graph regression datasets.

**Strengths:**

- The authors propose a unified framework to study the expressive power of various GNN-type architecture, unifying the landscape of different architectures. By that, they recover and extend several known results.
- Clear, although dense, presentation
- Proofs are clearly structured
- Good synthetic results, verifying the theoretical results

**Weaknesses:**

- The presentation is quite dense. It might be beneficial to push Subsection 3.3. to the appendix and expand on the other parts.
- The experimental study on real-world datasets is quite limited. It would be nice to see a more refined analysis, e.g., analyzing subgraph occurrences and see if an architecture's performance is correlated to its ability to count different subgraphs.

**Questions:**

- Is the connectedness assumption in Definition 3.1 necessary?

**Comments**
- On page 5, in the definition of $\mathsf{Clo_{ind}}$, there seems to be $G \in \mathcal{G}$ missing
- The results of Dell et al. were previously shown in

Zdeněk Dvořák. On recognizing graphs by numbers of homomorphisms. Journal of Graph
Theory, 64(4):330–342, August 2010. doi:10.1002/jgt.20461

it might be good to cite the above paper as well

---

> ### Author Response · Authors · 2023-11-20
> **Response to Reviewer 3wF7 (Part 1/2)**
>
> We sincerely thank Reviewer 3wF7 for the careful reading and positive feedback, and for raising an interesting technical question. Below, we would like to address each of your questions and concerns:
>
> > **Question.** Is the connectedness assumption in Definition 3.1 necessary?
>
> Thanks for raising this great technical question. Actually, the connectedness assumption in Definition 3.1 is not necessary, but it can greatly simplify the presentation of our theoretical results. We choose to add this assumption due to the following considerations:
> - Let $F$ be a disconnected graph that can be decomposed into several connected components $F_1,\cdots, F_m$. Then, for any graph $G$, we have $\mathsf{hom}(F,G)=\mathsf{hom}(F_1,G)\times \cdots \times \mathsf{hom}(F_m,G)$. It is straightforward to prove that $F\in\mathcal F^M$ iff $F_i\in\mathcal F^M$ for all $i\in[m]$. Consequently, there is no need to consider disconnected graphs.
> - For node/edge-level expressivity, things will become a bit complicated. Consider a disconnected rooted graph $F^u$ that can be decomposed into several connected components $F_1^u, F_2, \cdots, F_m$ where the root node is in $F_1$. Then, for any graph $G^v$, we have $\mathsf{hom}(F^u,G^v)=\mathsf{hom}(F_1^u,G^v)\times \mathsf{hom}(F_2,G)\times \cdots \times \mathsf{hom}(F_m,G)$. Analogously, it can be proved that $F^u\in\mathcal F_\mathsf{n}^M$ iff $F_1^u\in\mathcal F_\mathsf{n}^M$ and $F_i\in\mathcal F^M$ for all $i\in\\\{2.\cdots,m\\\}$. Again, there is no need to consider disconnected graphs.
> - The inclusion of disconnected graphs will complicate the definition of NED. In the original Definition 3.2, any ear $P_i$ ($i>1$) is nested on some previous ear, and the relationship between different ears forms a *tree* structure. When extending to disconnected graphs, Definition 3.2 should be modified wherein the relationship between different ears now forms a *forest* (i.e., there can be more than one ears not nested on any ear).
>
> > **Weakness.** The presentation is quite dense. It might be beneficial to push Subsection 3.3. to the appendix and expand on the other parts.
>
> Thanks for the suggestion. We fully agree with you that the page limit makes our presentation dense. Here, the inclusion of Section 3.3 (node/edge-level expressivity) is primarily motivated by its foundational role in Sections 4.2 (node/edge-level subgraph counting) and 4.3 (polynomial expressivity), where we have addressed pivotal open questions from previous research. However, we acknowledge the need for expanded exposition in other sections to enhance accessibility. We will definitely expand on other parts once our paper is accepted so that more space is allowed.
>
> > **Weakness.** The experimental study on real-world datasets is quite limited. It would be nice to see a more refined analysis, e.g., analyzing subgraph occurrences and see if an architecture's performance is correlated to its ability to count different subgraphs.
>
> Thanks for the suggestion. To further comprehend our experimental study, we have performed two additional tasks. First, we extended our experiments on real-world datasets to include the ZINC-full benchmark, which comprises 250K molecular graphs, a substantial increase in size compared to the ZINC-subset. The results have been presented in the updated paper. As before, all models obey the same computational budget of 500K parameters. Again, the correlation between model performance and homomorphism expressivity remains evident, with Local 2-FGNN exhibiting superior performance across all three real datasets.
>
> Next, we conduct experiments on real-world tasks to see if an architecture's performance is correlated to its ability to count different subgraphs. For the molecular property prediction task, it is well-known that the ability to count 6-cycles is crucial as 6-cycles cover important structures like benzene rings. We thus investigate whether different GNN models can count 6 cycles *on the ZINC dataset*. By examining the dataset, we found that the number of 6-cycles in most molecules ranges from 1 to 3. We then trained different models and report the MAE results in the following table. Here, we ran each experiment 10 times independently with different seeds and report the average performance as well as the standard deviation.
> | Model    | MAE |
> | -------- | ------- |
> | MPNN         | 1.357 ± 0.007 |
> | Subgraph GNN | 0.030 ± 0.001 |
> | Local 2-GNN  | 0.021 ± 0.001 |
> | Local 2-FGNN | 0.021 ± 0.001 |
>
> One can see that MPNN and Subgraph GNN perform much worse than Local 2-(F)GNN in counting 6-cycles, indicating that they cannot effectively encode the information of benzene rings on ZINC dataset. This is consistent with their performance in molecular property prediction as shown in our paper (Table 2).

---

> ### Author Response · Authors · 2023-11-20
> **Response to Reviewer 3wF7 (Part 2/2)**
>
> > **Comment.** Regarding $\mathsf{Clo_{\mathsf{ind}}}$.
>
> Thank you for pointing out this typo! We are delighted to update you on our recent advancements regarding $\mathsf{Clo_{\mathsf{ind}}}$. Specifically, we are able to remove the induced-subgraph-closure and yield a much cleaner result as follows:
> - **MPNN**: $\mathcal F^\mathsf{MP}=\\\{F:F\text{ is a tree}\\\}$;
> - **Subgraph GNN**: $\mathcal F^\mathsf{Sub}=\\\{F:F\text{ has an endpoint-shared NED}\\\}$;
> - **Local 2-GNN**: $\mathcal F^\mathsf{L}=\\\{F:F\text{ has a strong NED}\\\}$;
> - **Local 2-FGNN**: $\mathcal F^\mathsf{LF}=\\\{F:F\text{ has an almost-strong NED}\\\}$;
> - **2-FGNN**: $\mathcal F^\mathsf{F}=\\\{F:F\text{ has a NED}\\\}$.
>
> This is achieved by a slight modification of NED in Definition 3.2. Please see the General Response for more details and the significance of this result compared to the initial one.
>
> > **Comment.** The results of Dell et al. were previously shown in Zdeněk Dvořák. It might be good to cite the above paper as well.
>
> Thank you for pointing out this reference and we have cited it in our paper.

---

> > ### Comment · Reviewer_3wF7 · 2023-11-20
> >
> > Thanks for the thorough rebuttal; I still think that this paper should be accepted.

---

### Official Review · Reviewer_aNaP · 2023-11-09

**Soundness:** 3 good
**Presentation:** 4 excellent
**Contribution:** 3 good
**Rating:** 8
**Confidence:** 5

**Summary:**

In this paper, the authors study the set of substructures that various definitions of GNNs can distinguish. They consider the new notion of homomorphism expressivity (Definition 3.1), which intuitively means the ability of GNNs to count different substructures. In Theorem 3.3., which is the main contribution of the paper,  they use the concept of NED (Definition 3.2) to exactly express the power of GNNs in homomorphism counting for MPNNs, Subgraph GNNs, Local GNNs, and FGNNs. In a follow-up result, in Theorem 3.6., they extend their method to node-level functions on graphs. In Theorem 3.7. and Theorem 3.8., they find the homomorphism expressivity of Subgraph k-GNNs and Local k-GNNs using NEDs. In Proposition 4.4., they show how counting substructures is related to homomorphism expressivity. The paper is concluded with experiments.

**Strengths:**

- This paper is extremely well-written; the authors have spent a lot of time polishing the paper

- The theoretical problem is relevant to GNNs, while most theory works in GNNs are not necessarily applicable to practical scenarios

**Weaknesses:**

- it's better to have more examples of different kinds of substructures in Theorem 3.3 (and how they are different from each other).

- Missing discussion on some previous works: there are many papers proposed to count substructures, and it is not clear how your result is related to them.

**Questions:**

This is a well-written, interesting theoretical work on GNNs expressivity. I have a few questions:

- Why are homomorphism numbers important to us? Are there any concrete applications (e.g., in biology, etc.) that they appear? I know that subgraph counts are definitely important (from molecular biology), but not sure about homomorphisms. I'm asking because the main point of characterizing the power of GNNs is to have a better understanding for practical purposes.


- Under what conditions can the models in this paper express all subgraph counts of a  particular size (say less than k)? Indeed, how do you compare your method for counting with the other methods, like equivariant polynomials for GNNs and also recursive-based methods (like RNP-GNNs)? Is there a way to understand how recursion in RNP-GNNs can help counting all subgraphs using your theory (i.e., NEDs, etc.) and if there is any connections? I suggest discussing it a bit in the paper because recursion is a popular way to boost GNNs, and it's not clear if NEDs somewhere appear there or not.




--------------------------------
After the rebuttal: I appreciate the authors for their response and revision. As they fully addressed my questions/comments, and they added new stuff to the manuscript which I believe improved the quality of the paper, I decided to increase my score.

---

> ### Author Response · Authors · 2023-11-20
> **Response to Reviewer aNaP (Part 1/2)**
>
> We sincerely thank Reviewer aNaP for the detailed review and positive feedback. Below, we would like to address each of your concerns and suggestions:
>
> **Examples of different kinds of substructures in Theorem 3.3**. Thanks for the suggestion. We are sorry that owing to the space limit, in the initial submission we only give one example for each type of NED in the main text but have provided a comprehensive collection of examples in the Appendix (see Table 6). Based on Theorem 3, Table 6 lists all substructures within a moderate size and identifies whether each substructure belongs to each type of NED. We believe Table 6 is comprehensive enough to cover most substructures of interest in the GNN community. During the discussion period, we further add several representative examples in Figure 9. Additionally, in Table 4 we have presented the graph statistics, facilitating a quantitative comparison between different kinds of substructures. For example, among all 112 non-isomorphic graphs of 6 vertices:
> - 6 graphs are trees;
> - 51 graphs admit endpoint-shared NEDs;
> - 55 graphs admit strong NEDs;
> - 56 graphs admit almost-strong NEDs;
> - 56 graphs admit general NEDs.

---

> ### Author Response · Authors · 2023-11-20
> **Response to Reviewer aNaP (Part 2/2)**
>
> **Discussions on other work**. First, thank you for highlighting this valuable reference (RNP-GNN)! This paper gives a systematic way to design GNNs that are powerful for subgraph counting. We have cited it in the updated paper and made adequate discussions in our context (see the Introduction and Appendices A and B). We now address each of your specific questions:
>
> > Under what conditions can the models in this paper express all subgraph counts of a particular size (say less than k)?
>
> According to our theoretical framework, Subgraph GNN, Local 2-GNN, Local 2-FGNN, and 2-FGNN are capable of expressing all subgraph counts within 3 vertices but are unable to count 4-cliques. In a more general sense, Subgraph $(k-1)$-GNN and Local $k$-GNN can express all subgraph counts within $k+1$ vertices, though they fall short when counting $(k+2)$-cliques. It's worth noting that both Subgraph $(k-1)$-GNN and Local $k$-GNN exhibit a worst-case time complexity of $O(n^{k-1}m)$ for a graph with $n$ vertices and $m$ edges. The limitation in counting large cliques arises because the $k$−clique detection problem requires at least $n^{\Omega(k)}$ time [1].
>
> While counting cliques is intrinsically difficult, most models in this paper are still quite powerful in expressing subgraph counts for sparser graphs. For instance, as demonstrated in Table 5 of our paper, Subgraph GNN, Local 2-GNN, Local 2-FGNN, and 2-FGNN can express subgraph counts for all graphs within 5 edges.
>
> [1] Chen et al. Tight lower bounds for certain parameterized np-hard problems.
>
> > How do you compare your method for counting with the other methods, like equivariant polynomials for GNNs and RNP-GNNs?
>
> Homomorphism is closely connected to equivariant polynomials, and as elucidated in Section 4.3 and Appendix H, our findings offer valuable insights into polynomial expressivity across practical GNN architectures and answer an open problem in Puny et al. (2023). It is noteworthy that homomorphism expressivity is *complete*: $(\mathrm{i})$ whenever a model $M_1$ is more expressive than another model $M_2$ in **certain** aspect, homomorphism can capture it via $\mathcal F^{M_1}\backslash\mathcal F^{M_2}$. Conversely, $\mathcal F^{M_2}\subset\mathcal F^{M_1}$ will imply that model $M_1$ is more expressive than model $M_2$ in **any** aspect. However, it is not known whether this crucial property holds for polynomial expressivity.
>
> Regarding RNP-GNNs, understanding their homomorphism expressivity seems to be challenging. Indeed, RNP-GNNs with parameters $(r_1,\cdots,r_\tau)$ can be treated as a sparse version of $\tau$-order GNNs. Consequently, to characterize their homomorphism expressivity, a novel form of $\tau$-order ear must be identified, along with a specification detailing how different ears can nest upon each other. However, analyzing higher-order NED is quite challenging as shown in Appendix E. We only have some intuitions that the recursion may correspond to a "split" operation in a higher-order ear, which increases the number of paths by one (see Definition E.1). Nevertheless, we believe that there must exist a proper definition of NED that can precisely characterize the homomorphism expressivity of RNP-GNNs, and once the NED is found, the subgraph counting power of RNP-GNNs can be entirely characterized by using Theorem 4.5. We leave it as an interesting topic for future work.
>
> ---
>
> **Importance of homomorphism numbers**. We argue that homomorphism expressivity is a practical expressivity measure because $\mathcal F^M$ fully characterizes the subgraph counting power of GNN model $M$ as shown in Theorem 4.5 (please see the General Response for our new results). We use homomorphism count instead of subgraph count due to two main reasons:
>
> - The homomorphism expressivity $\mathcal F^M$ contains all information of the subgraph counting power, but the converse does not hold: even if the subgraph counting power of model $M$ is fully characterized, $\mathcal F^M$ is still not determined. In other words, homomorphism expressivity is a **complete** expressivity measure and contains more information than the incomplete measure of subgraph counting.
> - There exist elegant descriptions of the homomorphism expressivity for a wide range of models (e.g., using trees or certain types of NED). However, characterizing the subgraph counting power of popular GNN models is hard without resorting to homomorphism expressivity (see Arvind et al. (2020)). This possibly implies that homomorphism counting is more **fundamental** than subgraph counting. Indeed, subgraph is essentially a restricted form of homomorphism, namely, *injective homomorphism*.
>
> We hope our response as well as the updated paper can address your concerns. We are more than willing to delve further into any aspect of the aforementioned response and provide additional details as needed, and we look forward to your reply.

---

> ### Author Response · Authors · 2023-11-23
> **Looking forward to hear your thoughts on our response and revised manuscript**
>
> Dear Reviewer aNaP,
>
> We sincerely appreciate your valuable comments and constructive feedback on our paper. We have carefully responded to each of the questions you raised and followed your suggestion to revise our submission. As the discussion period will close in a few hours, we would greatly appreciate your confirmation on whether our responses adequately addressed your concerns. If you have any additional questions or open discussions, please don't be hesitant to leave more comments. Thank you for your time and consideration.
>
> Paper 6795 authors

---

### Author Response · Authors · 2023-11-20
**General Response: New results & Paper Updates**

We sincerely thank all the reviewers and the area chair for their great efforts in reviewing our paper. We would like to take this opportunity to highlight that, during the discussion period, we have significantly enhanced two key theoretical results originally presented in the initial submission, which we believe can further strengthen this work.

**Necessary and sufficient condition of subgraph counting**. In the initial submission, we prove that for all moderate-size graphs $F$ (i.e., $\le$ 6 vertices or $\le$ 8 edges), a model $M$ can subgraph-count $F$ iff $\mathsf{Spasm}(F)\subset \mathcal F^M$. We have now successfully extended the result to **all graphs** (see Theorem 4.5 in the updated paper). We are thus excited to see that homomorphism expressivity can **completely** characterize the subgraph counting power of different models!

**Improved description of homomorphism expressivity.** In the initial submission, the homomorphism expressivity of a GNN model $M$ is typically described in the following way: $\mathcal F^M=\mathsf{Clo}_\mathsf{ind}(\\\{F: F \text{ has a certain type of NED} \\\})$.

 A potential drawback of such a description is that it is still not easy to check whether a graph $F\notin F^M$ due to the presence of $\mathsf{Clo}_\mathsf{ind}$, since one needs to prove that for infinitely many $G$ where $F$ is an induced subgraph of $G$, $G$ does not have a certain type of NED.

This issue inspires us to think about whether it is possible to remove $\mathsf{Clo}_\mathsf{ind}$. We have recently succeeded in achieving this goal and obtained a cleaner description. The crux here is to slightly relax the definition of NED. In the previous definition, an ear $P_i$ is nested on another ear $P_j$ if *both* endpoints of $P_i$ lie in $P_j$; We now relax the definition that allows the situation where only one endpoint of $P_i$ lies in $P_j$ and the other endpoint is not in any previous ear. We note that this modification yields a novel extension of NED that does not appear in prior work. Under the new definition, we successfully proved the following results:

> **Theorem**: The homomorphism expressivity of the following architectures exists and can be described below:
>   - **Subgraph GNN**: $\mathcal F^\mathsf{Sub}=\\\{F:F\text{ has an endpoint-shared NED}\\\}$;
>   - **Local 2-GNN**: $\mathcal F^\mathsf{L}=\\\{F:F\text{ has a strong NED}\\\}$.
>   - **Local 2-FGNN**: $\mathcal F^\mathsf{LF}=\\\{F:F\text{ has an almost-strong NED}\\\}$.
>   - **2-FGNN**: $\mathcal F^\mathsf{F}=\\\{F:F\text{ has a NED}\\\}$.

One can see that the updated theorem is much simpler and more elegant than the previous one.

---

**Paper updates.** Below, we highlight the major updates of the revised submission.
- **Definition 3.2 and Theorems 3.3, 3.6, and 3.8.** The definition of NED has been subtly adjusted, and $\mathsf{Clo}_\mathsf{ind}$ has been removed in these theorems.
- **Theorem 4.5.** The condition restricting graphs to moderate-sizes has been removed. [Reviewer **TDGD**]
- **Section 5.** Add new experiments on ZINC-full dataset. [Reviewer **3wF7**]
- **Figure 9.** Add more examples and illustrations for different types of NED. [Reviewers **aNaP** and **wf9E**]
- **Appendix A.** Discussions on more related work. [Reviewer **aNaP**]
- In the Appendix, we have also revised the original proofs for the updated theorems.

---

### Meta-Review · Area_Chair_T2xu · 2023-12-15

**Metareview:**

All reviewers agreed that this is a strong paper presenting a novel theoretical framework for assessing the expressive power of graph neural networks.

**Justification For Why Not Higher Score:**

-

**Justification For Why Not Lower Score:**

strong paper, with both theoretical and experimental results. Theory has direct applications and extends our understanding on how to formalize the expressive power of GNNs.

---

### Decision · Program_Chairs · 2024-01-16

Accept (oral)